# NASH limits anti-tumour surveillance in immunotherapy-treated HCC

Hepatocellular carcinoma (HCC) can have viral or non-viral causes[1–5]. Non-alcoholic steatohepatitis (NASH) is an important driver of HCC. Immunotherapy has been approved for treating HCC, but biomarker-based stratification of patients for optimal response to therapy is an unmet need[6,7]. Here we report the progressive accumulation of exhausted, unconventionally activated CD8[+]PD1[+] T cells in NASH-affected livers. In preclinical models of NASH-induced HCC, therapeutic immunotherapy targeted at programmed death-1 (PD1) expanded activated CD8[+]PD1[+] T cells within tumours but did not lead to tumour regression, which indicates that tumour immune surveillance was impaired. When given prophylactically, anti-PD1 treatment led to an increase in the incidence of NASH–HCC and in the number and size of tumour nodules, which correlated with increased hepatic CD8[+]PD1[+]CXCR6[+], TOX[+], and TNF[+] T cells. The increase in HCC triggered by anti-PD1 treatment was prevented by depletion of CD8[+] T cells or TNF neutralization, suggesting that CD8[+] T cells help to induce NASH–HCC, rather than invigorating or executing immune surveillance. We found similar phenotypic and functional profiles in hepatic CD8[+]PD1[+] T cells from humans with NAFLD or NASH. A meta-analysis of three randomized phase III clinical trials that tested inhibitors of PDL1 (programmed death-ligand 1) or PD1 in more than 1,600 patients with advanced HCC revealed that immune therapy did not improve survival in patients with non-viral HCC. In two additional cohorts, patients with NASH-driven HCC who received anti-PD1 or anti-PDL1 treatment showed reduced overall survival compared to patients with other aetiologies. Collectively, these data show that non-viral HCC, and particularly NASH–HCC, might be less responsive to immunotherapy, probably owing to NASH-related aberrant T cell activation causing tissue damage that leads to impaired immune surveillance. Our data provide a rationale for stratification of patients with HCC according to underlying aetiology in studies of immunotherapy as a primary or adjuvant treatment.

Potentially curative treatments for HCC, such as liver transplantation, tumour resection, or ablation, are limited to early-stage tumours[1,2]. Multikinase inhibitors and anti-VEGF-R2 antibodies have been approved for use in advanced HCC[1,2]. Immunotherapy, which is thought to activate T cells or reinvigorate immune surveillance against cancer, showed response rates of 15–30% in patients with HCC[5,8–11]. Nivolumab and pembrolizumab (PD1-directed antibodies) have been approved for treatment of HCC[3,4], although phase III trials failed to reach their primary endpoints to increase survival[1,10,11]. A combination of atezolizumab (anti-PDL1) and bevacizumab (anti-VEGF) demonstrated increased overall and progression-free survival in a phase III trial, making it a first-line treatment for advanced HCC[5]. The efficacy of immunotherapy might be affected by different underlying HCC aetiologies, with diverse hepatic environments distinctly regulating HCC induction and immune responses[6]. Hence, we lack biomarkers that correlate with treatment response to allow patient stratification[12,13]. Non-alcoholic fatty liver disease (NAFLD) is an HCC-causing condition that affects more than 200 million people worldwide[14]. Approximately 10–20% of individuals with NAFLD progress over time from steatosis to NASH[14]. Innate and adaptive immune-cell activation[15–17], in combination with increased metabolites and endoplasmic reticulum stress[16,18], are believed to lead to a cycle of hepatic necro-inflammation and regeneration that potentially leads to HCC[19–21]. NASH has become an emerging risk factor for HCC[1,14,19], which led us to investigate the effects of immunotherapy in NASH–HCC[22–24].

## Hepatic CD8[+]PD1[+] T cells increase in NASH

We fed mice with diets that cause progressive liver damage and NASH over 3–12 months (Extended Data Fig. 1a–c), accompanied by an increase in the frequency of activated CD8[+] T cells expressing CD69, CD44 and PD1 (Extended Data Fig. 1d–g). Single-cell mapping of leukocytes showed altered immune-cell compositions in mice with NASH (Extended Data Fig. 1h, i) with strongly increased numbers of CD8[+]PD1[+] cells (Fig. 1a, b, Extended Data Fig. 1j–m, o). Similarly, elevated CD8[+] and PD1[+] cells were found in a genetic mouse model of NASH[17] (Extended Data Fig. 1n). Messenger RNA in situ hybridization and immunohistochemistry showed that increasing PDL1 expression in hepatocytes and

A list of authors and their affiliations appears online. ✉e-mail: m.heikenwaelder@dkfz-heidelberg.de

non-parenchymal cells correlated with the severity of NASH (Extended Data Fig. 1p). Mass spectrometric characterization of CD8+PD1+ T cells from NASH-affected livers indicated enrichment in pathways involved in ongoing T cell activation and differentiation, TNF signalling, and natural killer (NK) cell-like cytotoxicity (Fig. 1c). Single-cell RNA sequencing (scRNA-seq) of cells expressing T cell receptor β-chains (TCRβ) from the livers of mice with NASH showed that CD8+ T cells had gene expression profiles related to cytotoxicity and effector-function (for example, *Gzmk* and *Gzmm*) and inflammation markers (for example, *Ccl3*) with elevated exhaustion traits (for example, *Pdcd1* and *Tox*) (Fig. 1d, e). RNA-velocity analyses demonstrated enhanced transcriptional activity and differentiation from *Sell*-expressing CD8+ to CD8+PD1+ T cells (Extended Data Fig. 1q), indicating local differentiation. Thus, mice with NASH have increased hepatic abundance of CD8+PD1+ T cells with features of exhaustion and effector functions.

The high numbers of T cells in NASH suggest that anti-PD1-targeted immunotherapy may serve as an efficient therapy for NASH–HCC. Thirty per cent of C57BL/6 mice fed a choline-deficient high-fat diet (CD-HFD) for 13 months developed liver tumours with a similar load of genetic alterations to human NAFLD–HCC or NASH–HCC (Extended Data Fig. 2a, b). NASH mice bearing HCC (identified using MRI) were allocated to anti-PD1 immunotherapy or control arms (Fig. 1f). None of the pre-existing liver tumours regressed in response to anti-PD1 therapy (Fig. 1g, h, Extended Data Fig. 2c). Rather, we observed increased fibrosis, unchanged liver damage, slightly increased incidence of liver cancer and unaltered tumour loads and sizes after anti-PD1 treatment (Extended Data Fig. 2 d–h). In anti-PD1-treated mice, liver tumour tissue contained increased numbers of CD8+/PD1+ T cells and high levels of cells expressing *Cxcr6* or *Tnf* mRNA (Extended Data Fig. 2i–n). We found no regression of NASH-induced liver tumours upon anti-PDL1 immunotherapy (Extended Data Fig. 3a–f). By contrast, other (non-NASH) mouse models of liver cancer (with or without concomitant damage) reacted to PD1 immunotherapy with tumour regression[25], suggesting that lack of response to immunotherapy was associated specifically with NASH–HCC (Extended Data Fig. 3g–i). Thus, NASH precluded efficient anti-tumour surveillance in the context of HCC immunotherapy. Similarly, impaired immunotherapy has been described in mouse models with NASH and secondary liver cancer[25,26].

## CD8+ T cells promote HCC in NASH

As CD8+PD1+ T cells failed to execute effective immune surveillance, but rather showed tissue-damaging potential, we reasoned that CD8+ T cells might be involved in promoting NASH–HCC. We depleted CD8+ T cells in a preventive setting in mice with NASH but without liver cancer (CD-HFD fed for 10 months). CD8+ T cell depletion significantly decreased liver damage and the incidence of HCC in these mice (Fig. 2i, Extended Data Fig. 4a–j, n). Similar results were obtained after co-depletion of CD8+ and NK1.1+ cells (Fig. 2i, Extended Data Fig. 4a–f, n). This suggests that as well as lacking immune surveillance functions, liver CD8+ T cells also promote HCC in mice with NASH. Next, we investigated the effect of anti-PD1 therapy on HCC development in mice with NASH. Anti-PD1 immunotherapy aggravated liver damage (Fig. 2g, Extended Data Fig. 7c) and increased hepatic CD8+PD1+ T cells, with only minor changes in liver CD4+PD1+ T cells or other immune-cell populations (Extended Data Fig. 4a–o). Anti-PD1 immunotherapy also caused a marked increase in liver-cancer incidence, independent of changes in liver fibrosis (Fig. 2i). Mice lacking PD1 (*Pdcd1−/−*) showed an increase in incidence of, and earlier onset of, liver cancer, along with increased liver damage and elevated numbers of activated hepatic CD8+ T cells with increased cytokine expression (IFNγ, TNF) (Extended Data Fig. 5a–g). In summary, CD8+PD1+ T cells triggered the transition to HCC in mice with NASH, probably owing to impaired tumour surveillance and enhanced T cell-mediated tissue damage[27]. Despite a strong increase in CD8+PD1+ T cells within

tumours, therapeutic PD1- or PDL1-related immunotherapy failed to cause tumour regression in NASH–HCC.

We used an immune-mediated cancer field (ICF) gene-expression signature associated with the development of human HCC[28] to understand the tumour-driving mechanisms of anti-PD1 immunotherapy. Preventive anti-PD1 treatment was strongly associated with the pro-tumorigenic immunosuppressive ICF signature (for example, *Ifng*, *Tnf*, *Stat3*, *Tgfb1*), capturing the traits of T cell exhaustion, pro-carcinogenic signalling, and mediators of immune tolerance and inhibition. Depletion of CD8+ T cells led to significant down-regulation of the high-infiltrate ICF signature and diminished TNF in non-parenchymal cells (Extended Data Fig. 5h, i). Gene set enrichment analysis (GSEA), mRNA in situ hybridization, and histology of tumours developed in NASH mice that were treated prophylactically with anti-PD1 corroborated these data, showing increased CD8+ T cell abundance and enrichment for genes involved in inflammation-related signalling, apoptosis, and TGFβ signalling (Extended Data Fig. 5j–l). Anti-PD1 treatment triggered the expression of p62 (Extended Data Fig. 5m), which has been shown to drive hepatocarcinogenesis[29]. Array comparative genomic hybridization identified no significant differences in chromosomal deletions or amplifications between tumours from anti-PD1-treated mice or control mice (Extended Data Fig. 5n). In summary, hepatic CD8+PD1+ T cells did not cause tumour regression during NASH, but rather were linked to HCC development, which was enhanced by anti-PD1 immunotherapy.

We next analysed the hepatic T cell compartment for correlations with inflammation and hepatocarcinogenesis. Comparison of CD8+PD1+ T cells with CD8+ T cells by scRNA-seq showed that the former showed higher expression of genes associated with effector function (for example, increased *Gzma*, *Gzmb*, *Gzmk*, *Prf1*; reduced *Sell*, *Klf2*), exhaustion (for example, increased *Pdcd1*, *Tox*; reduced *Il7r*, *Tcf7*) and tissue residency (for example, increased *Cxcr6*, low levels of Ki-67) (Extended Data Fig. 6a–c). Notably, there was no difference in the transcriptome profiles of CD8+PD1+ T cells in NASH mice after anti-PD1 immunotherapy (Extended Data Fig. 6c), indicating that the number of T cells rather than their functional properties were changed. RNA-velocity blot analyses corroborated these data (Fig. 2a, Extended Data Fig. 6d–f). Similar patterns of markers (for example, *IL7r*, *Sell*, *Tcf7*, *Ccl5*, *Pdcd1*, *Cxcr6*, and *Rgs1*) correlated with latent time and overall transcriptional activity in NASH mice that received either treatment (Fig. 2a, b, Extended Data Fig. 6e, f). Mass spectrometry-based analyses of CD8+ or CD8+PD1+ T cells isolated from NASH mouse livers confirmed these findings (Fig. 2c, Extended Data Fig. 6g).

We characterized the transcriptome profiles of PD1+CD8+ T cells by uniform manifold approximation and projection (UMAP) analysis of high-parametric flow-cytometry data, dissecting the CD8+PD1+ and CD8+PD1− subsets (Fig. 2d). This revealed that CD8+PD1+ cells expressed high levels of effector (for example, *Gzmb*, *Ifng*, *Tnf*) and exhaustion markers (for example, *Eomes*, *Pdcd1*, Ki-67low). In particular, CD8+PD1+TNF+ cells were more abundant upon anti-PD1 treatment (Fig. 2e). Convolutional neural network analysis and manual gating validated this result (Fig. 2f, Extended Data Fig. 6j, k). CD8+PD1+ T cells were non-proliferative in anti-PD1-treated NASH mice; this result was supported by in vitro experiments, in which anti-PD1 treatment led to increased T cell numbers in the absence of proliferation (Extended Data Fig. 6l, m). Notably, CD8+PD1+ T cells from NASH mice showed reduced levels of FOXO1, which indicates an enhanced tissue-residency phenotype[30], potentially combined with boosted effector function, as indicated by higher calcium levels in CD8+PD1+ T cells (Extended Data Fig. 6n, o). Single-cell RNA-seq analysis also showed that CD8+PD1+ T cells from NASH mice had a tissue residency signature (Extended Data Fig. 6b). Thus, upon anti-PD1 immunotherapy in NASH mice, CD8+PD1+ T cells accumulated to high numbers in the liver, revealing a resident-like T cell character with increased expression of CD44, CXCR6, EOMES and TOX and low levels of CD244

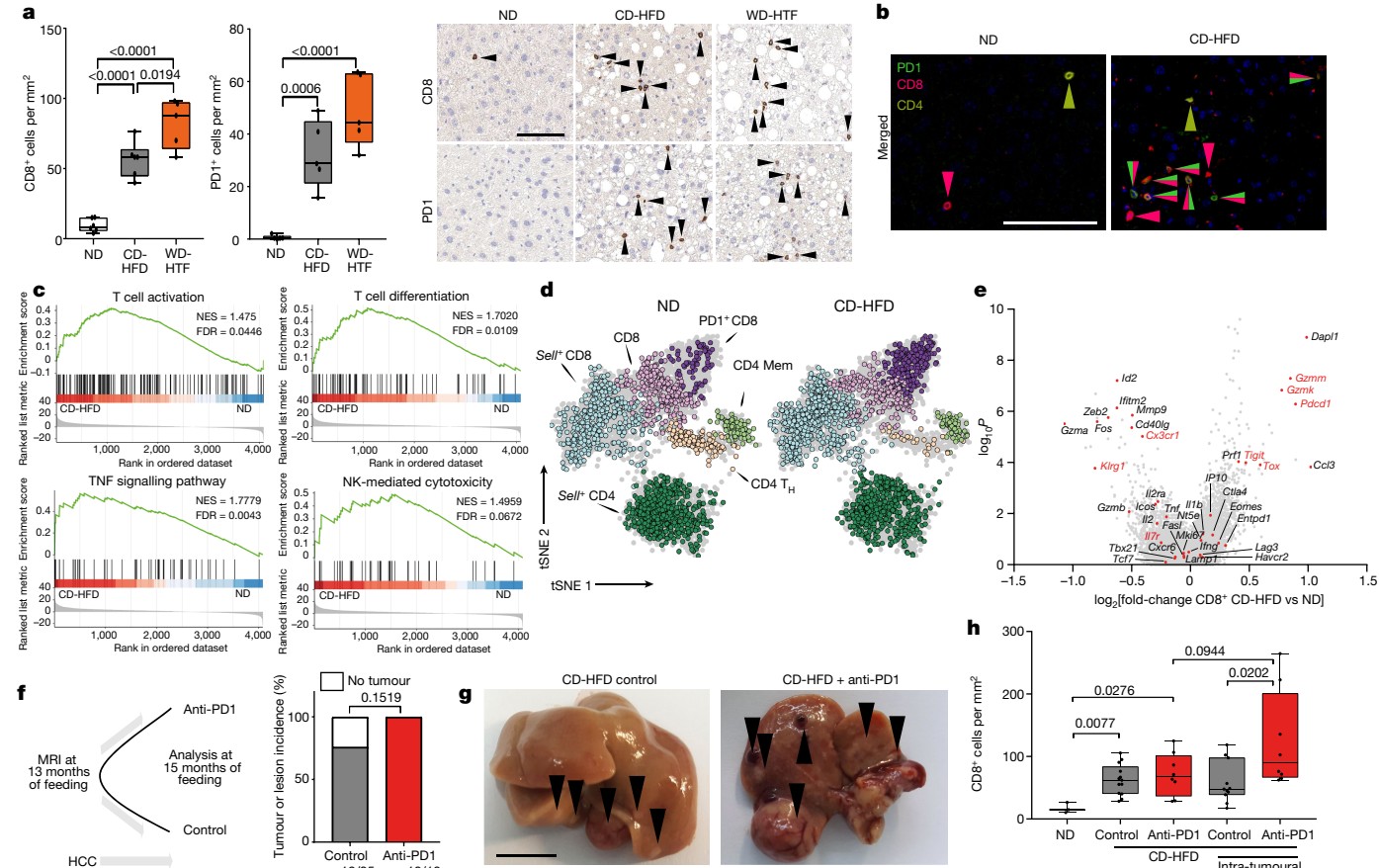

**Fig. 1 | NASH progression is associated with increased, activated CD8⁺PD1⁺ T cells. a**, CD8 and PD1 staining (right) and quantification (left) of T cells from mice fed normal diet (ND), CD-HFD or western-style diet with trans fat (WD-HTF) by immunohistochemistry. **b**, Immunofluorescence-based detection of PD1 (green), CD8 (red) and CD4 (yellow) cells. Scale bar, 100 µm. **c**, GSEA of hepatic CD8⁺PD1⁺ T cells sorted from TCRβ⁺ cells by mass spectrometry. FDR, false discovery rate; NES, normalized enrichment score. **d**–**f**, *t*-distributed stochastic neighbour embedding (tSNE) of TCRβ⁺ cells (**d**),

differential gene expression by scRNA-seq (**e**) and scheme for experiment (**f**): mice were fed CD-HFD for 13 months and then treated with anti-PD1 for 8 weeks before measurement of tumour incidence. Mem, memory CD4 T cells. **g**, Livers from treated and untreated mice after CD-HFD. Arrowheads, tumours or lesions. Scale bar, 10 mm. **h**, Quantification of CD8⁺ cell in liver by immunohistochemistry. Details of sample sizes, biological replicates and statistical tests are given in Methods and Source Data. **a**, **h**, *P* values shown above brackets.

---

expression, but lacking expression of TCF1/TCF7, CD62L, TBET, and CD127 (Extended Data Fig. 6p–u). In summary, anti-PD1 immuno-therapy increased the abundance of CD8⁺PD1⁺ T cells with a residency signature in the liver.

To investigate the mechanisms that drive the increased NASH–HCC transition in the preventive anti-PD1 treatment-setting, we treated NASH-affected mice with combinations of treatments. Both anti-CD8–anti-PD1 and anti-TNF–anti-PD1 antibody treatments ameliorated liver damage, liver pathology and liver inflammation (Fig. 2g, Extended Data Fig. 7), and decreased the incidence of liver cancer compared to anti-PD1 treatment alone (Fig. 2i). By contrast, anti-CD4–anti-PD1 treatment did not reduce the incidence of liver cancer, the NAFLD activity score (NAS), or the number of TNF-expressing hepatic CD8⁺ or CD8⁺PD1⁺CXCR6⁺ T cells (Fig. 2g–i, Extended Data Fig. 7). However, both the number of tumours per liver and tumour size were reduced, suggesting that depletion of CD4⁺ T cells or regulatory T cells might contribute to tumour control (Extended Data Fig. 8a, b). The incidence of tumours was directly correlated with anti-PD1 treatment, alanine aminotransferase (ALT), NAS, number of hepatic CD8⁺PD1⁺ T cells, and TNF expression (Extended Data Fig. 8c–e). These data suggested that CD8⁺PD1⁺ T cells lacked immune-surveillance and had tissue-damaging functions[27], which were increased by anti-PD1 treatment, possibly con-tributing to the unfavourable effects of anti-PD1 treatment on HCC development in NASH.

## Augmented CD8⁺PD1⁺ T cells in human-NASH

We next investigated CD8⁺ T cells from healthy or NAFLD/NASH-affected livers. In two independent cohorts of patients with NASH, we found enrichment of hepatic CD8⁺PD1⁺ T cells with a residency phenotype (by flow cytometry and mass cytometry) (Fig. 3a, b, Extended Data Fig. 9a–j, Supplementary Tables 1,2). The number of hepatic CD8⁺PD1⁺ T cells directly correlated with body-mass index and liver damage (Extended Data Fig. 9b). To investigate similarities between mouse and human T cells from livers with NASH, we analysed liver CD8⁺PD1⁺ T cells from patients with NAFLD or NASH by scRNA-seq. This identified a gene expression signature that was also found in liver T cells from NASH mice (for example, *PDCD1*, *GZMB*, *TOX*, *CXCR6*, *RGS1*, *SELL*) (Fig. 3c, d, Extended Data Fig. 9k, l). Differentially expressed genes were directly correlated between patient- and mouse-derived hepatic CD8⁺PD1⁺ T cells (Fig. 3d). Velocity-blot analyses identified CD8⁺ T cells expressing *TCF7*, *SELL* and *IL7R* as root cells, and CD8⁺PD1⁺ T cells as their endpoints (Fig. 3e, f), indicating a local developmental trajectory of CD8⁺ T cells into CD8⁺PD1⁺ T cells. The amount of gene expression and velocity magnitude, which indicate transcriptional activity, were increased in CD8⁺PD1⁺ T cells from mice and humans with NASH (Fig. 3e). The expres-sion of specific marker genes (for example, *IL7R*, *SELL*, *TCF7*, *CCL5*, *CCL3*, *PDCD1*, *CXCR6*, *RGS1* and *KLF2*) along the latent time in patients with NAFLD or NASH differed from that seen in control participants

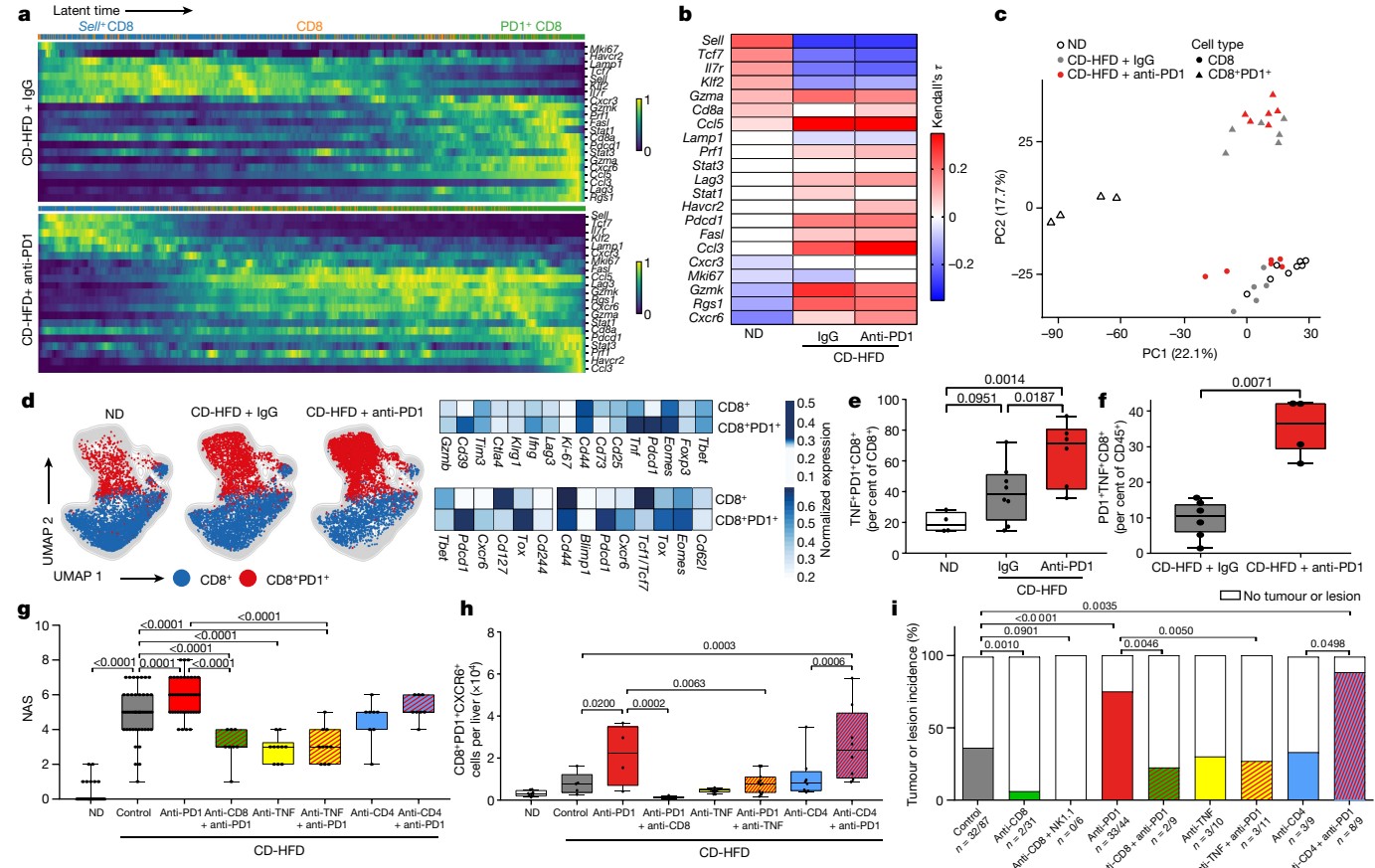

**Fig. 2 | Resident-like CD8⁺PD1⁺ T cells drive hepatocarcinogenesis in a TNF-dependent manner upon anti-PD1 treatment in NASH.**
**a, b**, RNA-velocity analyses of scRNA-seq data showing expression (**a**) and correlation of expression (**b**) along the latent time of selected genes in CD8⁺ T cells from mice with NASH. Latent time (pseudo-time by RNA velocity): dark colour, start of RNA velocity; yellow, end point of latent time. Kendall's τ, gene expression along latent time. **c**, Principal component analysis (PCA) plot of hepatic CD8⁺ and CD8⁺PD1⁺ T cells sorted by mass spectrometry from TCRβ⁺ cells from mice fed for 12 months with ND, CD-HFD or CD-HFD and treated for 8 weeks with anti-PD1 antibodies. **d, e**, UMAP representations showing FlowSOM-guided clustering (**d**, left), heat map showing median marker expression (**d**, right), and quantification of hepatic CD8⁺ T cells (**e**) from mice

fed for 12 months with ND or CD-HFD and treated for 8 weeks with IgG or anti-PD1 antibodies. **f**, Quantification of CellCNN-analysed flow cytometry data for hepatic CD8⁺ T cells from mice fed for 12 months with CD-HFD and treated for 8 weeks with IgG or anti-PD1 antibodies. **g, h**, NAS evaluation (**g**) and quantification of hepatic CD8⁺PD1⁺CXCR6⁺ T cells (**h**) from mice fed with ND for 12 months or fed with CD-HFD for 12 months and treated for 8 weeks with anti-PD1, anti-PD1 + anti-CD8, anti-TNF, anti-PD1 + anti-TNF, anti-CD4, or anti-PD1 + anti-CD4 antibodies. Kendall's τ, gene expression along latent time. **i**, Quantification of tumour incidence in mice as in **g, h**. Details of sample sizes, biological replicates and statistical tests are given in Methods and Source Data. **e–i**, P values shown above brackets.

(Fig. 3g), and correlated with the expression patterns seen in CD8⁺ T cells from NASH mice (Fig. 3h). Thus, scRNA-seq analysis demonstrated a resident-like liver CD8⁺PD1⁺ T cell population in patients with NAFLD or NASH that shared gene expression patterns with hepatic CD8⁺PD1⁺ T cells from NASH mice.

Different stages of NASH severity are considered to herald the development of liver cancer[31]. Indeed, different fibrosis stages (F0–F4) in patients with NASH correlated directly with the expression of *PDCD1*, *CCL2*, *IP10* and *TNF*, and the degree of fibrosis correlated with the numbers of CD4⁺, PD1⁺, and CD8⁺ T cells (Extended Data Fig. 10a–d, Supplementary Table 3). Moreover, PD1⁺ cells were absent from healthy livers but present in the livers of patients with NASH or NASH–HCC, but the number of these cells did not differ with the underlying fibrosis level (Extended Data Fig. 10e, Supplementary Tables 4–6). Species-specific effects, such as the absence in mice of cirrhosis or burnt-out NASH (a condition found in some patients with NASH–HCC[32]), and their possible influence on immunotherapy may make it difficult to translate findings from preclinical models of NASH to human NASH. However, in tumour tissue from patients with NASH-induced HCC—treated with anti-PD1 therapy—we found increased numbers of intra-tumoral PD1⁺

cells compared to patients with HCC and viral hepatitis (Extended Data Fig. 10f). Thus, we found a shared gene-expression profile and increased abundance of unconventionally activated hepatic CD8⁺PD1⁺ T cells in human NASH tissue.

## Lack of immunotherapy response in human NASH–HCC

To explore the concept of disrupted immune surveillance in NASH after anti-PD1 or anti-PDL1 treatment, we conducted a meta-analysis of three large randomized controlled phase III trials of immunotherapies in patients with advanced HCC (CheckMate-459[11], IMbrave150[5] and KEYNOTE-240[10]). Although immunotherapy improved survival in the overall population (hazard ratio (HR) 0.77; 95% confidence interval (CI) 0.63–0.94), survival was superior to the control arm in patients with HBV-related HCC (n = 574; P = 0.0008) and HCV-related HCC (n = 345; P = 0.04), but not in patients with non-viral HCC (n = 737; P = 0.39) (Fig. 4a, Extended Data Fig. 10g, Supplementary Table 7). Patients with viral aetiology (HBV or HCV infection) of liver damage and HCC showed a benefit from checkpoint inhibition (HR 0.64; 95% CI 0.48–0.94), whereas patients with HCC of a non-viral aetiology

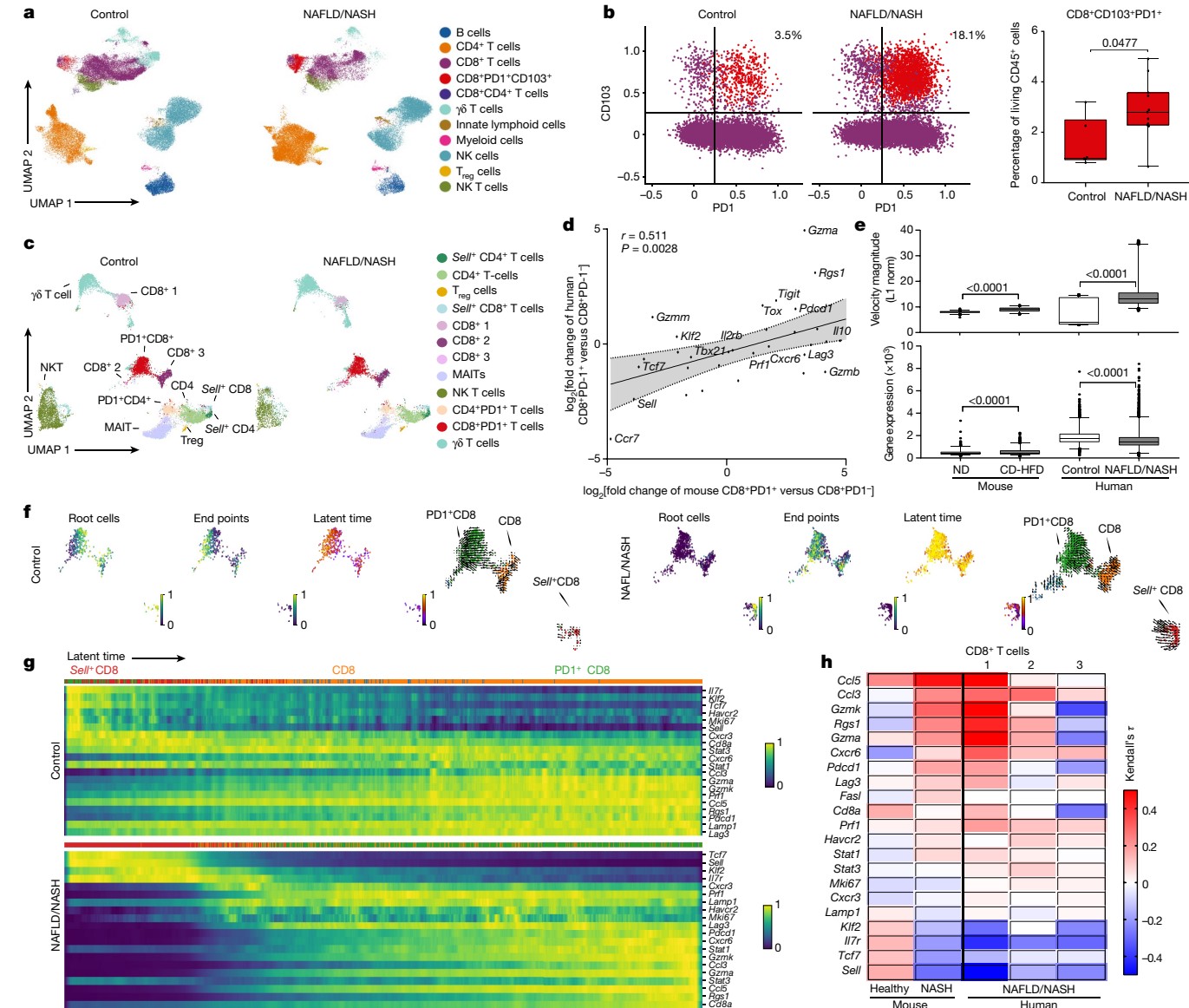

**Fig. 3 | Hepatic resident-like CD8⁺PD1⁺ T cells are increased in livers of patients with NAFLD patients. a**, **b**, UMAP representation showing the FlowSOM-guided clustering of CD45⁺ cells (**a**) and flow cytometry plots (**b**, left) and quantification (**b**, right) of CD8⁺PD1⁺CD103⁺ cells derived from hepatic biopsies of healthy individuals or patients with NAFLD or NASH (Supplementary Table 2). Populations in **b**: violet, CD8⁺; red, CD8⁺PD1⁺CD103⁺. T_reg cells, regulatory T cells. **c**, UMAP representations and analyses of differential gene expression by scRNA-seq of CD3⁺ cells from control individuals or patients with NAFLD or NASH. MAITs, mucosal-associated invariant T cells. **d**, Correlation of significant differentially expressed genes in liver-derived CD8⁺PD1⁺ T cells compared to CD8⁺PD1⁻ T cells from mice fed with CD-HFD for 12 months and patients with NAFLD/NASH. Shading shows 95% CI.

**e**–**h**, Expression (**e**) and transcriptional activity (**f**) of velocity analyses of scRNA-seq data, and gene expression (**g**) and correlation (**h**) of expression along the latent-time of selected genes along the latent-time of liver-derived CD8⁺ T cells from patients with NAFLD or NASH in comparison to control or NASH mouse liver-derived CD8⁺ T cells. Root cells: yellow, root cells; blue, cells furthest from the root by RNA velocity. End points: yellow, end-point cells; blue, cells furthest from defined end-point cells by RNA velocity. Latent time (pseudo-time by RNA velocity): dark colour, start of RNA velocity; yellow, end point of latent time. RNA velocity flow (top): blue cluster, start point; orange cluster, intermediate; green cluster, end point. Arrow indicates cell trajectory. Details of sample sizes, biological replicates and statistical tests are given in Methods and Source Data. **b**, **e**, *P* values shown above brackets.

did not (HR 0.92; 95% CI 0.77–1.11; *P* of interaction = 0.03 (Fig. 4a)). Subgroup analysis of first-line treatment compared to a control arm treated with sorafenib (*n* = 1,243) confirmed that immunotherapy was superior in patients with HBV-related (*n* = 473; *P* = 0.03) or HCV-related HCC (*n* = 281; *P* = 0.03), but not in patients with non-viral HCC (*n* = 489; *P* = 0.62; Extended Data Fig. 10h–j). We acknowledge that these results were derived from a meta-analysis of trials that included different lines of treatment and patients with heterogeneous liver damage, and did not differentiate between alcoholic liver disease and NAFLD or NASH. Nevertheless, the results of this meta-analysis supported the notion that stratification of patients according to the aetiology of their liver

damage and ensuing HCC identified patients who responded well to therapy.

To specifically characterize the effect of anti-PD(L)1 immunotherapy with respect to underlying liver disease, we investigated a cohort of 130 patients with HCC (patients with NAFLD *n* = 13; patients with other aetiologies *n* = 117) (Supplementary Table 8). NAFLD was associated with shortened median overall survival after immunotherapy (5.4 months (95% CI 1.8–9.0 months) versus 11.0 months (95% CI 7.5–14.5 months); *P* = 0.023), even though patients with NAFLD had less frequent macro-vascular tumour invasion (23% versus 49%), and immunotherapy was more often used as a first-line therapy in these patients (46% versus 23%;

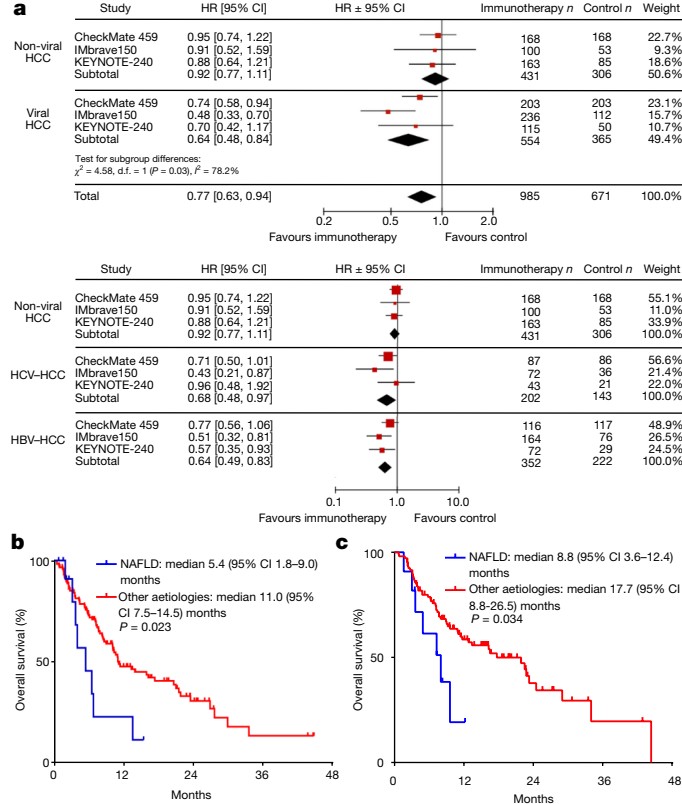

**Fig. 4 | PD1 and PDL1 targeted immunotherapy in advanced HCC has a distinct effect depending on disease aetiology. a**, Meta-analysis of 1,656 patients (Supplementary Table 7). Immunotherapy was initially assessed and then analysed according to disease aetiology: non-viral (NASH and alcohol intake) vs viral (HBV and HCV) (top). Heterogeneity: $\tau^2 = 0.00$; $\chi^2 = 0.14$, degrees of freedom (d.f.) = 2 ($P = 0.93$); $I^2 = 0\%$. Test for overall effect: $Z = 0.87$ ($P = 0.39$). Separate meta-analyses were subsequently performed for each of the three aetiologies: non-viral (NASH and alcohol intake), HCV and HBV (bottom). Heterogeneity: $\tau^2 = 0.03$; $\chi^2 = 3.67$, d.f. = 2 ($P = 0.16$); $I^2 = 46\%$. Test for overall effect: $Z = 3.13$ ($P = 0.002$). Diamonds represent estimated overall effect based on the meta-analysis random effect of all trials. Inverse variance and random effects methods were used to calculate HRs, 95% CIs, $P$ values, and the test for overall effect; calculations were two-sided. **b**, NAFLD is associated with a worse outcome in patients with HCC treated with PD(L)1-targeted immunotherapy. A total of 130 patients with advanced HCC received PD(L)1-targeted immunotherapy (Supplementary Table 8). **c**, Validation cohort of patients with HCC treated with PD(L)1-targeted immunotherapy. A total of 118 patients with advanced HCC received PD(L)1-targeted immunotherapy (Supplementary Table 10). **b**, **c**, Log-rank test. Details of sample sizes, biological replicates and statistical tests are given in Methods and Source Data.

Fig. 4b). After correction for potentially confounding factors that are relevant for prognosis, including severity of liver damage, macrovascular tumour invasion, extrahepatic metastases, performance status, and alpha-fetoprotein (AFP), NAFLD remained independently associated with shortened survival of patients with HCC after anti-PD1-treatment (HR 2.6; 95% CI 1.2–5.6; $P = 0.017$, Supplementary Table 9). This finding was validated in a further cohort of 118 patients with HCC who were treated with PD(L)1-targeted immunotherapy (patients with NAFLD $n = 11$; patients with other aetiologies $n = 107$) (Supplementary Table 10). NAFLD was again associated with reduced survival of patients with HCC (median overall survival 8.8 months, 95% CI 3.6–12.4 months) compared to other aetiologies of liver damage (median overall survival 17.7 months, 95% CI 8.8–26.5 months; $P = 0.034$) (Fig. 4c). Given the relatively small number of patients with NAFLD in both cohorts, these data need prospective validation. However, collectively these

results indicate that patients with underlying NASH did not benefit from checkpoint-inhibition therapy.

Liver cancer develops primarily on the basis of chronic inflammation. The latter can be activated by immunotherapy to induce tumour regression in a subset of patients with liver cancer. However, the identification of patients who will respond to immunotherapy for HCC remains difficult. Our data identify a non-viral aetiology of liver damage and cancer (that is, NASH) as a predictor of unfavourable outcome in patients treated with immune-checkpoint inhibitors. The better response to immunotherapy in patients with virus-induced HCC than in patients with non-viral HCC might be due to the amount or quality of viral antigens or to a different liver micro-environment, possibly one that does not impair immune surveillance. These results might also have implications for patients with obesity and NALFD or NASH who have cancer at other organ sites (for example, melanoma, colon carcinoma, or breast cancer) and are at risk for liver damage and the development of liver cancer in response to systemically applied immunotherapy. Overall, our results provide comprehensive mechanistic insight and a rational basis for the stratification of patients with HCC according to their aetiology of liver damage and cancer for the design of future trials of personalized cancer therapy.

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

**Dominik Pfister[1,82], Nicolás Gonzalo Núñez[2], Roser Pinyol[3], Olivier Govaere[4], Matthias Pinter[5,6], Marta Szydlowska[1], Revant Gupta[7,8], Mengjie Qiu[9], Aleksandra Deczkowska[10], Assaf Weiner[10], Florian Müller[1], Ankit Sinha[11,12], Ekaterina Friebel[2], Thomas Engleitner[13,14,15], Daniela Lenggenhager[16], Anja Moncsek[17], Danijela Heide[1], Kristin Stirm[1], Jan Kosla[1], Eleni Kotsiliti[1], Valentina Leone[1,18], Michael Dudek[19], Suhail Yousuf[9], Donato Inverso[20,21], Indrabahadur Singh[1,22], Ana Teijeiro[23], Florian Castet[3], Carla Montironi[3], Philipp K. Haber[24], Dina Tiniakos[4,25], Pierre Bedossa[4], Simon Cockell[26], Ramy Younes[4,27], Michele Vacca[28], Fabio Marra[29], Jörn M. Schattenberg[30], Michael Allison[31], Elisabetta Bugianesi[27], Vlad Ratziu[32], Tiziana Pressiani[33], Antonio D'Alessio[33], Nicola Personeni[33,34], Lorenza Rimassa[33,34], Ann K. Daly[4], Bernhard Scheiner[5,6], Katharina Pomej[5,6], Martha M. Kirstein[35,36], Arndt Vogel[35], Markus Peck-Radosavljevic[37], Florian Hucke[37], Fabian Finkelmeier[38], Oliver Waidmann[38], Jörg Trojan[38], Kornelius Schulze[39], Henning Wege[39], Sandra Koch[40], Arndt Weinmann[40], Marco Bueter[41], Fabian Rössler[41], Alexander Siebenhüner[42], Sara De Dosso[43], Jan-Philipp Mallm[44], Viktor Umansky[45,46], Manfred Jugold[47], Tom Luedde[48], Andrea Schietinger[49,50], Peter Schirmacher[51], Brinda Emu[1], Hellmut G. Augustin[20,21], Adrian Billeter[52], Beat Müller-Stich[52], Hiroto Kikuchi[53], Dan G. Duda[53], Fabian Kütting[54], Dirk-Thomas Waldschmidt[54], Matthias Philip Ebert[55], Nuh Rahbari[56], Henrik E. Mei[57], Axel Ronald Schulz[57], Marc Ringelhan[58,59,60], Nisar Malek[61], Stephan Spahn[61], Michael Bitzer[61], Marina Ruiz de Galarreta[24,62], Amaia Lujambio[24,62,63], Jean-Francois Dufour[64,65], Thomas U. Marron[24,66], Ahmed Kaseb[67], Masatoshi Kudo[68], Yi-Hsiang Huang[69,70], Nabil Djouder[23], Katharina Wolter[71,72], Lars Zender[71,72,73], Parice N. Marche[74,75], Thomas Decaens[74,75,76], David J. Pinato[77,78], Roland Rad[13,14,15], Joachim C. Mertens[17], Achim Weber[16,79], Kristian Unger[18], Felix Meissner[11], Susanne Roth[9], Zuzana Macek Jilkova[74,75,77], Manfred Claassen[7,8], Quentin M. Anstee[4,80], Ido Amit[10], Percy Knolle[19], Burkhard Becher[2], Josep M. Llovet[3,24,81]✉ & Mathias Heikenwalder[1]✉**

[1]Division of Chronic Inflammation and Cancer, German Cancer Research Center (DKFZ), Heidelberg, Germany. [2]Institute of Experimental Immunology, University of Zurich, Zurich, Switzerland. [3]Liver Cancer Translational Research Laboratory, Institut d'Investigacions Biomèdiques August Pi i Sunyer (IDIBAPS)-Hospital Clínic, Liver Unit, Universitat de Barcelona, Barcelona, Spain. [4]Translational and Clinical Research Institute, Faculty of Medical Sciences, Newcastle University, Newcastle, UK. [5]Division of Gastroenterology and Hepatology, Department of Internal Medicine III, Medical University of Vienna, Vienna, Austria. [6]Liver Cancer (HCC) Study Group Vienna, Medical University of Vienna, Vienna, Austria. [7]Internal Medicine I, University Hospital Tübingen, Faculty of Medicine, University of Tübingen, Tübingen, Germany. [8]Department of Computer Science, University of Tübingen, Tübingen, Germany. [9]Department of General, Visceral and Transplantation Surgery, Universitätsklinikum Heidelberg, Heidelberg, Germany. [10]Department of Immunology, Weizmann Institute of Science, Rehovot, Israel. [11]Experimental Systems Immunology Laboratory, Max-Planck Institute of Biochemistry, Munich, Germany. [12]Institute of Translational Cancer Research and Experimental Cancer Therapy, Klinikum rechts der Isar, Technical University Munich, Munich, Germany. [13]Center for Translational Cancer Research (TranslaTUM), Technical University Munich, Munich, Germany. [14]Department of Medicine II, Klinikum Rechts der Isar, Technical University Munich, Munich, Germany. [15]German Cancer Consortium (DKTK), German Cancer Research Center (DKFZ), Munich, Germany. [16]Department of Pathology and Molecular Pathology, University and University Hospital Zurich, Zurich, Switzerland. [17]Department of Gastroenterology and Hepatology, University Hospital Zurich, Zurich, Switzerland. [18]Research Unit of Radiation Cytogenetics, Helmholtz Zentrum Munich, Munich, Germany. [19]Institute of Molecular Immunology and Experimental Oncology, Technical University Munich, Munich, Germany. [20]Division of Vascular Oncology and Metastasis, German Cancer Research Center (DKFZ-ZMBH Alliance), Heidelberg, Germany. [21]European Center of Angioscience (ECAS), Medical Faculty Mannheim, Heidelberg University, Heidelberg, Germany. [22]Emmy Noether Research Group Epigenetic Machineries and Cancer, Division of Chronic Inflammation and Cancer, German Cancer Research Center (DKFZ), Heidelberg, Germany. [23]Cancer Cell Biology Programme, Growth Factors, Nutrients and Cancer Group, Spanish National Cancer Research Centre, CNIO, Madrid, Spain. [24]Mount Sinai Liver Cancer Program, Division of Liver Diseases, Icahn School of Medicine at Mount Sinai, New York, NY, USA. [25]Department of Pathology, Aretaieion Hospita, National and Kapodistrian University of Athens, Athens, Greece. [26]Bioinformatics Support Unit, Faculty of Medical Sciences, Newcastle University, Newcastle, UK. [27]Department of Medical Sciences, Division of Gastro-Hepatology, A.O. Città della Salute e della Scienza di Torino, University of Turin, Turn, Italy. [28]University of Cambridge Metabolic Research Laboratories, Wellcome-MRC Institute of Metabolic Science, Addenbrooke's Hospital, Cambridge, UK. [29]Dipartimento di Medicina Sperimentale e Clinica, University of Florence, Florence, Italy. [30]Metabolic Liver Research Program, I. Department of Medicine, University Medical Center Mainz, Mainz, Germany. [31]Liver Unit, Department of Medicine, Cambridge Biomedical Research Centre, Cambridge University NHS Foundation Trust, Cambridge, UK. [32]Assistance Publique-Hôpitaux de Paris, Hôpital Beaujon, University Paris-Diderot, Paris, France. [33]Medical Oncology and Hematology Unit, Humanitas Cancer Center, Humanitas Clinical and Research Center-IRCCS, Milan, Italy. [34]Department of Biomedical Sciences, Humanitas University, Milan, Italy. [35]Department of Gastroenterology, Hepatology and Endocrinology, Hannover Medical School, Hannover, Germany. [36]University Medical Center Schleswig-Holstein, Schleswig-Holstein, Germany. [37]Department of Internal Medicine and Gastroenterology (IMuG), Hepatology, Endocrinology, Rheumatology and Nephrology including Centralized Emergency Department (ZAE), Klinikum Klagenfurt am Wörthersee, Klagenfurt, Austria. [38]Department of Gastroenterology, Hepatology and Endocrinology, University Hospital Frankfurt, Frankfurt, Germany. [39]Department of Internal Medicine, Gastroenterology & Hepatology, University Medical Center Hamburg-Eppendorf, Hamburg, Germany. [40]Department of Internal Medicine, University Medical Center of the Johannes Gutenberg University Mainz, Mainz, Germany. [41]Department of Surgery and Transplantation, University Hospital Zurich, Zurich, Switzerland. [42]Department of Medical Oncology and Hematology, University Hospital Zurich and University of Zurich, Zurich, Switzerland. [43]Oncology Institute of Southern Switzerland, Bellinzona, Switzerland. [44]Division of Chromatin Networks, German Cancer Research Center (DKFZ) and Bioquant, Heidelberg, Germany. [45]Clinical Cooperation Unit Dermato-Oncology, German Cancer Research Center (DKFZ), Heidelberg, Germany. [46]Department of Dermatology, Venereology and Allergology, University Medical Center Mannheim, Ruprecht-Karl University of Heidelberg, Heidelberg, Germany. [47]Core Facility Small Animal Imaging, German Cancer Research Center Heidelberg, Heidelberg, Germany. [48]Department of Gastroenterology, Hepatology and Infectious Diseases, Medical Faculty, Heinrich-Heine-University, Düsseldorf, Germany. [49]Immunology Program, Memorial Sloan Kettering Cancer Center, New York, NY, USA. [50]Immunology and Microbial Pathogenesis Program, Weill Cornell Graduate School of Medical Sciences, New York, NY, USA. [51]Institute of Pathology, University Hospital Heidelberg, Heidelberg, Germany. [52]Department of General, Visceral and Transplantation Surgery, Heidelberg University Hospital, Heidelberg, Germany. [53]Edwin L. Steele Laboratories for Tumor Biology, Department of Radiation Oncology, Massachusetts General Hospital, Boston, MA, USA. [54]Department of Gastroenterology and Hepatology, University of Cologne, Cologne, Germany. [55]Department of Medicine II, Medical Faculty Mannheim, Heidelberg University, Heidelberg, Germany. [56]Department of Surgery at University Hospital Mannheim, Medical Faculty Mannheim, Heidelberg University, Heidelberg, Germany. [57]Mass Cytometry Lab, Deutsches Rheumaforschungszentrum Berlin, a Leibniz Institute, Berlin, Germany. [58]Institute of Virology, Technical University Munich/Helmholtz Zentrum Munich, Munich, Germany. [59]Department of Internal Medicine II, University Hospital rechts der Isar, Technical University Munich, Munich, Germany. [60]German Center for Infection Research (DZIF), partner site Munich, Munich, Germany. [61]Medical University Hospital Department of Internal Medicine I, Tübingen, Germany. [62]Department of Oncological Sciences, Icahn School of Medicine at Mount Sinai, New York, NY, USA. [63]The Precision Immunology Institute, Icahn School of Medicine at Mount Sinai, New York, NY, USA. [64]University Clinic for Visceral Surgery and Medicine, Inselspital, Bern, Switzerland. [65]Hepatology, Department of Biomedical Research, University of Bern, Bern, Switzerland. [66]Department of Medicine, Division of Hematology/Oncology, Tisch Cancer Institute, Mount Sinai Hospital, New York, NY, USA. [67]Department of Gastrointestinal Medical Oncology, The University of Texas MD Anderson Cancer Center, Houston, TX, USA. [68]Department of Gastroenterology and Hepatology, Kindai University Faculty of Medicine, Osaka-, Sayama, Japan. [69]Institute of Clinical Medicine, National Yang-Ming University, Taipei, Taiwan. [70]Division of Gastroenterology and Hepatology, Taipei Veterans General Hospital, Taipei, Taiwan. [71]Department of Medical Oncology and Pneumology (Internal Medicine VIII), University Hospital Tübingen, Tübingen, Germany. [72]Cluster of Excellence 'Image Guided and Functionally Instructed Tumor Therapies' (iFIT), Eberhard-Karls University of Tübingen, Tübingen, Germany. [73]German Consortium for Translational Cancer Research (DKTK), Partner Site Tübingen, German Cancer Research Center (DKFZ), Tübingen, Germany. [74]Université Grenoble Alpes, Grenoble, France. [75]Institute for Advanced Biosciences, Research Center UGA/Inserm U 1209/CNRS 5309, Grenoble, France. [76]Service d'hépato-gastroentérologie, Pôle Digidune, CHU Grenoble Alpes, Grenoble, France. [77]Department of Surgery & Cancer, Imperial College London, Hammersmith Hospital, London, UK. [78]Division of Oncology, Department of Translational Medicine, University of Piemonte Orientale, Novara, Italy. [79]Institute of Molecular Cancer Research (IMCR), University of Zurich, Zurich, Switzerland. [80]Newcastle NIHR Biomedical Research Centre, Newcastle upon Tyne Hospitals NHS Trust, Newcastle, UK. [81]Institució Catalana de Recerca i Estudis Avançats (ICREA), Barcelona, Spain. [82]Present address: Liver Disease Research, Global Drug Discovery, Novo Nordisk A/S, Malov, Denmark. ✉e-mail: jmllovet@clinic.cat; m.heikenwaelder@dkfz-heidelberg.de

## Methods

### Mice, diets, and treatments

Standard mouse diet feeding (ad libitum water and food access) and treatment regimens were as described previously[17]. Male mice were housed at the German Cancer Research Center (DKFZ) (constant temperature of 20–24 °C and 45–65% humidity with a 12-h light–dark cycle). Mice were maintained under specific pathogen-free conditions and experiments were performed in accordance with German law and the governmental bodies, and with approval from the Regierungspräsidium Karlsruhe (G11/16, G129/16, G7/17). Tissues from inducible knock-in mice expressing the human unconventional prefoldin RPB5 interactor were received from N. Djouder[17,33]. The plasmids for hydrodynamic tail-vein delivery have been described previously[34–37]. For interventional studies, male mice fed a CD-HFD were treated with bi-weekly for 8 weeks by intravenous injection of 25 µg CD8-depleting antibody (Bioxcell, 2.43), 50 µg NK1.1-depleting antibody (Bioxcell, PK136), 300 µg anti-PDL1 (Bioxcell, 10F.9G2), 200 µg anti-TNF (Bioxcell, XT3.11), 100 µg anti-CD4 (Bioxcell, GK1.5), or 150 µg anti-PD1 (Bioxcell, RMP1-14). PD1$^{-/-}$ mice were kindly provided by G. Tiegs and K. Neumann. Mice for Extended Data Fig. 3g were treated with anti-PD1 antibody (Bioxcell, RMP1-14) or isotype control (Bioxcell, 2A3) at an initial dose of 500 µg intraperitoneally (i.p.) followed by doses of 200 µg i.p. bi-weekly for 8 weeks. Mice for Extended Data Fig. 3h were treated i.p. with anti-PD1 (200 µg, Bioxcell, RMP1-14) or IgG (200 µg, Bioxcell, LTF-2). The treatment regimen for Extended Data Fig. 3i was as described elsewhere[38].

Intraperitoneal glucose tolerance test and measurement of serum parameters were as described previously[17].

### Magnetic resonance Imaging

MRI was done in the small animal imaging core facility in DKFZ using a Bruker BioSpec 9.4 Tesla (Ettlingen). Mice were anaesthetized with 3.5% sevoflurane, and imaged with T2-weighted imaging using a T2_TurboRARE sequence: TE = 22 ms, TR = 2,200 ms, field of view (FOV) 35 × 35 mm, slice thickness 1 mm, averages = 6, scan time 3 min 18 s, echo spacing 11 ms, rare factor 8, slices 20, image size 192 × 192 pixels, resolution 0.182 × 0.182 mm.

### Multiplex ELISA

Liver homogenates were prepared as for western blotting[17] and cytokines or chemokines were analysed on a customized ELISA according to the manufacturer's manual (Meso Scale Discovery, U-PLEX Biomarker group 1, K15069L-1).

### Flow cytometry and FACS

**Isolation and staining of lymphocytes.** After perfusion and mechanical dissection, livers were incubated for up to 35 min at 37 °C with collagen IV (60 U final concentration (f.c.)) and DNase I (25 µg/ml f.c.), filtered at 100 µm, and washed with RPMI1640 (11875093, Thermo Fisher). Next, samples underwent a two-step Percoll gradient (25%/50% Percoll/HBSS) and centrifugation for 15 min at 1,800g and 4 °C. Enriched leukocytes were then collected, washed, and counted. For re-stimulation, cells were incubated for 2 h at 37 °C under 5% $CO_2$ with 1:500 Biolegend´s Cell Activation Cocktail (with brefeldin A) (423304) and 1:1,000 Monensin Solution (420701). Live/dead discrimination was done using DAPI or ZombieDyeNIR according to the manufacturer's instructions with subsequent staining of titrated antibodies (Supplementary Tables 12–14). Samples for flow cytometric-activated cell sorting (FACS) were sorted and samples for flow cytometry were fixed using eBioscience IC fixation (00-8222-49) or FOXP3 Fix/Perm kit (00-5523-00) according to the manufacturer's instructions. Intracellular staining was performed in eBioscience Perm buffer (00-8333-56). Cells were analysed using BD FACSFortessa or BD FACSSymphony and data were analysed using FlowJo (v10.6.2). For sorting, FACS Aria II and FACSAria FUSION were used in collaboration with the DKFZ FACS core facility.

For UMAP and FlowSOM plots, BD FACSymphony data (mouse and human) were exported from FlowJo (v10). Analyses were performed as described elsewhere[39].

### Single-cell RNA-seq and metacell analysis (mouse)

Single-cell capturing for scRNA-seq and library preparation were done as described previously[40]. Libraries (pooled at equimolar concentration) were sequenced on an Illumina NextSeq 500 at a median sequencing depth of ~40,000 reads per cell. Sequences were mapped to the mouse genome (mm10), using HISAT (version 0.1.6); reads with multiple mapping positions were excluded. Reads were associated with genes if they were mapped to an exon, using the Ensembl gene annotation database (Ensembl release 90). Exons of different genes that shared a genomic position on the same strand were considered to represent a single gene with a concatenated gene symbol. The level of spurious unique molecular identifiers (UMIs) in the data was estimated by using statistics on empty MARS-seq wells and excluded rare cases with estimated noise >5% (median estimated noise overall for experiments was 2%). Specific mitochondrial genes, immunoglobulin genes, genes linked with poorly supported transcriptional models (annotated with the prefix "Rp-"), and cells with fewer than 400 UMIs were removed. Gene features were selected using Tvm = 0.3 and a minimum total UMI count >50. We carried out hierarchical clustering of the correlation matrix between those genes (filtering genes with low coverage and computing correlation using a down-sampled UMI matrix) and selected the gene clusters that contained anchor genes. We used $K$ = 50, 750 bootstrap iterations, and otherwise standard parameters. Subsets of T cells were obtained by hierarchical clustering of the confusion matrix and supervised analysis of enriched genes in homogeneous groups of metacells[41].

### Velocity and correlation analyses of scRNA-seq data

Velocyto (0.6) was used to estimate the spliced and unspliced counts from the pre-aligned bam files[42]. RNA velocity, latent time, root, and terminal states were calculated using the dynamical velocity model from scvelo (0.2.2)[43]. Kendall's rank correlation coefficient ($\tau$) was used to correlate the expression patterns of biologically significant genes with latent time.

### Preparation for mass spectrometry, data acquisition, and data analysis

After FACS purification, cells were resuspended in 50% (vol/vol) 2,2,2-trifluoroethanol in PBS pH 7.4 buffer and lysed by repeated sonication and freeze–thaw cycles. Proteins were denatured at 60 °C for 2 h, reduced using dithiothreitol at a final concentration of 5 mM (30 min at 60 °C), cooled to room temperature, alkylated using iodoacetamide at 25 mM (30 min at room temperature in the dark), and diluted 1:5 using 100 mM ammonium bicarbonate, pH 8.0. Proteins were digested overnight by trypsin (1:100 ratio, 37 °C), desalted using C18-based stage-tips, dried under vacuum, resuspended in 20 µl HPLC-grade water with 0.1% formic acid, and measured using A380.

We used 0.5 µg of peptides for proteomic analysis on a C18 column using a nano liquid chromatography system (EASY-nLC 1200, Thermo Fisher Scientific). Peptides were eluted using a gradient of 5–30% buffer B (80% acetonitrile and 0.1% formic acid) at a flow rate of 300 nl/min at a column temperature of 55 °C. Data were acquired by data-dependent Top15 acquisition using a high-resolution orbitrap tandem mass spectrometer (QExactive HFX, Thermo Scientific). All MS1 scans were acquired at 60,000 resolution with AGC target of $3 × 10^6$, and MS2 scans were acquired at 15,000 resolution with AGC target of $1 × 10^5$ and maximum injection time of 28 ms. Analyses were performed using MaxQuant (1.6.7.0), mouse UniProt Isoform fasta (Version: 2019-02-21, number of sequences 25,233) as a source for protein sequences. One per cent FDR was used for controlling at the peptide and protein levels, with a minimum of two peptides needed for consideration of analysis. GSEA was performed using ClusterProfiler (3.18)[44] and gene

sets obtained from WikiPathway (https://www.wikipathways.org/) and MSigDB (https://broadinstitute.org/msigdb)[45–47].

## Histology, immunohistochemistry, scanning, and automated analysis

Histology, immunohistochemistry, scanning, and automated analysis have been described previously[17]. Antibodies used in this manuscript are described in Supplementary Table 12. For immunofluorescence staining, established antibodies were used, coupled with the AKOYA Biosciences Opal fluorophore kit (Opal 520 FP1487001KT, Opal 540 FP1494001KT, Opal 620 FP1495001KT). For mRNA in situ hybridization, freshly non-baked 5 μm formalin-fixed paraffin-embedded sections were cut and stained according to the manufacturer's (ACD biotech) protocol for manual assay RNAscope, using probes PDL1 (420501), TNF (311081) and CXCR6 (871991).

## Isolation of RNA and library preparation for bulk RNA sequencing

RNA isolation[17] and library preparation for bulk 3′-sequencing of poly(A)-RNA was as described previously[48]. Gencode gene annotations version M18 and the mouse reference genome major release GRCm38 were derived from https://www.gencodegenes.org/. Dropseq tools v1.12[49] were used for mapping the raw sequencing data to the reference genome. The resulting UMI-filtered count matrix was imported into R v3.4.4. Before differential expression analysis with Limma v3.40.6[50] sample-specific weights were estimated and used as coefficients alongside the experimental groups as a covariate during model fitting with Voom. $t$-test was used for determining differentially ($P < 0.05$) regulated genes between all possible experimental groups. GSEA was conducted with the pre-ranked GSEA method[46] within the MSigDB Reactome, KEGG, and Hallmark databases (https://broadinstitute.org/msigdb). Raw sequencing data are available at European Nucleotide Archive (https://www.ebi.ac.uk/ena/browser/home) under the accession number PRJEB36747.

## Stimulation of CD8 T cells

Stimulation of CD8 T cells was as described elsewhere[27].

## Flow cytometry of human biopsies

Analysis of patient material (Supplementary Table 1) was performed on liver tissue (needle biopsies or resected tissue, BIOFACS Study KEK 2019-00114), which were obtained from the patient collection nAC-2019-3627 (CRB03) from the biological resource centre of CHU Grenoble-Alpes (nBRIF BB-0033-00069). Tissue samples were minced using scalpels, incubated (with 1 mg/ml collagenase IV (Sigma Aldrich), 0.25 μg/ml DNase (Sigma Aldrich), 10% FCS (Thermo Fisher Scientific), RPMI 1640 (Seraglob)) for 30 min at 37 °C, stopping enzymatic reactions with 2 mM EDTA (StemCell Technologies, Inc.) in PBS. After filtering through a 100-μm cell strainer, cells were resuspended in FACS buffer (PBS, EDTA 2 mM, FCS 0.5%) with Human TruStain FcX (Fc Receptor Blocking Solution) (Biolegend), incubated for 15 min at 4 °C and stained with antibodies (Supplementary Table 13).

Flow cytometry of human samples (Extended Data Fig. 9f) was approved by the local ethical committee (AC-2014-2094 n 03).

## High-throughput RNA-seq of human samples

As previously reported, RNA-seq analysis was performed using the data from 206 snap-frozen biopsy samples from 206 patients diagnosed with NAFLD in France, Germany, Italy, and the UK and enrolled in the European NAFLD Registry (GEO accession GSE135251)[51,52]. Samples were scored for NAS by two pathologists[53]. Alternate diagnoses were excluded, including excessive alcohol intake (30 g per day for males, 20 g for females), viral hepatitis, autoimmune liver diseases, and steatogenic medication use. Patient samples were grouped: NAFL ($n = 51$) and NASH with fibrosis stages of F0/1 ($n = 34$), F2 ($n = 53$),

F3 ($n = 54$) and F4 ($n = 14$). Collection and use of data of the European NAFLD Registry were approved by the relevant local and/or national Ethical Review Committee[51]. A correction for sex, batch, and centre effects was implemented. Pathway enrichment and visualization were as described elsewhere[52,54,55].

## Immunohistochemistry of NAFLD/NASH cohort

Sixty-five human FFPE biopsies from patients with NAFLD were included (Supplementary Table 3). Sequential slides were immunostained with antibodies against human CD8 (Roche, SP57, ready-to-use), PD1 (Roche; NAT105, ready-to-use), and CD4 (Abcam, ab133616, 1:500). All staining was performed on the VENTANA BenchMark autostainer at 37 °C. Immunopositive cells were quantified at 400× magnification in the portal tract and the adherent parenchyma.

## Isolation of cells for scRNA-seq data analysis (human)

Analyses used liver samples from patients undergoing bariatric surgery at the Department of Surgery at Heidelberg University Hospital (S-629/2013). Samples were preserved by FFPE for pathological evaluation and single cells were generated by mincing, using the Miltenyi tumour dissociation kit (130-095-929) per the manufacturer's instructions, filtering through a 70-μm cell strainer and washing. ACK lysis using the respective buffer (Thermo Fischer Scientific A1049201) was performed, and samples were stored in FBS with 20% DMSO until further processing (scRNA-seq analysis and mass cytometry).

Cells were thawed in a 37 °C water bath, washed with PBS + 0.05 mM EDTA (10 min, 300$g$ at 4 °C), Fc receptor-block (10 min at 4 °C), stained with CD45-PE (3 μl, Hl30, 12-0459-42) and Live/Dead discrimination (1:1,000, Thermofischer, L34973), washed and sorted on a FACSAria FUSION in collaboration with the DKFZ FACS. Library generation was performed according to the manufacturer's protocol (Chromium Next EM Single Cell 3′GEM, 10000128), and sequencing was performed on an Illumina NovaSeq 6000. De-multiplexing and barcode processing were performed using the Cell Ranger Software Suite (Version 4.0.0) and reads were aligned to human GRCh38[56]. A gene–barcode matrix containing cell barcodes and gene expression counts was generated by counting the single-cell 3′ UMIs, which were imported into R (v4.0.2), where quality control and normalization were executed using Seurat v3[57]. Cells with more than 10% mitochondrial genes, fewer than 200 genes per cell, or more than 6,000 genes per cell were excluded. Matrices from 10 samples were integrated with Seurat v3 to remove batch effects across samples. PCA analysis of filtered gene–barcode matrices of all CD3+ cells, visualized by UMAP (top 50 principal components), and identification of major cell types using the highly variable features and indicative markers were performed. Pairwise comparisons of CD4+ T cells versus CD4+PD1+ T cells and CD8+ T cells versus CD8+PD1+ T cells were performed using the results of differential expression analysis by DESeq2 (v1.28.1)[58], setting CD4+/CD8+ T cells as controls. Volcano plots were then generated using EnhancedVolcano (v1.6.0)[59] to visualize the results of differential expression analysis.

## Mass cytometry data analysis (human)

Antibody conjugates for mass cytometry were purchased from Fluidigm, generated in-house using antibody labelling kits (Fluidigm X8, MCP9), or as described before[60,61]. Antibody cocktails for mass cytometry were cryopreserved as described before[62]. Isolation of cells is described in 'Isolation of cells for scRNA-seq data analysis (human)'. Cells were thawed, transferred into RPMI + benzonase (14 ml RPMI + 0.5 μl benzonase), and centrifuged for 5 min at 500$g$. The cell pellet was resuspended in 1 ml CSM-B (CSM (PBS 0.5% BSA 0.02% sodium azide) +1 μl benzonase), filtered through a 30-μm cell strainer, adjusted to 3 ml, counted, resuspended in 35 μl CSM-B and incubated for 45 min at 4 °C, and 100 μl CSM-B was added. Cells were pooled and stained with a surface antibody cocktail (Supplementary Table 15) for 30 min at 4 °C. Dead cell discrimination was performed with mDOTA-103Rh

(5 min, room temperature). For intracellular staining, the FOXP3 intracellular staining kit from Miltenyi Biotec was used per the manufacturer's instructions, followed by staining for intracellular targets for 30 min at room temperature. Cells were washed, resuspended in 1 ml of iridium intercalator solution, and incubated for 25 min at room temperature. Cells were washed with CSM, PBS, and MilliQ water, adjusted to a final concentration of $7.5 \times 10^5$ cells/ml and supplemented with 4-element EQ beads. The sample was acquired on a Helios mass cytometer and raw data were EQ-Bead-normalized using Helios mass cytometer and Helios instrument software (version 6.7). Compensation was performed in CATALYST (v1.86)[63] and FlowCore (1.50.0). De-barcoding and gating of single, live CD45[+] cells were performed using FlowJo (v10.6.2). Then, data from CD45[+] cells were imported into Cytosplore 2.3.1 and transformed using the arcsinh(5) function. Major immune cell lineages were identified at the first level of a two-level hierarchical stochastic neighbour embedding (HSNE) analysis with default perplexity and iteration settings. HSNE with the same parameters was run on CD3[+] cells to identify T cell phenotypes. Gaussian mean shift clustering was performed in Cytosplore and a heat map of arcsinh(5)-transformed expression values of all antibody targets was generated. Cell type identification was based on the transformed expression values and clusters showing high similarity were merged manually.

## Histological and immunohistochemical analysis of NASH–HCC cohort

Four healthy samples, 16 samples from patients with NASH cases, and non-tumoral tissue adjacent to HCC tumours from patients of the following aetiologies were selected: NASH ($n = 26$), viral hepatitis ($n = 19$ HCV, $n = 3$ HBV), alcohol ($n = 5$), and other ($n = 2$). All samples were obtained from International Genomic HCC Consortium with IRB approval. After heat-induced antigen retrieval (10 mM sodium citrate buffer (pH 6.0) or Universal HIER antigen retrieval reagent (ab208572) for 15 min ($3 \times 5$ min), the reaction was quenched using 3% hydrogen peroxide. Samples were washed with PBS and incubated with anti-CD8 (Cell Signaling, Danvers, MA) or anti-PD1 (NAT105, ab52587). DAB (3,3′-diaminobenzidine) was used as a detection system (EnVision+ System-HRP, Dako). PD1-positive cases were defined by considering median positivity by immunohistochemistry[64] and using a cutoff of ≥1% of PD1-positive lymphocytes among all lymphocytes present on each slide. Analysis of human samples from the Department of Pathology and Molecular Pathology, University Hospital Zurich (Extended Data Fig. 10), was approved by the local ethics committee (Kantonale Ethikkommission Zürich, KEK-ZH-Nr. 2013-0382 and BASEC-Nr. PB_2018-00252).

## Search strategy, selection criteria, and meta-analysis of phase III clinical trials

The literature search was done through MEDLINE on PubMed, Cochrane Library, Web of Science, and clinicaltrials.gov, using the following searches: 'checkpoint inhibitors', 'HCC', 'phase III', between January 2010 and January 2020, and complemented by manual searches of conference abstracts and presentations. Single-centre, non-controlled trials, studies with insufficient data to extract HRs or 95% confidence intervals, and trials including disease entities other than HCC were excluded. As conference abstracts were not excluded, quality assessment of the included studies was not performed. Three studies[5,10,11] fulfilled the criteria and were included in the quantitative synthesis (Extended Data Fig. 10). The primary outcome of the meta-analysis was overall survival, defined as the time from randomization to death. HRs and CIs related to overall survival were extracted from the papers or conference presentations[5,10,11]. Pooled HRs were calculated using the random-effects model and we used the DerSimonian–Laird method to estimate $\tau^2$, and the generic inverse variance was used for calculating weights [65]. To evaluate heterogeneity among studies, Cochran's $Q$ test and $I^2$ index were used. $P < 0.10$ in the $Q$-test was considered to indicate substantial heterogeneity. $I^2$ was interpreted as suggested in the literature:

0% to 40% might not represent significant heterogeneity; 30% to 60% may represent moderate heterogeneity; 50% to 90% may represent substantial heterogeneity; 75% to 100% represents considerable heterogeneity. All statistical pooled analyses were performed using RevMan 5.3 software.

## A cohort of patients with HCC treated with PD(L)1-targeted immunotherapy

The retrospective analysis was approved by local Ethics Committees. Data from this cohort were published previously[66]. Patients with liver cirrhosis and advanced-stage HCC treated with PD(L)1-targeted immune checkpoint blockers from 12 centres in Austria, Germany, Italy, and Switzerland were included. The $\chi^2$ test or Fisher's exact test were used to compare nominal data. Overall survival was defined as the time from the start of checkpoint inhibitor treatment until death. Patients who were still alive were censored at the date of the last contact. Survival curves were calculated by the Kaplan–Meier method and compared by using the log-rank test. Multivariable analysis was performed by a Cox regression model. Statistical analyses were performed using IBM SPSS Statistics version 25 (SPSS Inc., Chicago, IL).

## A validation cohort of patients with HCC treated with PD1-targeted immune checkpoint blockers

A multi-institutional dataset that included 427 patients with HCC treated with immune checkpoint inhibitors between 2017 and 2019 in 11 tertiary-care referral centres specialized in the treatment of HCC was analysed. Clinical outcomes of this patient cohort have been reported elsewhere[67,68]. Inclusion criteria were: 1) diagnosis of HCC made by histopathology or imaging criteria according to American Association for the Study of Liver Disease and European Association for the Study of the Liver guidelines; 2) systemic therapy with immune checkpoint inhibitors for HCC that was not amenable to curative or loco-regional therapy following local multidisciplinary tumour board review; 3) measurable disease according to RECIST v1.1 criteria at commencement of treatment with immune checkpoint inhibitors. One hundred and eighteen patients with advanced-stage HCC were recruited with Child–Pugh A liver functional reserve, and documented radiologic or clinical diagnosis of cirrhosis. Ethical approval to conduct this study was granted by the Imperial College Tissue Bank (reference number R16008).

## Statistical analyses

No statistical methods were used to predetermine sample size. The experiments were not randomized and the investigators were not blinded to allocation during experiments and outcome assessment. Data were collected in Microsoft Excel. Mouse data are presented as the mean ± s.e.m. Pilot experiments and previously published results were used to estimate the sample size, such that appropriate statistical tests could yield significant results. Statistical analysis was performed using GraphPad Prism software version 7.03 (GraphPad Software). Exact $P$ values lower than $P < 0.1$ are reported and specific tests are indicated in the legends.

## Sample sizes, biological replicates and statistical tests

Fig. 1a: PD1, $n = 5$ mice/group; CD8, ND $n = 6$ mice; CD-HFD $n = 6$ mice; WD-HTF $n = 5$ mice. Scale bar, 100 µm. Fig. 1b: $n = 3$ mice/group. Scale bar, 100 µm. Fig. 1c: ND $n = 4$ mice, CD-HFD $n = 6$ mice. Fig. 1d, e: $n = 3$ mice/group. Fig. 1f: tumour incidence: CD-HFD, $n = 19$ tumours/lesions in 25 mice; CD-HFD + anti-PD1, $n = 10$ tumours/lesions in 10 mice. Fig. 1h: ND, $n = 3$ mice; CD-HFD, $n = 13$ mice; CD-HFD + anti-PD1, $n = 8$ mice; intra-tumoral staining: CD-HFD, $n = 11$ mice; CD-HFD + anti-PD1, $n = 8$ mice. Data in Fig. 1a, h were analysed by two-tailed Student's $t$-test. Data in Fig. 1f were analysed by two-sided Fisher's exact test.

Fig. 2a, b: $n = 3$ mice/group. Fig. 2c: CD8[+]: ND, $n = 6$ mice; CD-HFD + IgG, $n = 5$ mice; CD-HFD + anti-PD1, $n = 6$ mice; CD8[+]PD1[+]: ND, $n = 4$ mice, CD-HFD

+ IgG, $n = 6$ mice; CD-HFD + anti-PD1, $n = 6$ mice. Fig. 2d, e: ND, $n = 4$ mice; CD-HFD + IgG, $n = 8$ mice; CD-HFD + anti-PD1, $n = 6$ mice. Fig. 2f: CD-HFD + IgG, $n = 6$ mice; CD-HFD + anti-PD1, $n = 4$ mice. Fig. 2g: ND, $n = 30$ mice; CD-HFD, $n = 47$ mice; CD-HFD + anti-PD1, $n = 35$ mice; CD-HFD + anti-PD1/anti-CD8, $n = 9$ mice; CD-HFD + anti-TNF, $n = 10$ mice; CD-HFD + anti-PD1/anti-TNF, $n = 11$ mice; CD-HFD + anti-CD4, $n = 8$ mice; CD-HFD + anti-PD1/anti-CD4, $n = 8$ mice. Fig. 2h: CD8$^+$PD1$^+$CXCR6$^+$: ND, $n = 30$ mice; CD-HFD, $n = 47$ mice; CD-HFD + anti-PD1, $n = 35$ mice; CD-HFD + anti-PD1/anti-CD8, $n = 9$ mice; CD-HFD + anti-TNF, $n = 10$ mice; CD-HFD + anti-PD1/anti-TNF, $n = 11$ mice; CD-HFD + anti-CD4, $n = 8$ mice; CD-HFD + anti-PD1/anti-CD4, $n = 8$ mice. Fig. 2j: tumour incidence: CD-HFD, $n = 32$ tumours/lesions in 87 mice; CD-HFD + anti-CD8, $n = 2$ tumours/lesions in 31 mice; CD-HFD + anti-CD8/NK1.1, $n = 0$ tumours/lesions in 6 mice; CD-HFD + anti-PD1, $n = 33$ tumours/lesions in 44 mice; CD-HFD + anti-PD1/anti-CD8, $n = 2$ tumours/lesions in 9 mice; CD-HFD + anti-TNF, $n = 3$ tumours/lesions in 10 mice; CD-HFD + anti-PD1/anti-TNF, $n = 3$ tumours/lesions in 11 mice; CD-HFD + anti-CD4, $n = 3$ tumours/lesions in 9 mice; CD-HFD + anti-PD1/anti-CD4, $n = 8$ tumours/lesions in 9 mice. All data are shown as mean ± s.e.m. Data in Fig. 2e, g, h were analysed by one-way ANOVA and Fisher's LSD test. Data in Fig. 2f were analysed by two-tailed Mann–Whitney test. Data in Fig. 2j were analysed by two-sided Fisher's exact test.

Fig. 3a, b: control, $n = 6$ patients; NAFLD/NASH, $n = 11$ patients. Fig. 3c: control, $n = 4$ patients; NAFLD/NASH, $n = 7$ patients. Fig. 3d–h: mouse, $n = 3$; human, $n = 3$. All data are shown as mean ± s.e.m. Data in Fig. 3b, f were analysed by two-tailed Mann–Whitney test. Data in Fig. 3d were analysed by two-tailed Spearman's correlation.

Fig. 4a: Hazard ratios are represented by squares, the size of the square represents the weight of the trial in the meta-analysis. Cochran's $Q$-test and $I^2$ were used to calculate heterogeneity. Fig. 4b: Kaplan–Meier curve displays overall survival of patients with NAFLD versus those with any other aetiology; all 130 patients were included in these survival analyses (NAFLD $n = 13$; any other aetiology $n = 117$). Fig. 4c: Kaplan–Meier curve displays overall survival of patients with NAFLD versus those with any other aetiology (NAFLD $n = 11$; any other aetiology $n = 107$). Data in Fig. 4b, c were analysed by Kaplan–Meier method and compared using log rank test.

### Reporting summary
Further information on research design is available in the Nature Research Reporting Summary linked to this paper.

### Data availability
The proteomics data described in this article are available at the PRIDE database, under the identifier PXD017236 or through the dataset website (http://www.ebi.ac.uk/pride/archive/projects/PXD017236). The bulk RNA-seq data described in this article are available at the European Nucleotide Archive (ENA) under accession number PRJEB36747. The scRNA-seq data described in this article are available at GEO under accession GSE144635. The array of comparative genomic hybridization data described in this article is available at GEO under accession GSE144875. The results here are in whole or part based upon data generated by the TCGA Research Network (https://www.cancer.gov/tcga). The human scRNA-seq data described in this article are available at GEO under accession GSE159977. Databases used in this manuscript are WikiPathways (https://www.wikipathways.org/) and MSigDB (https://broadinstitute.org/msigdb). Source data are provided with this paper.

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

**Acknowledgements** We thank the Newcastle Molecular Pathology Node Proximity Laboratory for their technical support; P. Sinn for technical support; the DKFZ FACS core facility for support for sorting; G. Tiegs and K. Neumann for support with *Pdcd1*⁻/⁻ mice; V. Eichwald for help with non-invasive imaging of mice; J. Schmid for help with human liver tissue sampling for flow cytometry; A. Teufel for human cohort gathering; K. Inoue and Z. Ammozzar for help with the non-NASH mouse model; T. O'Connor, C. Groth, M. Matter, L. Terracciano, R. Kaeser, T. Boettler, M. Thimme, T. Longerich and B. Müllhaupt for guidance and discussions; and S. Prokosch, U. Rothermel, J. Janzen, J. Hetzer, C. Gropp, S. Jung, L. Beideck, S. Torrecilla, K. E. Lindblad, E. Rist and T.-W. Kang for technical support. D. Pfister was supported by the Helmholtz Future topic Inflammation and Immunology. I.S. is funded by the 'Deutsche Forschungsgemeinschaft' (DFG, Bonn Germany) through Emmy Noether program (SI 2620/1-1). A. Sinha is supported by EMBO LT fellowship

(ALTF 539-2018). F. Meissner is supported in this project by SFB 1335 and BMBF 031B0686B. Q.M.A., O.G., D.T., P.B., S.C., R.Y., M.V., F. Marra, J.M.S., M.A., E.B., V.R. and A.K.D. are supported by the EPoS (Elucidating Pathways of Steatohepatitis) consortium funded by the Horizon 2020 Framework Program of the European Union under Grant Agreement 634413, the LITMUS (Liver Investigation: Testing Marker Utility in Steatohepatitis) consortium funded by the Innovative Medicines Initiative (IMI2) Program of the European Union under Grant Agreement 777377, and are collaborators in the European NAFLD Registry. Q.M.A. is also supported by the Newcastle NIHR Biomedical Research Centre. I.A. is supported by the Chan Zuckerberg Initiative (CZI), an HHMI international scholar award, European Research Council consolidator grant (ERC-COG) 724471-HemTree2.0, the Thompson Family Foundation, an MRA established investigator award (509044), the Israel Science Foundation (703/15), the Ernest and Bonnie Beutler Research Program for Excellence in Genomic Medicine, a Helen and Martin Kimmel award for innovative investigation, a NeuroMac DFG/Transregional Collaborative Research Center grant, International Progressive MS Alliance/NMSS PA-1604-08459, an Adelis Foundation grant, and the SCA award of the Wolfson Foundation. I.A. is the incumbent of the Alan and Laraine Fischer Career Development Chair. A.D. is supported by Steven and Eden Romick. B.S. received travel support from AbbVie and Gilead. J.M.L. is supported through a partnership between Cancer Research UK, Fondazione AIRC, and Fundación Científica de la Asociación Española Contra el Cáncer (HUNTER, ref. C9380/A26813), by the European Commission (EC)/Horizon 2020 Program (HEPCAR, Ref. 667273-2), US Department of Defense (CA150272P3), NCI Cancer Center Support Grant, National Cancer Institute, Tisch Cancer Institute (P30-CA196521), Samuel Waxman Cancer Research Foundation, Spanish National Health Institute (SAF2016-76390 and PID2019-105378RB-I00) and the Generalitat de Catalunya/AGAUR (SGR-1358). R.P. is supported by HEPCAR and AECC. C.M. is supported by a Rio Hortega grant (CM19/00039) from the ISCIII and the European Social Fund. F.C. is supported by grant funding from AECC. A. Weber is supported by a grant from the Swiss National Science Foundation (SNF). M.H. was supported by an ERC Consolidator grant (HepatoMetaboPath), SFBTR179 Project-ID 272983813, SFB/TR 209 Project-ID 314905040, SFBTR1335 Project-ID 360372040, the Wilhelm Sander-Stiftung, the Rainer Hoenig Stiftung, a Horizon 2020 grant (Hepcar), Research Foundation Flanders (FWO) under grant 30826052 (EOS Convention MODEL-IDI), Deutsche Krebshilfe projects 70113166 and 70113167, German-Israeli Cooperation in Cancer Research (DKFZ-MOST) and the Helmholtz-Gemeinschaft, Zukunftsthema 'Immunology and Inflammation' (ZT-0027). P.K.H. is supported by the fellowship grant of the German Research Foundation (HA 8754/1-1). V.U. is supported by the 'Deutsche Forschungsgemeinschaft' (DFG; 259332240/RTG 2099) and Cooperation Program in Cancer Research of the Deutsches Krebsforschungszentrum (DKFZ) and Israel's Ministry of Science, Technology and Space (MOST) (CA181). D.J.P. is supported by grant funding from the Wellcome Trust Strategic Fund (PS3416), ASCO/Conquer Cancer Foundation Global Oncology Young Investigator Award 2019 (14704), Cancer Research UK (C57701/A26137), CW+ and the Westminster Medical School Research Trust (JRC SG 009 2018-19) and received infrastructural support from the Imperial Experimental Cancer Medicine Centre, Cancer Research UK Imperial Centre, the Imperial College Healthcare NHS Trust Tissue Bank and the Imperial College BRC. M.Q., S.Y. and S.R. were supported by German Cancer Aid grants (70112720 and 70113167). J.-F.D. is supported by the Swiss National Foundation and the Swiss Foundation against Liver Cancer. H.E.M. and A.R.S. received support from DFG Me3644/5-1 and the Elke-Kröner-Fresenius foundation. This work was supported by the Deutsche Forschungsgemeinschaft (FOR2314, SFB-TR209, Gottfried Wilhelm Leibniz Program) and the German Ministry for Education and Research (BMBF). Further funding was provided by the DFG under Germany's excellence strategy EXC 2180-390900677 (Image Guided and Functionally Instructed Tumour Therapies (iFIT)), the Landesstiftung Baden-Wuerttemberg, the European Research Council (CholangioConcept) and the German Cancer Research Center (DKTK). B.B., N.G.N. and E.F. are supported by the Swiss National Science Foundation (grants 733 310030_170320, 316030_150768 and CRSII5_183478) and the Swiss Cancer League. N.G.N. is a recipient of a University Research Priority Program (URPP) postdoctoral fellowship. J.-P.M. is supported by SNF Project Grant 310030 182679, Canica Holding Research Grant, Norwegian PSC Research Center, Stiftung zur Krebsbekämpfung, Bangerter-Rhyner Stiftung, Dangel Stiftung.

**Author contributions** Design of the study: D.P., M.H. Performed breeding and housing of mice: D.P., E.K., V.L. Performed flow cytometry experiments: D.P., N.G.N., M.S., E.F., K. Stirm, J.K., M.D., E.K. Histological staining and analyses: D.P., M.S., D.H., F. Müller, V.L. Bulk or scRNA-seq: R.G., M.Q., A.D., A. Weiner, T.E., S.Y., I.S. Proteome analyses: A. Sinha. Meta-analyses: R.P. Human cohorts: O.G., M.P., D.L, D.I., F.C., C.M., D.T., Z.M.J. Immunotherapy treatment of genetic model of HCC: K.W., M.R.d.G., A.T., K.U. Designed/performed the clinical case study, provided tissue samples or mouse strains and/or scientific input: A.M., P.K.H., P.B., S.C., R.Y., M.V., F. Marra, J.M.S., M.A., E.B., V.R., T.P., A.D'A., N.P., L.R., A.K.D., B.S., K.P., M.M.K., A.V., M.P.-R., F.H., F.F., O.W., J.T., K. Schulze, H.W., S.K., H.K., D.G.D., F.K., D.-T.W., M.P.E., A. Weinmann, M. Bueter, F.R., A. Siebenhüner, S.D.D., J.-P.M., V.U., M.J., T.L., A. Schietinger, P.S., H.G.A., A.B., B.M.-S., L.Z., H.E.M., A.R.S., M.R., N.M., S.S., M. Bitzer, A.L., N.R., J.-F.D., T.U.M., A.K., M.K., Y.-H.H., N.D., A. Weber, P.N.M., D.J.P., T.D., R.R., J.C.M., F. Meissner, S.R., M.C., Q.M.A., I.A., P.K., B.B., J.M.L., M.H. All authors analysed data. D.P., P.K., J.M.L. and M.H. wrote the manuscript, and all authors contributed to writing and provided feedback.

**Funding** Open access funding provided by Deutsches Krebsforschungszentrum (DKFZ) (1052).

**Competing interests** M.P. is an investigator for Bayer, BMS, Lilly, and Roche; has received speaker honoraria from Bayer, BMS, Eisai, Lilly, and MSD; is a consultant for Bayer, BMS, Eisai, Ipsen, Lilly, MSD, and Roche; and has received travel support from Bayer and BMS. D.P. works currently for Novo Nordisk. M. Szydlowksa works currently for Astra Zeneca. M.K. received honoraria from BMS as consultant and is an investigator for AstraZeneca and BMS. A.V. has served as consultant for Roche, Bayer, Lilly, BMS, Eisai, and Ipsen; has received speaking fees form Roche, Bayer, Lilly, BMS, Eisai, and Ipsen; and is an investigator for Roche, Bayer, Lilly, BMS, Eisai, and Ipsen. F.H. has received travel support from Bayer, Abbvie, and Gilead. M.P.-R. is an advisor/consultant for Astra Zeneca, Bayer, BMS, Eisai, Ipsen, Lilly, and MSD; has served as a speaker for Bayer, Eisai, and Lilly; and is an investigator for Bayer, BMS, Exelixis, and Lilly. F.F. has received travel support from Abbvie and Novartis. O.W. has served as consultant for Amgen, Bayer, BMS, Celgene, Eisai, Merck, Novartis, Roche, Servier, and Shire; has served as a speaker for Abbvie, Bayer, BMS, Celgene, Falk, Ipsen, Novartis, Roche, and Shire; and has received travel support from Abbvie, BMS, Ipsen, Novartis, and Servier. J.T. has served as consultant for Amgen, Bayer, BMS, Eisai, Lilly, Merck Serono, MSD, Ipsen, and Roche; has received travel support from BMS and Ipsen; has received speaking fees from Amgen, Bayer, BMS, Eisai, Lilly, Merck Serono, MSD, Ipsen, and Roche; and is an investigator for Amgen, Bayer, BMS, Eisai, Lilly, Merck Serono, MSD, Ipsen, and Roche. K.S. has served as consultant for Ipsen and Bayer; and conducts studies for Bayer, Roche, Lilly, MSD, and BMS. H.W. has served as speaker for Bayer, Eisai, and Ipsen; has served as a consultant for Bayer, Eisai, Lilly, BMS, Roche, and Ipsen; and conducts studies for Bayer, Roche, Lilly, MSD, and BMS. A. Weberis an advisor for BMS, Wako, Eisai, Roche, and Amgen. J.C.M. has received consulting honoraria from Abbvie, Bayer, BMS, Eisai, Gilead, Incyte, Intercept and MSD for work performed outside the current study. J.M.L. is receiving research support from Bayer HealthCare Pharmaceuticals, Eisai Inc, Bristol-Myers Squibb, Boehringer-Ingelheim and Ipsen, and consulting fees from Eli Lilly, Bayer HealthCare Pharmaceuticals, Bristol-Myers Squibb, Eisai Inc, Celsion Corporation, Exelixis, Merck, Ipsen, Genentech, Roche, Glycotest, Leerink Swann LLC, Fortress Biotech, Nucleix, Can-Fite Biopharma, Sirtex, Mina Alpha Ltd and AstraZeneca. J.M.S. serves as a consultant for Intercept Pharmaceuticals, Genfit, Gilead Sciences, BMS, Madrigal, Novartis, Pfizer, Roche, and Siemens-Healthineers; and has received research funding from Gilead Sciences. D.J.P. has received lecture fees from ViiV Healthcare and Bayer Healthcare; travel expenses from BMS and Bayer Healthcare; consulting fees from Mina Therapeutics, EISAI, Roche and Astra Zeneca; and research funding (to institution) from MSD and BMS. J.-F.D. has served on advisory committees for Abbvie, Bayer, Bristol-Myers Squibb, Falk, Genfit, Genkyotex, Gilead Sciences, HepaRegenix, Intercept, Lilly, Merck, and Novartis; and has spoken or taught at Bayer, Bristol-Myers Squibb, Intercept, Genfit, Gilead Sciences, Novartis, and Roche. L.R. has received consulting fees from Amgen, ArQule, Astra Zeneca, Basilea, Bayer, Celgene, Eisai, Exelixis, Hengrui, Incyte, Ipsen, Lilly, MSD, Nerviano Medical Sciences, Roche, and Sanofi; lectures fees from AbbVie, Amgen, Eisai, Gilead, Incyte, Ipsen, Lilly, Roche, Sanofi; travel expenses from Ipsen; and institutional research funding from Agios, ARMO BioSciences, AstraZeneca, BeiGene, Eisai, Exelixis, Fibrogen, Incyte, Ipsen, Lilly, MSD, and Roche. N.P. has received consulting fees from Amgen, Merck Serono, and Servier; lectures fees from AbbVie, Gilead and Lilly; travel expenses from Amgen and ArQule; and institutional research funding from Basilea, Merck Serono and Servier. T.P. has received institutional research funding from Lilly. D.G.D. has received consultant fees from Bayer, Simcere, Surface Oncology and BMS; and research grants from Bayer, Exelixis and BMS. The remaining authors declare no competing interests.

**Additional information**
**Correspondence and requests for materials** should be addressed to J.M.L. or M.H.

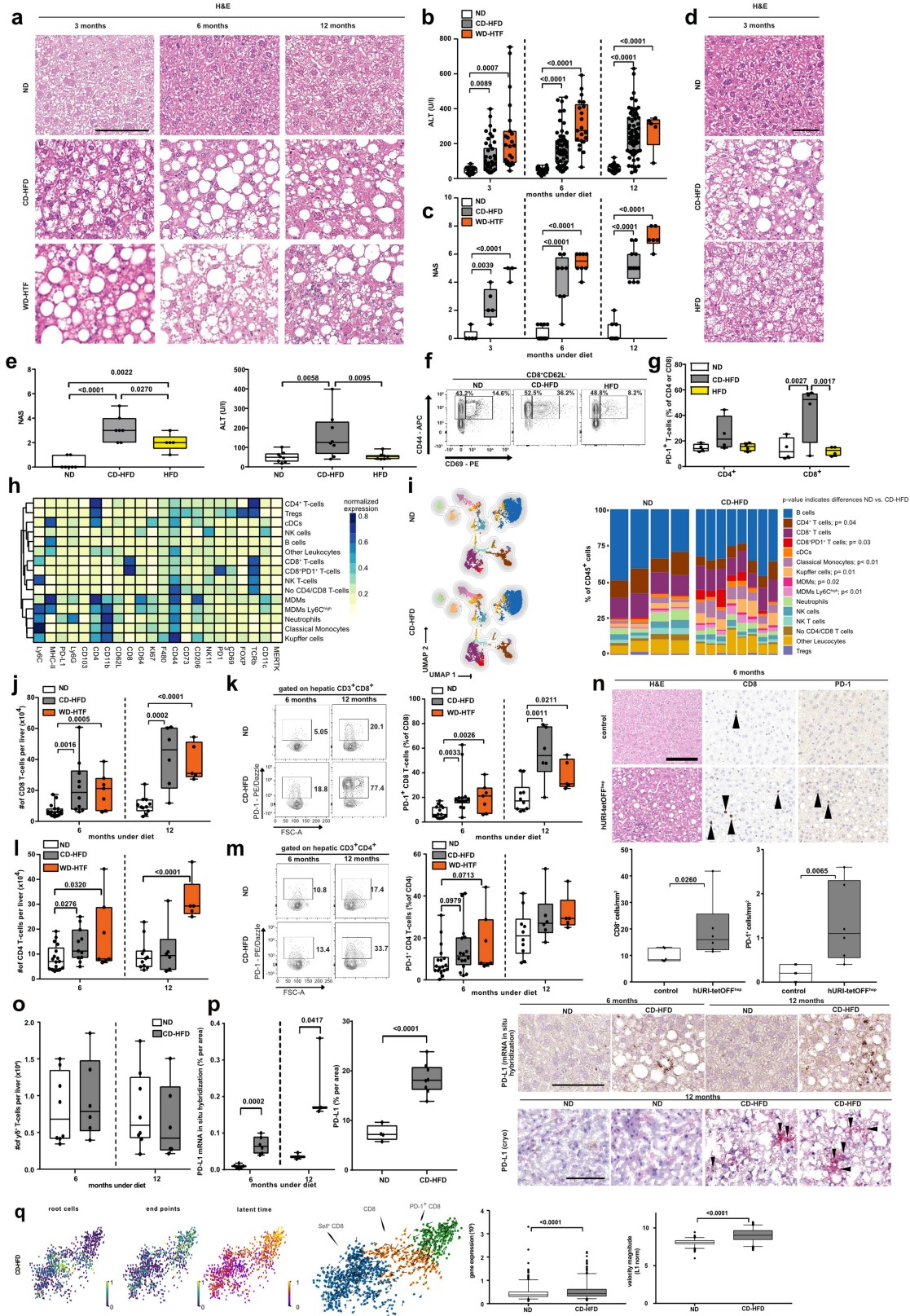

**Extended Data Fig. 1** | See next page for caption.

**Extended Data Fig. 1 | T cell activation and hepatic abundance correlate with NASH pathology. a**–**c**, Time kinetics of haematoxylin and eosin (H&E) staining of liver tissue (**a**), ALT (**b**), and NAS (**c**) in mice fed ND, CD-HFD, or WD-HTF ($n \geq 5$ mice per group). Scale bar, 100 μm. H&E 3 months: ND $n = 5$ mice; CD-HFD $n = 5$ mice; WD-HTF $n = 3$ mice; 6 months: ND $n = 16$ mice; CD-HFD $n = 8$ mice; WD-HTF $n = 8$ mice; 12 months: ND $n = 9$ mice; CD-HFD $n = 12$ mice; WD-HTF $n = 6$ mice; ALT 3 months: ND $n = 15$ mice; CD-HFD $n = 46$ mice; WD-HTF $n = 23$ mice; 6 months: ND $n = 46$ mice; CD-HFD $n = 59$ mice; WD-HTF $n = 21$ mice; 12 months: ND $n = 25$ mice; CD-HFD $n = 69$ mice; WD-HTF $n = 5$ mice; NAS 3 months: ND $n = 5$ mice; CD-HFD $n = 5$ mice; WD-HTF $n = 3$ mice; 6 months: ND $n = 16$ mice; CD-HFD $n = 8$ mice; WD-HTF $n = 8$ mice; 12 months: ND $n = 9$ mice; CD-HFD $n = 12$ mice; WD-HTF $n = 6$ mice. **d**, **e**, H&E staining (**d**) with NAS evaluation by H&E (**e**, left) and ALT (**e**, right) of mice fed with ND, HFD or CD-HFD for 3 months. NAS: ND $n = 7$ mice; CD-HFD $n = 7$ mice; HFD $n = 5$ mice; ALT: ND $n = 8$ mice; CD-HFD $n = 8$ mice; HFD $n = 7$ mice. Scale bar, 50 μm. **f**, **g**, Representative flow cytometry plots (**f**) and PD1 expression (**g**) of hepatic T cells from mice fed for 3 months with ND, HFD or CD-HFD ($n = 4$ mice per group). **h**, Heat map showing the median marker expression of the defined CD45[+] subsets displayed in **i** by flow cytometry of cells from mice fed for 12 months with ND or CD-HFD (ND $n = 4$ mice; CD-HFD $n = 8$ mice). **i**, UMAP representation of FlowSOM-guided clustering and quantification of hepatic immune cell composition of mice fed for 12 months with ND or CD-HFD (ND $n = 4$ mice; CD-HFD $n = 8$ mice). **j**, **k**, Abundance (**j**), flow cytometry plots (**k**, left) and PD1 expression (**k**, right) of hepatic CD8[+] T cells from mice fed for 6 or 12 months with ND or CD-HFD (abundance of CD8 6 months: ND $n = 17$ mice; CD-HFD $n = 10$ mice; WD-HTF $n = 7$ mice; 12 months: ND $n = 11$ mice; CD-HFD $n = 6$ mice; WD-HTF $n = 5$ mice; PD1 expression in CD8[+] T cells 6 months: ND $n = 15$ mice; CD-HFD $n = 14$ mice; WD-HTF $n = 7$ mice; 12 months: ND $n = 10$ mice; CD-HFD $n = 6$ mice; WD-HTF $n = 5$ mice). **l**, **m**, Abundance (**l**), flow cytometry plots (**m**, left) and PD1 expression (**m**, right) of hepatic CD4[+] T cells from mice fed for 6 or 12 months with ND or CD-HFD (abundance of CD4 6 months: ND $n = 17$ mice; CD-HFD $n = 10$ mice; WD-HTF $n = 7$ mice; 12 months: ND $n = 11$ mice; CD-HFD $n = 6$ mice; WD-HTF $n = 5$ mice; PD1 expression in CD4[+] T cells 6 months: ND $n = 15$ mice; CD-HFD $n = 14$ mice; WD-HTF $n = 7$ mice; 12 months: ND $n = 10$ mice; CD-HFD $n = 6$ mice; WD-HTF $n = 5$ mice). **n**, H&E, CD8 and PD1 hepatic staining (top), and quantification of CD8[+] cells and PD1[+] cells by immunohistochemistry (bottom) from 32-week-old hURI-tetOFFhep and non-transgenic littermate control mice ($n = 6$ mice/group). Arrowheads, specific positive-staining cells. Scale bar, 100 μm. **o**, Hepatic abundance of TCRγδ T cells from mice fed for 6 or 12 months with ND or CD-HFD (6 months ND $n = 8$ mice; CD-HFD $n = 6$ mice; 12 months ND $n = 8$ mice; CD-HFD $n = 6$ mice). **p**, Left, quantification of hepatic *Cd274*[+] expression by mRNA in situ hybridization of mice fed for 6 or 12 months with ND or CD-HFD (6 months: ND $n = 6$ mice; CD-HFD $n = 6$ mice; 12 months: ND $n = 3$ mice; CD-HFD $n = 3$ mice). Middle, quantification of hepatic PDL1[+] expression by immunohistochemistry of mice fed for 12 months with ND or CD-HFD (ND $n = 8$ mice; CD-HFD $n = 6$ mice). Right, mRNA in situ hybridization (top) and PD1-stained micrographs (bottom). Scale bars, 100 μm. **q**, RNA velocity indicating transcriptional activity, gene expression, and the trajectory of CD8[+] cells by scRNA-seq from 12 months ND or CD-HFD-fed mice. Root cells: yellow; blue cells: farthest away from root. End points: yellow indicates end point; blue cells: farthest away from defined end point. Latent time: pseudo-time by RNA velocity, dark color: start of velocity, yellow: end point of latent time. RNA velocity flow: Blue cluster: start point; orange cluster: intermediate; green cluster: end point. Arrows: cell trajectory (n=3 mice/group). All data are shown as mean ± s.e.m. **a**, **b**, **j**–**m**, **o**, **p**, Two-tailed Student's *t*-test. **d**–**g**, One-way ANOVA and Fisher's LSD test. **i**, **n**, **q**, Two-tailed Mann–Whitney test.

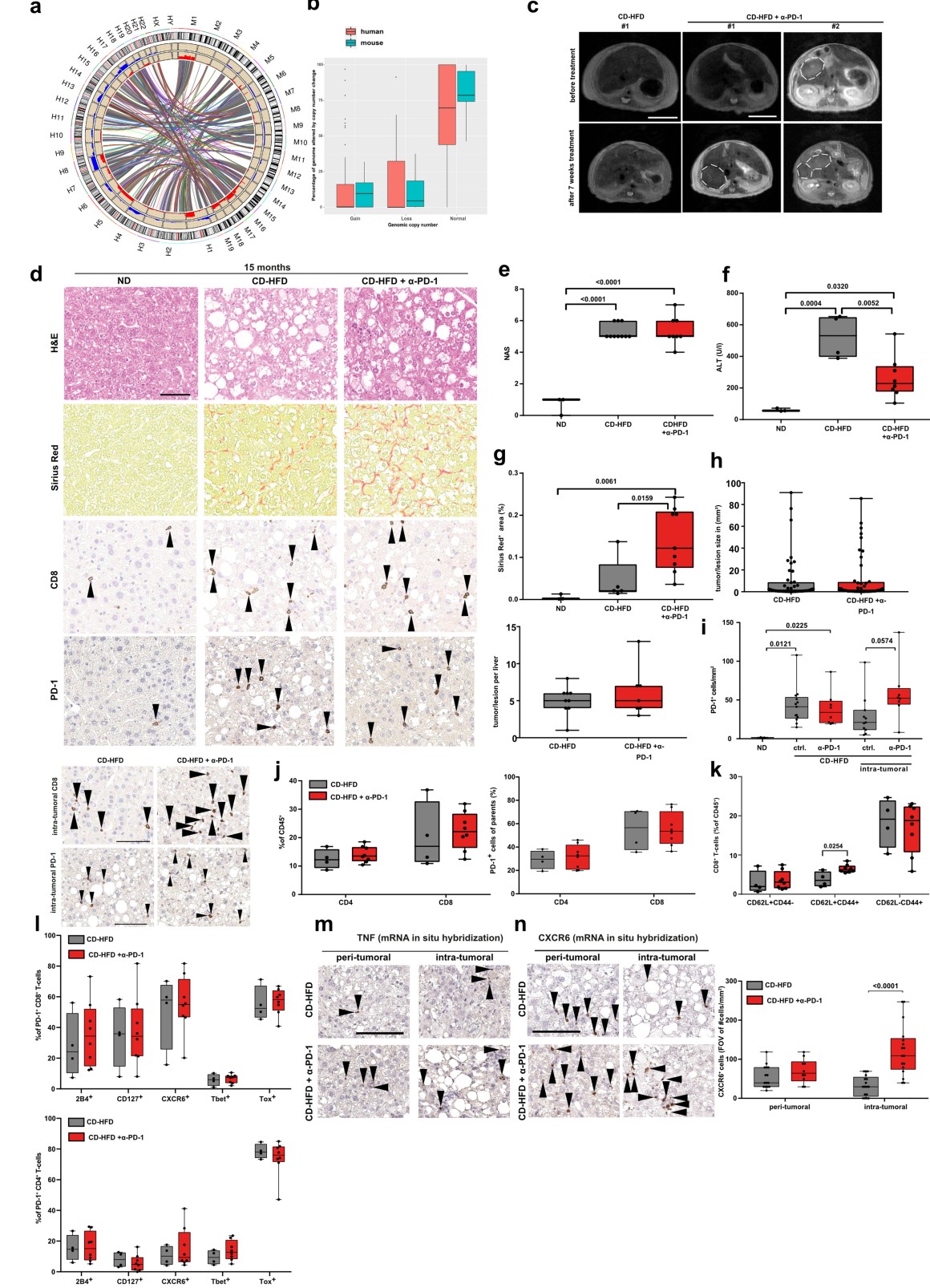

**Extended Data Fig. 2** | See next page for caption.

**Extended Data Fig. 2 | Anti-PD1 treatment does not achieve anti-tumour effects in NASH-induced tumours. a**, **b**, Synteny analysis of mouse HCC (**a**) and quantification of genomic aberrations by array-based comparative genomic hybridization (aCGH) for mice after 12 months on CD-HFD ($n = 19$) and for human NALFD/NASH–HCC ($n = 78$). The results here are in whole or part based upon data generated by the TCGA Research Network (https://www.cancer.gov/tcga). **c**, MRI images of mouse liver after 13 months on CD-HFD followed by 7 weeks with or without treatment with anti-PD1 antibodies ($n = 3$ mice per group). Dashed outlines indicate tumour nodules. Scale bars, 10 mm. **d**, Histological staining of hepatic tissue with H&E, Sirius Red, CD8 and PD1 of mice fed for 15 months ND or CD-HFD and either untreated or treated for 8 weeks with anti-PD1 antibodies (H&E: ND $n = 3$ mice; CD-HFD $n = 10$ mice; CD-HFD + anti-PD1 $n = 8$ mice; Sirius Red: ND $n = 3$ mice; CD-HFD $n = 5$ mice; CD-HFD + anti-PD1 $n = 9$ mice; CD8, PD1: ND $n = 3$ mice; CD-HFD $n = 13$ mice; CD-HFD + anti-PD1 $n = 8$ mice). Scale bar, 50 μm. Arrowheads, CD8$^+$ or PD1$^+$ cells. **e**, NAS evaluation by H&E staining of hepatic tissue from mice fed for 15 months with ND or CD-HFD and either untreated or treated for 8 weeks with anti-PD1 antibodies (ND $n = 3$ mice; CD-HFD $n = 10$ mice; CD-HFD + anti-PD1 $n = 8$ mice). **f**, ALT levels mice as in **e** (ND $n = 3$ mice; CD-HFD $n = 4$ mice; CD-HFD + anti-PD1 $n = 8$ mice). **g**, Quantification of fibrosis by Sirius Red staining of hepatic tissue from mice as in **e** (ND $n = 3$ mice; CD-HFD $n = 5$ mice; CD-HFD + anti-PD1 $n = 9$ mice). **h**, Quantification of tumour/lesion size and tumour load in livers from mice as in **e** (tumour/lesion size and tumour load: CD-HFD $n = 9$ mice; CD-HFD + anti-PD1 $n = 7$ mice; tumour incidence: CD-HFD $n = 17$ tumours/lesions in 22 mice; CD-HFD + anti-PD1 $n = 10$ tumours/lesions in 10 mice). **i**, Staining for CD8 and quantification of PD1$^+$ cells in hepatic tissue by immunohistochemistry for mice as in **e** (ND $n = 3$ mice; CD-HFD $n = 13$ mice; CD-HFD + anti-PD1 $n = 8$ mice; intra-tumoral staining: CD-HFD $n = 11$ mice; CD-HFD + anti-PD1 $n = 8$ mice). Scale bar, 100 μm. **j**, **k**, Quantification and expression of PD1 in hepatic CD4$^+$ and CD8$^+$ T cells (**j**) and polarization of CD8$^+$ T cells (**k**) by flow cytometry for mice fed for 15 months with CD-HFD and either untreated or treated for 8 weeks with anti-PD1 antibodies (CD-HFD $n = 4$ mice; CD-HFD + anti-PD1 $n = 8$ mice). **l**, Quantification of hepatic PD1$^+$ CD4$^+$ and PD1$^+$ CD8$^+$ T cells by flow cytometry for mice as in **j** (CD-HFD $n = 4$ mice; CD-HFD + anti-PD1 $n = 8$ mice). **m**, **n**, Expression of *Tnf* (**m**) and *Cxcr6* (**n**, left) in hepatic intra-tumoral and peri-tumoral tissue from mice as in **j** with quantification of *Cxcr6*-expressing cells (**n**, right) (quantification of CXCR6: peri-tumoral: CD-HFD $n = 15$ fields of view (FOV) in 6 tumours from 2 mice; CD-HFD + anti-PD1 $n = 10$ FOV in 6 tumours from 2 mice; intra-tumoral: CD-HFD $n = 17$ FOV in 6 tumours from 2 mice; CD-HFD + anti-PD1 $n = 17$ FOV in 6 tumours from 2 mice). Scale bars, 100 μm. Arrowheads, positive cells. All data are shown as mean ± s.e.m. **b**, Mann–Whitney test. **e**–**g**, One-way ANOVA and Fisher's LSD test. **h**–**l**, **n**, Two-tailed Student's *t*-test.

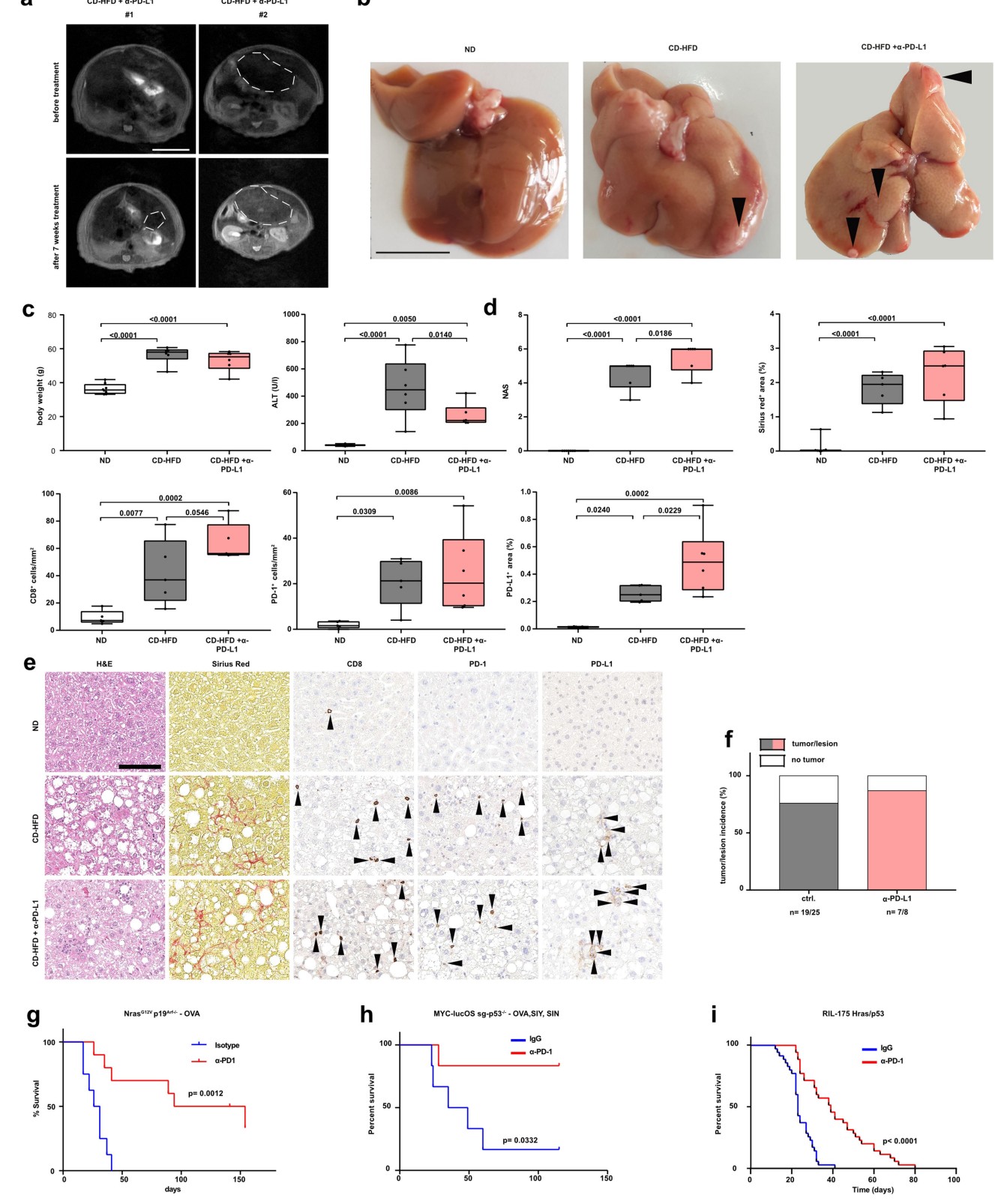

**Extended Data Fig. 3** | See next page for caption.

**Extended Data Fig. 3 | Anti-PDL1 treatment does not achieve anti-tumour effects in NASH-induced tumours, but in non-NASH livers PD1-targeted immunotherapy leads to prolonged survival. a**, MRI images of livers of mice after 13 months CD-HFD either untreated or after 7 weeks of treatment with anti-PDL1 antibodies (CD-HFD $n$ = 6 mice; CD-HFD + anti-PDL1 $n$ = 8 mice). Dashed outlines indicate tumour nodules. Scale bar, 10 mm. **b**, Livers of mice fed with ND or CD-HFD for 13 months and either untreated or treated for 8 weeks with anti-PDL1 antibodies. Arrowheads, tumours or lesions. Scale bar, 10 mm. **c**, Body weight and ALT of mice as in **b** (ND $n$ = 8 mice; CD-HFD $n$ = 6 mice; CD-HFD + anti-PDL1 $n$ = 6 mice). **d**, **e**, NAS evaluation by H&E, quantification of fibrosis by Sirius Red, and quantification of CD8, PD1 and PDL1 staining of hepatic tissue by immunohistochemistry (**d**) and corresponding micrographs (**e**) of mice fed for 13 months with ND or CD-HFD and untreated or treated for 8 weeks with anti-PDL1 antibodies (NAS: ND $n$ = 7 mice; CD-HFD $n$ = 6 mice; CD-HFD + anti-PDL1 $n$ = 6 mice; Sirius Red: ND $n$ = 7 mice; CD-HFD $n$ = 5 mice; CD-HFD + anti-PDL1 $n$ = 6 mice; CD8: ND $n$ = 5 mice; CD-HFD $n$ = 5 mice; CD-HFD + anti-PDL1 $n$ = 5 mice; PD1 and PDL1: ND $n$ = 5 mice; CD-HFD $n$ = 5 mice; CD-HFD + anti-PDL1 $n$ = 6 mice). Scale bar, 100 μm. Arrowheads, positive cells. **f**, Tumour or lesion incidence in mice fed with CD-HFD for 15 months and untreated or treated for 8 weeks with anti-PDL1 antibodies (CD-HFD $n$ = 19 tumours/lesions in 25 mice; CD-HFD + anti-PDL1 $n$ = 7 tumours/lesions in 8 mice). **g**, Survival analysis of mice with hydrodynamically delivered $Nras^{G12V}p19^{Arf-/-}$ liver tumours with OVA as antigen, treated with isotype or anti-PD1 antibodies (control $n$ = 8 mice; anti-PD1 $n$ = 10 mice). **h**, Survival analysis of a non-NASH model of HCC in which tumours are generated autochthonally in the liver by hydrodynamic injection of genetic elements (OVA, SIY, SIN and MYC-lucOS, in a CRISPR-based vector with tumour suppressor p53 deleted (sg-p53), and a transposase-expressing vector (SB13)). Mice were treated on days 7, 9 and 11 with IgG or anti-PD1 (control $n$ = 6 mice; anti-PD1 $n$ = 6 mice). **i**, Survival analysis of mice with RIL-175 $Hras/P53$-mutant hydrodynamically induced liver tumours, treated with IgG or anti-PD1 ($n$ = 35 mice per group). All data are shown as mean ± s.e.m. **c**, **d**, One-way ANOVA and Fisher's LSD test. **f**, Two-sided Fisher's exact test. **i**, Two-tailed Student's $t$-test. **g**–**i**, Two-sided $\chi^2$ test.

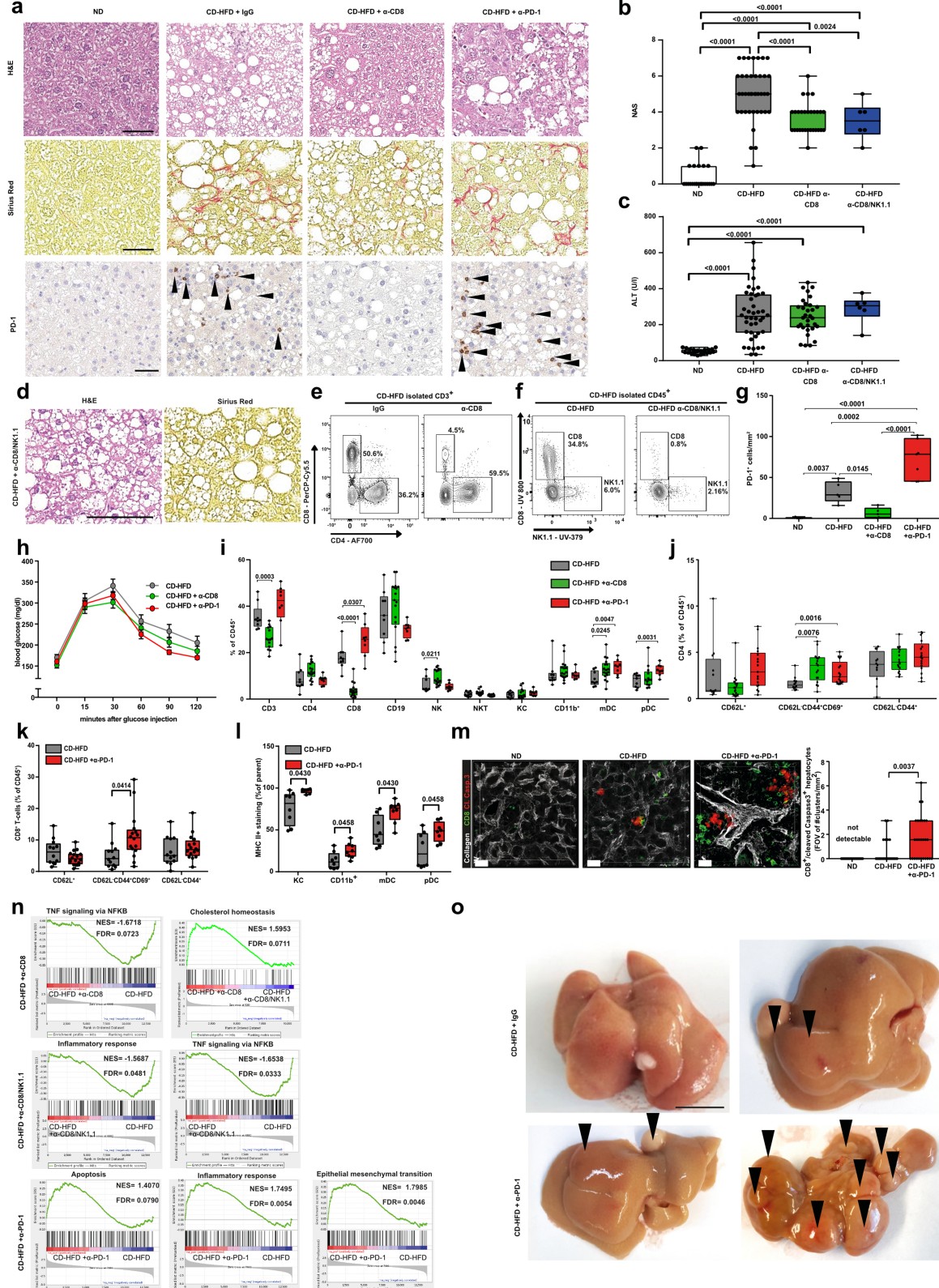

**Extended Data Fig. 4** | See next page for caption.

**Extended Data Fig. 4 | Preventive anti-PD1 treatment drives hepatocarcinogenesis in a CD8-dependent manner in NASH. a**, Histological staining of hepatic tissue with H&E, Sirius Red and PD1 from mice fed for 12 months with ND or CD-HFD and treated for 8 weeks with IgG, anti-CD8 or anti-PD1 antibodies (H&E: ND $n = 24$ mice; CD-HFD $n = 40$ mice; CD-HFD + anti-CD8 $n = 29$ mice; CD-HFD + anti-PD1 $n = 36$ mice; Sirius Red: ND $n = 19$ mice; CD-HFD $n = 53$ mice; CD-HFD + anti-CD8 $n = 24$ mice; CD-HFD + anti-PD1 $n = 33$ mice; PD1: ND $n = 5$ mice; CD-HFD $n = 5$ mice; CD-HFD + anti-CD8 $n = 5$ mice; CD-HFD + anti-PD1 $n = 7$ mice). Arrowheads, PD1$^+$ cells. Scale bars, 50 μm. **b**–**d**, NAS evaluation by H&E (**b**), ALT levels (**c**) and histological staining of hepatic tissue by H&E and Sirius Red (**d**) of mice fed for 12 months with ND or CD-HFD, and untreated or treated for 8 weeks with anti-CD8 or anti-CD8 + anti-NK1.1 antibodies (fibrosis ND $n = 19$ mice; CD-HFD $n = 53$ mice; CD-HFD + anti-CD8 $n = 27$ mice; CD-HFD + anti-CD8/NK1.1 $n = 6$ mice; NAS: ND $n = 24$ mice; CD-HFD $n = 40$ mice; CD-HFD + anti-CD8 $n = 29$ mice; CD-HFD + anti-CD8/NK1.1 $n = 6$; ALT: ND $n = 22$ mice; CD-HFD $n = 42$ mice; CD-HFD + anti-CD8 $n = 31$ mice; CD-HFD + anti-CD8/NK1.1 $n = 6$). Scale bar, 100 μm. **e**, **f**, Flow cytometry plots of hepatic cells from mice fed for 12 months with ND or CD-HFD and treated for 8 weeks with anti-CD8 (**e**) or anti-CD8 + anti-NK1.1 (**f**) antibodies. **g**, Quantification by immunohistochemistry of PD1$^+$ cells in hepatic tissue from mice fed for 12 months with ND or CD-HFD and untreated or treated for 8 weeks treatment with anti-CD8 or anti-PD1 antibodies (ND $n = 5$ mice; CD-HFD $n = 5$ mice; CD-HFD + anti-CD8 $n = 5$ mice; CD-HFD + anti-PD1 $n = 7$ mice). **h**, Assessment of metabolic tolerance by intraperitoneal glucose tolerance test of mice as in **g** (CD-HFD $n = 8$ mice; CD-HFD + anti-CD8 $n = 10$ mice; CD-HFD + anti-PD1 $n = 9$ mice). **i**, Relative quantification of hepatic leukocytes of mice as in **g** (CD3, NK T: CD-HFD $n = 9$ mice; CD-HFD + anti-CD8 $n = 14$ mice; CD-HFD + anti-PD1 $n = 8$ mice; CD4, CD8, CD19, NK, CD11b$^+$, mDC: CD-HFD $n = 9$ mice; CD-HFD + anti-CD8 $n = 17$ mice; CD-HFD + anti-PD1 $n = 8$ mice; pDC: CD-HFD $n = 9$ mice; CD-HFD + anti-CD8 $n = 13$ mice; CD-HFD + anti-PD1 $n = 8$ mice; Kupffer cells (KC): CD-HFD $n = 9$ mice; CD-HFD + anti-CD8 $n = 12$ mice; CD-HFD + anti-PD1 $n = 8$ mice). More MHCII$^+$ myeloid cells were found in the respective sub-populations. **j**, Flow cytometry analysis for polarization of hepatic CD4$^+$ T cells from mice as in **g** (CD-HFD $n = 12$ mice; CD-HFD + anti-CD8 $n = 17$ mice; CD-HFD + anti-PD1 $n = 17$ mice). **k**, Flow cytometric analysis for polarization of hepatic myeloid cells of mice fed for 12 months with CD-HFD and untreated or treated for 8 weeks with anti-PD1 antibodies (CD-HFD $n = 8$ mice; anti-PD1 + CD-HFD $n = 12$ mice). **l**, Flow cytometric analysis for polarization of hepatic CD8$^+$ T cells from mice as in **k** (CD-HFD $n = 10$ mice; anti-PD1 + CD-HFD $n = 14$ mice). **m**, Confocal analyses revealed clusters of CD8$^+$ T cells with adjacent cleaved caspase 3$^+$ hepatocytes that were strongly increased by anti-PD1-related immunotherapy in liver tissue from mice fed for 12 months with ND or CD-HFD and treated for 8 weeks with IgG or anti-PD1 antibodies, suggesting increased necro-inflammation in the vicinity of CD8$^+$ T cells ($n = 27$ FOV in 3 mice per group). Scale bars, 30 μm. **n**, GSEA of RNA-seq data for hepatic tissue from mice fed for 12 months with CD-HFD and treated for 8 weeks with anti-CD8, anti-CD8 + anti-NK1.1 or anti-PD1 antibodies ($n = 5$ mice per group) revealed enrichment for TNF signalling via NF-κB and inflammatory responses. Deletion of NK1.1$^+$ cells altered the cholesterol homeostasis-related signature, suggesting a link between NK T cells and aberrant cholesterol metabolism. Moreover, tissue from mice treated with anti-PD1 antibodies revealed positive enrichment of apoptosis, inflammatory responses and epithelial–mesenchymal transition, indicating a pro-inflammatory, pro-carcinogenic liver environment upon anti-PD1 treatment. **o**, Livers from mice fed for 12 months with CD-HFD and treated for 8 weeks with IgG or anti-PD1 antibodies. Arrowheads, tumours or lesions. Scale bar, 10 mm. All data are shown as mean ± s.e.m. **b**, **c**, One-way ANOVA and Fisher's LSD test. **h**, Two-way ANOVA and Sidak's multiple comparison test. **i**–**l**, Two-tailed Student's $t$-test. **m**, Two-tailed Mann–Whitney test.

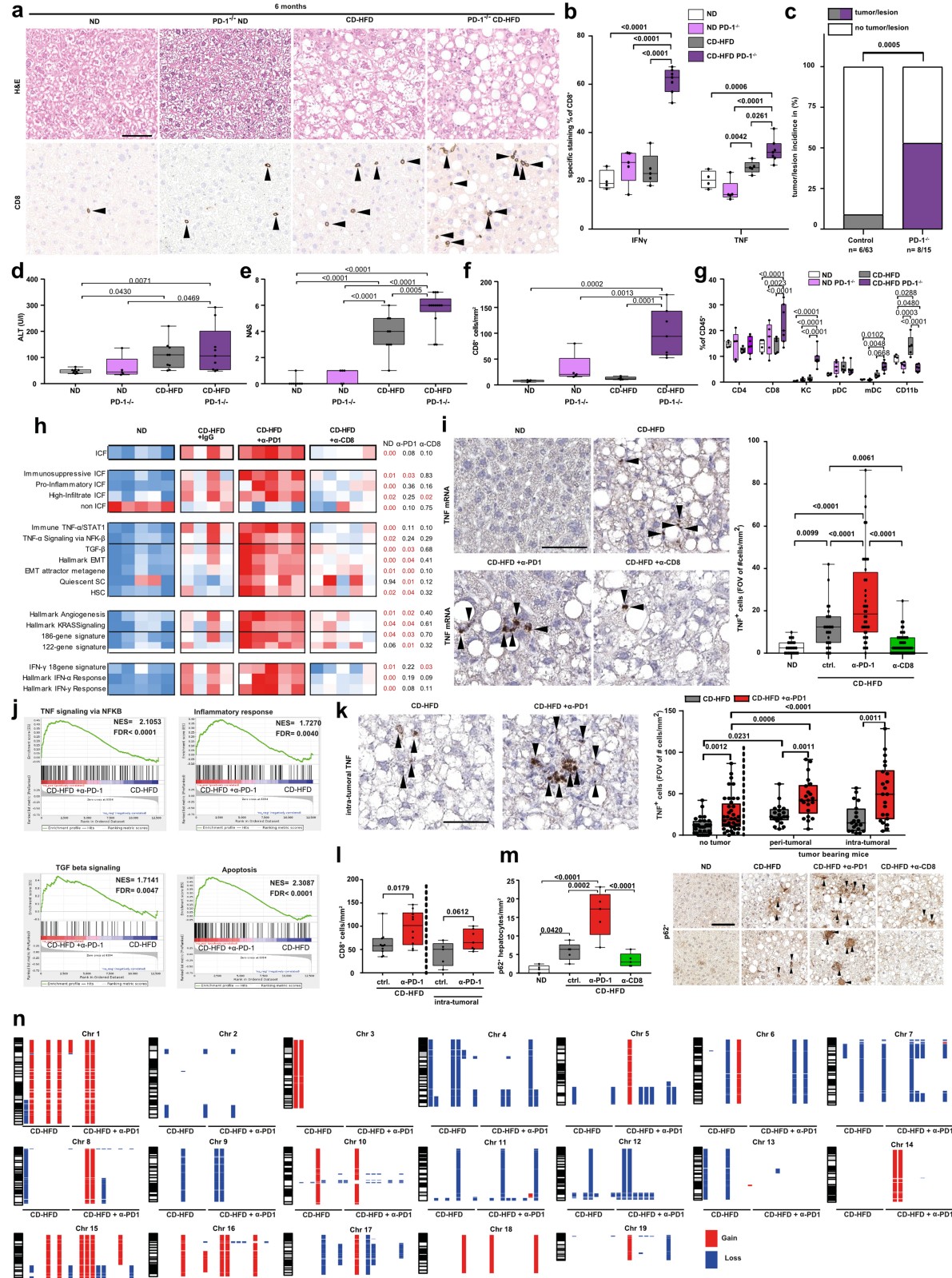

**Extended Data Fig. 5** | See next page for caption.

**Extended Data Fig. 5 | Anti-PD1 treatment drives hepatocarcinogenesis by enhancing an inflammatory and pro-tumorigenic liver microenvironment.**
**a**, Histological staining with H&E and CD8 of hepatic tissue from wild-type or *Pdcd1*[−/−] mice fed for 6 months with ND or CD-HFD (H&E: ND $n$ = 8 mice; *Pdcd1*[−/−] ND $n$ = 5 mice; CD-HFD $n$ = 9 mice; *Pdcd1*[−/−] CD-HFD $n$ = 13 mice; CD8: ND $n$ = 4 mice; CD-HFD $n$ = 5 mice; *Pdcd1*[−/−] CD-HFD $n$ = 7 mice). Arrowheads, CD8[+] cells. Scale bar, 50 μm. **b**, Cytokine expression of hepatic CD8[+] T cells from mice as in **a** (ND $n$ = 4 mice; *Pdcd1*[−/−] ND $n$ = 5 mice; CD-HFD $n$ = 5 mice; *Pdcd1*[−/−] CD-HFD $n$ = 6 mice). **c**, Tumour or lesion incidence in wild-type or *Pdcd1*[−/−] mice fed for 6 months with CD-HFD (CD-HFD $n$ = 6 tumours/lesions in 63 mice; *Pdcd1*[−/−] CD-HFD $n$ = 6 tumours/lesions in 13 mice). **d**, ALT levels for mice as in **a** (ND $n$ = 9 mice; *Pdcd1*[−/−] ND $n$ = 5 mice; CD-HFD $n$ = 9 mice; *Pdcd1*[−/−] CD-HFD $n$ = 10 mice). **e**, NAS evaluation by H&E of mice as in **a** (ND $n$ = 8 mice; *Pdcd1*[−/−] ND $n$ = 5 mice; CD-HFD $n$ = 9 mice; *Pdcd1*[−/−] CD-HFD $n$ = 13 mice). **f**, Quantification of CD8[+] cells in hepatic tissue by immunohistochemistry of mice as in **a** (ND $n$ = 4 mice; *Pdcd1*[−/−] ND $n$ = 5 mice; CD-HFD $n$ = 5 mice; *Pdcd1*[−/−] CD-HFD $n$ = 7 mice). **g**, Relative quantification of hepatic leukocytes in mice as in **a** (ND $n$ = 4 mice; *Pdcd1*[−/−] ND $n$ = 5 mice; CD-HFD $n$ = 5 mice; *Pdcd1*[−/−] CD-HFD $n$ = 6 mice). **h**, Immune cancer field (ICF) and ICF patterns of RNA-seq data for hepatic tissue from mice fed for 12 months with ND or CD-HFD and treated for 8 weeks treatment with IgG, anti-PD1 or anti-CD8 antibodies (ND, CD-HFD + anti-PD1, CD-HFD + anti-CD8 $n$ = 5 mice per group; CD-HFD $n$ = 4 mice) through single-sample GSEA. **i**, mRNA in situ hybridization (left) and quantification (right) for hepatic TNF[+] cells from mice as in **h** (ND $n$ = 25 FOV in 3 mice; CD-HFD $n$ = 27 FOV in 3 mice; CD-HFD + anti-PD1 $n$ = 40 FOV in 3 mice; CD-HFD + anti-CD8 $n$ = 55 FOV in 3 mice). Arrowheads, TNF[+] cells. Scale bar, 20 μm. **j**, GSEA of RNA-seq data for hepatic tissue comparing tumour-bearing mice fed for 12 months with CD-HFD and untreated or treated for 8 weeks with anti-PD1 antibodies ($n$ = 5 mice per group). **k**, mRNA in situ hybridization (left) and quantification (right) for hepatic TNF[+] cells from mice fed for 12 months with CD-HFD and untreated or treated for 8 weeks with anti-PD1 antibodies, with or without tumours (without tumours: CD-HFD $n$ = 30 FOV in 3 mice; CD-HFD + anti-PD1 $n$ = 40 FOV in 3 mice; peri-tumoural: CD-HFD $n$ = 20 FOV in 3 mice; CD-HFD + anti-PD1 $n$ = 21 FOV in 3 mice; intra-tumoural: CD-HFD $n$ = 19 FOV in 3 mice; CD-HFD + anti-PD1 $n$ = 22 FOV in 3 mice). Arrowheads, TNF[+] cells. Scale bar, 20 μm. **l**, Quantification of CD8 staining by immunohistochemistry of peri- and intra-tumoural hepatic tissue from mice fed for 12 months with CD-HFD and untreated or treated for 8 weeks with anti-PD1 antibodies (peri-tumoural: CD-HFD $n$ = 11 mice; CD-HFD + anti-PD1 $n$ = 10 mice; intra-tumoural: CD-HFD $n$ = 5 mice; CD-HFD + anti-PD1 $n$ = 7 mice). **m**, Histological staining for p62 (right) and quantification (left) of liver tumour tissue from mice fed for 12 months with ND or CD-HFD and untreated or treated for 8 weeks with anti-PD1 antibodies or anti-CD8 antibodies ($n$ = 5 mice per group). Scale bar, 100 μm. **n**, Genomic aberrations by array comparative genomic hybridization (aCGH) of tumour tissue from mice fed for 12 months with CD-HFD and untreated ($n$ = 9) or treated for 8 weeks with anti-PD1 antibodies ($n$ = 12). All data are shown as mean ± s.e.m. **b**, **d**–**i**, **m**, One-way ANOVA and Fisher's LSD test. **c**, Two-sided Fisher's exact test. **k**, **l**, Two-tailed Student's $t$-test.

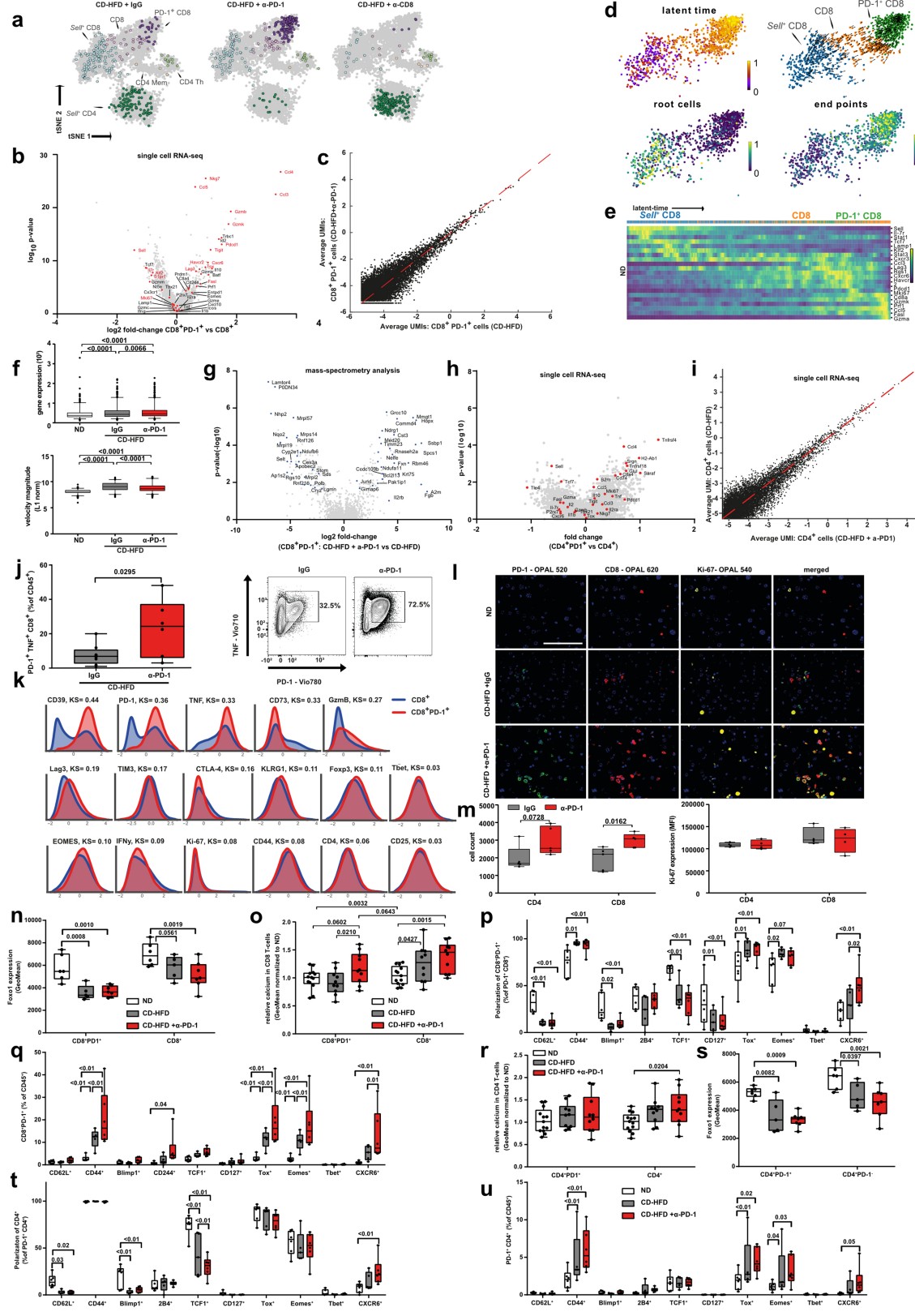

**Extended Data Fig. 6** | See next page for caption.

**Extended Data Fig. 6 | CD8⁺PD1⁺ TOX^high T cells with a resident-like character are enriched and are cellular drivers of hepatic necroinflammation and increased hepatocarcinogenesis upon anti-PD1 treatment in mice with NASH. a**–**c**, scRNA-seq analysis of hepatic TCRβ⁺ cells (**a**), expression of selected markers in hepatic CD8⁺ T cells by scRNA-seq comparing CD8⁺ with CD8⁺PD1⁺ T cells (**b**), and average UMI comparison (**c**) of hepatic CD8⁺PD1⁺ T cells from mice fed for 12 months with CD-HFD and treated for 8 weeks with IgG, anti-PD1 antibodies or anti-CD8 antibodies (*n* = 3 mice per group). **d**, Velocity analyses on scRNA-seq data from CD8⁺ cells from mice fed for 12 months with ND or CD-HFD and treated for 8 weeks with anti-PD1 antibodies (*n* = 3 mice). Yellow, root cells; yellow; blue, farthest from root. End points: yellow, end point cells; blue, farthest from defined end point. RNA velocity flow: blue cluster, start point; orange cluster, intermediate; green cluster, end point. Arrow shows trajectory of cells. **e**, Velocity analyses of scRNA-seq data showing correlation of expression of selected genes along the latent time of ND-fed mice (*n* = 3 mice). Latent time (pseudo-time by RNA velocity): dark colour, start of RNA velocity; yellow, end point of latent time. **f**, RNA velocity analyses by scRNA-seq indicating transcriptional activity and gene expression of CD8⁺ cells from mice fed for 12 months with ND or CD-HFD and untreated or treated for 8 weeks with anti-PD1 antibodies (*n* = 3 mice per group). **g**, Expression of selected markers in hepatic CD8⁺PD1⁺ T cells sorted from TCRβ⁺ cells by mass spectrometry from mice fed for 12 months with CD-HFD and untreated or treated for 8 weeks with anti-PD1 antibodies (*n* = 6 mice per group). **h**, Analyses of CD4⁺ and CD4⁺PD1⁺ T cells derived from livers of NASH mice with or without anti-PD1 treatment indicate minor differences in expression of selected markers in hepatic CD4⁺ T cells sorted from TCRβ⁺ cells by scRNA-seq comparing CD4⁺ with CD4⁺PD1⁺ T cells from mice fed for 12 months with CD-HFD and treated for 8 weeks with IgG, anti-PD1 or anti-CD8 antibodies (*n* = 3 mice per group). **i**, Comparison of average UMIs for hepatic CD4⁺ T cells from mice fed for 12 months with CD-HFD and treated for 8 weeks with IgG or anti-PD1 antibodies (*n* = 3 mice per group). **j**, Quantification of manual gating (left) and flow cytometry plots (right) for hepatic CD8⁺PD1⁺TNF⁺ cell abundance in mice as in **i** (CD-HFD *n* = 8 mice; CD-HFD + anti-PD1 *n* = 6 mice). **k**, CellCNN-analysed flow cytometry data for hepatic CD8⁺ T cells from mice as in **i** (CD-HFD + IgG *n* = 6 mice; CD-HFD + anti-PD1 *n* = 4 mice). **l**, Immunofluorescence staining for PD1, CD8 and Ki-67 of liver tissue from mice fed for 12 months with ND or CD-HFD and treated for 8 weeks with IgG or anti-PD1 antibodies (*n* = 2 mice per group). Scale bar, 100 μm. **m**, In vitro stimulated splenic CD8 T cells from C57Bl/6 mice were treated with anti-PD1 antibody for 72 h. Cell count (left), *n* = 5 experiments per group; Ki-67 (right), *n* = 4 experiments per group. **n**–**p**, Quantification of intracellular FOXO1 (**n**), calcium levels (**o**), and polarization (**p**) in CD8⁺ T cells isolated by flow cytometry from mice fed for 12 months with ND or CD-HFD and untreated or treated for 8 weeks with anti-PD1 antibodies (FOXO1: ND *n* = 6 mice; CD-HFD *n* = 5 mice; CD-HFD + anti-PD1 *n* = 7 mice; calcium: ND *n* = 13 mice; CD-HFD *n* = 10 mice; CD-HFD + anti-PD1 *n* = 10 mice; polarization: ND *n* = 6 mice; CD-HFD *n* = 5 mice; CD-HFD + anti-PD1 *n* = 6 mice). **q**, Relative quantification by flow cytometry of hepatic CD8⁺PD1⁺ cells from mice as in **n** (ND *n* = 6 mice; CD-HFD *n* = 5 mice; CD-HFD + anti-PD1 *n* = 6 mice). **r**–**t**, Quantification of intracellular calcium (**r**), FOXO1 (**s**) and polarization (**t**) in CD4⁺ T cells isolated by flow cytometry from mice as in **n** (FOXO1: ND *n* = 6 mice; CD-HFD *n* = 5 mice; CD-HFD + anti-PD1 *n* = 7 mice; calcium: ND *n* = 13 mice; CD-HFD *n* = 10 mice; CD-HFD + anti-PD1 *n* = 10 mice; polarization: ND *n* = 6 mice; CD-HFD *n* = 5 mice; CD-HFD + anti-PD1 *n* = 6 mice). **u**, Relative quantification by flow cytometry of hepatic CD4⁺PD1⁺ T cells from mice as in **n** (ND *n* = 6 mice; CD-HFD *n* = 5 mice; CD-HFD + anti-PD1 *n* = 6 mice). All data are shown as mean ± s.e.m. **f**, Two-tailed Mann–Whitney test. **j**, **m**, Two-tailed Student's *t*-test. **n**–**u**, Two-way ANOVA and Fisher's LSD test.

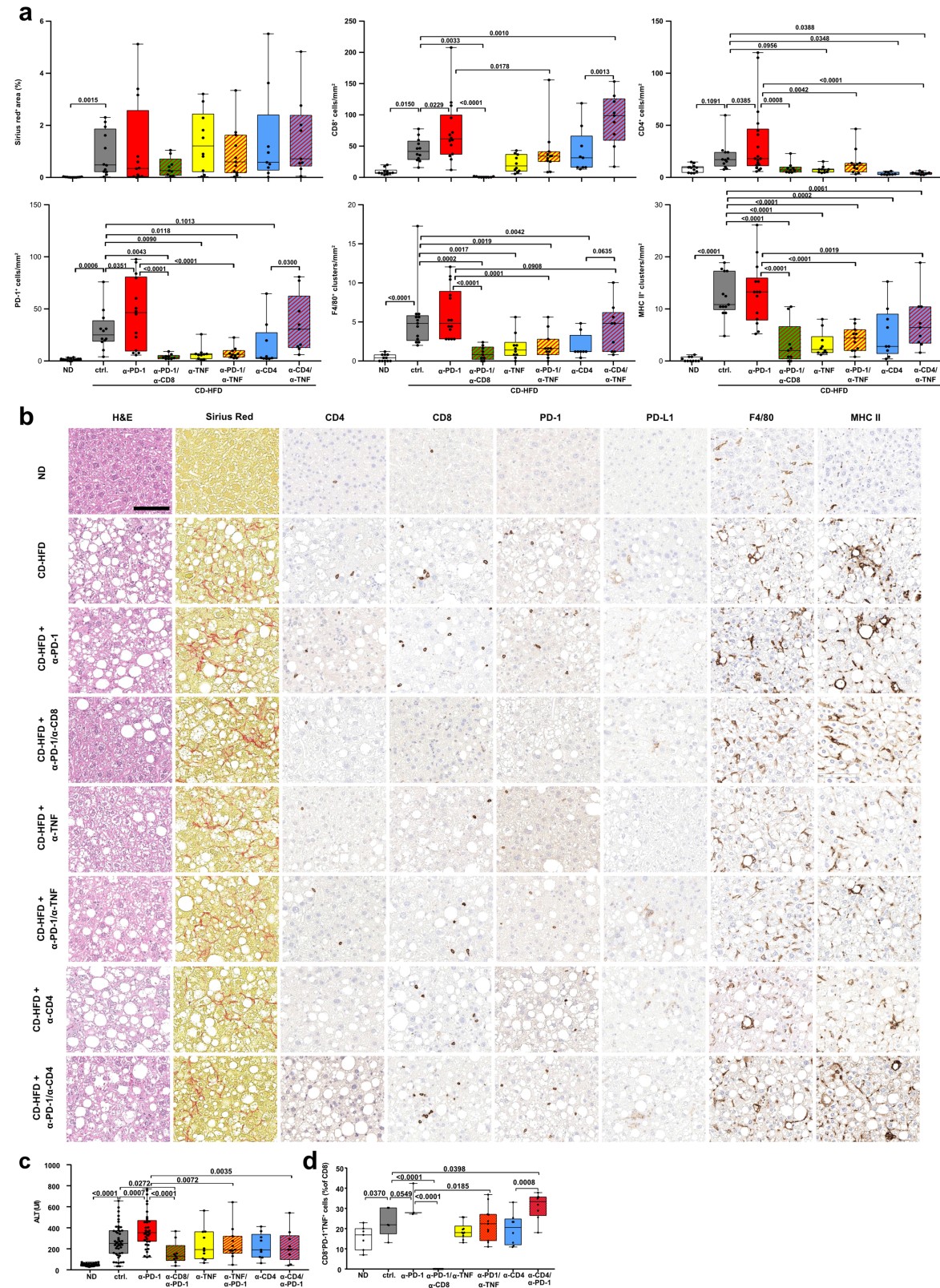

**Extended Data Fig. 7** | See next page for caption.

**Extended Data Fig. 7 | CD8[+] T cells drive hepatic inflammation and subsequent liver cancer in a TNF-dependent manner upon PD1-targeted immunotherapy. a**, **b**, Histological evaluation (**a**) and representative micrographs (**b**) of Sirius Red, CD4, CD8, PD1, PDL1, F4/80, and MHC-II staining of mice fed for 12 months with ND or CD-HFD and untreated or treated for 8 weeks with anti-PD1, anti-PD1 + anti-CD8, anti-TNF, anti-PD1 + anti-TNF, anti-CD4 or anti-PD1 + anti-CD4 antibodies (Sirius Red: ND $n$ = 11 mice; CD-HFD $n$ = 12 mice; CD-HFD + anti-PD1 $n$ = 12 mice; CD-HFD + anti-PD1 + anti-CD8 $n$ = 9 mice; CD-HFD + anti-TNF $n$ = 10 mice; CD-HFD + anti-PD1 + anti-TNF $n$ = 11 mice; CD-HFD + anti-CD4 $n$ = 8 mice; CD-HFD + anti-PD1 + anti-CD4 $n$ = 8 mice; CD4: ND $n$ = 10 mice; CD-HFD $n$ = 11 mice; CD-HFD + anti-PD1 $n$ = 14 mice; CD-HFD + anti-PD1 + anti-CD8 $n$ = 9 mice; CD-HFD + anti-TNF $n$ = 10 mice; CD-HFD + anti-PD1 + anti-TNF $n$ = 11 mice; CD-HFD + anti-CD4 $n$ = 8 mice; CD-HFD + anti-PD1 + anti-CD4 $n$ = 8 mice; CD8: ND $n$ = 10 mice; CD-HFD $n$ = 12 mice; CD-HFD + anti-PD1 $n$ = 14 mice; CD-HFD + anti-PD1 + anti-CD8 $n$ = 9 mice; CD-HFD + anti-TNF $n$ = 10 mice; CD-HFD + anti-PD1 + anti-TNF $n$ = 11 mice; CD-HFD + anti-CD4 $n$ = 8 mice; CD-HFD + anti-PD1 + anti-CD4 $n$ = 8 mice; PD1: ND $n$ = 12 mice; CD-HFD $n$ = 12 mice; CD-HFD + anti-PD1 $n$ = 14 mice; CD-HFD + anti-PD1 + anti-CD8 $n$ = 8 mice; CD-HFD + anti-TNF $n$ = 10 mice; CD-HFD + anti-PD1 + anti-TNF $n$ = 10 mice; CD-HFD + anti-CD4 $n$ = 8 mice; CD-HFD + anti-PD1 + anti-CD4 $n$ = 8 mice; PDL1: ND $n$ = 10 mice; CD-HFD $n$ = 11 mice; CD-HFD + anti-PD1 $n$ = 14 mice; CD-HFD + anti-PD1 + anti-CD8 $n$ = 9 mice; CD-HFD + anti-TNF $n$ = 10 mice; CD-HFD + anti-PD1 + anti-TNF $n$ = 11 mice; CD-HFD + anti-CD4 $n$ = 8 mice; CD-HFD + anti-PD1 + anti-CD4 $n$ = 8 mice; F4/80: ND $n$ = 11 mice; CD-HFD $n$ = 12 mice; CD-HFD + anti-PD1 $n$ = 14 mice; CD-HFD + anti-PD1 $n$ = 14 mice; CD-HFD + anti-PD1 + anti-CD8 $n$ = 9 mice; CD-HFD + anti-TNF $n$ = 10 mice; CD-HFD + anti-PD1 + anti-TNF $n$ = 11 mice; CD-HFD + anti-CD4 $n$ = 8 mice; CD-HFD + anti-PD1 + anti-CD4 $n$ = 8 mice; MHC-II: ND $n$ = 11 mice; CD-HFD $n$ = 13 mice; CD-HFD + anti-PD1 $n$ = 14 mice; CD-HFD + anti-PD1 $n$ = 14 mice; CD-HFD + anti-PD1 + anti-CD8 $n$ = 9 mice; CD-HFD + anti-TNF $n$ = 10 mice; CD-HFD + anti-PD1 + anti-TNF $n$ = 11 mice CD-HFD + anti-CD4 $n$ = 8 mice; CD-HFD + anti-PD1 + anti-CD4 $n$ = 8 mice). Scale bar, 100 μm. **c**, **d**, ALT (**c**) and quantification (**d**) of hepatic CD8[+]PD-1[+]TNF[+] T cells from mice fed for 12 months with ND or CD-HFD and untreated or treated for 8 weeks with anti-PD-1, anti-PD-1 + anti-CD8, anti-TNF, anti-PD-1 + anti-TNF, anti-CD4, or anti-PD-1 + anti-CD4 antibodies (ALT: ND $n$ = 30 mice; CD-HFD $n$ = 47 mice; CD-HFD + anti-PD-1 $n$ = 35 mice; CD-HFD + anti-PD-1 + anti-CD8 $n$ = 9 mice; CD-HFD + anti-TNF $n$ = 10 mice; CD-HFD + anti-PD-1 + anti-TNF $n$ = 11 mice; CD-HFD + anti-CD4 $n$ = 8 mice; CD-HFD + anti-PD-1 + anti-CD4 $n$ = 8 mice; CD8[+]PD-1[+]TNF[+]: ND $n$ = 8 mice; CD-HFD $n$ = 5 mice; CD-HFD + anti-PD-1 $n$ = 3 mice; CD-HFD + anti-PD-1 + anti-CD8 $n$ = 9 mice; CD-HFD + anti-TNF $n$ = 10 mice; CD-HFD + anti-PD-1 + anti-TNF $n$ = 11 mice; CD-HFD + anti-CD4 $n$ = 8 mice; CD-HFD + anti-PD-1 + anti-CD4 $n$ = 8 mice). All data are shown as mean ± s.e.m. All data were analysed by one-way ANOVA and Fisher's LSD test.

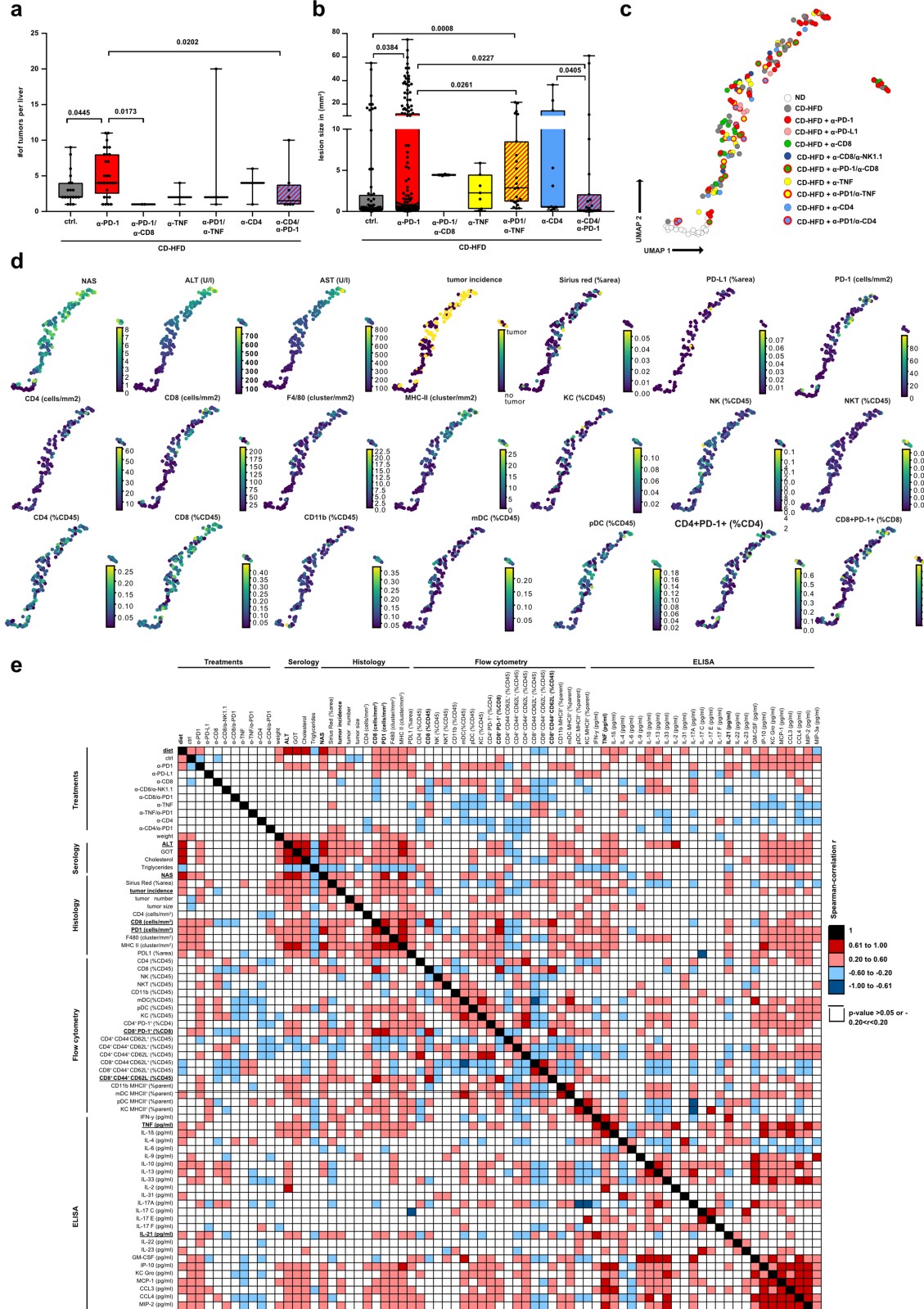

**Extended Data Fig. 8** | See next page for caption.

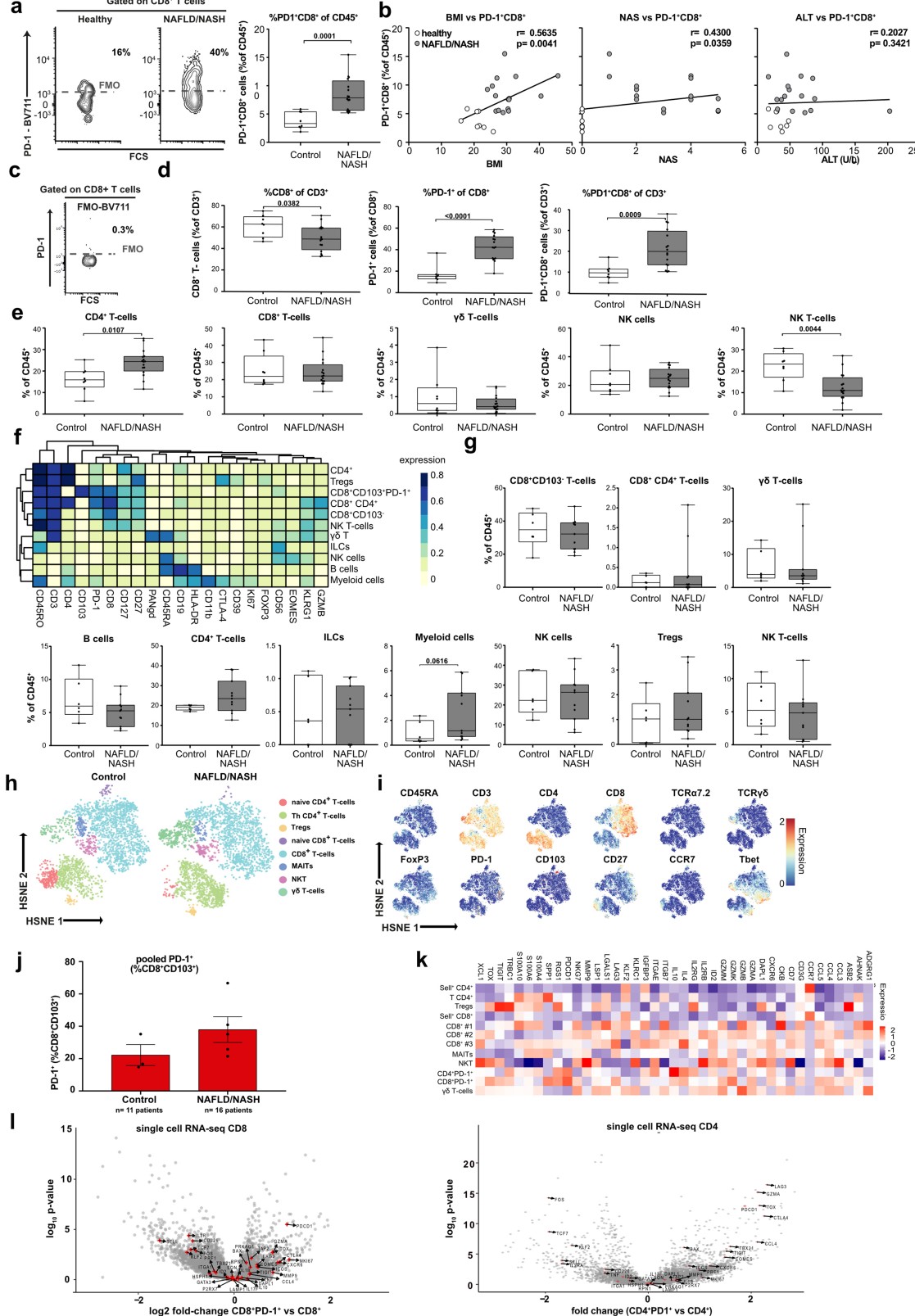

**Extended Data Fig. 9** | See next page for caption.

**Extended Data Fig. 9 | An inflammatory cellular polarization of T cells can be found in liver biopsies from patients with NAFLD or NASH. a**, **b**, Flow cytometry plots and quantification of patient-liver-derived PD1$^+$CD8$^+$ T cells (**a**), and correlation of PD1$^+$CD8$^+$ T cells with BMI, NAS and ALT for healthy participants and patients with NAFLD or NASH (**b**) (Supplementary Table 1; healthy controls $n = 8$; NAFLD/NASH $n = 16$ patients). **c**–**e**, Flow cytometry plot of FMO control (**c**), quantification of patient-liver-derived PD1$^+$CD8$^+$ T cells (**d**), and quantification of CD4, CD8, γδ, NK and NK T cells from healthy participants or patients with NAFLD or NASH (**e**) (Supplementary Table 1: healthy controls $n = 8$; NAFLD/NASH $n = 16$ patients). **f**, **g**, Heat map showing median marker expression (**f**) and quantification of the defined CD45$^+$ subsets from Fig. 3c (**g**) by flow cytometry derived from hepatic biopsies from control participants and patients with NAFLD or NASH to define distinct marker expression (Supplementary Table 2: control individuals $n = 6$; NAFLD/NASH $n = 11$ patients). **h**–**j**, HSNE representation of defined T cell subsets (**h**), marker expression (**i**) and quantification of CD8$^+$CD103$^+$PD1$^+$ cells (**j**) in liver-derived T cells from control individuals and patients with NAFLD or NASH analysed by cytometry by time of flight (CyTOF) (control $n = 11$ individuals pooled in 3 analyses; NAFLD/NASH $n = 16$ patients pooled in 5 analyses). **k**, **l**, Selected average marker expression in CD4$^+$ and CD8$^+$ T cell subsets (**k**) and differential gene expression of CD8$^+$PD1$^+$ versus CD8$^+$ T cells and CD4$^+$PD1$^+$ versus CD4$^+$ T cells by scRNA-seq (**l**) for control individuals and patients with NAFLD or NASH (control $n = 4$ individuals; NAFLD/NASH $n = 7$ patients). All data are shown as mean ± s.e.m. All data were analysed by two-tailed Mann–Whitney test.

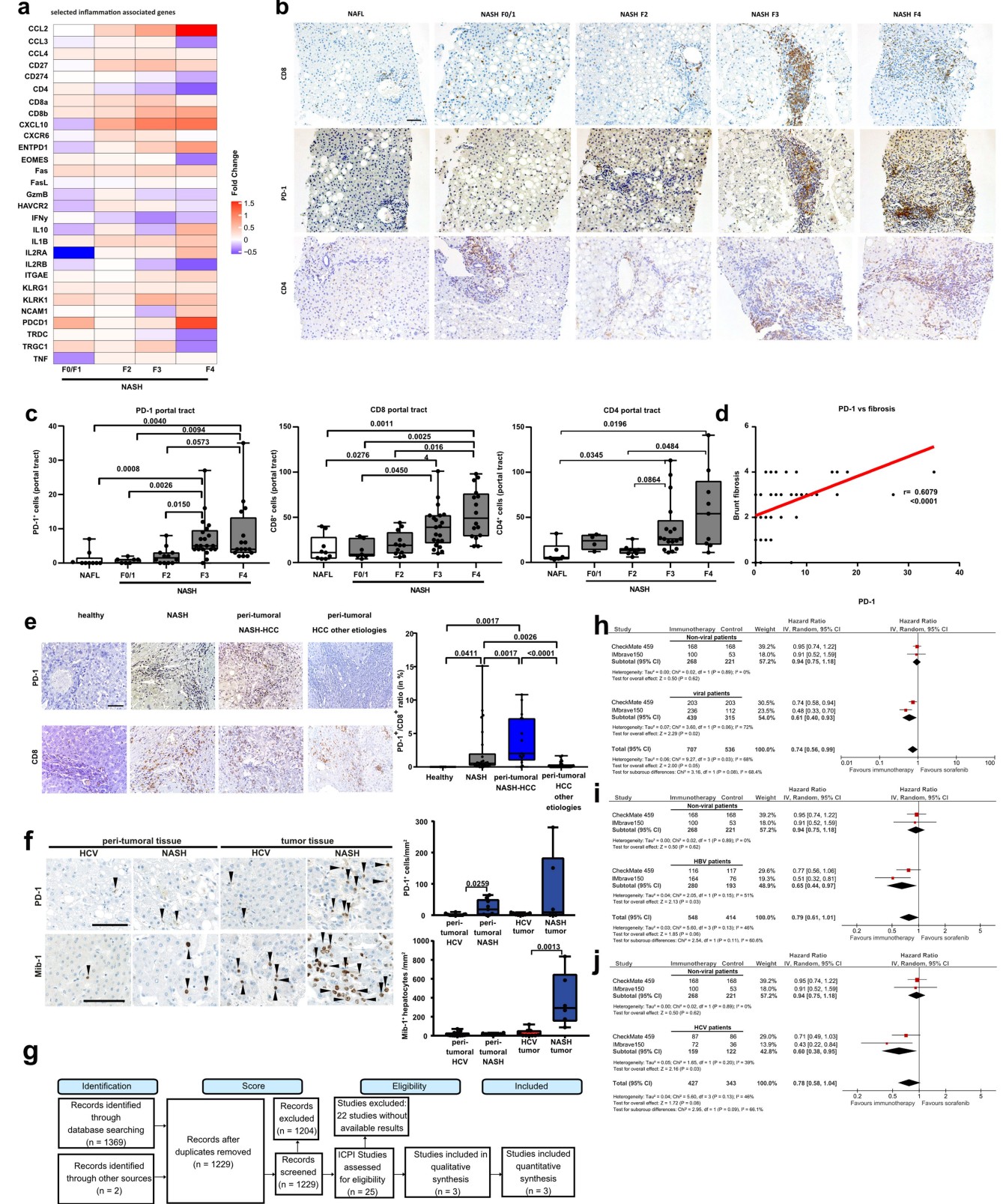

**Extended Data Fig. 10** | See next page for caption.

**Extended Data Fig. 10 | PD1 and PDL1 targeted immunotherapy in advanced HCC has a distinct effect depending on disease aetiology. a**, Comparison of RNA-seq data from patients with NASH with varying degrees of fibrosis (F0–F4, Brunt classification) normalized to data from patients with NAFLD from a total of $n = 206$ patients with NAFLD or NASH. **b**, **c**, Immunohistochemical staining (**b**) and quantification (**c**) of hepatic PD1[+], CD8[+], and CD4[+] cells from patients with NAFLD or NASH with varying degrees of fibrosis (Supplementary Table 3) (NAFLD $n = 9$ patients; NASH F0/1 $n = 7$ patients; NASH F2 $n = 12$ patients; NASH F3 $n = 21$ patients; NASH F4 $n = 16$ patients; CD4: NAFL $n = 6$ patients; NASH F0/1 $n = 4$ patients; NASH F2 $n = 8$ patients; NASH F3 $n = 17$ patients; NASH F4 $n = 9$ patients). Scale bar, 100 μm. **d**, Correlation analysis of PD1 expression against fibrosis grade by immunohistochemical staining (NAFLD/NASH $n = 65$ patients). **e**, Immunohistochemical staining and quantification of ratio of PD1[+]/CD8[+] cells in immunohistochemical staining of samples from patient cohort in Supplementary Tables 4–6 (healthy individuals $n = 4$, NASH $n = 26$ patients, peri-tumoural NASH–HCC $n = 16$ patients, peri-tumoural HCC other aetiologies $n = 29$ patients). Scale bar, 100 μm. **f**, Immunohistochemical staining and quantification of PD1[+] cells and MIB1[+] hepatocytes in peri-tumoural and intra-tumoural samples from patients with HCV- or NASH-induced HCC (PD1: peri-tumoural HCV $n = 16$ tissues from 7 patients; peri-tumoural NASH $n = 9$ tissues from 2 patients; intra-tumoural HCV $n = 10$ HCCs from 7 patients; intra-tumoural NASH $n = 6$ HCCs from 2 patients; MIB1: peri-tumoural HCV $n = 16$ tissues from 7 patients; peri-tumoural NASH $n = 9$ tissues from 2 patients; intra-tumoural HCV $n = 10$ HCCs from 7 patients; intra-tumoural NASH $n = 6$ HCCs from 2 patients). Arrowheads, PD1[+] or MIB1[+] cells. Scale bars, 100 μm. **g**, PRISMA flow chart of the systematic review of targeted immunotherapy in HCC and the selection of articles assessing the clinical outcome of immune checkpoint inhibitors in advanced HCC for inclusion in the systematic review and meta-analysis. ICPI, immune checkpoint inhibitor. A total of 1,243 patients were included in two first-line trials comparing PD1- or PDL1-targeted immunotherapy to sorafenib. In these trials, 707 patients received an immune checkpoint inhibitor (either anti-PD1 or anti-PDL1). **h**–**j**, HCV and HBV were pooled into a separate category, termed 'viral', and a subsequent meta-analysis comparing viral ($n = 754$) and non-viral HCC ($n = 489$; mostly NASH and alcohol intake) was performed (**h**). A subgroup analysis studying the specific effects of non-viral aetiologies ($n = 489$) on the magnitude of effect of immunotherapy is presented, when compared to HBV (**i**; $n = 473$) or HCV (**j**; $n = 281$). HRs for each trial are represented by squares; the size of the square represents the weight of the trial in the meta-analysis. The horizontal line crossing the square represents the 95% CI. The diamonds represent the estimated overall effect based on the meta-analysis random effect of all trials. Inverse variance (IV) and random effects methods (Random) were used to calculate HRs, 95% CIs, $P$ values, and the test for overall effect; these calculations were two-sided. Cochran's $Q$-test and $I^2$ were used to calculate heterogeneity. All data are shown as mean ± s.e.m. **c**, **e**, **f**, One-way ANOVA and Dunn's multiple comparison test. **d**, Two-tailed Spearman's correlation.

# nature research

|---|---|

# Reporting Summary

Nature Research wishes to improve the reproducibility of the work that we publish. This form provides structure for consistency and transparency in reporting. For further information on Nature Research policies, see Authors & Referees and the Editorial Policy Checklist.

## Statistics

For all statistical analyses, confirm that the following items are present in the figure legend, table legend, main text, or Methods section.

| n/a | Confirmed | |
|---|---|---|
| ☐ | ☒ | The exact sample size ($n$) for each experimental group/condition, given as a discrete number and unit of measurement |
| ☐ | ☒ | A statement on whether measurements were taken from distinct samples or whether the same sample was measured repeatedly |
| ☐ | ☒ | The statistical test(s) used AND whether they are one- or two-sided *Only common tests should be described solely by name; describe more complex techniques in the Methods section.* |
| ☐ | ☒ | A description of all covariates tested |
| ☐ | ☒ | A description of any assumptions or corrections, such as tests of normality and adjustment for multiple comparisons |
| ☐ | ☒ | A full description of the statistical parameters including central tendency (e.g. means) or other basic estimates (e.g. regression coefficient) AND variation (e.g. standard deviation) or associated estimates of uncertainty (e.g. confidence intervals) |
| ☐ | ☒ | For null hypothesis testing, the test statistic (e.g. $F$, $t$, $r$) with confidence intervals, effect sizes, degrees of freedom and $P$ value noted *Give P values as exact values whenever suitable.* |
| ☒ | ☐ | For Bayesian analysis, information on the choice of priors and Markov chain Monte Carlo settings |
| ☒ | ☐ | For hierarchical and complex designs, identification of the appropriate level for tests and full reporting of outcomes |
| ☐ | ☒ | Estimates of effect sizes (e.g. Cohen's $d$, Pearson's $r$), indicating how they were calculated |

*Our web collection on statistics for biologists contains articles on many of the points above.*

## Software and code

Policy information about availability of computer code

| Data collection | Data were collected with Microsoft Excel. |
|---|---|
| Data analysis | Flowcytometry: Cells were analyzed using BD FACSFortessa or BD FACSSymphony and data were analyzed using FlowJo (v10.6.2). For sorting, FACS Aria II and FACSAria FUSION in collaboration with the DKFZ FACS core facility were used. For UMAP/FlowSOM plots, BD FACSymphony data (mouse and human) were exported from FlowJo (v10). Analyses was performed as described elsewhere46. scRNA-seq (mouse): Single-cell capturing for scRNA-seq and library preparation was described previously47. Libraries (pooled at equimolar concentration) were sequenced on an Illumina NextSeq 500 at a median sequencing depth of ~40,000 reads/cell. Sequences were mapped to the mouse (mm10), using HISAT (version 0.1.6); reads with multiple mapping positions were excluded. Reads were associated with genes if they were mapped to an exon, using the Ensembl gene annotation database (Embl release 90). Exons of different genes that shared a genomic position on the same strand were considered as a single gene with a concatenated gene symbol. The level of spurious UMIs in the data was estimated by using statistics on empty MARS-seq wells and excluded rare cases with estimated noise > 5% (median estimated noise overall experiments was 2%). Removal of specific mitochondrial genes, immunoglobulin genes, genes linked with poorly supported transcriptional models (annotated with the prefix "Rp-"), and cells with less than 400UMIs. Gene features were selected using Tvm=0.3 and a minimum total UMI count > 50. Hierarchical clustering of the correlation matrix between those genes (filtering genes with low coverage and computing correlation using a down-sampled UMI matrix) and selected the gene clusters that contained anchor genes. We used K=50, 750 bootstrap iterations, and otherwise standard parameters. Subsets of T-cells were obtained by hierarchical clustering of the confusion matrix and supervised analysis of enriched genes in homogeneous groups of metacells48. scRNA-seq (human): De-multiplexing and barcode processing was performed using the Cell Ranger Software Suite (Version 4.0.0) and reads were aligned to human GRCh3863. Gene-barcode matrix containing cell barcodes and gene expression counts was generated by counting the single-cell 3' UMIs, imported into R (v4.0.2) where quality control and normalization were executed using Seurat v364. Cells with more than 10% mitochondrial genes, fewer than 200 genes per cell, or more than 6000 genes per cell were excluded. Matrices from 10 samples were integrated with Seurat v3 to remove batch effects across samples. PCA analysis of filtered gene-barcode matrices of all CD3+ cells, visualized by UMAP (top 50 principal components) and identification of major cell types using the highly variable features and indicative markers was performed. Besides, pairwise combinations of CD4+ T-cells vs CD4+PD-1+ T-cells and CD8+ T-cells vs CD8+PD-1+ T-cells were performed using the results of differential expression analysis by DESeq2 (v1.28.1)65, setting CD4+/CD8+ T-cells as controls. |

Volcano plots were then generated using EnhancedVolcano (v1.6.0)66 to visualize the results of differential expression analysis. Velocity and correlation analyses of scRNA-seq data: Velocyto (0.6) was used to estimate the spliced/unspliced counts from the pre-aligned bam files49. RNA velocity, latent time, root, and terminal states were calculated using the dynamical velocity model from scvelo (0.2.2)50. Kendall's rank correlation coefficient was used to correlate the expression patterns of biologically significant genes with latent time.

Mass spectrometry: Analyses was performed using MaxQuant (1.6.7.0), mouse UniProt Isoform fasta (Version: 2019-02-21, number of sequences 25,233) as a source for protein sequences. 1% FDR was used for controlling at the peptide and protein level, with a minimum of two peptides needed for consideration of analysis. Gene set enrichment analysis was performed using ClusterProfiler (3.18)51 and gene sets obtained from WikiPathway (wikipathways.org) and MSigDB (broadinstitute.org/msigdb)52–54.

Isolation of RNA and library preparation for bulk RNA sequencing: RNA isolation23and library preparation for bulk 3'-sequencing of poly(A)-RNA was described previously55. Using the with the Feature Extraction software (11.0.1.1, Agilent Technologies), gencode gene annotations version M18 and the mouse reference genome major release GRCm38 were derived from (https:// www.gencodegenes.org/). Dropseq tools v1.1256 were used for mapping the raw sequencing data to the reference genome. Resulting UMI filtered count matrix was imported into R v3.4.4. Prior differential expression analysis with Limma v3.40.657 sample-specific weights were estimated and used as coefficients alongside the experimental groups as a covariate during model fitting with Voom. T-test was used for determining differentially (p-value below 0.05) regulated genes between all possible experimental groups. Gene set enrichment analysis was conducted with the pre-ranked GSEA method53 within the MSigDB Reactome, KEGG, and Hallmark databases (broadinstitute.org/msigdb). Raw sequencing data are available under the accession number PRJEB36747.

Mass cytometry: The sample was acquired on a Helios mass cytometer and raw data were EQ-Bead-normalized using Helios mass cytometer and Helios instrument software (version 6.7). Compensation was performed in CATALYST (v1.86)70 and FlowCore (1.50.0). De-barcoding and gating of single, live CD45+ cells were performed using FlowJo (v10.6.2). Then, data of CD45+ cells were imported into Cytosplore 2.3.1 and transformed using the arcsinh(5) function. Major immune cell lineages were identified at the first level of a 2-level hierarchical stochastic neighbor embedding (HSNE) analysis with default perplexity and iteration settings. HSNE with the same parameters was run on CD3+ cells to identify T-cell phenotypes. Gaussian mean shift clustering was performed in Cytosplore and a heatmap of arcsinh(5)-transformed expression values of all antibody targets was generated. Cell type identification was based on the transformed expression values and clusters showing high similarity were merged manually.

Search strategy, selection criteria, and meta-analysis of phase III clinical trials: The literature search was done through MEDLINE on PubMed, Cochrane Library, Web of Science, and clinicaltrials.gov, using the following searches: "checkpoint inhibitors", "HCC", "phase III", between January 2010 and January 2020, and complemented by hand searches of conference abstracts/presentations. Single-center, non-controlled trials, studies with insufficient data to extract hazard ratios (HR), 95% confidence intervals, or trials including disease entities other than HCC were excluded. As conference abstracts were not excluded, quality assessment of the included studies was not performed. Three studies5,13,14 fulfilled the criteria and were included in the quantitative synthesis (Extended Data 10). The primary outcome of the meta-analysis was OS, defined as the time from randomization to death. HRs and CIs related to OS were extracted from the papers/conference presentations5,13,14. Pooled HRs were calculated using the random-effects model (Der Simonian and Laird), and the generic inverse variance was used for calculating weights72. To evaluate heterogeneity among studies, Cochran's Q test and I2 index were used. A p-value < 0.10 in the Q-test was considered to indicate substantial heterogeneity. I2 was interpreted as suggested in the literature: 0% to 40% might not represent significant heterogeneity; 30% to 60% may represent moderate heterogeneity, 50% to 90% may represent substantial heterogeneity, 75% to 100% represents considerable heterogeneity. All statistical pooled analyses were performed using the RevMan 5.3 software.

Statistical analyses: Data was collected in Microsoft Excel. Mouse data are presented as the mean±SEM. Pilot experiments and previously published results were used to estimate the sample size, such that appropriate statistical tests could yield significant results. Statistical analysis was performed using GraphPad Prism software version 7.03 (GraphPad Software). Exact p-values lower than p< 0.1 are reported and specific tests are indicated in the legends. Survival analyses were performed using IBM SPSS Statistics version 25 (SPSS Inc., Chicago, IL).

For manuscripts utilizing custom algorithms or software that are central to the research but not yet described in published literature, software must be made available to editors/reviewers. We strongly encourage code deposition in a community repository (e.g. GitHub). See the Nature Research guidelines for submitting code & software for further information.

# Data

Policy information about availability of data

All manuscripts must include a data availability statement. This statement should provide the following information, where applicable:
- Accession codes, unique identifiers, or web links for publicly available datasets
- A list of figures that have associated raw data
- A description of any restrictions on data availability

The proteomics data described in this article is available at ProteomeXchange Consortium via the PRIDE under the identifier XD017236 login/password: /37ScigLe. The bulk-RNA-seq data described in this article is available at PRJEB36747. The single-cell RNA-seq data described in this article is available at GEO Submission (GSE144635). The array of comparative genomic hybridization data described in this article is available at GEO Submission (GSE144875). The results here are in whole or part based upon data generated by the TCGA Research Network: https://www.cancer.gov/tcga. The human single-cell RNA-seq data described in this article is available at GEO Submission (GSE159977). Databases used in this manuscript are WikiPathways (wikipathways.org), MSigDB (www.broadinstitute.org/msigdb).

# Field-specific reporting

Please select the one below that is the best fit for your research. If you are not sure, read the appropriate sections before making your selection.

☒ Life sciences    ☐ Behavioural & social sciences    ☐ Ecological, evolutionary & environmental sciences

For a reference copy of the document with all sections, see nature.com/documents/nr-reporting-summary-flat.pdf

# Life sciences study design

All studies must disclose on these points even when the disclosure is negative.

| | |
|---|---|
| Sample size | Pilot experiments and previous published results were used to estimate the sample size such that appropriate statistical tests could yield significant results (DOI: 10.1016/j.ccell.2014.09.003; DOI: 10.1038/s41591-019-0379-5). The exact n numbers used in the study are indicated in the source data file. |
| Data exclusions | No data exlusion of mice and human. |
| Replication | All experiments presented were conducted with sufficient mouse numbers to ensure statistical significance could be reached, particularly for experiments involving tumor studies.<br>Biochemical or image based data were reproduced in multiple mice: e.g. Weight analysis of mice, Measuring transaminase levels, NASH phenotype characterization, HCC development characterization, flow cytometry analyses, immunohistochemical staining, confocal analysis.<br>All attempts of replicating data were successfull. |
| Randomization | 5-week-old C57Bl/6 mice were randomly allocated into different groups and were fed with appropriated diet or treatment regimens.<br>No prospective trial was performed, therefore randomization of human specimen was not relevant for the performed experiments. |
| Blinding | Experiments with different genotypes on the same diet were blinded.<br>For the dissection between ND or NASH mouse experiments this was not possible since the type of diet and the systemic obesity were visible to the researcher, e.g., the color of the normal diet is brown and color of the NASH-diet is green.<br>The analysis of the transaminase level was blinded and measured with code labeling. The NASH phenotype analysis was blinded - still the morphology of the normal/chow-fed liver is different from a NASH liver. |

# Reporting for specific materials, systems and methods

We require information from authors about some types of materials, experimental systems and methods used in many studies. Here, indicate whether each material, system or method listed is relevant to your study. If you are not sure if a list item applies to your research, read the appropriate section before selecting a response.

## Materials & experimental systems

| n/a | Involved in the study |
|---|---|
| ☐ | ☒ Antibodies |
| ☒ | ☐ Eukaryotic cell lines |
| ☒ | ☐ Palaeontology |
| ☐ | ☒ Animals and other organisms |
| ☐ | ☒ Human research participants |
| ☐ | ☒ Clinical data |

## Methods

| n/a | Involved in the study |
|---|---|
| ☒ | ☐ ChIP-seq |
| ☐ | ☒ Flow cytometry |
| ☒ | ☐ MRI-based neuroimaging |

## Antibodies

| | |
|---|---|
| Antibodies used | Description of all antibodies (clone and supplier) used in the study are provided in the Materials&Methods Table 11-15. |
| Validation | Validation of commercial antibodies was done on a regular quality control of each lot by the manufacturer (e.g. Biolegend "The antibody was purified by affinity chromatography and conjugated with PE under optimal conditions."; "Each lot of this antibody is quality control tested by immunofluorescent staining with flow cytometric analysis."; "Every lot of product is quality tested against a "gold standard" reference lot. A new lot is only released based on our defined QC specifications to ensure lot to lot reproducibility and reliability. BioLegend guarantees the stability and performance of all our products shipped at room temperature". or Bioxcell for in vivo treatment antibodies "Binding Validation: Western Blot data shown below confirms that htis clone binds to its target antigen"). |

## Animals and other organisms

Policy information about studies involving animals; ARRIVE guidelines recommended for reporting animal research

| | |
|---|---|
| Laboratory animals | All animals are described in the Supplementary Materials.<br>All mice were male and on a C57Bl/6 background and were put from 5-8 weeks of age onwards in the respective diets and/or treatment regimens.<br>Following strains were used for the experiment described in the manuscript: C57Bl6/J and Pdcd1-/-. |
| Wild animals | not used |

October 2018

| Field-collected samples | not used |
|---|---|
| Ethics oversight | Regierungspräsidium Karlsruhe, Karlsruhe, Germany |

Note that full information on the approval of the study protocol must also be provided in the manuscript.

# Human research participants

Policy information about studies involving human research participants

| Population characteristics | Human characteristics are reported in Supplementary Materials Table 1-10. |
|---|---|
| Recruitment | No active recruitment was done in the frame of this study Baseline characteristics of analyzed human specimen are listed in Supplementary Tables 1-10. The human survival analyses are retrospective studies, they need prospective validation. |
| Ethics oversight | The respective ethics committee is listed in the Material&Method section. Flow cytometry of patient material (Figure 3a,b ) was performed under ethics oversight: BIOFACS Study KEK 2019-00114. Flow cytometry of human samples (Extended Data 9d) was approved by the local ethical committee (AC-2014-2094 n 03). High-throughput RNA sequencing of human specimen: As previously reported, RNA sequencing analysis was performed using the data from 206 snap-frozen biopsy samples from 206 patients diagnosed with NAFLD in France, Germany, Italy, and the UK and enrolled in the European NAFLD Registry (GSE135251)58,59. Collection and use of data of the European NAFLD Registry were approved by the relevant local and/or national Ethical Review Committee58. Isolation of cells for single-cell RNA-seq data analysis (human): Analyses of liver samples from patients undergoing bariatric surgery at the Department of Surgery at Heidelberg University Hospital (S-629/2013). Histological and immunohistochemical analysis of NASH/HCC cohort: All samples were obtained from International Genomic HCC Consortium with IRB approval. A cohort of patients with HCC treated with PD-(L)1-targeted immunotherapy: The retrospective analysis was approved by local Ethics Committees. Data from this cohort were published previously73. Validation cohort: Ethical approval to conduct this study was granted by the Imperial College Tissue Bank (Reference Number R16008). Analysis of human samples from the Department of Pathology and Molecular Pathology, University Hospital Zurich, was approved by the local ethics committee ('Kantonale Ethikkommission Zürich', KEK-ZH-Nr. 2013-0382 and BASEC-Nr. PB_2018-00252). |

Note that full information on the approval of the study protocol must also be provided in the manuscript.

# Clinical data

Policy information about clinical studies

All manuscripts should comply with the ICMJE guidelines for publication of clinical research and a completed CONSORT checklist must be included with all submissions.

| Clinical trial registration | This study was performed not in active clinical trials. |
|---|---|
| Study protocol | This study was performed not in active clinical trials. |
| Data collection | The respective data collection (fresh/retrospective cohort analysis/analyses of published data) process is listed in the Material&Method section. |
| Outcomes | This study was performed not in active clinical trials. |

# Flow Cytometry

## Plots

Confirm that:

☒ The axis labels state the marker and fluorochrome used (e.g. CD4-FITC).

☒ The axis scales are clearly visible. Include numbers along axes only for bottom left plot of group (a 'group' is an analysis of identical markers).

☒ All plots are contour plots with outliers or pseudocolor plots.

☒ A numerical value for number of cells or percentage (with statistics) is provided.

## Methodology

| Sample preparation | Please see Supplementary Materials. Mice were transcardially perfused with PBS, and livers were dissected. Livers were incubated for up to 35min in 37°C with digestion buffer (Collagen IV 1:10 (60 U f.c.) and DNase I 1:100 (25µg/ml f.c.)) and subsequently passed through a 100µm filter. Livers were washed with RPMI1640 (#11875093) medium and subsequently centrifuged for 7min/300g/4°C. Lymphocyte enrichment was achieved by a 2-step Percoll gradient (20ml 25% Percoll/HBSS underlay with 20ml 50% Percoll/HBSS) and centrifugation for 15min/1800g/4°C (Acc:1 Dcc:0). Leukocytes were collected, washed with HBSS, centrifuged for 10min/700g/4° |
|---|---|

C, counted and transferred to a 15ml Falcon for a final washing step with FACS buffer (PBS supplemented with v/v 0.4% 0.5M EDTA pH= 8 and w/v 0.5% albumin fraction V (#90604-29-8)). Isolation of splenic lymphocytes was done by passing spleens through a 100μm mesh and subsequent washing. Afterwards, an erythrocyte lysis using ACK-buffer 1x 2ml for 5 min RT and then a wash was performed. For T-cell re-stimulation, cells were incubated for 2h, 37°C, 5% CO2 in RPMI 1640 supplemented with v/v 2% fetal calf serum using 1:500 Biolegend´s Cell Activation Cocktail (with Brefeldin A) (#423304) and 1:1000 Monensin Solution (1,000X) (#420701). Staining was performed using Live/Dead discrimination by using DAPI or ZombieDyeNIR according to the manufacturer´s instructions. After washing (~400g, 5min, 4°C), cells were stained in 25μl of titrated antibody master mix for 20min at 4°C and washed again (antibodies shown in Table 7-9). Samples for flow cytometric activated cell sorting (FACS) were then sorted. Samples for flow cytometry were fixed using eBioscience IC fixation (#00-8222-49) or Foxp3 Fix/Perm kit (#00-5523-00) according to the manufacturer´s instruction. Intracellular staining was performed in eBioscience Perm buffer (#00-8333-56). Cells were analyzed using BD FACSFortessa or BD FACSSymphony and data were analyzed using FlowJo. For sorting, a FACS Aria II and a FACSAria FUSION in collaboration with the DKFZ FACS core facility were used.

Analysis of human material (Table 1) was performed on human liver tissue samples (needle biopsies or resected tissue, BIOFACS Study KEK 2019-00114), that were processed within 4 hours after collection. Tissue samples were minced using scalpels, incubated (1 mg/mL collagenase IV (Sigma Aldrich), 0.25 μg/mL DNase (Sigma Aldrich), 10% FCS (Thermo Fisher Scientific), RPMI 1640 (Seraglob)) for 30 min at 37oC with continuous shaking. The enzymatic reaction was stopped with 2 mM EDTA (StemCell Technologies, Inc) in PBS. The homogenized cell suspension was filtered through 100 μm cell strainer and centrifuged to get a cell pellet. Next cells were resuspended in FACS Buffer (PBS, EDTA 2mM, FCS 0.5%) with Human TruStain FcX™ (Fc Receptor Blocking Solution) (Biolegend) and incubated for 15 min at 4oC. After cells were spun without washing and stained with the flow cytometry antibodies mix and Zombie Aqua™ Fixable Viability Kit (Biolegend).

| | |
|---|---|
| Instrument | Cells were analyzed using BD FACSFortessa or BD FACSSymphony. For sorting, a FACS Aria II and a FACSAria FUSION in collaboration with the DKFZ FACS core facility were used. |
| Software | Collected data was analyzed by FlowJo V10.2. |
| Cell population abundance | Absolute quantification by using CountBright™ Absolute Counting Beads. |
| Gating strategy | Debry esclusion byFSC-A/SSC-A. Doublets were excluded by using FSC-A/ FSC-H and SSC-A/SSC-H gates. Life/Dead exclusion was performed. Remaining cells were analyzed according to displayed markers. |

☒ Tick this box to confirm that a figure exemplifying the gating strategy is provided in the Supplementary Information.

