## [Peer Review File · Nature]

Manuscript Title: NASH limits anti-tumor surveillance in immunotherapy-treated HCC

Editorial Notes: *none*

Reviewer Comments & Author Rebuttals

Reviewer Reports on the Initial Version:

Referee #1 (Remarks to the Author):

Using two different mouse models of NASH-induced HCC as well as data from patients with NASH-associated HCC, the authors suggest the concept that CD8+PD1+ T cells promote NASH development and that treatment with checkpoint inhibitors may release the brake in these NASH-promoting cells, resulting in disease exacerbation and more HCC, which they proposed is confirmed by their findings of absent response to checkpoint inhibitors Nivolumab and Pembroluzimab in patients with NASH-associated HCC but not in patients with HCC due to other causes. While the analyses are carefully performed and raise the question of harmful effects of checkpoints in NASH-associated HCC, both the mouse and patient studies have major limitations, and it cannot be excluded that this paper sends the wrong message to the community and will negatively impact the field.

1. The NASH-HCC mouse models represent a major weakness of this paper and may lead to premature conclusions on the effect of PD-1 therapy in NASH-associated HCC. While the employed mouse models may be among the best to study various aspects of NASH, there are several limitations that preclude them from serving as useful preclinical models for HCC:

1a. Many mouse models of cancer are simply not responsive to checkpoint inhibition because of low mutational load and lacking tumor antigens/neoantigens. The authors do not provide evidence that the employed models have a mutational load that is at least as high as in that seen in HCC patients.

1b. The mouse model - albeit taking over a year - is not comparable to HCC development in patients, which takes decades and mostly occurs in the setting of advanced fibrosis or cirrhosis (even though a subset of NASH-associated HCC patients do not have cirrhosis, most of them have advanced fibrosis). Importantly, in most of these patients the underlying NASH is much less activate than in earlier disease stages/burnt out - meaning that the risk of increasing NASH activity and thereby worsening not only NASH but also increasing NASH-HCC is much lower and possibly not even relevant. The authors' conclusions would be relevant if one employed checkpoint inhibitors for HCC prevention but are likely not applicable to patients except for those, in whom HCC develops in the absence of cirrhosis and with high NAS.

2. In relation to above-described limitations of the model, the paper does not sufficiently focus on dual functions of CD8+PD1+ T cells, promoting NASH but possibly also restricting HCC. These functions are likely to occur at different stages in patients.

3. The data on the NASH- and NASH-HCC-promoting role of CD8+ T cells is similar to a previous study from the last author (Wolf et al, Cancer Cell). Hence a number of the findings presented in this manuscript are incremental with, adding PD1 into this context, with somewhat expected results, as well as novel techniques such as scRNA-seq.

4. The human data are based on a very small and poorly analyzed cohort of patients with NASH-associated HCC (n=10-11). While the underlying question is important, pairing data from this small cohort with the data from the mouse model with its above-described limitations and confounders may send a wrong and potentially deleterious message to the community, and much more careful analysis as well as larger cohorts are needed to put the provided message on a solid scientific foundation: The authors should analyzed outcomes for NASH-HCC patients with or without cirrhosis to account for the possibility of worsened NASH in patients without cirrhosis (for which the cohort is much too small).

A. A cohort of n=10-11 NASH-associated HCC patients is unacceptable. Many of the parameters

such as PFS are not significant and it cannot be excluded that inclusion of a larger number of NASH-HCC patients may change the data significantly.

B. The authors do not answer the question whether the differences in survival are due to failed checkpoint therapy or due to other differences between the two cohorts. Most likely, the differences in survival would persist if the authors removed all responders from the "other etiologies" group. Control groups that did not receive checkpoint inhibitors are missing to determine if survival is different between NASH and non-NASH HCC in patients who did not receive checkpoint inhibitors.

C. Is there any indication of increase NASH activity in patients receiving Pembro or Nivo?

D. There is no proper analysis of confounding factors.

E. Another problem is mixing Pembro and Nivo groups. Even though the target is the same, the authors need to provide subgroup analysis for this and increase the number far beyond what they have to make any meaningful conclusions in these subgroups.

F. Characterization of patients is insufficient - how were other liver diseases excluded, including ALD, which is not trivial, and especially important in such small cohorts?

5. Do the authors get the same results when blocking CTLA-4 - which was, even though not approved for HCC - the first approach and published study to show efficacy of checkpoint inhibitors in HCC?

Referee #2 (Remarks to the Author):

In their manuscript, Pfister and colleagues aim to show that CD8+PD-1+ T cells expand during progressing, diet-induced NAFLD and, upon treatment with anti-PD-1 antibodies, that these cells can promote carcinogenesis by establishing an inflammatory tumor microenvironment in a diet-induced, murine model of advanced NAFLD. Additionally, the authors observe a similar, intratumoral CD8+CD103+PD-1+ T cell subset in NASH-induced human HCC patients and claim that patients with NASH-induced HCC respond worse to anti-PD-1 therapy compared to HCC of other origin.

While the seminal observation in this paper is intriguing, namely that anti-PD-1 treatment can exacerbate tumorigenesis in a murine model of NASH-induced HCC, the authors fail to demonstrate clear causal relationships between the implicated cell types, liver inflammation and tumor development in the vast amount of the data they present, which therefore remain largely correlative. I will highlight my major concerns below.

1. In the reporting summary, the authors state that "Exclusion criteria was pre-established and the CD-HFD fed mice which did not show the NASH phenotype, high ALT, AST and body weight, were excluded from the analysis". I fail to understand why this decision was taken as these mice offer valuable insight in the author's proposed mechanism. Do CD-HFD mice without overt signs of NASH have reduced CD8+PD-1+ T cells? Do these mice also less frequently grow tumors upon anti-PD-1 blockade? Do the T cells in the livers of these mice fail display an enhanced effector phenotype? Aside from the valuable experimental insights that could be gained from these mice, the decision to exclude these CD-HFD but non-NASH mice from analysis also invalidates any claim that links a given diet to a given phenotype since mice that did not fit the authors' desired phenotype were excluded.

2. The data presented by the authors fail to demonstrate clear causal relationships. As an example, the authors note in lines 341-343 that a pro-inflammatory hepatic environment is created by TNF upon anti-PD-1 treatment, yet fail to show supporting evidence that this indeed drives "necro-inflammation" and accelerated hepatocarcinogenesis. The authors should neutralize TNF in their in vivo models to determine whether this molecule is indeed required for their phenotype, i.e., inflammatory microenvironment, liver damage and increased tumorigenicity.

3. Based on the authors' presented data, this problem can be further expanded. In Figure S9d and S9m, the authors show an increase in the number of antigen-presenting cells and increased MHC-

II expression. Are these recruited upon liver inflammation? Are they required for liver inflammation?

4. In Figure S11 the authors show an increase in many inflammatory mediators upon anti-PD-1 therapy; which of these are required for the accelerated carcinogenesis? While the authors propose a mechanism based on liver inflammation leading to increased hepatocarcinogenesis upon anti-PD-1 blockade, they provide little if any conclusive evidence for this hypothesis.

5. Some of the data the authors present seems internally inconsistent. As an example, the authors postulate that the pro-inflammatory hepatic environment is responsible for the increase in liver cancer incidence in anti-PD-1-treated mice, which they underscore by an increase in inflammatory cytokines in the liver microenvironment (Figure S11). However, they also show that upon CD8 depletion, which reduces cancer incidence, the inflammatory cytokines do not significantly reduce compared to the CD-HFD diet mice alone. This implies that the inflammatory microenvironment is not actually responsible for increased cancer incidence. How do the authors harmonize these findings?

6. Crucially, and related to my previous point, the authors also did not perform CD8 depletion in the context of anti-PD-1 treatment to show that CD8 cells are indeed the cells that are responsible for increased carcinogenesis upon anti-PD-1 therapy.

7. At times, the authors are (highly) selective in the data they choose to discuss and interpret. As an example, regarding Figure 1i, the authors describe the CD8⁺ T cells in CD-HFD mice to demonstrate profiles of cytotoxicity and effector function because of increased expression of GzmK/M and Pcd1. However, in the same plot shows that these cells have reduced expression of GzmA/B, Klrp1, Il2ra, TNF and Il2; all markers of effector/cytotoxicity. How do the authors harmonize these observations?

8. Regarding Figure 1e, the authors state that CD-HFD contain a significantly altered immune composition that mainly affects the CD8⁺ T cell compartment. However, this finding was not significant ($p=0.09$ for CD8⁺PD-1⁺ T cells and ns for CD8⁺ T cells). In this plot, the authors do show significant differences in frequency of CD4⁺ T cells ($p<0.01$), classical monocytes ($p<0.01$) and MDMs Ly6CHigh ($p=0.01$). Why are these cell types not regarded as interesting? Are these cells responsible for the authors' proposed phenotype? In line 259 the authors state that there are only minor differences in the CD4 compartment, yet when looking at the data (Figure S9h and Figure S9f) the difference in the CD4 subset of CD62L-CD44⁺CD69⁺ upon anti-PD-1 blockade is as strong as, if not stronger than, in the same subset of CD8 T cells, which the authors do deem interesting.

9. Along these lines, in line 387 the authors state that consistent with previous results, effects on the CD4⁺PD-1⁺ T cell compartment remained minor, yet the differences observed for matching analyses (i.e. S17a vs S17g, S17b vs S17f, S17i vs S17j) of CD4 and CD8 populations show similar, if not stronger, effects for the CD4 T cell population. Why are these differences disregarded by the authors?

10. Similarly, in Figure 5a, the authors claim that a CD8⁺PD-1⁺ T cell population arises upon NASH. However, there is a, perhaps even stronger, depletion of an Eomes⁺ gamma-delta T cell subset. Additionally, a very strong induction of a CD4⁺CD27⁺ population is observed in NASH samples. Why are these not discussed? Can these populations also be identified in the authors' murine models? Do these contribute to the authors' described phenotype? The authors should deplete CD4 T cells and gamma-delta T cells in their murine models, as these cell types may, at the very least, contribute to what occurs in patients.

11. The patient data is not convincing, but also does not match their murine models. In Figure 5a, the authors show that CD8⁺GzmB⁺ cells are specifically lost in NASH samples which seems to counteract the claim made by the authors that inflammatory CD8 T cells cause liver inflammation and associated carcinogenesis. The authors similarly show in S19a that IFN γ , Ccl3 and PD-L1 are in fact reduced in advanced NASH samples; does the loss of these inflammatory genes not counteract the claims made in Figure 3g, S4d, S10, S11 and S13a?

12. Lastly, the majority of patient data are not significant and show weak effect sizes; is it fair to draw strong conclusions on the basis of these data as the authors do?

Minor points:

- Figure 1j lacks a color scale bar and proper description. How does one interpret the difference between ND and CD-HFD in this plot?
- Where is the ND + PD-1^{-/-} in Figure 3b? Do these mice also get accelerated carcinogenesis?
- There is no color scale bar in Figure 3e.
- In Figure 5k, shouldn't progression-free survival and time to progression plots yield the exact same data, but inversed? Why don't these curves match?
- In Figure S1i, what is the parent population?
- In Figure S4a, how does one distinguish ND from CD-HFD mice? The y-axis lacks a label.
- Figure 5c is plotted in a confusing manner (as the z-score scale is red independent of whether it goes up or down), but it seems that the TNF signaling gene sets are actually decreasing in expression.
- Why do the PD-1^{-/-} mice still express PD-1 (Fig. S12e)?
- In Figure S13k, the authors should present cleaved Caspase 3 and cleaved Caspase 8 if they want to conclude something about cell death, as total, uncleaved levels of these proteins do not indicate cell death.
- In Figure S16f, the FACS plot does not match the quantification on the left.
- Regarding Figure S17b, the authors claim an increase in calcium levels in line 383 of their manuscript, but this difference is not significant.
- In Figure S18b, how does one interpret the difference between healthy, borderline NASH or NASH patients? There is no explanation of the color scale bar. Also, what are "randomly chosen CD45+ cells" as mentioned in the corresponding Figure Legend?
- Figure S19b is not legible.
- In lines 237-246 the authors describe that NK1.1-based depletion of immune populations did not result in changed liver pathology, body weight, fibrosis ALT, hepatic cytokines and hepatic chemokines. However, the animals who underwent this depletion also completely lacked liver cancer development. How does this happen if the authors did not detect any changes? The authors should perform NK1.1 depletion by itself to see if NK1.1+ cells, potentially depending on CD8 cells, are in fact responsible for the authors' phenotype.
- Sentence 289-292 is unclear.
- When discussing GSEA, the authors frequently use the wording 'reduced enrichment (e.g. line 241)' when talking about enrichment in the opposite phenotype. This is incorrect, as the absolute amount of enrichment is often similar just, as mentioned, in the opposite direction.

Referee #3 (Remarks to the Author):

This full article manuscript is novel, and the experimentation to support the conclusions is exhaustive and solid for the most part. In essence, the findings indicate that, in NASH livers, there is an accumulation/expansion of a pathogenic CD8 T cell population that expresses PD-1 and exacerbates NASH pathology and fosters hepatocellular carcinogenesis and progression. The inflammatory and tissue-damaging functions of this pathogenic CD8 T cells are repressed by PD-1 blockade that is common clinical practice for second-line treatment of advanced HCC and is under clinical trials for earlier stages of the disease. In fact, PD-L1 blockade plus anti-VEGF will soon become the standard of treatment for advanced HCC in first line. According to the findings in this paper upon PD-1 blockade, authors document an exacerbation of carcinogenesis and liver damage that questions the indication of PD-1 blockade in NASH-associated liver cancer. A balanced presentation of preclinical and supportive clinical results in patient specimens very much enhances the significance of this study.

Questions and comments:

1. TNF seems to be an actionable therapeutic target for the observed harmful effects of this CD8 T cell population. It would be interesting to know if TNF could be blocked preserving anti-cancer immunity (especially under checkpoint inhibition therapy) but preventing tissue damage and

carcinogenesis promotion.

2. Would PD-L1 blockade enhance liver cancer and tissue damage as well? Which cells are expressing PD-L1 in the system. This becomes important given the recent approval of atezolizumab + bevacizumab.
3. Results on NASH in human samples are compelling and supportive of the relevance of the findings. It would be interesting to know in such livers which cells express PD-L1.
4. What do you think is the fibrogenic factor/s promoted by pathogenic CD8 cells? Any candidates from the extensive transcriptomic analyses?
5. Are Kupffer cells involved in the CD8-dependent pathogenesis mechanisms?
6. Obesity and response to PD-1 associations have been reported (PMID: 30420753 and PMID: 30813970). According to these studies, obesity relates to T cell dysfunction that PD-1 blockade derepresses and results in better responsiveness. The models of NASH should suffer overweight as well as perhaps the patients in the reported series. This point should be addressed if possible and at least discussed. Authors may gain insight with their comparisons of the models with and without choline in the diet. As a potential consequence, would it be the case that in HCC patients, obese patients respond worse to treatment contrary to other indications? Of clinical note, advanced HCC patients frequently experience cachexia but perhaps less frequently so those with presumed or documented NASH etiology.
7. The retrospective series of patients with advanced HCC treated cannot be considered conclusive at this point and only hypothesis-generating. The wording there needs to be carefully down-toned.
8. An important message of this paper is that progression following PD-(L) treatment in NASH patients could be the development of a second primary malignancy rather than from the same one. Can this point be addressed in the models? Is multifocal cancer more common in those cases? The more CD8 pathogenic T cells in the infiltrate, the more multifocal the tumors?
9. The companion back to back paper shows more data on the physiology of the pathogenic CD8 T cells that I would otherwise ask to this article. Therefore, proper cross-reference of those findings is needed at least in discussion.

Referee #4 (Remarks to the Author):

This is an interesting and quite original study of the role of immunity in promoting liver cancer. There are data from the mouse models presented which show that CD8+ T cells can contribute to the pathology of NASH and the risk of cancers. The implication is that checkpoint blockade which can accentuate the function of CD8 populations can worsen disease. There are also some human data which are fairly consistent with this idea. It is perhaps not surprising that checkpoint inhibition might worsen an inflammatory condition, although inducing a cancer risk is very interesting.

Overall the authors do a very good job in describing the cellular responses and the impact of depletion/blockade. There seemed to be a bit of a gap around defining the mechanisms in terms of how the CD8+ T cell population induced cancer. Also it was somewhat unclear what the specificity of these T cells was and what was triggering their initial responsiveness in NASH. So although a strong case is made for the pro-tumor role the actual pathways to disease were less concrete.

Figure 1: There do not appear to be any iNKT cells in the UMAP or tSNE plots – these are discussed later in the text. That seems a little surprising as they are quite dominant in the mouse liver and have a clear transcriptional profile. Could the authors clarify where these cells lie. It would be also useful to know whether other unconventional cell subsets including GD T cells and MAIT cells are incorporated in this, although they are likely much rarer. The latter may be relevant even if rare as they have been linked to liver fibrosis. The same questions would also apply to the scRNAseq of the human samples

Figure 1e: What are the p values on the right referencing? The difference in the PD1+ population does not appear to be significant. How valid is the PD1+ subset as a subcluster and also what are

the critical significant differences apart from elevated PD1 expression – some justification for this early on would be helpful. Often PD1 expression is more of a gradient (even within PD1+ cells) so a binary distinction needs a bit more justification. Does this group of cells have distinct TCRs from the non-PD1 (or lower PD1) subset or are they the same population with distinct expression? Some data on this would address the question about specificity – although this would be better addressed by defining actual TCR-specific (or independent) functionality.

Figure 1f: The stains are both single stains. It should be possible to show a double staining CD8+PD1+ population and enumerate them as this seems like the critical part of the study.

Figure 1j: One of the most upregulated genes in the PD1+ subset is IL-10. Do the authors have any data on whether this is secreted by this subset. Although the subset is labelled as “PD1+” it is not the top upregulated gene here (as above). A side-by-side broader functional study would add a bit of resolution here and if they do secrete IL-10 this may impact on the overall interpretation. The interpretations about function are all via the screening approaches so some further specific back up by FACS/ELISA would be helpful in confirming functionality, especially in the context of an “exhausted” phenotype – this would clarify the statement on line 199 about “potential effector function”. Such an experiment would also be valuable in the anti-PD1 treated mice in later parts of the manuscript.

Figure 2: It was not that clear why depleting CD8s had no impact on ALT, suggesting they are not playing a role in vivo, while blocking PD1 had some impact (AST is not shown for the anti-CD8 treatment).

Line 202 – lack of impact of anti-PD1. Is there a control for this experiment? The implication is that this lack of impact is aetiology-specific but it may also be that the intervention does not work well in other HCC models.

Figure 5b and the text are presented in a slightly confusing way. It would be easier to understand the disease associations of %CD8 (of CD3), and % PD1+ (or MFI) of CD3+CD8+ first. The association of CD103 with tissue residency in the liver is not as good as other tissues, so a broader look at the CD8+PD1+ population by flow would be better as well as some caution in interpretation.

Figure 5e could include some study of CD4s as well for reference. That subset has been linked to NASH pathogenesis as well. As above, it should be possible to perform some dual CD8 and PD1 staining to map the subset of interest.

Figure 5f is not really that convincing of a relationship with TNF – the r-squared value would be better to illustrate and would be very low. If the authors think TNF secretion is critical it would be possible to explore this further in the mouse model.

For Figure 5G some disease controls would be valuable.

Line 493+: This sentence is perhaps overstating the data, which were not significant in all those parameters. It is likely quite hard to make the firmest comparisons, especially in such a retrospective analysis, where the heterogeneous group of patients with eg viral aetiologies will be on effective therapies - the actual aetiologies were not obvious in the supplementary data. This interpretation could be a bit more cautious throughout (eg it is in the abstract).

Author Rebuttals to Initial Comments

FULL AUTHOR REBUTTAL

(please note that the authors have quoted the reviewers in black and responded in blue)

**Referee #1 (Remarks to the Author):**

Using two different mouse models of NASH-induced HCC as well as data from patients with
NASH-associated HCC, the authors suggest the concept that CD8+PD1+ T-cells promote
NASH development and that treatment with checkpoint inhibitors may release the brake in
these NASH-promoting cells, resulting in disease exacerbation and more HCC, which they
proposed is confirmed by their findings of absent response to checkpoint inhibitors Nivolumab
and Pembrolizumab in patients with NASH-associated HCC but not in patients with HCC due
to other causes. While the analyses are carefully performed and raise the question of harmful
effects of checkpoints in NASH-associated HCC, both the mouse and patient studies have
major limitations, and it cannot be excluded that this paper sends the wrong message to the
community and will negatively impact the field.

We thank Referee #1 for appreciating that our experiments have been “carefully performed”
experiments as well as for outlining the potential clinical impact of our study on PD-1 targeted
immunotherapy in HCC. Also, we thank Referee #1 for pointing out the current limitations of
the applied mouse models and clinical cohorts of our study, which we have taken utmost
seriously and improved both. Statements on the role of checkpoint inhibitors in non-viral
etiologies in HCC have been tempered, but nonetheless reflect the results of the meta-
analysis, which is aligned with the pre-clinical findings.

In short:

(i) We have added a third preclinical mouse model of NASH with NASH to HCC transition
(Gomes et al., 2016; Tummala et al., 2014). Analysis of this model corroborated the link
between CD8+PD1+ T-cells and NASH development

(ii) We have extended our preclinical experiments with six novel treatment groups and
performed in detail analyses on the mechanism and functional link of liver damage,
inflammation, and responsiveness to anti-PD1-targeted immunotherapy in liver cancer.

(iii) We have added human clinical data sets (with 1656 HCC patients on immunotherapy
involving the important clinical trials - IMbrave 150; Checkmate 459; Keynote
240), enlarged our initial retrospective clinical cohort, and validated results
obtained from this cohort in a second cohort of HCC patients under
immunotherapy. Moreover, we corroborated our findings of CD8+PD1+
increasing by NASH in now in total 3 independent patient cohorts across
Europe by flow cytometry or single-cell RNA-seq.

Furthermore, we have performed CYTOF and scRNA Seq analysis of lymphocytes from livers
derived from human NAFLD/NASH and steatosis and compared these data with our preclinical
models – corroborating our data.

We hereby address the Referee’s concerns in the following section point-by-point.

We agree with the Referee that additional analyses of patient cohorts and mouse experiments
have been necessary to strengthen and corroborate our data.

We believe that we have achieved this in the new version of our manuscript by examining a
very large number of HCC patients on immunotherapy with viral and non-viral/NASH/NAFLD
origin – adding both individual cohorts from independent centers as well as a meta-analysis
from the most important published trials on immunotherapy on HCC. Furthermore, we have
strongly increased our *in vivo* analyses applying several different treatments in combination
with anti-PD1 treatment, and a third NASH mouse model, validating further the reliability of our
pre-clinical mouse models.

In particular, we have now added a meta-analysis including 1656 HCC patients with different
underlying etiologies (viral and non-viral) treated with immunotherapy derived from three large
clinical trials (included in **Figure 6, Extended Data 30-32** and **Rebuttal Figure 1, 2**).
(Comment from our side: The total number of patients in the combined cohort is 1656.
However, one patient in the CheckMate-459 had unknown etiology, and could therefore not be
included in the quantitative meta-analysis). We conducted this meta-analysis to support the
experimental data suggesting that anti-PD1/anti-PDL1 checkpoint inhibitors would have a
distinct effect in non-viral (NASH-related) HCC as opposed to viral-related HCC (included in
**Figure 6, Extended Data 30-32** and **Supplementary Table 7** and **Rebuttal Figure 1, 2**). Out
of eight studies identified in the search, only three fulfill the pre-established criteria (included
in **Extended data 30** and **Rebuttal Figure 1a, b**), including a total of 1656 HCC patients.

These randomized controlled trials (RCT) included **A**) CheckMate-459 (Yau et al., 2019), a
first-line, randomized, sorafenib-controlled trial testing nivolumab (an anti-PD1 monoclonal
antibody) in monotherapy (n=742), **B**) IMbrave150 (Finn et al., 2020), a first-line, randomized,
sorafenib-controlled trial testing the combination of atezolizumab (an anti-PD-L1 monoclonal
antibody) and bevacizumab (an anti-VEGF-A monoclonal antibody) (n=501), **C**) KEYNOTE-
240 (Finn et al., 2019), a second-line, randomized, placebo-controlled trial testing
pembrolizumab (an anti-PD1 monoclonal antibody) monotherapy.

All three trials reported a subgroup analysis of survival data stratified according to disease
etiology: hepatitis B virus (HBV), hepatitis C virus (HCV), and non-viral, which mostly includes
both NASH and alcohol intake.

First, we analyzed whether checkpoint inhibitors were effective in each of three etiologies
(HBV, HCV, and non-viral) and then compared the efficacy by categorizing patients with viral
vs non-viral etiology HCC in all three phase III studies including a total of 1656 patients.
Immunotherapy was superior to the control arm in both HBV (n= 574; p=0.0008) and HCV-
related HCC patients (n= 350; p=0.04), **but not in non-viral** HCCs (n=737; p=0.39). The
magnitude of the benefit with checkpoint treatment according to etiology was significantly
better in viral etiology (pooled HBV and HCV cases) [HR: 0.64; 95%CI 0.48-0.94] than non-
viral etiology [HR: 0.92; 95%CI 0.77-1.11]; p of interaction= 0.03 (Rebuttal Figure 1d). Then,
we dissected the specific effect by each viral type in a subgroup analysis. Comparison of
magnitude of effect was significant comparing HBV vs. non- viral etiology (n=1311; p
interaction= 0.03), and there was a non-significant trend for HCV vs. non-viral etiology
(n=1082; p of interaction=0.14) (Rebuttal Figure 2a,b).

Second, considering that two out of three RCT were conducted in first-line treatment of
advanced HCC with a homogeneous control arm (sorafenib), we conducted a subgroup
analysis specifically with these two studies (n= 1234). This approach allowed us to control for
biases related to the study population and distinct control arms. Immunotherapy was superior
to sorafenib in both HBV (n= 473; p=0.03) and HCV-related HCC patients (n= 281; p=0.03),
but not in non-viral HCC (n=489; p=0.62). (Rebuttal Figure 2d,e). The magnitude of the
checkpoint treatment effect vs sorafenib according to etiology showed a non-significant trend
favoring viral etiology (n=754; HR: 0.61 (95%CI 0.40-0.93)] when compared to non-viral
etiology [n=489; HR: 0.94 (95%CI 0.75-1.18) (p of interaction= 0.08) (Rebuttal Figure 2c). As
a result, we have included these data in the main text and main figure (**Figure 6**) of the
resubmitted manuscript.

Based on these data we want to point out that it is - as indicated by Referee#1 - of the highest
importance to us to specifically define/tone down appropriately the message of our manuscript:
Our manuscript does not indicate that immunotherapy is not beneficial for HCC patients at all.
Our manuscript rather demonstrates that HCC patients with viral etiologies do respond well
and achieve survival benefits - however, that patients with non-viral etiologies (e.g. NASH) do
not achieve a significant outcome benefit.

We thus propose to stratify HCC patients who are very likely to profit from immunotherapy and
strengthen the argumentation to use immunotherapy in specific cohorts of HCC patients. We

agree with Referee#1 that this information needs to be articulated in the paper appropriately
 not to deliver wrong messages but to be very specific.
 We truly believe that these are important clinical data, also providing the basis to test our
 hypotheses in prospective studies on non-significantly beneficial effects in terms of OS for
 immunotherapy in HCC patients with non-viral and NAFLD/NASH etiology, in particular.

 **Rebuttal Figure 1**

(a) Selection of articles assessing the clinical outcome of immune checkpoint inhibitors in
 advanced HCC for inclusion in the systematic review and meta-analysis. ICPI: Immune
 checkpoint inhibitor. (b) Pooled baseline characteristics of the patients included in the meta-
 analysis (total n= 1656). (c) A total of 1656 patients were included in all three randomized trials,

Rebuttal Figure 2

A total of 1656 patients were included in all three randomized trials, and 985 patients received
a checkpoint inhibitor. Subgroup analysis was performed to study the specific effects of
immunotherapy comparing non-viral etiologies (n=737) with (a) HBV (n=574) or (b) HCV
(n=345). A total of 1243 patients were included in two first-line trials comparing PD-1 or PD-L1
targeted immunotherapy to sorafenib. 707 patients received an immune checkpoint inhibitor
(either PD-1 or anti-PD-1). (c) HCV and HBV were pooled into a separate category, termed
“viral”, and a subsequent meta-analysis comparing viral (n=754) and non-viral (n=489), mostly
NASH and alcohol intake, was performed. A subgroup analysis studying the specific effects of
non-viral etiologies (n=489) on the magnitude of effect of immunotherapy are presented, when
compared to (d) HBV (n=473) or (e) HCV (n=281). Hazard ratios for each trial are represented
by squares, the size of the square represents the weight of the trial in the meta-analysis. The
horizontal line crossing the square represents the 95% confidence interval (CI). The diamonds
represent the estimated overall effect based on the meta-analysis random effect of all trials.

Specific points:

1. The NASH-HCC mouse models represent a major weakness of this paper and may lead to
premature conclusions on the effect of PD-1 therapy in NASH-associated HCC. While the
employed mouse models may be among the best to study various aspects of NASH, several
limitations preclude them from serving as useful preclinical models for HCC:

We thank Referee #1 for appreciating the used NASH-HCC models as “among the best to
study various aspect of NASH”, and we agree in general that studies in preclinical models have
their limitations, especially in the context of chronic inflammation-induced cancer. These
limitations of preclinical models are pronounced if mouse models are not used chronically (e.g.
≥ 1 year).

However, we would like to point out that the model(s) used in our paper reflect sporadic liver
cancer development with similar immune cell signature, pathophysiology, and the
heterogeneous genetic landscape found in humans (Ma et al., 2016; Malehmir et al., 2019;
Wolf et al., 2014 - and the data reported in this manuscript). In response to Referee #1, we
have performed synteny analyses comparing HCC nodules from individual mice with human
HCC (included in **Extended Data 6** and **Rebuttal Figure 3**). These data indicated no
significant changes in genomic aberrations and thus a comparable character between human
HCC and mouse liver tumors.

**Rebuttal Figure 3**

(a) Synteny analysis of mouse-HCC and (b) quantification of genomic aberrations by array
comparative genomic hybridization (aCGH) after 12 months on CD-HFD (n= 19) and human
NALFD/NASH-HCC (n= 78).

1a. Many mouse models of cancer are simply not altered responsive to checkpoint inhibition because
of low mutational load and lacking tumor antigens/neo-antigens. The authors do not provide
evidence that the employed models have a mutational load that is at least as high as in that
seen in HCC patients.

We thank and agree with Referee #1 for pointing out the possible unresponsiveness of clinical
models to checkpoint inhibition due to low mutational load. The mutational load HCC of most
conventional preclinical models is indeed very low, or lower compared to human HCC. This is
the case, in particular when taking into account liver cancer models triggered through
transgenesis, e.g. c-myc transgenic mice or preclinical mouse models with hydrodynamic tail
vein injection (HTDVi) of oncogenic drivers and tumor suppressors. In those models, pre-
existing genetic drivers and tumor suppressor deficiencies can be a major drawback
concerning additional mutations and increased mutational load.

In a chronic model of liver inflammation, we could show that mutational load increases over
time - comparing 9, 12, and 15 months (Finkin et al., 2015).

Our chronic, spontaneous NASH-HCC models develop liver cancer in the absence of specific
genetic drivers – but rather through chronic liver damage triggering DNA instability, ER and
mitochondrial stress, accumulating genetic hits over time stochastically triggering liver cancer
formation, like has been shown in human NASH (Boege et al., 2017).

In light of the important question of Referee #1, we have now included a further genetic
screening of 19 mouse HCC nodules in our revised manuscript and compared them to human
HCC nodules and their mutational landscape (included in **Extended Data 6** and **Rebuttal**
**Figure 3**). Data from this study confirm that quality, degree of heterogeneity, and load of
chromosomal aberrations (gains and deletions) of the used NASH to HCC mouse model is
similar to human HCC (Wolf et al., 2014 and this manuscript). Strikingly, also the immune cell
populations revealed by scRNA Seq are comparable in mouse and human NASH underscoring
that the used NASH-HCC mouse model reflects the basic immune landscape of NASH and
subsequently NASH-HCC transition.

Furthermore, we would like to point out, that overall in human HCC so far a responder rate of
17-20% for PD-1-targeted monotherapy was observed, potentially due to a generally low
amount or lack of broad-scale tumor antigens in HCC (El-Khoueiry et al., 2017; Zhu et al.,
2018).

1b. The mouse model - albeit taking over a year - is not comparable to HCC development in
patients, which takes decades and mostly occurs in the setting of advanced fibrosis or cirrhosis
(even though a subset of NASH-associated HCC patients do not have cirrhosis, most of them
have advanced fibrosis). Importantly, in most of these patients, the underlying NASH is much
less activate than in earlier disease stages/burnt out - meaning that the risk of increasing NASH
activity and thereby worsening not only NASH but also increasing NASH-HCC is much lower
and possibly not even relevant. The authors' conclusions would be relevant if one employed
checkpoint inhibitors for HCC prevention but are likely not applicable to patients except for
those, in whom HCC develops in the absence of cirrhosis and with high NAS.

We thank Referee #1 to point out the limitations of preclinical models in comparison to patient-
derived data. We agree that preclinical models do not take decades to develop HCC (averages
mouse life-time ~ 2 years), however, mouse models have helped in the identification of
molecular and cellular mechanisms leading to liver cancer (Ringelhan et al., 2018) - and if used
in a long term fashion - up to 2 years - they do recapitulate in part the chronicity of inflammatory
etiologies driving liver cancer. Moreover, mouse liver cancer occurs in age comparable to the
life-span of patients (we applied 12 - 15 months of NASH-diet feeding months from 2 months
of age onwards), which is comparable with the 4th to 5th life decade in humans regarding the
age of HCC onset/HCC disease (Llovet et al., 2016).

We would like to highlight, that preclinical models implemented in our study develop fibrosis to
different degrees (mostly mild peri-cellular fibrosis to periportal streets and cirrhosis (Malehmir
et al., 2019; Wolf et al., 2014)).

Thus, we agree with Referee #1, that the preclinical model might represent a subgroup of
patients developing HCC in the background of fibrosis.

Moreover, we agree with Referee#1, that underlying NASH in HCC patients might be less
activated compared to earlier stages and burnt-out.

Of note, clinical state-of-the-art care includes the use of corticosteroids for the treatment of
adverse effects (Weiler-Normann and Lohse, 2016), which can also induce NASH-like
pathologies. Thus, understanding mechanisms of underlying NASH in NASH-HCC in
preclinical models is of vital interest. Furthermore, current studies explore checkpoint inhibitors
for HCC as prevention of recurrence (Kudo, 2018).

We take this point of Referee #1 utmost seriously and devised importance for this critique in
the discussion section of our manuscript. We toned down our interpretations from human
cohorts analyzed in a retrospective design, although we believe the points raised in our
manuscript address important points like a potential stratification for etiology, the need for
biomarkers, and clinical awareness of potential unfavorable side-effects of checkpoint inhibitor
usage (e.g. similar to hyper progressive disease during PD-1 blockade in advanced HCC (Kim
et al., 2020)).

In line with the suggestion of Referee #1 to explore the limitations of our mouse models and to
understand the link between liver inflammation and tumor development better, we have re-
analyzed our mouse data sets to dissect potential correlations of fibrosis, tumor size, tumor
nodule number, flow cytometry data of livers, ALT, NAS, CD8, and PD-1 expression using
artificial intelligence, machine learning and neuronal networking (included in **Figures 1** and
**Extended Data 4 and 24** and **Rebuttal Figure 4a,b, 5**).

Moreover, we have added a third NASH-HCC mouse model, which corroborates the link
between the amount of CD8+, PD1+ T-cells, and NASH (included in **Extended Data 3e** and
**Rebuttal Figure 4c**).

Of note, we now underlined that our preclinical NASH models recapitulate in part the alterations
of hepatic immune cells in NASH by performing correlative analyses and machine learning of
liver-derived lymphocytes of NASH patients by CYTOF, classical flow cytometry, and scRNA-
seq (included in **Figure 5, Extended Data 25-27** and **Rebuttal Figure 6-9**). Data from these
analyses demonstrate that the pro-tumorigenic T cell population found in our preclinical NASH
mouse models livers (CD8+PD1+CXCR6+) are also found in / and correlate with NASH in
human livers (CD8+PD1+CD103+).

Research for a Life without Cancer

**Rebuttal Figure 4**

(a) UMAP representation of 63 parameters (serology, flow cytometry, histology) indicating
 NASH pathology severity measured of 12 months ND or CD-HFD fed mice (ND n= 22 mice;
 CD-HFD n= 31 mice). (b) Data gathered from hepatic tissue analyses was binary correlated
 with each other of 6- or 12-months ND or CD-HFD fed mice (ND n= 47 mice; CD-HFD n= 72
 mice). (c) H&E, CD8 and PD-1 staining, evaluation by NAS and quantification of CD8+ cells

and PD-1+ expressing cells by immunohistochemistry of 32-weeks old hURI-tetOFFhep and
non-transgenic litter control mice (n=6 mice/group). Arrowheads indicate specific staining
positive cells. Scale bar: 100 μ m.

Research for a Life without Cancer

Rebuttal Figure 5

(a) UMAP representation of 63 parameters (serology, flow cytometry, histology) and (b)
selected display of analyzed parameters indicating NASH pathology severity measured of 12
266 months ND or CD-HFD fed mice (ND n= 22 mice; CD-HFD n= 31 mice; CD-HFD + α -PD-1 n=
41 mice; CD-HFD + α -PD-L1 n= 6 mice; CD-HFD + α -CD8 n= 24 mice; CD-HFD + α -
CD8/NK1.1 n= 6 mice; CD-HFD + α -PD-1/ α -CD8 n= 9 mice; CD-HFD + α -TNF n= 10 mice;
CD-HFD + α -PD-1/ α -TNF n= 11 mice; CD-HFD + α -CD4 n= 9 mice; CD-HFD + α -PD-1/ α -CD4
n= 9 mice). (c) Data gathered from hepatic tissue analyses was binary correlated with each
other of 6- or 12-months ND, CD-HFD or CD-HFD + 8 weeks treatment of α -CD8, α -CD8/ α -
NK1.1; α -PD-1, α -PD-1/ α -CD8, α -TNF, α -PD-1/ α -TNF, α -CD4, or α -PD-1/ α -CD4 fed mice (ND
n= 47 mice; CD-HFD n= 72 mice; CD-HFD + α -PD-1 n= 41 mice; CD-HFD + α -PD-L1 n= 6
mice; CD-HFD + α -CD8 n= 29 mice; CD-HFD + α -CD8/NK1.1 n= 6 mice; CD-HFD + α -PD-1/ α -
CD8 n= 9 mice; CD-HFD + α -TNF n= 10 mice; CD-HFD + α -PD-1/ α -TNF n= 11 mice; CD-HFD
+ α -CD4 n= 9 mice; CD-HFD + α -PD-1/ α -CD4 n= 9 mice).

Rebuttal Figure 6

(a) Flow cytometry plots, quantification of patient-liver-derived PD-1+CD8+ T-cells, and (b)
correlation of PD-1+CD8+ T-cells with BMI, NAS and ALT of healthy or NAFLD/NASH patients
(Supplementary Table 1: healthy n= 8 patients; NAFLD/NASH n= 16 patients). (c) UMAP

Research for a Life without Cancer

representation of randomly chosen CD45+ cells and (b) flow cytometry plots and quantification
of CD8+PD-1+CD103+ derived from hepatic biopsies of control, or NAFLD/NASH patients
(Supplementary Table 2: control n= 6 patients; NAFLD/NASH n= 11 patients) Populations:
CD8+ (violet), CD8+PD-1+CD103+ (red). (e) UMAP representation of CD3+ cells and (f)
analyses of differential gene expression by scRNA-seq of control, or NAFLD/NASH patients
(control n= 4 patients; NAFLD/NASH n= 7 patients). (f) Correlation of significant differentially
expressed genes in liver-derived CD8+PD-1+ compared to CD8+PD-1- T-cells subsets of 12
289 months CD-HFD fed mice and NAFLD/NASH patients (mouse: n= 3 mice; human: n= 3
patients). (g) RNA Velocity analyses of scRNA-seq data showing (h) expression,
transcriptional activity, (i) gene expression and (j) correlation of expression along the latent-
time of selected genes along the latent-time of patient-liver-derived CD8+ T-cells of control, or
NAFLD/NASH patients in comparison to mouse-liver-derived CD8+ T-cells (patients:
NAFLD/NASH n= 3 patients; mouse: n= 3 mice/group).

**Rebuttal Figure 7**

(a) Flow cytometry plot of FMO control, (b) quantification of patient-liver-derived PD-1+CD8+
T-cells, and (c) quantification of CD4, CD8, γδ, NK and NKT cells healthy or NAFLD/NASH
patients (Supplementary Table 1: healthy n= 8 patients; NAFLD/NASH n= 16 patients). (d)
Analysis of randomly chosen CD45+ cells and (e) average marker expression of defined
CD45+ subsets by flow cytometry derived from hepatic biopsies of control and NAFLD/NASH
patients to define distinct marker expression (Supplementary Table 2: control n= 6 patients;
NAFLD/NASH n= 11 patients). (f) Definition of cellular subsets, (g) relative quantification of
defined cellular subsets of randomly chosen CD45+ cells, (h) polarization of CD8+ T-cells and
(i) quantification of CD4+CD27+, or γδ TCR+Eomes+, T-cells by flow cytometry derived from
hepatic biopsies of healthy and NAFLD/NASH patients (Supplementary Table 2: control n= 6
patients; NAFLD/NASH n= 11 patients).

**Rebuttal Figure 8**

(a) tSNE representation, (b) marker expression, (c) average marker expression of defined T-
cell subsets of patient-liver-derived T-cells analyzed by CYTOF of control and NAFLD/NASH
patients (control n= 11 patients pooled in 3 analyses; NAFLD/NASH n= 16 patients pooled in
5 analyses). (d) Composition, (e) HSNE representation of defined T-cell subsets and (f)
quantification of CD8+CD103+PD-1+ cells of patient-liver-derived T-cells analyzed by CyTOF
of control and NAFLD/NASH patients (control n= 11 patients pooled in 3 analyses;
NAFLD/NASH n= 16 patients pooled in 5 analyses).

Research for a Life without Cancer

Rebuttal Figure 9

(a) NAS and BMI of patients used for scRNA-seq analyses of patient-liver-derived T-cells of
control and NAFLD/NASH patients (control n= 4 patients; NAFLD/NASH n= 7 patients). (b)
UMAP representation, marker expression, (c) relative quantification and (d), (e), (f) polarization
of defined T-cell subsets of defined T-cell subsets of patient-liver-derived T-cells by scRNA-
seq of control and NAFLD/NASH patients (control n= 4 patients; NAFLD/NASH n= 7 patients).
(g) Differential gene expression of CD4+PD-1+ vs CD4+ T-cells and (h) selected average
marker expression in CD4+ and CD8+ T-cell subsets of by scRNA-seq of control and
NAFL/NA2SH patients (control n= 4 patients; NAFLD/NASH n= 7 patients).

2. In relation to above-described limitations of the model, the paper does not sufficiently focus
on dual functions of CD8+PD1+ T-cells, promoting NASH but possibly also restricting HCC.
These functions are likely to occur at different stages in patients.

We thank Referee #1 for this important concern. We agree that the effects of CD8+PD1+ cells
are executed at different time points. However, we would like to draw attention to the point that
immunotherapy is considered to boost pre-existing inflammation (determined e.g. by
evaluation of liver infiltration by immune cells using immunohistochemistry or flow cytometry
for CD3, CD8, and PD-L1). Our data rather indicate that this certain population has no impact
in restricting HCC development - in the context of NASH - and even immunotherapy. In fact,
we show that depletion of CD8+ T-cells in NASH prevents NASH to HCC transition.

Thus, CD8+PD1+ T cells drive NASH which is exacerbated in the context of anti-PD1-related
immunotherapy. We have now pointed this out more clearly, executed novel experiments to
underline this point of early (NASH) and late time points (NASH to HCC transition) and have
further discussed this in the discussion section (see also below) as well analyzed these cells
in the context of human NASH.

To mirror the clinical status of the majority of patients at the time of diagnosis, we performed
PD-1-targeted checkpoint inhibition in mice with pre-existing liver tumors (**Extended Data 6**
**and 7** and **Rebuttal Figure 10, 11**) and performed now MRI-guided follow up.

Our data clearly show, that anti-PD1 or anti-PDL1-related immunotherapy does not stop or
revert tumor burden but rather supports further tumor abundance. In contrast, when anti-CD8
antibody therapy was applied, it decreased tumor incidence and thus development (**Figure 2,**
**Extended Data 8** and **Rebuttal Figure 12q**). Furthermore, we underlined the importance of
hepatic CD8+ T-cells abundance driving NASH-induced hepatocarcinogenesis by antibody-
based treatments in our mouse model (anti-CD8/anti-NK1.1, anti-CD4, anti-TNF; included in
**Figure 2 and 4, Extended Data 8, 9, 20-23** and **Rebuttal Figure 12b-d, 13-18**), as well as
cross-referencing to the co-submitted manuscript Dudek et al., which describes molecular

mechanisms of CD8+ T-cell-mediated liver damage. Additionally, we dissected CD8+ T-cell
mediated mechanisms driving NASH-induced hepatocarcinogenesis in PD1-targeted
immunotherapy by antibody-based treatments (anti-CD8/anti-PD1, anti-TNF/anti-PD1, anti-
CD4/anti-PD1; included in **Figure 4, Extended Data 20-23** and **Rebuttal Figure 12b-d, 15-
18**). These data indicated that the abundance of CD8+ T-cells, as well as CD8+ T-cell-derived
TNF plays an important role in boosting liver cancer in the context of NASH/HCC related
immunotherapy. Of note, velocity analyses of scRNA-seq for transcriptional activation, or
proteome analyses of sorted cells could not detect different phenotypes between CD8+PD1+
T-cells derived from mice fed CDHFD with NASH or CDHFD treated with an anti-PD1 related
therapy in the context of HCC development, indicating that the main proportion of CD8+PD1+
T-cells in our preclinical models drive hepatocarcinogenesis and do not restrict HCC (included
in **Figure 4, Extended Data 4 and 24** and **Rebuttal Figure 4b, 5c, 19**).

Further, our data show that anti-PDL1 therapy lead (included in **Extended Data 7** and **Rebuttal
Figure 11**) to the same effects as observed in the anti-PD1 therapy (included in **Extended
Data 6** and **Rebuttal Figure 10**) or in the context of our analyses using PD1 knock-out mice
developing NASH/HCC (included in **Figure 3, Extended Data 14** and **Rebuttal Figure 20**).

Data that have not been included in the initial submission of the manuscript indicate that PD-1
targeted immunotherapy-induced hepatic inflammation triggers the enrichment of central
memory-like cells (CD44+CD62L+CD8+) but not T-cells with a naïve character (CD62L+CD8+)
(included in **Extended Data 6** and **Rebuttal Figure 10I**). This enrichment of memory-like
CD44+CD62L+CD8+ T-cells can be explained by one of two options: these cells might be
expanded and infiltrate the liver upon the anti-PD-1 targeted immunotherapy to either drive
hepatic inflammation or these memory-like T-cells might be indicative of a subset of T-cells
reactive to tumor-associated antigens and thus of CD8+ T-cells of a dual role (included in
**Extended Data 6** and **Rebuttal Figure 10I**). In respect of the co-submitted manuscript Dudek
et al., CD8+ T-cells drive liver damage and subsequently liver cancer in NASH in an antigen-
independent manner, thus the enrichment of memory-like CD44+CD62L+CD8+ T-cells upon
PD-1 targeted immunotherapy might argue in favor of a dual role of CD8 T-cells. However,
tumor size, tumor number per liver, and tumor incidence are not affected by increased
CD44+CD62L+CD8+ T-cells, arguing against a tumor restricting function of CD8 T-cells in this
context.

Finally, we would like to draw again the attention to the improved cross-referencing of the
revised manuscript to the co-submitted manuscript Dudek et al..

Data described in this manuscript demonstrate that the NASH-induced microenvironment
drives hepatic inflammation in a TCR-independent manner and thus rather describes a

Research for a Life without Cancer

mechanism that activates CD8+T-cells downstream of the TCR through environmental
signaling (e.g. acetate, IL21 signaling), arguing against a tumor antigen-specific CD8+ T-cells
mediated HCC restriction in the context of NASH. It is exactly these CD8+ T-cells which –
altered by the NASH liver microenvironment acquired a pro-tumorigenic phenotype – which
we can detect also by analysis of the ICF signature – predictive of inflammation triggered liver
cancer in humans. Notably, CD8 depletion eliminates this signature – strongly underlining that
CD8 T cells are the main source of driving the pro-tumorigenic environment.

Rebuttal Figure 10

(a) MRI pictures of liver of mice after 13- months CD-HFD-fed mice followed by 7 weeks treatment of CD-HFD or CD-HFD + 7 weeks by α -PD-1 antibodies (CD-HFD n= 6 mice; CD-

406 HFD + α -PD-1 n= 4 mice). Lines indicate tumor nodule. Scale bar: 10 mm. (b) Histological
staining of hepatic tissue by H&E, Sirius Red and CD8 of 15 months ND, CD-HFD or CD-HFD
+ 8 weeks treatment of α -PD-1 (H&E: ND n= 3 mice; CD-HFD n= 10 mice; CD-HFD + α -PD-1
n= 8 mice; Sirius Red: ND n= 3 mice; CD-HFD n= 5 mice; CD-HFD + α -PD-1 n= 9 mice; CD8:
ND n= 5 mice; CD-HFD n= 5 mice; CD-HFD + α -PD-1 n= 3 mice). Scale bar: 50 μ m.
Arrowheads indicate CD8+ cells. (c) Body weight of 15 months ND, CD-HFD or CD-HFD + 8
412 weeks treatment of α -PD-1 (ND n= 5 mice; CD-HFD n= 4 mice; CD-HFD + α -PD-1 n= 9 mice).
(d) NAS evaluation by H&E of 15 months ND, CD-HFD or CD-HFD + 8 weeks treatment of α -
PD-1 (ND n= 3 mice; CD-HFD n= 10 mice; CD-HFD + α -PD-1 n= 8 mice). (e) Fibrosis
evaluation of Sirius Red staining of 15 months ND, CD-HFD or CD-HFD + 8 weeks treatment
of α -PD-1 (ND n= 3 mice; CD-HFD n= 5 mice; CD-HFD + α -PD-1 n= 9 mice). (f) ALT levels of
15 months ND, CD-HFD or CD-HFD + 8 weeks treatment of α -PD-1 (ND n= 3 mice; CD-HFD
n= 4 mice; CD-HFD + α -PD-1 n= 8 mice). (g) Quantification of CD8 and (h) PD-1 staining of
hepatic tissue by immunohistochemistry of 15 months ND, CD-HFD or CD-HFD + 8 weeks
treatment of α -PD-1 (ND n= 3 mice; CD-HFD n= 4 mice; CD-HFD + α -PD-1 n= 8 mice; intra-
tumoral staining: CD-HFD n= 3 mice; CD-HFD + α -PD-1 n= 8 mice). (i) Quantification and (j)
expression of PD-1 of hepatic CD4+ and CD8+ T-cells by flow cytometry of 15 months CD-
HFD or CD-HFD + 8 weeks treatment of α -PD-1 (CD-HFD n= 4 mice; CD-HFD + α -PD-1 n= 8
mice). (k) Macroscopic images of liver of 15 months ND, CD-HFD or CD-HFD + 8 weeks
treatment of α -PD-1. Arrowheads indicate tumor/lesions. Scale bar: 10 mm. (l) Quantification
of CD8+ T-cells by flow cytometry of 15 months CD-HFD or CD-HFD + 8 weeks treatment of
α -PD-1 (ND n= 3 mice; CD-HFD n= 4 mice; CD-HFD + α -PD-1 n= 8 mice). (m) Quantification
of tumor/lesion size, (n) tumor load and (o) tumor incidence of 15 months CD-HFD or CD-HFD
+ 8 weeks treatment of α -PD-1 (tumor/lesion size and tumor load: CD-HFD n= 9 mice; CD-
HFD + α -PD-1 n= 7 mice; tumor incidence: CD-HFD n= 17 tumors/lesions in 22 mice; CD-HFD
+ α -PD-1 n= 10 tumors/lesions in 10 mice).

Rebuttal Figure 11

(a) MRI pictures of liver of mice after 13 months CD-HFD followed by 7 weeks treatment to
 CD-HFD or CD-HFD-fed mice + 7 weeks by α -PD-L1 antibodies (CD-HFD n= 6 mice; CD-HFD
 + α -PD-L1 n= 8 mice). Lines indicate tumor nodule. Scale bar: 10 mm. (b) Macroscopic images
 of liver of 12 months ND, CD-HFD or CD-HFD + 8 weeks treatment of α -PD-L1. Arrowheads
 indicate tumor/lesions. Scale bar: 10 mm. (c) Body weight, ALT levels of 12 months ND, CD-
 HFD or CD-HFD + 8 weeks treatment of α -PD-L1 (Body weight, ALT, : ND n= 8 mice; CD-HFD
 n= 6 mice; CD-HFD + α -PD-L1 n= 6 mice) (d) and (e) NAS evaluation by H&E, Fibrosis
 evaluation of Sirius Red staining, quantification of CD8, PD-1 and PD-L1 staining of hepatic
 tissue by immunohistochemistry of 12 months ND, CD-HFD or CD-HFD + 8 weeks treatment
 of α -PD-L1 (NAS: ND n= 7 mice; CD-HFD n= 6 mice; CD-HFD + α -PD-L1 n= 6 mice; Sirius
 Red: ND n= 7 mice; CD-HFD n= 5 mice; CD-HFD + α -PD-L1 n= 6 mice; CD8, : ND n= 5 mice;
 CD-HFD n= 5 mice; CD-HFD + α -PD-L1 n= 5 mice; PD-1, PD-L1: ND n= 5 mice; CD-HFD n=

5 mice; CD-HFD + α -PD-L1 n= 6 mice). Scale bar: 100 μ m. (f) Tumor/Lesion incidence in CD-
HFD or CD-HFD + 8 weeks treatment of α -PD-L1 fed mice (CD-HFD n= 19 tumors/lesions in
25 mice; CD-HFD + α -PD-L1 n= 7 tumors/lesions in 8 mice). Arrowheads indicate specific
staining positive cells.

Rebuttal Figure 12

(a) Quantification of tumor incidence of 12 months CD-HFD or CD-HFD + 8 weeks treatment
of α -CD8, co-depletion of α -CD8/NK1, or α -PD-1 (tumor incidence: CD-HFD n= 32
tumors/lesions in 87 mice; CD-HFD + α -CD8 n= 2 tumors/lesions in 31 mice; CD-HFD + α -
CD8/NK1.1 n= n= 0 tumors/lesions in 6 mice; CD-HFD + α -PD-1 n= 33 tumors/lesions in 44
mice). (b) ALT and (c) NAS evaluation of 12 months ND, CD-HFD, CD-HFD + 8 weeks
treatment of α -PD-1, α -PD-1/ α -CD8, α -TNF, or α -PD-1/ α -TNF fed mice (ND n= 30 mice; CD-
HFD n= 47 mice; CD-HFD + α -PD-1 n= 35 mice; CD-HFD + α -PD-1/ α -CD8 n= 9 mice; CD-
HFD + α -TNF n= 10 mice; CD-HFD + α -PD-1/ α -TNF n= 11 mice). (d) Quantification of tumor
incidence of 12 months CD-HFD or CD-HFD + 8 weeks treatment of α -CD8, α -CD8/NK1.1, α -
PD-1, α -PD-1/ α -CD8, α -TNF, α -PD-1/ α -TNF fed mice, α -CD4, or α -PD-1/ α -CD fed mice 1
(tumor incidence: CD-HFD n= 32 tumors/lesions in 87 mice; CD-HFD + α -CD8 n= 2
tumors/lesions in 31 mice; CD-HFD + α -CD8/NK1.1 n= 0 tumors/lesions in 6 mice; CD-HFD +
α -PD-1 n= 33 tumors/lesions in 44 mice; CD-HFD + α -PD-1/ α -CD8 n= 2 tumors/lesions in 9
mice; CD-HFD + α -TNF n= 3 tumors/lesions in 10 mice; CD-HFD + α -PD-1/ α -TNF n= 3
tumors/lesions in 11 mice); CD-HFD + α -CD4 n= 3 tumors/lesions in 9 mice; CD-HFD + α -PD-
1/ α -CD4 n= 8 tumors/lesions in 9 mice).

**472 Rebuttal Figure 13**

(a) Body weight of 12 months ND, CD-HFD or CD-HFD + 8 weeks treatment of α -CD8 (ND n=
15 mice; CD-HFD n= 28 mice; CD-HFD + α -CD8 n= 28 mice). (b) Assessment of metabolic
tolerance by intra peritoneal glucose tolerance test of 12 months CD-HFD or CD-HFD + 8
476 weeks treatment of α -CD8 (CD-HFD n= 8 mice; CD-HFD + α -CD8 n= 10 mice). (c)
Quantification of CD8 staining of hepatic tissue by immunohistochemistry of 12 months ND,
CD-HFD or CD-HFD + 8 weeks treatment of α -CD8 fed mice (ND n= 6 mice; CD-HFD n= 6
mice; CD-HFD + α -CD8 n= 5 mice). (d) Absolute and (e) relative quantification of hepatic
leukocytes of 12 months CD-HFD or CD-HFD + 8 weeks treatment of α -CD8 fed mice (CD-
HFD n= 9 mice; CD-HFD + α -CD8 n= 12 mice). (f) Cytokine expression for polarization of
hepatic CD8+ T-cells of 12 months CD-HFD or CD-HFD + 8 weeks treatment of α -CD8 fed
mice (GzmB, IFN γ , TNF: CD-HFD n= 13 mice; α -CD8 + CD-HFD n= 17 mice; IL-10: CD-HFD
n= 7 mice; α -CD8 + CD-HFD n= 9 mice). (g) Expression of PD-1 of hepatic CD4+ and CD8+
T-cells by flow cytometry of 12 months CD-HFD or CD-HFD + 8 weeks treatment of α -CD8 fed
mice (CD-HFD n= 11 mice; α -CD8 + CD-HFD n= 17 mice). (h) Flow cytometry analysis for
polarization of hepatic myeloid cells of 12 months CD-HFD or CD-HFD + 8 weeks treatment of
α -CD8 fed mice (CD-HFD n= 8 mice; α -CD8 + CD-HFD n= 12 mice). (i) Flow cytometric
analysis for polarization of hepatic CD4+ T-cells of 12 months CD-HFD or CD-HFD + 8 weeks
treatment of α -CD8 fed mice (CD-HFD n= 12 mice; α -CD8 + CD-HFD n= 17 mice). (j) Cytokine
expression of hepatic CD4+ T-cells of 12 months CD-HFD or CD-HFD + 8 weeks treatment of
α -CD8 fed mice (GzmB, IFN γ , TNF: CD-HFD n= 13 mice; CD-HFD + α -CD8 n= 17 mice; IL-
10, Foxp3: CD-HFD n= 7 mice; CD-HFD + α -CD8 n= 9 mice). (k) Cytokine expression for
polarization of hepatic NK and NKT-cells of 12 months CD-HFD or CD-HFD + 8 weeks
treatment of α -CD8 fed mice (CD-HFD n= 4 mice; α -CD8 + CD-HFD n= 5 mice). (l) Gene set
enrichment analysis of RNA sequencing data of hepatic tissue comparing CD-HFD with CD-
HFD + α -CD8 of 12 months ND, CD-HFD or CD-HFD + 8 weeks treatment of α -CD8 fed mice
(n= 5 mice/group).

Rebuttal Figure 14

(a) H&E and Sirius Red staining, (b) body weight, (c) NAS evaluation by H&E, (d) fibrosis
 evaluation of Sirius Red and (e) ALT levels of 12 months ND, CD-HFD, CD-HFD + 8 weeks
 treatment of α -CD8 or CD-HFD + 8 weeks co-depletion of α -CD8/NK1.1 (body weight: ND n =
 15 mice; CD-HFD n = 28 mice; CD-HFD + α -CD8 n = 28 mice; fibrosis ND n = 19 mice; CD-HFD
 n = 53 mice; CD-HFD + α -CD8 n = 27 mice; CD-HFD + α -CD8/NK1.1 n = 6 mice; NAS: ND n =
 24 mice; CD-HFD n = 40 mice; CD-HFD + α -CD8 n = 29 mice; CD-HFD + α -CD8/NK1.1 n = 6;
 ALT: ND n = 22 mice; CD-HFD n = 42 mice; CD-HFD + α -CD8 n = 31 mice; CD-HFD + α -
 CD8/NK1.1 n = 6). Scale bar: 100 μ m. (f) Flow cytometry plots and (g) quantification of hepatic
 NK1.1 abundance of 12 months ND, CD-HFD, CD-HFD + 8 weeks treatment of α -CD8 or CD-
 HFD + 8 weeks co-depletion of α -CD8/NK1.1 (ND n = 4 mice; CD-HFD n = 8 mice; CD-HFD +
 α -CD8 n = 7 mice; CD-HFD + α -CD8/NK1.1 n = 6 mice). (h) Gene set enrichment analysis of
 RNA sequencing data of hepatic tissue comparing CD-HFD with CD-HFD + co-depletion of α -
 CD8/NK1.1 of 12 months ND, CD-HFD or CD-HFD + co-depletion of α -CD8/NK1.1 (n = 5
 mice/group). (i) Gene set enrichment analysis of RNA sequencing data of hepatic tissue

comparing or CD-HFD + 8 weeks treatment of α -CD8 fed mice with CD-HFD + co-depletion of
α -CD8/NK1.1 of 12 months ND, CD-HFD, CD-HFD + 8 weeks treatment of α -CD8 fed or CD-
HFD + co-depletion of α -CD8/NK1.1 (n= 5 mice/group)

Rebuttal Figure 15

(a) Body weight, AST, and histological evaluation by (b) Sirius red, CD4, CD8, PD-1, PD-L1,
F4/80, MHC-II and (c) staining of ND, CD-HFD, or CD-HFD-fed mice + 8 weeks treatment by
α -PD-1, α -PD-1/ α -CD8, α -TNF, α -PD-1/ α -TNF antibodies (body weight: ND n= 16 mice; CD-
HFD n= 29 mice; CD-HFD + α -PD-1 n= 23 mice; CD-HFD + α -PD-1/ α -CD8 n= 9 mice; CD-
HFD + α -TNF n= 10 mice; CD-HFD + α -PD-1/ α -TNF n= 11 mice; AST: body weight: ND n= 30
mice; CD-HFD n= 40 mice; CD-HFD + α -PD-1 n= 30 mice; CD-HFD + α -PD-1/ α -CD8 n= 9
mice; CD-HFD + α -TNF n= 10 mice; CD-HFD + α -PD-1/ α -TNF n= 11 mice; Sirius red: ND n=
11 mice; CD-HFD n= 12 mice; CD-HFD + α -PD-1 n= 12 mice; CD-HFD + α -PD-1/ α -CD8 n= 9
mice; CD-HFD + α -TNF n= 10 mice; CD-HFD + α -PD-1/ α -TNF n= 11 mice; CD4: ND n= 10
mice; CD-HFD n= 11 mice; CD-HFD + α -PD-1 n= 14 mice; CD-HFD + α -PD-1/ α -CD8 n= 9
mice; CD-HFD + α -TNF n= 10 mice; CD-HFD + α -PD-1/ α -TNF n= 11 mice; CD8: ND n= 10
mice; CD-HFD n= 12 mice; CD-HFD + α -PD-1 n= 14 mice; CD-HFD + α -PD-1 n= 14 mice; CD-
HFD + α -PD-1/ α -CD8 n= 9 mice; CD-HFD + α -TNF n= 10 mice; CD-HFD + α -PD-1/ α -TNF n=
11 mice; PD-1: ND n= 12 mice; CD-HFD n= 12 mice; CD-HFD + α -PD-1 n= 14 mice; CD-HFD
+ α -PD-1/ α -CD8 n= 8 mice; CD-HFD + α -TNF n= 10 mice; CD-HFD + α -PD-1/ α -TNF n= 10
mice; PD-L1: ND n= 10 mice; CD-HFD n= 11 mice; CD-HFD + α -PD-1 n= 14 mice; CD-HFD +
α -PD-1/ α -CD8 n= 9 mice; CD-HFD + α -TNF n= 10 mice; CD-HFD + α -PD-1/ α -TNF n= 11 mice;
F4/80: ND n= 11 mice; CD-HFD n= 12 mice; CD-HFD + α -PD-1 n= 14 mice; CD-HFD + α -PD-
1 n= 14 mice; CD-HFD + α -PD-1/ α -CD8 n= 9 mice; CD-HFD + α -TNF n= 10 mice; CD-HFD +
α -PD-1/ α -TNF n= 11 mice; MHC-II: ND n= 11 mice; CD-HFD n= 13 mice; CD-HFD + α -PD-1
n= 14 mice; CD-HFD + α -PD-1 n= 14 mice; CD-HFD + α -PD-1/ α -CD8 n= 9 mice; CD-HFD +
α -TNF n= 10 mice; CD-HFD + α -PD-1/ α -TNF n= 11 mice).

Rebuttal Figure 16

(a) Quantification of hepatic immune cell composition and (b) CD8⁺PD-1⁺TNF⁺ T-cells by flow cytometry of 12 months ND, CD-HFD, or CD-HFD-fed mice + 8 weeks treatment by α-PD-1, α-PD-1/α-CD8, α-TNF, α-PD-1/α-TNF antibodies (Hepatic immune cell composition: ND n= 8

mice; CD-HFD n= 5 mice; CD-HFD + α -PD-1 n= 4 mice; CD-HFD + α -PD-1/ α -CD8 n= 9 mice;
 CD-HFD + α -TNF n= 10 mice; CD-HFD + α -PD-1/ α -TNF n= 11 mice; CD8+PD-1+TNF+: ND
 n= 8 mice; CD-HFD n= 5 mice; CD-HFD + α -PD-1 n= 3 mice; CD-HFD + α -PD-1/ α -CD8 n= 9
 mice; CD-HFD + α -TNF n= 10 mice; CD-HFD + α -PD-1/ α -TNF n= 11 mice). (c) and (d)
 multiplex ELISA of hepatic inflammation associated cytokines and (e) chemokines of 12
 554 months ND, CD-HFD or CD-HFD-fed mice + 8 weeks treatment by α -PD-1, α -PD-1/ α -CD8, α -
 555 TNF, α -PD-1/ α -TNF antibodies (ND n= 10 mice; CD-HFD n= 14 mice; CD-HFD + α -PD-1 n=
 13 mice; CD-HFD + α -PD-1/ α -CD8 n= 9 mice; CD-HFD + α -TNF n= 10 mice; CD-HFD + α -PD-
 1/ α -CD8 n= 9 mice; CD-HFD + α -TNF n= 10 mice; CD-HFD + α -PD-
 1/ α -TNF n= 11 mice).

Rebuttal Figure 17

(a) Body weight, ALT, AST, NAS, and histological evaluation by (b) Sirius Red, CD4, CD8, PD-1, PD-L1, F4/80, MHC-II and (c) staining of ND, CD-HFD, or CD-HFD-fed mice + 8 weeks

treatment by α -PD-1, α -CD4, α -PD-1/ α -CD4 antibodies (body weight: ND n= 16 mice; CD-HFD
n= 29 mice; CD-HFD + α -PD-1 n= 23 mice; CD-HFD + α -CD4 n= 9 mice; CD-HFD + α -PD-
1/ α -CD4 n= 9 mice; ALT ND n= 30 mice; CD-HFD n= 47 mice; CD-HFD + α -PD-1 n= 35 mice;
CD-HFD + α -CD4 n= 9 mice; CD-HFD + α -PD-1/ α -CD4 n= 9 mice; AST: ND n= 30 mice; CD-
HFD n= 40 mice; CD-HFD + α -PD-1 n= 30 mice; CD-HFD + α -CD4 n= 9 mice; CD-HFD + α -
PD-1/ α -CD4 n= 9 mice; NAS: ND n= 31 mice; CD-HFD n= 46 mice; CD-HFD + α -PD-1 n= 40
mice; CD-HFD + α -CD4 n= 8 mice; CD-HFD + α -PD-1/ α -CD4 n= 8 mice; Sirius red: ND n= 11
mice; CD-HFD n= 12 mice; CD-HFD + α -PD-1 n= 12 mice; CD-HFD + α -CD4 n= 9 mice; CD-
HFD + α -PD-1/ α -CD4 n= 9 mice; CD4: ND n= 10 mice; CD-HFD n= 11 mice; CD-HFD + α -PD-
1 n= 14 mice; CD-HFD + α -CD4 n= 10 mice; CD-HFD + α -PD-1/ α -CD4 n= 11 mice; CD8: ND
n= 10 mice; CD-HFD n= 12 mice; CD-HFD + α -PD-1 n= 14 mice; CD-HFD + α -CD4 n= 9 mice;
CD-HFD + α -PD-1/ α -CD4 n= 9 mice; PD-1: ND n= 13 mice; CD-HFD n= 12 mice; CD-HFD +
α -PD-1 n= 14 mice; CD-HFD + α -CD4 n= 9 mice; CD-HFD + α -PD-1/ α -CD4 n= 9 mice; PD-L1:
ND n= 12 mice; CD-HFD n= 12 mice; CD-HFD + α -PD-1 n= 14 mice; CD-HFD + α -CD4 n= 9
mice; CD-HFD + α -PD-1/ α -CD4 n= 9 mice; F4/80: ND n= 11 mice; CD-HFD n= 13 mice; CD-
HFD + α -PD-1 n= 14 mice; CD-HFD + α -CD4 n= 8 mice; CD-HFD + α -PD-1/ α -CD4 n= 9 mice;
MHC-II: ND n= 11 mice; CD-HFD n= 13 mice; CD-HFD + α -PD-1 n= 14 mice; CD-HFD + α -
PD-1 n= 14 mice; CD-HFD + α -CD4 n= 9 mice; CD-HFD + α -PD-1/ α -CD4 n= 9 mice). Scale
581 bar: 100 μ m. All data are shown as mean \pm SEM.

Rebuttal Figure 18

(a) Quantification of hepatic immune cell composition and (b) CD8+PD-1+TNF+ T-cells by flow cytometry of 12 months ND, CD-HFD, or CD-HFD-fed mice + 8 weeks treatment by α -PD-1, α -CD4, α -PD-1/ α -CD4 antibodies (Hepatic immune cell composition: ND n= 8 mice; CD-HFD n= 5 mice; CD-HFD + α -PD-1 n= 4 mice; CD-HFD + α -CD4 n= 8 mice; CD-HFD + α -PD-1/ α -CD4 n= 8 mice; CD8+PD-1+TNF+: ND n= 8 mice; CD-HFD n= 5 mice; CD-HFD + α -PD-1 n=

3 mice; CD-HFD + α -CD4 n= 8 mice; CD-HFD + α -PD-1/ α -CD4 n= 8 mice). (c) and (d) multiplex
 ELISA of hepatic inflammation associated cytokines and (e) chemokines of 12 months ND,
 CD-HFD or CD-HFD-fed mice + 8 weeks treatment by α -PD-1, α -CD4, α -PD-1/ α -CD4
 antibodies (ND n= 10 mice; CD-HFD n= 14 mice; CD-HFD + α -PD-1 n= 13 mice; CD-HFD +
 α -CD4 n= 9 mice; CD-HFD + α -PD-1/ α -CD4 n= 9 mice).

Rebuttal Figure 19

(a) scRNA-seq analysis of hepatic TCR β ⁺ cells of 12 months CD-HFD + IgG or CD-HFD-fed mice + 8 weeks treatment by α -PD-1 or α -CD8 antibodies (n= 3 mice/group). (b) Selected
 marker expression in hepatic CD8⁺ T-cells by scRNA-seq comparing CD8⁺ with CD8⁺PD-1⁺
 T-cells of 12 months CD-HFD + IgG or CD-HFD-fed mice + 8 weeks treatment by α -PD-1
 antibodies (n= 3 mice/group). (c) Average UMI comparison of hepatic CD8⁺PD-1⁺ T-cells of
 12 months CD-HFD + IgG or CD-HFD-fed mice + 8 weeks treatment by α -PD-1 antibodies (n=
 3 mice/group). (d) RNA velocity analyses of scRNA-seq data showing expression and (e)
 correlation of expression along the latent-time of selected genes along the latent-time (n= 3
 mice/group). Root cells: yellow cells indicate root cells, blue cells indicate cells farthest away
 from root by RNA velocity. End points: yellow cells indicate end point cells, blue cells indicate

Research for a Life without Cancer

cells farthest away from defined end point cells by RNA velocity. Latent time: pseudo-time by
RNA velocity, dark color indicate start of RNA velocity, yellow color indicate end point of latent
time. RNA velocity flow: Blue cluster defined as start point, orange cluster as intermediate,
green cluster as end point. Arrows indicate trajectory of cells. (f) PCA plot of hepatic CD8+ or
CD8+PD-1+ T-cells sorted TCR β + cells by mass spectrometry of 12 months ND, CD-HFD or
CD-HFD-fed mice + 8 weeks treatment by α -PD-1 antibodies (CD8+: ND n= 6 mice, CD-HFD
+ IgG n= 5 mice; CD-HFD + α -PD-1 n= 6 mice; CD8+PD-1+: ND n= 4 mice, CD-HFD + IgG n=
6 mice; CD-HFD + α -PD-1 n= 6 mice). (g) UMAP representation showing the FlowSOM-guided
clustering, heatmap showing the median marker expression, and (h) quantification of hepatic
CD8+ T-cells of 12 months ND, CD-HFD + IgG or CD-HFD-fed mice + 8 weeks treatment by
α -PD-1 antibodies (ND n= 4 mice; CD-HFD + IgG n= 8 mice; CD-HFD + α -PD-1 n= 6 mice). (i)
Quantification of CellCNN analyzed flow cytometry data of hepatic CD8+ T-cells of 12 months
CD-HFD + IgG or CD-HFD-fed mice + 8 weeks treatment by α -PD-1 antibodies (CD-HFD +
IgG n= 6 mice; CD-HFD + α -PD-1 n= 4 mice). (j) UMAP representation showing the FlowSOM-
guided clustering, the expression intensity of the indicated marker and heatmap showing the
median marker expression of flow cytometry data of hepatic CD8+PD-1+ T-cells of 12 months
ND, CD-HFD or CD-HFD-fed mice + 8 weeks treatment by α -PD-1 antibodies (ND n= 6 mice;
CD-HFD n= 5 mice; CD-HFD + α -PD-1 n= 6 mice).

Rebuttal Figure 20

(a) Histological staining of hepatic tissue by H&E and CD8 of 6 months ND, CD-HFD or PD-1^{-/-} CD-HFD-fed mice (H&E: ND n= 8 mice; PD-1^{-/-} ND n= 5 mice; CD-HFD n= 9 mice; PD-1^{-/-} CD-HFD n= 13 mice; CD8: ND n= 4 mice; CD-HFD n= 5 mice; PD-1^{-/-} CD-HFD n= 7 mice).

Arrowheads indicate CD8+ cells. Scale bar: 50 μ m. (b) Cytokine expression of hepatic CD8+
T-cells of 6 months ND, PD-1-/- ND, CD-HFD or PD-1-/- CD-HFD-fed mice (ND n= 4 mice; PD-
1-/- ND n= 5 mice; CD-HFD n= 5 mice; PD-1-/- CD-HFD n= 6 mice). (c) Tumor/lesion incidence
of 6 months CD-HFD or PD-1-/- CD-HFD-fed mice (tumor incidence: CD-HFD n= 6
tumors/lesions in 63 mice; PD-1-/- CD-HFD n= 6 tumors/lesions in 13 mice). (d) Body weight
of 6 months ND, PD-1-/- ND, CD-HFD or PD-1-/- CD-HFD-fed mice (ND n= 5 mice; PD-1-/-
ND n= 3 mice; CD-HFD n= 5 mice; PD-1-/- CD-HFD n= 10 mice). (e) ALT levels of ND, PD-1-/-
ND, CD-HFD or PD-1-/- CD-HFD (ND n= 9 mice; PD-1-/- ND n= 5 mice; CD-HFD n= 9 mice;
PD-1-/- CD-HFD n= 10 mice). (f) NAS evaluation by H&E of ND, PD-1-/- ND, CD-HFD or PD-
1-/- CD-HFD fed mice (ND n= 8 mice; PD-1-/- ND n= 5 mice; CD-HFD n= 9 mice; PD-1-/- CD-
HFD n= 13 mice). (g) CD8 staining of hepatic tissue by immunohistochemistry of 6 months ND,
PD-1-/- ND, CD-HFD or PD-1-/- CD-HFD fed mice (ND n= 4 mice; PD-1-/- ND n= 5 mice; CD-
HFD n= 5 mice; PD-1-/- CD-HFD n= 7 mice). (h) – (j) Characterization of hepatic T-cells by
flow cytometry of 6 months ND, PD-1-/- ND, CD-HFD or PD-1-/- CD-HFD fed mice (ND n= 4
mice; PD-1-/- ND n= 5 mice; CD-HFD n= 5 mice; PD-1-/- CD-HFD n= 6 mice). (k) Relative
quantification of hepatic leukocytes of 6 months CD-HFD or PD-1-/- CD-HFD fed mice (ND n=
4 mice; PD-1-/- ND n= 5 mice; CD-HFD n= 5 mice; PD-1-/- CD-HFD n= 6 mice). (l) Histological
staining of hepatic tissue by H&E of CD-HFD or PD-1-/- CD-HFD fed mice (ND n= 8 mice; CD-
HFD n= 9 mice; PD-1-/- CD-HFD n= 13 mice). Dotted line indicates tumor/lesion border. Scale
648 bar: 100 μ m.

3. The data on the NASH- and NASH-HCC-promoting role of CD8+ T-cells is similar to a
previous study from the last author (Wolf et al, Cancer Cell). Hence a number of the findings
presented in this manuscript are incremental with, adding PD1 into this context, with somewhat
expected results, as well as novel techniques such as scRNA-seq.

We thank Referee #1 for the opinion on the progress we tried to achieve with this manuscript
as a follow-up study (Wolf et al., 2014). We politely disagree with the statement of Referee #1
– that indicates “...are incremental with, adding PD1 into this context, with somewhat expected
results, as well as novel techniques such as scRNA-seq.”, because:

(i) Our presented data show for the first time that CD8+PD1+ T-cells and their behavior in
the context of immunotherapy and metabolic syndrome affect liver cancer in an unexpected
manner – CD8+PD1+ T cells are pro-tumorigenic in this context – which very likely has clinical
implications.

Identification of increased hepatic abundance of unconventional activated resident-like
CD8+PD-1+ (e.g. CXCR6+, TOX+, TNF+), but not a change of quality in these cells are the
hepatocarcinogenesis-driver in the context of NASH is novel – and can be found also in the
human situation (e.g. two IHC-cohorts across Europe comparing viral vs. NAFLD/NASH-HCC,
one IHC cohort dissecting the abundance of cells depending on NASH pathology severity; also
comparing control vs NAFLD/NASH patient samples by scRNA Seq, CYTOF and flow
cytometry).

(ii) Our data expand current knowledge of NASH pathology-associated mechanisms (e.g.
auto-aggression in a TCR-independent manner with the co-submitted manuscript Dudek et al.,
corroborating the data in total 3x preclinical models of NASH). Furthermore, we tested this
mechanism hypothesis on a functional level by various antibody-based treatments (PD-L1-
targeted immunotherapy; combination therapy of anti-TNF/anti-PD-1, anti-CD4/anti-PD-1, anti-
CD8/anti-PD1) and now identify that it indeed is TNF and CD8 T cells that promote liver cancer
in the context of PD1-related immunotherapy.

(iii) Novel comparison/corroboration and in-depth analysis of T-cell populations in human
and mouse NASH by scRNA, flow cytometry and CYTOF. We did not expect a link between
resident-like CD8+PD1+ cells in the progression of NASH pathology and NASH-induced
hepatocarcinogenesis, as well as the correlation of preclinical model to patient data, identifying
NASH as an etiology of unfavorable predictor of response (e.g. the meta-analysis of 1656
patients corroborates non-viral (NASH-related) HCC compared to viral-HCC as less
responsive to immunotherapy (included in **Figure 6, Extended Data 30-32** and **Rebuttal**
**Figure 1, 2**), as well as our own small retrospective NASH-HCC vs other-etiological-HCC
cohort, which was validated in a second validation cohort of HCC-patients under
immunotherapy (included in **Figure 6** and **Rebuttal Figure 21**).

**Rebuttal Figure 21**

(a) Nonalcoholic fatty liver disease (NAFLD) is associated with a worse outcome in patients
 with hepatocellular carcinoma (HCC) treated with PD-(L)1-targeted immunotherapy. A total of
 130 patients with advanced HCC received PD-(L)1-targeted immunotherapy (Supplementary
 Table 8). Kaplan-Meier curve display overall survival of patients with NAFLD vs. those with
 any other etiology; all 130 patients were included in these survival analyses (NAFLD n=13, any
 other etiology n=117). (b) Validation cohort of patients with HCC treated with PD-(L)1-targeted
 immunotherapy. A total of 1180 patients with advanced HCC received PD-(L)1-targeted
 immunotherapy (Supplementary Table 10). Kaplan-Meier curve display overall survival of
 patients with NAFLD vs. those with any other etiology; all 118 patients were included in these

survival analyses (NAFLD n=11, any other etiology n=107). (c) Multivariate analysis of
prognostic factors in HCC patients treated with anti-PD-(L)1-based immunotherapy.

4. The human data are based on a very small and poorly analyzed cohort of patients with
NASH-associated HCC (n=10-11). While the underlying question is important, pairing data
from this small cohort with the data from the mouse model with its above-described limitations
and confounders may send a wrong and potentially deleterious message to the community,
and much more careful analysis as well as larger cohorts are needed to put the provided
message on a solid scientific foundation: The authors should analyzed outcomes for NASH-
HCC patients with or without cirrhosis to account for the possibility of worsened NASH in
patients without cirrhosis (for which the cohort is much too small).

We thank Referee #1 and fully agree, that the presented retrospective
Nivolumab/Pembrolizumab-treated NAFLD/NASH-associated HCC cohort – although unique
for Europe where treatment is not officially licensed - is too small for subgroup analysis for
patients.

We have taken this point raised utmost seriously. Thus, we have strengthened our hypothesis
of non-viral (NASH-related) HCC being less responsive to immunotherapy by a meta-analysis
including patients of the three most important clinical trials (1656 patients, included in **Figure**
**6, Extended Data 31-33** and **Rebuttal Figure 1, 2**).

Moreover, we have increased the number of patients in our initial clinical cohort from 65 to 130
HCC patients under anti-PD(L)1-targeted immunotherapy and validated our results in a second
cohort of 118 HCC patients under PD(L)1-targeted immunotherapy (included in **Figure 6** and
**Rebuttal Figure 21**).

A disadvantage by nature of a retrospective analysis of cohort across multiple centers is, that
clinical material that would have the potential to characterize in patient subgroups (e.g.
worsened NASH) was not sampled. Furthermore, no paired biopsies or other biological
materials (e.g. blood or serum) before/after immunotherapy were taken in this cohorts for HCC
patients, making characterization of treatment response at the single patient resolution and
thus subgroups impossible in this retrospective cohort. Therefore, we decided to investigate
the outcomes for BCLC-C NAFLD/NASH-HCC vs other-etiological-HCC patients with cirrhosis
and observed, that NAFLD/NASH-HCC have significantly reduced overall survival compared
to other-etiological-HCC in this retrospective study. Of note, multivariate analyses identified
NAFLD/NASH as an independent factor for treatment response (included in **Supplementary**
**Table 9** and **Rebuttal Figure 21**). We validated these results in a second independent cohort
of 118 under PD1-targeted immunotherapy based in North America, which included additional

n= 11 patients with NASH-HCC under immunotherapy, corroborating that NASH/NAFLD is a
negative predictor to immunotherapy (main text).

We toned down the conclusions of our retrospective cohort in the manuscript and would like
to point out, that larger cohorts and prospective clinical trials are of utmost importance for the
scientific community and to investigate the points of Referee #1.

740 A. A cohort of n=10-11 NASH-associated HCC patients is unacceptable. Many of the
741 parameters such as PFS are not significant and it cannot be excluded that inclusion of a larger
number of NASH-HCC patients may change the data significantly.

We agree with Referee #1, however we would like to point out attention, that prominent trends
or effects can also be seen in small retrospective cohorts as well. Although unique for Europe,
where treatment is not officially licensed yet, the complete cohort we have gathered is too small
for subgroup analysis for patients.

We decided to leave out the non-significant data of TTP and PFS in our manuscript. Moreover,
upon recruiting the validation cohort of 118 HCC-patients under immunotherapy we decided
to not show TTP and PFS, but instead the multivariate analysis (included in **Supplemental**
**Table 9**). However, we are in line, that an increased patient cohort allows a more sophisticated
analysis. Thus, as mentioned in the previous comment, we increased our patient cohort (from
65 HCC-patients to 130 HCC-patients) and validated the results in the second cohort of 118
HCC-patients under PD(L)1-targeted immunotherapy. Furthermore, we would like to highlight
the message from the performed meta-analysis of 1656 patients, also pointing towards
identifying NAFLD/NASH as a negative predictor of immunotherapy response in HCC. Still, the
cohorts are small, and thus, we toned down the conclusions drawn from this retrospective
cohort analyses (added in the main text, **Figure 6**).

B. The authors do not answer the question whether the differences in survival are due to failed
checkpoint therapy or due to other differences between the two cohorts. Most likely, the
differences in survival would persist if the authors removed all responders from the “other
etiologies” group. Control groups that did not receive checkpoint inhibitors are missing to
determine if survival is different between NASH and non-NASH HCC in patients who did not
receive checkpoint inhibitors.

We thank Referee #1 for raising this important point of potential differences in survival due to
potential confounders. To address these issues, we have submitted our data to multivariate

analyses, which we included in an updated **Supplementary Table 9**. When we excluded
patients with a complete or partial response from the 112 patients with at least one follow-up
imaging, 86 patients were available for analysis (NAFLD, n=9; other etiologies, n=77). Median
OS was significantly shorter in the NAFLD group (5.4 (95%CI, 1.7-9.1) months vs. 10.3
(95%CI, 8.2-12.4) months; p=0.006), as was median TTP (2.4 (95%CI, 2.1-2.7) months vs. 3.9
(95%CI, 2.5-5.4) months; p=0.008), and median PFS (2.4 (95%CI, 1.9-3.0) months vs. 3.7
(2.3-5.1) months; p=0.035). These data suggest that the improved outcome of non-NAFLD
patients is not only driven by the better response rate observed in these patients. However,
the interpretation of these data due to the size of the underlying cohorts needs to be taken with
caution.

Like mentioned before, we have now included a meta-analysis with appropriate control
cohorts, identifying immunotherapy vs control for viral HCC as favorable treatment (HR(viral)=
0.64), in contrast, non-viral-HCC show less benefit (HR(non-viral)= 0.92). In this meta-analysis
patients with NASH-HCC and Non-NASH HCC who did not receive checkpoint inhibitors are
included as receiving either sorafenib (in RCT of front-line) or placebo (in RCT in second-line).
We thank Referee #1 for pointing out the lack of appropriate control groups (e.g. NASH-HCC
vs. different etiology-induced HCC under Sorafenib/different multi-kinase inhibitors as a
second/third-line therapy). Although of extreme interest for public health and public knowledge,
we described this important issue in our discussion and to the best of our knowledge there are
no NASH-HCC treated cohorts available (apart from, possibly, inside of the big pharma-
industry), which would allow an adequate control arm.

Available cohorts (El-Khoueiry et al., 2017; Finn et al., 2019, 2020) are only differentiating
between viral vs. non-viral etiologies, which combine ASH and NASH-induced HCC.

C. Is there any indication of increase NASH activity in patients receiving Pembro or Nivo?

We thank Referee #1 for this important comment. We have added baseline AST and ALT in
the pre-existing and novel cohorts (included in **Supplementary Table 8**). Like previously
mentioned, the character of the retrospective studies did not allow to obtain paired biopsies
before/after immunotherapy, and bigger cohorts of prospective clinical trials are needed.

D. There is no proper analysis of confounding factors.

We thank Referee #1 for pointing out this lack of analyses in our initial submission. We have
now performed multivariate analyses, which we included in the main text and in an updated
**Supplementary Tables 8 and 9.**

In short: Macrovascular invasion, a negative prognostic factor in HCC, was less frequent in
NAFLD patients (23% vs 49%). NAFLD patients received immunotherapy more often as first-
line therapy (46% vs. 23%), and the proportion of patients receiving the combination of
atezolizumab plus bevacizumab, the only immunotherapy-based treatment that has
succeeded in a phase III trial of advanced-stage HCC so far, was higher in the NAFLD cohort
(23% vs. 5%). Despite these more favorable characteristics, immunotherapy was less effective
in patients with NAFLD, which translated into a worse overall survival (OS) for the NAFLD
cohort: 5.4 (95%CI, 1.8-9.0) months vs. 11.0 (95%CI, 7.5-14.5) months (p=0.023). Adjusting
for other well-known prognostic factors (Child-Pugh class, macrovascular invasion,
extrahepatic metastases, performance status, and alpha-fetoprotein (AFP)), NAFLD remained
independently associated with worse survival (HR 2.6 (95%CI, 1.2-5.6; p=0.017). These data
indicate that PD-1-targeted immunotherapy in HCC patients with concomitant NASH might
lead to unfavorable effects.

E. Another problem is mixing Pembro and Nivo groups. Even though the target is the same,
the authors need to provide subgroup analysis for this and increase the number far beyond
what they have to make any meaningful conclusions in these subgroups.

We thank Referee#1 for this comment. Nivolumab and pembrolizumab are mostly considered
comparable in solid tumors. Performing a subgroup analysis based on Nivolumab and
pembrolizumab is simply not feasible nor realistic in HCC, even more so in NASH-HCC.

We would like to draw attention to other studies performed in solid tumors (NSCLC (Cui et al.,
2020), and Melanoma (Moser et al., 2020)) that show a similar efficacy (although the overall
level of evidence is low):

We agree with this point of Referee #1, which we so far have not been able to make clear.
Similar to the previous point (4A.), our retrospective analyses of the patient cohorts is too small
to address these concerns in an in-depth manner.

We agree with Referee #1, that both Nivolumab and Pembrolizumab are targeting the molecule
PD-1, with similar response rates of 17-20% as monotherapy in HCC (El-Khoueiry et al., 2017;
Zhu et al., 2018). The consensus in the literature is to combine both PD-1 targeting antibodies
and pool their results. Moreover, we validated these results in the second cohort of 118 treated
immunotherapy treated HCC-patients, including n= 11 NASH-HCC patients.

F. Characterization of patients is insufficient - how were other liver diseases excluded,
including ALD, which is not trivial, and especially important in such small cohorts?

We thank Referee #1 for raising this important point and would like to draw the attention, that
criteria for the retrospective patient cohort are described elsewhere (Scheiner et al., 2019).

We have especially analyzed the parameters to identify NAFLD/NASH from viral (e.g. patient
history, liver histology, MRI, obesity). It should be indicated that the differences between NASH
and BASH are indeed difficult to account for – less so when differentiating between NASH and
ASH. Furthermore, we toned down our statement regarding the effects of immunotherapy in
our patient cohorts/case reports in the revised manuscript.

5. Do the authors get the same results when blocking CTLA-4 - which was, even though not
approved for HCC - the first approach and published study to show efficacy of checkpoint
inhibitors in HCC?

We thank Referee #1 for this important question and would like to draw the attention to a phase
II trial combining TACE with Tremelimumab that did not differentiate between underlying
etiology for the patient outcome or immune population (Agdashian et al., 2019; Duffy et al.,
2016). This phase II trial showed a similar response rate (21-26%) compared to the 17-20%
response rate for PD-1 targeted monotherapy (El-Khoueiry et al., 2017; Zhu et al., 2018).
Clinical consensus for immunotherapy indicates increased hepatotoxicity of CTLA-4-
compared to PD-1-targeting immunotherapy (Zen and Yeh, 2018), arguing in favor of PD-
1/PD-L1-targeting immunotherapies for the future.

Although we observed in human Tregs cells CTLA-4 positivity by scRNA-seq and flow
cytometry, in our manuscript CTLA-4 expression was not identified as significantly different
between treatments as shown by scRNA-seq (**Figure 1**: CTLA-4 expression in CD8+ T-cells
comparing ND vs CD-HFD: FC= 0.1894, p= 0.0642; **Extended Data 5**: CTLA-4 expression in
CD4+ T-cells comparing ND vs CD-HFD: FC= 0.2173, p= 0.1431; **Figure 4 and Extended
Data 18**). In our mass spectrometry-based data set, we found no significant change of CTLA-
4 abundance (**Extended Data 5 and 18 and Rebuttal Figure 22**), corroborating our flow
cytometry-based analysis, which had also low CTLA-4 expression in mouse or human
(**Figures 4 and 5, Extended Data 18 and 25 and Rebuttal Figure 22**). Thus, we believe that
the application of CTLA-4-targeted immunotherapy is unlikely to cause a positive effect in our
preclinical model.

We have discussed the potential use of targeting rather T-cell activation (anti-CTLA-4) than
 exhaustion (anti-PD-1 or anti-PD-L1) in combination, or together with a potential generation of
 tumor antigens by ablation strategies (e.g. TACE).

Rebuttal Figure 22

(a) Selected average marker expression in T-cell subsets of CD8+ and (b) CD4+ sorted TCR β +
by scRNA-seq of 12 months ND or CD-HFD-fed mice (n= 3 mice/group). (c) Selected marker
expression in hepatic CD8+ T-cells by scRNA-seq comparing CD8+ with CD8+PD-1+ T-cells
of 12 months CD-HFD + IgG or CD-HFD-fed mice + 8 weeks treatment of α -PD-1 (n= 3
mice/group). (d) Selected marker expression in hepatic CD4+ T-cells by scRNA-seq comparing
CD4+ with CD4+PD-1+ T-cells of 12 months CD-HFD + IgG or CD-HFD-fed mice + 8 weeks
treatment of α -PD-1 fed mice (n= 3 mice/group). (e) Selected marker expression in hepatic
CD8+PD-1+ T-cells by mass- spectrometry of 12 months ND or CD-HFD-fed mice (ND n= 4
mice, CD-HFD n= 6 mice). (f) Selected marker expression in hepatic CD8+PD-1+ T-cells
sorted TCR β + cells by mass- spectrometry of 12 months CD-HFD or CD-HFD-fed + 8 weeks
treatment of α -PD-1 fed mice (n= 6 mice/group). Candidates developing steady in-/decrease
from ND to CD-HFD to CD-HFD-fed mice + 8 weeks treatment of α -PD-1 are indicated in red.
(n= 6 mice/group). (g) Analysis of 5000 randomly chosen TCR β + CD8+ cells of flow cytometry
data to define distinct marker expression of 12 months ND, CD-HFD + IgG, CD-HFD-fed mice
+ 8 weeks treatment of α -PD-1 (ND n= 4 mice; CD-HFD n= 8 mice; CD-HFD + α -PD-1 n= 6
mice). (h) Analysis of CD45+ cells by flow cytometry derived from hepatic biopsies of control
and NAFLD/NASH patients to define distinct marker expression (Supplementary Table 2:
control n= 6 patients; NAFLD/NASH n= 11 patients).

**Referee #2 (Remarks to the Author):**

In their manuscript, Pfister and colleagues aim to show that CD8+PD-1+ T-cells expand during
progressing, diet-induced NAFLD and, upon treatment with anti-PD-1 antibodies, that these
cells can promote carcinogenesis by establishing an inflammatory tumor microenvironment in
a diet-induced, murine model of advanced NAFLD. Additionally, the authors observe a similar,
intratumoral CD8+CD103+PD-1+ T-cell subset in NASH-induced human HCC patients and
claim that patients with NASH-induced HCC respond worse to anti-PD-1 therapy compared to
HCC of other origin.

While the seminal observation in this paper is intriguing, namely that anti-PD-1 treatment can
exacerbate tumorigenesis in a murine model of NASH-induced HCC, the authors fail to
demonstrate clear causal relationships between the implicated cell types, liver inflammation
and tumor development in the vast amount of the data they present, which therefore remain
largely correlative. I will highlight my major concerns below.

We thank Referee #2 for the concise and detailed comments and understanding of our aimed
key points to be delivered in the manuscript. Also, we thank Referee #2 for pointing out the
limitations of our study of correlative data interpretation rather than functional dissection. We
appreciate Referee's #2 opinion, that our human cohort results lead to indications of a worse
response rate of NAFLD/NASH-induced HCC compared to non-NAFLD/NASH-HCC upon PD-
1 targeted immunotherapy. We would like to address the referee's concerns in the following
section point-by-point by new experimental data, rephrasing of the text, and re-analysis of the
underlying as well as novel data-sets.

1. In the reporting summary, the authors state that "Exclusion criteria was pre-established and
the CD-HFD fed mice which did not show the NASH phenotype, high ALT, AST and body
weight, were excluded from the analysis". I fail to understand why this decision was taken as
these mice offer valuable insight in the author's proposed mechanism. Do CD-HFD mice
without overt signs of NASH have reduced CD8+PD-1+ T-cells? Do these mice also less
frequently grow tumors upon anti-PD-1 blockade? Do the T-cells in the livers of these mice fail
display an enhanced effector phenotype? Aside from the valuable experimental insights that
could be gained from these mice, the decision to exclude these CD-HFD but non-NASH mice
from analysis also invalidates any claim that links a given diet to a given phenotype since mice
that did not fit the authors' desired phenotype were excluded.

We thank Referee #2 for the above questions. All mice were included in the respective
treatment – as stated in the paper, indicated by the large mouse data sets in **Figure 1-4** in
NAS, ALT, AST, and body weight. Thus, the statement “Exclusion criteria ...” is inappropriate
and a mistake made on our side and is corrected in an updated Reporting Summary. We fully
agree with Referee #2 that these mice “offer valuable insight in the proposed mechanism” and
this is actually why we have included all of them in our analyses.

To display the experimental range of mice fed 12 months CD-HFD, we have now performed
correlations of a large number of integrated parameters of each mouse (e.g. tumor incidence,
tumor size, tumor nodule number, immune-histochemistry, serology, flow cytometry data;
included now in **Figures 1 and 4, Extended Data 4 and 24 and Rebuttal Figure 23, 24**): In
more detail, we have - for example - re-analyzed our data sets to dissect the potential
correlations of CD8+ T-cells, PD-1+ T-cells, ALT, fibrosis, and NAS, as well as tumor
incidence, tumor nodule size, and effector phenotype – by artificial intelligence and machine
learning clustering. We have now included these analyses in our revised manuscript.

We did not analyze the hepatic environment at time points 10, but after 12 months under diet,
after treatment finished, thus a paired analysis of mice with reduced CD8+PD-1+ T-cells and
their reaction to PD-1-targeted immunotherapy is not possible. In 12 months, CD-HFD-fed mice
CD8 (%CD45) and effector CD8 cells (CD8+CD44+CD62L-) correlate positively with markers
of severity of NASH pathology (e.g. ALT, AST, NAS), as well as tumor incidence (included in
**Extended Data 4 and Rebuttal Figure 23**). In 12 months CD-HFD-fed mice polarization by
PD-1 of these CD8+ T-cells (CD8+PD-1+(%CD8)) correlate positively with ALT, AST, but not
significantly with NAS or tumor incidence, indicating that the hepatic abundance of CD8+PD-
1+ cells is important for NASH (e.g. CD8+PD-1+ (%CD45) correlates (Spearman correlation
$r= 0.3844$, $p= 0.0058$) with NAS, not reported in the paper).

Correlation data included in **Extended Data 24 and Rebuttal Figure 24** shows, that PD-1-
targeted immunotherapy correlates positively with markers of severity of NASH pathology (e.g.
ALT, AST, NAS), with tumor incidence and tumor numbers per liver, and hepatic CD8 T-cells
(e.g. by histology and flow cytometry), effector CD8 cells (CD8+CD44+CD62L-), as well as the
polarization of CD8+PD-1+(%CD8). These data indicate similar to the Referee’s comment,
that mice with reduced hepatic CD8 T-cells and thus also less effector CD8 cells
(CD8+CD44+CD62L-) develop fewer tumors, and that in our data set reduced numbers of
hepatic CD8+PD1+ T-cells result in lower NAS and lower tumor incidence upon PD-1-targeted
immunotherapy (included in **Extended Data 24 and Rebuttal Figure 24**).

We agree with Referee #2, that these data allowed us to gain valuable insights understanding
the phenotype, why some mice develop milder NAFLD/NASH when compared to experimental

controls submitted to similar times of diet feeding, and how this affected PD-1 blockade. We
 would like to point out that mice develop NAFLD/NASH at 12 months post-diet start with an
 incidence of 100% (please also see **Figures 1** and **Rebuttal Figure 25**).

**Rebuttal Figure 23**
 **(a)** UMAP representation of 63 parameters (serology, flow cytometry, histology) indicating
 NASH pathology severity measured of 12 months ND or CD-HFD fed mice (ND n= 22 mice;
 CD-HFD n= 31 mice). **(b)** Data gathered from hepatic tissue analyses was binary correlated
 with each other of 6- or 12-months ND or CD-HFD-fed mice (ND n= 47 mice; CD-HFD n= 72
 mice). **(c)** H&E, CD8, and PD-1 staining, evaluation by NAS and quantification of CD8+ cells
 and PD-1+ expressing cells by immunohistochemistry of 32-weeks old hURI-tetOFFhep and
 non-transgenic litter control mice (n=6 mice/group). Arrowheads indicate specific staining
 positive cells. Scale bar: 100 μm .

Research for a Life without Cancer

**Rebuttal Figure 24**

(a) UMAP representation of 63 parameters (serology, flow cytometry, histology) and (b)
selected display of analyzed parameters indicating NASH pathology severity measured of 12
984 months ND or CD-HFD fed mice (ND n= 22 mice; CD-HFD n= 31 mice; CD-HFD + α-PD-1 n=
41 mice; CD-HFD + α-PD-L1 n= 6 mice; CD-HFD + α-CD8 n= 24 mice; CD-HFD + α-
CD8/NK1.1 n= 6 mice; CD-HFD + α-PD-1/α-CD8 n= 9 mice; CD-HFD + α-TNF n= 10 mice;
CD-HFD + α-PD-1/α-TNF n= 11 mice; CD-HFD + α-CD4 n= 9 mice; CD-HFD + α-PD-1/α-CD4
n= 9 mice). (c) Data gathered from hepatic tissue analyses was binary correlated with each
other of 6- or 12-months ND, CD-HFD or CD-HFD + 8 weeks treatment of α-CD8, α-CD8/
NK1.1; α-PD-1, α-PD-1/α-CD8, α-TNF, α-PD-1/α-TNF, α-CD4, or α-PD-1/α-CD4 fed mice (ND
n= 47 mice; CD-HFD n= 72 mice; CD-HFD + α-PD-1 n= 41 mice; CD-HFD + α-PD-L1 n= 6
mice; CD-HFD + α-CD8 n= 29 mice; CD-HFD + α-CD8/NK1.1 n= 6 mice; CD-HFD + α-PD-1/α-
CD8 n= 9 mice; CD-HFD + α-TNF n= 10 mice; CD-HFD + α-PD-1/α-TNF n= 11 mice; CD-HFD
+ α-CD4 n= 9 mice; CD-HFD + α-PD-1/α-CD4 n= 9 mice).

**Rebuttal Figure 25**

(a) Histological staining of hepatic tissue by H&E of 3, 6 or 12 months ND, CD-HFD or WD-
HTF fed mice (H&E: 3 months: ND n= 5 mice; CD-HFD n= 5 mice; WD-HTF n= 3 mice; 6
1000 months: ND n= 16 mice; CD-HFD n= 8 mice; WD-HTF n= 8 mice; 12 months: ND n= 9 mice;
CD-HFD n= 12 mice; WD-HTF n= 6 mice). Scale bar: 50 μm. (b) Body weight of 3, 6 or 12
1002 months ND, CD-HFD or WD-HTF mice (3 months: ND n= 8 mice; CD-HFD n= 8 mice; WD-
1003 HTF n= 3 mice; 6 months: ND n= 14 mice; CD-HFD n= 8 mice; WD-HTF n= 8 mice; 12 months:
ND n= 8 mice; CD-HFD n= 8 mice; WD-HTF n= 6 mice). (c) ALT levels of 3, 6 or 12 months
ND, CD-HFD or WD-HTF mice (3 months: ND n= 15 mice; CD-HFD n= 46 mice; WD-HTF n=
23 mice; 6 months: ND n= 46 mice; CD-HFD n= 59 mice; WD-HTF n= 21 mice; 12 months:
ND n= 25 mice; CD-HFD n= 69 mice; WD-HTF n= 5 mice). (d) NAS evaluation by H&E of 3, 6
or 12 months ND, CD-HFD or WD-HTF mice (3 months: ND n= 5 mice; CD-HFD n= 5 mice;
WD-HTF n= 3 mice; 6 months: ND n= 16 mice; CD-HFD n= 8 mice; WD-HTF n= 8 mice; 12

1010 months: ND n= 9 mice; CD-HFD n= 12 mice; WD-HTF n= 6 mice). (e) UMAP representation
of 5000 randomly chosen CD45+ cells and quantification of hepatic immune cell composition
by flow cytometry of 12 months ND or CD-HFD fed mice (ND n= 4 mice; CD-HFD n= 8 mice).

2. The data presented by the authors fail to demonstrate clear causal relationships. As an
example, the authors note in lines 341-343 that a pro-inflammatory hepatic environment is
created by TNF upon anti-PD-1 treatment, yet fail to show supporting evidence that this indeed
drives “necro-inflammation” and accelerated hepatocarcinogenesis. The authors should
neutralize TNF in their in vivo models to determine whether this molecule is indeed required
for their phenotype, i.e., inflammatory microenvironment, liver damage and increased
tumorigenicity.

We thank Referee #2 for this very important point. We agree with the comment of Referee #2
and therefore have performed anti-TNF treatment in NASH mice with/without PD-1 targeted
immunotherapy (included in **Figure 4, Extended Data 20 and 21** and **Rebuttal Figure 26-28**).
Of note, data from these experiments demonstrate that TNF, derived from CD8+ T-cells is the
main driver of the pro-tumorigenic effects of T-cells in the context of immunotherapy in NASH
(included in **Figure 3** and **Rebuttal Figure 29**).

Furthermore, we would like to highlight, that our manuscript correlates increased hepatic
abundance of CD8+PD-1+ T-cells upon PD-1-targeted immunotherapy as crucial for driving
hepatocarcinogenesis. Besides, we have now performed additional scRNA-seq and velocity
blot analyses for human patients with NAFLD/NASH or steatosis and compared those with
mouse immune cells. These data demonstrate high similarities between CD8+ PD1+ T-cells
derived from human and mouse NASH livers.

Moreover, we would like to draw the attention of this Referee to the improved cross-referencing
to the co-submitted manuscript Dudek et al., in which the authors also show that TNF is one
key molecule driving increased CD8-dependent hepatic pathogenesis.

**Rebuttal Figure 26**
(a) ALT and (b) NAS evaluation of 12 months ND, CD-HFD, CD-HFD-fed mice + 8 weeks
treatment of alpha-PD-1, alpha-PD-1/alpha-CD8, alpha-TNF, or alpha-PD-1/alpha-TNF (ND n= 30 mice; CD-HFD n= 47
mice; CD-HFD + alpha-PD-1 n= 35 mice; CD-HFD + alpha-PD-1/alpha-CD8 n= 9 mice; CD-HFD + alpha-TNF

n= 10 mice; CD-HFD + α -PD-1/ α -TNF n= 11 mice). (c) Quantification of tumor incidence of 12
1043 months CD-HFD or CD-HFD-fed mice + 8 weeks treatment of α -CD8, α -CD8/NK1.1, α -PD-1,
α -PD-1/ α -CD8, α -TNF, α -PD-1/ α -TNF, α -CD4, or α -PD-1/ α -CD4 (tumor incidence: CD-HFD n=
32 tumors/lesions in 87 mice; CD-HFD + α -CD8 n= 2 tumors/lesions in 31 mice; CD-HFD + α -
CD8/NK1.1 n= 0 tumors/lesions in 6 mice; CD-HFD + α -PD-1 n= 33 tumors/lesions in 44 mice;
CD-HFD + α -PD-1/ α -CD8 n= 2 tumors/lesions in 9 mice; CD-HFD + α -TNF n= 3 tumors/lesions
in 10 mice; CD-HFD + α -PD-1/ α -TNF n= 3 tumors/lesions in 11 mice); CD-HFD + α -CD4 n= 3
tumors/lesions in 9 mice; CD-HFD + α -PD-1/ α -CD4 n= 8 tumors/lesions in 9 mice).

**1053 Rebuttal Figure 27**

(a) Body weight, AST, and histological evaluation by (b) Sirius red, CD4, CD8, PD-1, PD-L1,
F4/80, MHC-II and (c) staining of ND, CD-HFD, or CD-HFD-fed mice + 8 weeks treatment by
α -PD-1, α -PD-1/ α -CD8, α -TNF, α -PD-1/ α -TNF antibodies (body weight: ND n= 16 mice; CD-
HFD n= 29 mice; CD-HFD + α -PD-1 n= 23 mice; CD-HFD + α -PD-1/ α -CD8 n= 9 mice; CD-
HFD + α -TNF n= 10 mice; CD-HFD + α -PD-1/ α -TNF n= 11 mice; AST: body weight: ND n= 30
mice; CD-HFD n= 40 mice; CD-HFD + α -PD-1 n= 30 mice; CD-HFD + α -PD-1/ α -CD8 n= 9
mice; CD-HFD + α -TNF n= 10 mice; CD-HFD + α -PD-1/ α -TNF n= 11 mice; Sirius red: ND n=
11 mice; CD-HFD n= 12 mice; CD-HFD + α -PD-1 n= 12 mice; CD-HFD + α -PD-1/ α -CD8 n= 9
mice; CD-HFD + α -TNF n= 10 mice; CD-HFD + α -PD-1/ α -TNF n= 11 mice; CD4: ND n= 10
mice; CD-HFD n= 11 mice; CD-HFD + α -PD-1 n= 14 mice; CD-HFD + α -PD-1/ α -CD8 n= 9
mice; CD-HFD + α -TNF n= 10 mice; CD-HFD + α -PD-1/ α -TNF n= 11 mice; CD8: ND n= 10
mice; CD-HFD n= 12 mice; CD-HFD + α -PD-1 n= 14 mice; CD-HFD + α -PD-1 n= 14 mice; CD-
HFD + α -PD-1/ α -CD8 n= 9 mice; CD-HFD + α -TNF n= 10 mice; CD-HFD + α -PD-1/ α -TNF n=
11 mice; PD-1: ND n= 12 mice; CD-HFD n= 12 mice; CD-HFD + α -PD-1 n= 14 mice; CD-HFD
+ α -PD-1/ α -CD8 n= 8 mice; CD-HFD + α -TNF n= 10 mice; CD-HFD + α -PD-1/ α -TNF n= 10
mice; PD-L1: ND n= 10 mice; CD-HFD n= 11 mice; CD-HFD + α -PD-1 n= 14 mice; CD-HFD +
α -PD-1/ α -CD8 n= 9 mice; CD-HFD + α -TNF n= 10 mice; CD-HFD + α -PD-1/ α -TNF n= 11 mice;
F4/80: ND n= 11 mice; CD-HFD n= 12 mice; CD-HFD + α -PD-1 n= 14 mice; CD-HFD + α -PD-
1 n= 14 mice; CD-HFD + α -PD-1/ α -CD8 n= 9 mice; CD-HFD + α -TNF n= 10 mice; CD-HFD +
α -PD-1/ α -TNF n= 11 mice; MHC-II: ND n= 11 mice; CD-HFD n= 13 mice; CD-HFD + α -PD-1
n= 14 mice; CD-HFD + α -PD-1 n= 14 mice; CD-HFD + α -PD-1/ α -CD8 n= 9 mice; CD-HFD +
α -TNF n= 10 mice; CD-HFD + α -PD-1/ α -TNF n= 11 mice). Scale bar: 100 μ m.

Rebuttal Figure 28

(a) Quantification of hepatic immune cell composition and (b) CD8⁺PD-1⁺TNF⁺ T-cells by flow cytometry of 12 months ND, CD-HFD, or CD-HFD-fed mice + 8 weeks treatment by α-PD-1, α-PD-1/α-CD8, α-TNF, α-PD-1/α-TNF antibodies (Hepatic immune cell composition: ND n= 8 mice; CD-HFD n= 5 mice; CD-HFD + α-PD-1 n= 4 mice; CD-HFD + α-PD-1/α-CD8 n= 9 mice;

CD-HFD + α -TNF $n=10$ mice; CD-HFD + α -PD-1/ α -TNF $n=11$ mice; CD8+PD-1+TNF+: ND
$n=8$ mice; CD-HFD $n=5$ mice; CD-HFD + α -PD-1 $n=3$ mice; CD-HFD + α -PD-1/ α -CD8 $n=9$
mice; CD-HFD + α -TNF $n=10$ mice; CD-HFD + α -PD-1/ α -TNF $n=11$ mice). (c) and (d)
multiplex ELISA of hepatic inflammation associated cytokines and (e) chemokines of 12
1086 months ND, CD-HFD or CD-HFD-fed mice + 8 weeks treatment by α -PD-1, α -PD-1/ α -CD8, α -
1087 TNF, α -PD-1/ α -TNF antibodies (ND $n=10$ mice; CD-HFD $n=14$ mice; CD-HFD + α -PD-1 $n=$
13 mice; CD-HFD + α -PD-1/ α -CD8 $n=9$ mice; CD-HFD + α -TNF $n=10$ mice; CD-HFD + α -PD-
1/ α -TNF $n=11$ mice).

**Rebuttal Figure 29**

(a) Quantification of RNA in situ hybridization for hepatic TNF+ cells of 12 months ND, CD-
HFD or CD-HFD-fed mice + 8 weeks treatment of α -CD8 or α -PD-1 (ND $n=25$ FOV in 3 mice;
CD-HFD $n=27$ FOV in 3 mice; CD-HFD + α -PD-1 $n=40$ FOV in 3 mice; CD-HFD + α -CD8 $n=$
55 FOV in 3 mice). Arrowheads indicate TNF+ cells. Scale bar: 20 μ m.

3. Based on the authors' presented data, this problem can be further expanded. In Figure S9d
and S9m, the authors show an increase in the number of antigen-presenting cells and
increased MHC-II expression. Are these recruited upon liver inflammation? Are they required
for liver inflammation?

We thank Referee #2 for raising the point about myeloid cells in the context of chronic
inflammation and would like to interpret the data shown in **Extended Data 11** and **Rebuttal**
**Figure 30** in comparison to **Extended Data 8** and **Rebuttal Figure 31**, which now indicates,
that antigen-presenting cells and increased MHC-II expression are a result of increased liver
inflammation upon PD-1 targeted immunotherapy.

We would like to highlight our previous study (Malehmir et al., 2019), which demonstrated, that
myeloid cells are correlated with liver inflammation and are recruited as a consequence of
NASH development. Moreover, we have shown by depletion of antigen-presenting cells,
including Kupffer cells (by chlodronate encapsulating liposomes) abrogates or prevents NASH
development.

To address the point raised by Referee #2 more experimentally, we analyzed our mouse
cohorts in total by AI, which indicates that hepatic MHCII+ cells correlate positively with NASH
pathology (weight, NAS, ALT, AST, cholesterol, fibrosis by Sirius Red staining, hepatic

concentrations of MCP-1, CCL3, MIP-2, and IL-21) and MHCII+ as a marker of myeloid
activation on different subsets correlated predominantly in CD11b+CD11c+ (myeloid dendritic
cells (CD11b+CD11c+) with ALT, GOT, NAS in 12 months CD-HFD-fed mice (included in
**Extended Data 4** and **Rebuttal Figure 23**). To dissect the Referees question in our
experimental functional antibody-treatment experiments (included in **Extended Data 24** and
**Rebuttal Figure 24**). MHCII+ cells correlate positively with CD-HFD and CD-HFD+PD-1-
targeted immunotherapy, as well as NASH pathology (weight, NAS, ALT, AST, cholesterol,
fibrosis by Sirius Red staining, hepatic concentrations of MCP-1, CCL3, CCL4, MIP-2, and IL-
21) in 12 months old mice. Moreover, MHCII+ as a marker of myeloid activation on different
subsets correlated for CD11b+MHCII+ and mDC+MHCII+ positive with PD-1-targeted
immunotherapy, ALT, AST, NAS CCL4, and MIP-2. pDC+MHCII+ and KC+MHCII+ cells
correlated negatively in CD8-depleted and CD8+NK1.1 co-depleted animals. The latter
myeloid subset correlates positively with fibrosis and tumor incidence when pooling the data
of all treatments.

We would like to highlight our previous study (Malehmir et al., 2019), which showed, that
myeloid cells are correlated with liver inflammation and are recruited as a consequence of
NASH development. However, a genetic study using CCR2-/- mice (impaired myeloid
recruitment upon inflammation) developed NASH and NASH-induced tumors; in contrast,
Rag1-/- mice with functional myeloid but impaired adaptive immune compartments were
protected from NASH and NASH-induced tumors (Wolf et al., 2014). These data argue, that
myeloid cells are recruited to the liver, extend, and fine-tune liver inflammation.

Rebuttal Figure 30

(a) Body weight of 12 months ND, CD-HFD or CD-HFD-fed mice + 8 weeks treatment by α -PD-1 antibodies (ND n= 15 mice; CD-HFD n= 28 mice; CD-HFD + α -PD-1 n= 26 mice).

(b) Assessment of metabolic tolerance by intra peritoneal glucose tolerance test of 12 months CD-HFD or CD-HFD-fed mice + 8 weeks treatment by α -PD-1 antibodies (n= 9 mice/group).

(c) Expression of PD-1 of hepatic CD4⁺ and PD-1⁺ T-cells by flow cytometry of 12 months CD-HFD or CD-HFD-fed mice + 8 weeks treatment by α -PD-1 antibodies (CD-HFD n= 10 mice; α -PD-1 + CD-HFD n= 13 mice).

(d) Absolute and (e) relative quantification of hepatic leukocytes

(f) Absolute and (g) relative quantification of CD8⁺ T-cells in CD62L⁺, CD62L⁺CD44⁺CD69⁺, and CD62L⁺CD44⁺ populations.

(h) Absolute and (i) relative quantification of CD4⁺ T-cells in CD62L⁺, CD62L⁺CD44⁺CD69⁺, and CD62L⁺CD44⁺ populations.

(j) Tim-3⁺ cells (% of parent) in CD4⁺ and CD8⁺ populations.

of 12 months CD-HFD or CD-HFD-fed mice + 8 weeks treatment by α -PD-1 antibodies (CD3:
CD-HFD n= 6 mice; CD-HFD + α -PD-1 n= 10 mice; CD4, CD8, CD19, NK, NKT, CD11b+,
mDC, pDC: CD-HFD n= 10 mice; CD-HFD + α -PD-1 n= 12 mice, KC: CD-HFD n= 6 mice; CD-
HFD + α -PD-1 n= 4 mice). (f) Flow cytometric analysis for polarization of hepatic CD8+ T-cells
of 12 months CD-HFD or CD-HFD-fed mice + 8 weeks treatment by α -PD-1 antibodies (CD-
HFD n= 10 mice; α -PD-1 + CD-HFD n= 14 mice). (g) Cytokine expression of hepatic CD4+ T-
cells of 12 months CD-HFD or CD-HFD-fed mice + 8 weeks treatment by α -PD-1 antibodies
(CD-HFD n= 13 mice; CD-HFD + α -PD-1 n= 14 mice). (h) Flow cytometry analysis for
polarization of hepatic CD4+ T-cells of 12 months CD-HFD or CD-HFD-fed mice + 8 weeks
treatment by α -PD-1 antibodies (CD-HFD n= 12 mice; α -PD-1 + CD-HFD n= 17 mice). (i)
Cytokine expression of hepatic CD4+ T-cells of 12 months CD-HFD or CD-HFD-fed mice + 8
1156 weeks treatment by α -PD-1 antibodies (GzmB, IFN γ , TNF: CD-HFD n= 13 mice; CD-HFD + α -
1157 PD-1 n= 14 mice; IL-10, Foxp3: CD-HFD n= 7 mice; CD-HFD + α -PD-1 n= 9 mice). (j)
Expression of Tim-3 of hepatic CD4+ and CD8+ T-cells by flow cytometry of 12 months CD-
HFD or CD-HFD-fed mice + 8 weeks treatment by α -PD-1 antibodies (CD-HFD n= 4 mice; α -
PD-1 + CD-HFD n= 9 mice). (k) Cytokine expression for polarization of hepatic NK and (l) NKT-
cells of 12 months CD-HFD or CD-HFD-fed mice + 8 weeks treatment by α -PD-1 antibodies
(n= 5 mice/group). (m) Flow cytometric analysis for polarization of hepatic myeloid cells of 12
1163 months CD-HFD or CD-HFD-fed mice + 8 weeks treatment by α -PD-1 antibodies (CD-HFD n=
8 mice; α -PD-1 + CD-HFD n= 12 mice).

**1167 Rebuttal Figure 31**

(a) Body weight of 12 months ND, CD-HFD or CD-HFD-fed mice + 8 weeks treatment by α -
CD8 antibodies (ND n= 15 mice; CD-HFD n= 28 mice; CD-HFD + α -CD8 n= 28 mice). (b)
Assessment of metabolic tolerance by intra peritoneal glucose tolerance test of 12 months CD-
HFD or CD-HFD-fed mice + 8 weeks treatment by α -CD8 antibodies (CD-HFD n= 8 mice; CD-
HFD + α -CD8 n= 10 mice). (c) Quantification of CD8 staining of hepatic tissue by
immunohistochemistry of 12 months ND, CD-HFD or CD-HFD-fed mice + 8 weeks treatment
by α -CD8 antibodies (ND n= 6 mice; CD-HFD n= 6 mice; CD-HFD + α -CD8 n= 5 mice). (d)
Absolute and (e) relative quantification of hepatic leukocytes of 12 months CD-HFD or CD-
HFD-fed mice + 8 weeks treatment by α -CD8 antibodies (CD-HFD n= 9 mice; CD-HFD + α -
CD8 n= 12 mice). (f) Analyses of cytokine expression for polarization of hepatic CD8+ T-cells
of 12 months CD-HFD or CD-HFD-fed mice + 8 weeks treatment by α -CD8 antibodies (GzmB,
IFN γ , TNF: CD-HFD n= 13 mice; α -CD8 + CD-HFD n= 17 mice; IL-10: CD-HFD n= 7 mice; α -
CD8 + CD-HFD n= 9 mice). (g) Expression of PD-1 of hepatic CD4+ and CD8+ T-cells by flow
cytometry of 12 months CD-HFD or CD-HFD-fed mice + 8 weeks treatment by α -CD8
antibodies (CD-HFD n= 11 mice; α -CD8 + CD-HFD n= 17 mice). (h) Flow cytometry analysis
for polarization of hepatic myeloid cells of 12 months CD-HFD or CD-HFD-fed mice + 8 weeks
treatment by α -CD8 antibodies (CD-HFD n= 8 mice; α -CD8 + CD-HFD n= 12 mice). (i) Flow
cytometric analysis for polarization of hepatic CD4+ T-cells of 12 months CD-HFD or CD-HFD-
fed mice + 8 weeks treatment by α -CD8 antibodies (CD-HFD n= 12 mice; α -CD8 + CD-HFD
n= 17 mice). (j) Cytokine expression of hepatic CD4+ T-cells of 12 months CD-HFD or CD-
HFD-fed mice + 8 weeks treatment by α -CD8 antibodies (GzmB, IFN γ , TNF: CD-HFD n= 13
mice; CD-HFD + α -CD8 n= 17 mice; IL-10, Foxp3: CD-HFD n= 7 mice; CD-HFD + α -CD8 n=
9 mice). (k) Cytokine expression for polarization of hepatic NK and NKT-cells of 12 months
CD-HFD or CD-HFD-fed mice + 8 weeks treatment by α -CD8 antibodies (CD-HFD n= 4 mice;
α -CD8 + CD-HFD n= 5 mice). (l) Gene set enrichment analysis of RNA sequencing data of
hepatic tissue comparing CD-HFD with CD-HFD-fed mice + α -CD8 of 12 months ND, CD-HFD
or CD-HFD-fed mice + 8 weeks treatment by α -CD8 antibodies (n= 5 mice/group).

4. In Figure S11 the authors show an increase in many inflammatory mediators upon anti-PD-
1 therapy; which of these are required for the accelerated carcinogenesis? While the authors
propose a mechanism based on liver inflammation leading to increased hepatocarcinogenesis
upon anti-PD-1 blockade, they provide little if any conclusive evidence for this hypothesis.

We thank Referee #2 for asking this important question. We believe that the inflammatory
mediators for increased hepatocarcinogenesis stem from the increase of CD8+ T-cells upon
anti-PD1 immunotherapy. Importantly, by performing depletion experiments of different T-cell
subsets – anti-CD8 or anti-CD4, we can demonstrate that the CD8+ T-cells but not CD4+ T-
cells are needed for driving hepatocarcinogenesis and driving the pro-tumorigenic effect of
anti-PD1-related immunotherapy (included in **Figure 4**, **Extended Data 20-23** and **Rebuttal**
**Figure 32-34**).

Of note, PD-1-targeted immunotherapy increases the hepatic abundance of CD8+PD1+ T-
cells in vivo (included in e.g. **Extended Data 11** and **Rebuttal Figure 35a, b**), as well as
increases the number of CD8+PD1+ cells in vitro (included **Extended Data 18** and **Rebuttal**

**Figure 35c**). To understand the nuances of the observed necro-inflammation, anti-PD1-related
immunotherapy, and liver cancer formation, we perform correlations analysis of fibrosis, tumor
nodule number, tumor size, ALT, NAS, CD8, and PD-1 expression by machine learning and
neuronal networking (included in **Figures 1 and 4, Extended Data 4 and 24 and Rebuttal**
**Figure 23, 24**).

We have analyzed the inflammatory environment looking into a specific signature (ICF) on the
transcriptional level in NASH mice with and without anti-PD1-related immunotherapy (included
in **Figure 3 and Rebuttal Figure 35d**). This transcriptional ICF signature is a predictor of liver
cancer formation triggered through inflammation in humans. It can be stated that the altered
inflammatory signature of NASH livers in the context of anti-PD1-related immunotherapy
overlaps with a signature that from human patients is known to have a bad prognosis and high
correlation with inflammation triggered liver cancer. Importantly, upon CD8+ T cell depletion
the intrahepatic ICF signature is downregulated – demonstrating that CD8+ T cell-derived
inflammatory mediators might be linked with liver cancer formation.

Moreover, to identify factors secreted in relation to CD8+ T-cells in NASH livers (as identified
by their reduction upon anti-CD8 treatment) we have performed *in situ* RNA hybridization
analyses for several cytokines. Further, we have performed flow cytometry and RNA-seq of
hepatic tissues as well as scRNA-seq from human and mouse immune cells. Doing so, we
have identified T-cell derived TNF as a possible, important candidate for increased
hepatocarcinogenesis upon PD1-targeted immunotherapy.

To test this hypothesis on a functional level, we performed an anti-PD1/anti-TNF as well as an
anti-TNF treatment alone. These experiments demonstrate that TNF is a functionally important
cytokine contributing to the anti-PD1 antibody treatment mediated pro-carcinogenic effect.

Besides, we would like to draw attention to the improved cross-referencing to the co-submitted
manuscript Dudek et al., which shows that TNF and IL-15, a target downstream of IL-21 - both
upregulated upon anti-PD-1 therapy - are crucial mediators of CD8-mediated hepatic cell
death.

In line, literature highlight the crucial role of TNF for hepatocarcinogenesis (Nakagawa et al.,
2014; Park et al., 2011; Pikarsky et al., 2004) and that anti-TNF treatment uncouples the
toxicity of CTLA-4/PD-1-targeted immunotherapy (Perez-Ruiz et al., 2019).

12 months CD-HFD + IgG or CD-HFD-fed mice + 8 weeks treatment by α -PD-1 antibodies (n= 3 mice/group). (d) RNA velocity analyses of scRNA-seq data showing expression and (e) 1250 correlation of expression along the latent-time of selected genes along the latent-time (n= 3 1251 mice/group). Root cells: yellow cells indicate root cells, blue cells indicate cells farthest away 1252 from root by RNA velocity. End points: yellow cells indicate end point cells, blue cells indicate 1253 cells farthest away from defined end point cells by RNA velocity. Latent time: pseudo-time by 1254 RNA velocity, dark color indicate start of RNA velocity, yellow color indicate end point of latent 1255 time. RNA velocity flow: Blue cluster defined as start point, orange cluster as intermediate, 1256 green cluster as end point. Arrows indicate trajectory of cells. (f) PCA plot of hepatic CD8+ or 1257 CD8+PD-1+ T-cells sorted TCR β + cells by mass spectrometry of 12 months ND, CD-HFD or 1258 CD-HFD-fed mice + 8 weeks treatment by α -PD-1 antibodies (CD8+: ND n= 6 mice, CD-HFD 1259 + IgG n= 5 mice; CD-HFD + α -PD-1 n= 6 mice; CD8+PD-1+: ND n= 4 mice, CD-HFD + IgG n= 1260 6 mice; CD-HFD + α -PD-1 n= 6 mice). (g) UMAP representation showing the FlowSOM-guided 1261 clustering, heatmap showing the median marker expression, and (h) quantification of hepatic 1262 CD8+ T-cells of 12 months ND, CD-HFD + IgG or CD-HFD-fed mice + 8 weeks treatment by 1263 α -PD-1 antibodies (ND n= 4 mice; CD-HFD + IgG n= 8 mice; CD-HFD + α -PD-1 n= 6 mice). (i) 1264 Quantification of CellCNN analyzed flow cytometry data of hepatic CD8+ T-cells of 12 months 1265 CD-HFD + IgG or CD-HFD-fed mice + 8 weeks treatment by α -PD-1 antibodies (CD-HFD + 1266 IgG n= 6 mice; CD-HFD + α -PD-1 n= 4 mice). (j) UMAP representation showing the FlowSOM- 1267 guided clustering, the expression intensity of the indicated marker and heatmap showing the 1268 median marker expression of flow cytometry data of hepatic CD8+PD-1+ T-cells of 12 months 1269 ND, CD-HFD or CD-HFD-fed mice + 8 weeks treatment by α -PD-1 antibodies (ND n= 6 mice; 1270 CD-HFD n= 5 mice; CD-HFD + α -PD-1 n= 6 mice). (k) ALT and (l) NAS evaluation of 12 months 1271 ND, CD-HFD, CD-HFD-fed mice + 8 weeks treatment by α -PD-1, α -PD-1/ α -CD8, α -TNF, or α - 1272 PD-1/ α -TNF antibodies (ND n= 30 mice; CD-HFD n= 47 mice; CD-HFD + α -PD-1 n= 35 mice; 1273 CD-HFD + α -PD-1/ α -CD8 n= 9 mice; CD-HFD + α -TNF n= 10 mice; CD-HFD + α -PD-1/ α -TNF 1274 n= 11 mice). (m) Quantification of hepatic CD8+PD-1+CXCR6+ T-cells ND, CD-HFD, CD- 1275 HFD-fed mice + 8 weeks treatment by α -PD-1, α -PD-1/ α -CD8, α -TNF, α -PD-1/ α -TNF, α -CD4, 1276 or α -PD-1/ α -CD4 antibodies (ND n= 30 mice; CD-HFD n= 47 mice; CD-HFD + α -PD-1 n= 35 1277 mice; CD-HFD + α -PD-1/ α -CD8 n= 9 mice; CD-HFD + α -TNF n= 10 mice; CD-HFD + α -PD- 1278 1/ α -TNF n= 11 mice); CD-HFD + α -CD4 n= 8 mice; CD-HFD + α -PD-1/ α -CD4 n= 8 mice). (n) 1279 Quantification of tumor incidence of 12 months CD-HFD or CD-HFD-fed mice + 8 weeks 1280 treatment by α -CD8, α -CD8/NK1.1, α -PD-1, α -PD-1/ α -CD8, α -TNF, α -PD-1/ α -TNF, α -CD4, or 1281 α -PD-1/ α -CD4 antibodies (tumor incidence: CD-HFD n= 32 tumors/lesions in 87 mice; CD- 1282 HFD + α -CD8 n= 2 tumors/lesions in 31 mice; CD-HFD + α -CD8/NK1.1 n= 0 tumors/lesions in 1283 6 mice; CD-HFD + α -PD-1 n= 33 tumors/lesions in 44 mice; CD-HFD + α -PD-1/ α -CD8 n= 2 1284 tumors/lesions in 9 mice; CD-HFD + α -TNF n= 3 tumors/lesions in 10 mice; CD-HFD + α -PD- 1285 1/ α -TNF n= 3 tumors/lesions in 11 mice); CD-HFD + α -CD4 n= 3 tumors/lesions in 9 mice; CD- 1286 HFD + α -PD-1/ α -CD4 n= 8 tumors/lesions in 9 mice). 1287

Rebuttal Figure 33

(a) Body weight, ALT, AST, NAS, and histological evaluation by (b) Sirius Red, CD4, CD8, PD-1, PD-L1, F4/80, MHC-II and (c) staining of ND, CD-HFD, or CD-HFD-fed mice + 8 weeks treatment by α-PD-1, α-CD4, α-PD-1/α-CD4 antibodies (body weight: ND n= 16 mice; CD-HFD n= 29 mice; CD-HFD + α-PD-1 n= 23 mice; CD-HFD + α-CD4 n= 9 mice; CD-HFD + α-PD-1/α-CD4 n= 9 mice; ALT ND n= 30 mice; CD-HFD n= 47 mice; CD-HFD + α-PD-1 n= 35 mice; CD-HFD + α-CD4 n= 9 mice; CD-HFD + α-PD-1/α-CD4 n= 9 mice; AST: ND n= 30 mice; CD-HFD n= 40 mice; CD-HFD + α-PD-1 n= 30 mice; CD-HFD + α-CD4 n= 9 mice; CD-HFD + α-PD-1/α-CD4 n= 9 mice; NAS: ND n= 31 mice; CD-HFD n= 46 mice; CD-HFD + α-PD-1 n= 40 mice; CD-HFD + α-CD4 n= 8 mice; CD-HFD + α-PD-1/α-CD4 n= 8 mice; Sirius red: ND n= 11 mice; CD-HFD n= 12 mice; CD-HFD + α-PD-1 n= 12 mice; CD-HFD + α-CD4 n= 9 mice; CD-HFD + α-PD-1/α-CD4 n= 9 mice; CD4: ND n= 10 mice; CD-HFD n= 11 mice; CD-HFD + α-PD-1 n= 14 mice; CD-HFD + α-CD4 n= 10 mice; CD-HFD + α-PD-1/α-CD4 n= 11 mice; CD8: ND n= 10

Research for a Life without Cancer

mice; CD-HFD n= 12 mice; CD-HFD + α -PD-1 n= 14 mice; CD-HFD + α -CD4 n= 9 mice; CD-
HFD + α -PD-1/ α -CD4 n= 9 mice; PD-1: ND n= 13 mice; CD-HFD n= 12 mice; CD-HFD + α -
PD-1 n= 14 mice; CD-HFD + α -CD4 n= 9 mice; CD-HFD + α -PD-1/ α -CD4 n= 9 mice; PD-L1:
ND n= 12 mice; CD-HFD n= 12 mice; CD-HFD + α -PD-1 n= 14 mice; CD-HFD + α -CD4 n= 9
mice; CD-HFD + α -PD-1/ α -CD4 n= 9 mice; F4/80: ND n= 11 mice; CD-HFD n= 13 mice; CD-
HFD + α -PD-1 n= 14 mice; CD-HFD + α -CD4 n= 8 mice; CD-HFD + α -PD-1/ α -CD4 n= 9 mice;
MHC-II: ND n= 11 mice; CD-HFD n= 13 mice; CD-HFD + α -PD-1 n= 14 mice; CD-HFD + α -
PD-1 n= 14 mice; CD-HFD + α -CD4 n= 9 mice; CD-HFD + α -PD-1/ α -CD4 n= 9 mice). Scale
1310 bar: 100 μ m.

Rebuttal Figure 34

(a) Quantification of hepatic immune cell composition and (b) CD8+PD-1+TNF+ T-cells by flow cytometry of 12 months ND, CD-HFD, or CD-HFD-fed mice + 8 weeks treatment by α-PD-1, α-CD4, α-PD-1/α-CD4 antibodies (Hepatic immune cell composition: ND n= 8 mice; CD-HFD n= 5 mice; CD-HFD + α-PD-1 n= 4 mice; CD-HFD + α-CD4 n= 8 mice; CD-HFD + α-PD-1/α-CD4 n= 8 mice; CD8+PD-1+TNF+: ND n= 8 mice; CD-HFD n= 5 mice; CD-HFD + α-PD-1 n=

3 mice; CD-HFD + α -CD4 n= 8 mice; CD-HFD + α -PD-1/ α -CD4 n= 8 mice). (c) and (d) multiplex
 ELISA of hepatic inflammation associated cytokines and (e) chemokines of 12 months ND,
 CD-HFD or CD-HFD-fed mice + 8 weeks treatment by α -PD-1, α -CD4, α -PD-1/ α -CD4
 antibodies (ND n= 10 mice; CD-HFD n= 14 mice; CD-HFD + α -PD-1 n= 13 mice; CD-HFD +
 α -CD4 n= 9 mice; CD-HFD + α -PD-1/ α -CD4 n= 9 mice).

Rebuttal Figure 35

(a) Absolute and (b) relative quantification of hepatic leukocytes of 12 months CD-HFD or CD-
 HFD-fed mice + 8 weeks treatment of α -PD-1 (CD3: CD-HFD n= 6 mice; CD-HFD + α -PD-1
 n= 10 mice; CD4, CD8, CD19, NK, NKT, CD11b+, mDC, pDC: CD-HFD n= 10 mice; CD-HFD
 + α -PD-1 n= 12 mice, KC: CD-HFD n= 6 mice; CD-HFD + α -PD-1 n= 4 mice). (c) In vitro
 stimulated splenic CD8 T cells from C57Bl/6 mice were treated with α -PD-1 antibody for 72
 1330 hours (cell count: n= 5 experiments/group; Ki-67: n= 4 experiments/group). (d) Immune-related
 gene expression patterns of RNA sequencing data of hepatic tissue of 12 months ND, CD-
 HFD or CD-HFD—fed mice + 8 weeks treatment of α -PD-1 or α -CD8 (ND, CD-HFD + α -PD-1,
 CD-HFD + α -CD8 n= 5 mice/group; CD-HFD n= 4 mice).

5. Some of the data the authors present seems internally inconsistent. As an example, the
authors postulate that the pro-inflammatory hepatic environment is responsible for the increase
in liver cancer incidence in anti-PD-1-treated mice, which they underscore by an increase in
inflammatory cytokines in the liver microenvironment (Figure S11). However, they also show
that upon CD8 depletion, which reduces cancer incidence, the inflammatory cytokines do not
significantly reduce compared to the CD-HFD diet mice alone. This implies that the
inflammatory microenvironment is not actually responsible for increased cancer incidence.
How do the authors harmonize these findings?

We thank Referee #2 for his comment on the bivalence of cellular and micro-environmental
induced cell death, inflammation, and liver cancer formation. However, we firmly state, that our
data is not internally inconsistent, and have added several experiments that clarify the
mechanisms of action.

We state, that anti-PD-1 therapy induces an increased hepatic inflammatory
microenvironment, indicated by a) increased abundance of hepatic immune cells (mainly CD8+
and CD8+PD-1+ cells) (included in **Figure 2 and Extended Data 11 and Rebuttal Figure 30,**
**36**); b) by increased inflammation-associated cytokines (e.g. IFN γ , TNF, IL-21, IP10, MCP-1,
CCL3) (included in **Extended Data 13 and Rebuttal Figure 37**); c) on mRNA expression levels
we actually clearly see the increase in all pathways relevant for inflammation induced liver
cancer – as analyzed by the ICF-signature (included in **Figure 3 and Rebuttal Figure 35d**).
Thus, we think, that there are 2 components (first cells, like CD8+ T-cells and second, the
inflammatory liver environment) responsible for (increased) liver cancer incidence.

We agree with Referee #2 that initially this appears not logic – but we believe that a liver tissue
homogenate analysis cannot uncover the CD8+-T cell restricted cytokine changes, as other
immune cells will still produce inflammatory immune cells. This is indicated for example in
**Figure 3 and Rebuttal Figure 29**, which shows, that upon CD8 depletion TNF+ cells are
significantly reduced by *in situ* hybridization. Again, effects of the CD8 depletion manifests
strongly on mRNA expression level as pathways relevant for inflammation induced liver cancer
are strongly reduced– as analyzed by the ICF-signature (included in **Figure 3 and Rebuttal**
**Figure 35d**).

Moreover, as stated by the Referee it appears that anti-CD8 treatment alone did not, but anti-
CD8/anti-PD-1 did reduce several chemokines indicative of a hepatic inflammatory
environment on protein level, that are responsible for myeloid cell attraction like MCP-1, CCL2,

CCL3, MIP-3a, or alarmins like IL-33 (included in **Extended Data 10+21** and **Rebuttal Figure**
**28c-d, 31**).

Moreover, we want to point out that our data are also confirmed by the co-submitted manuscript
Dudek et al., revealing that the mechanisms of CD8+ T-cell mediated cell death is 1) CD8+ T-
cell dependent, 2) TCR independent, and 3) TNF is a crucial cytokine sensitizing the CD8+ T-
cell to get auto-aggressive and thus starts to mediate cell death.

We demonstrate that TNF is a marker of a pro-inflammatory, pro-carcinogenic hepatic
environment and that it is increased upon PD-1-targeted immunotherapy and remains high in
CD8+ depleted mice (included in **Extended Data 10** and **Rebuttal Figure 31**). However, CD8
depleted mice lack tumor development (included in **Figure 2** and **Rebuttal Figure 36j**). In line
with Referee #2 and the co-submitted manuscript Dudek et al., we think, that the presence of
CD8+ T-cells is essential to drive hepatocarcinogenesis. We thus have performed the above
mentioned CD8 depletion combined with PD-1 targeted immunotherapy to underline that CD8+
T-cells are essential for increased hepatocarcinogenesis upon PD-1-targeted immunotherapy
compared to control mice under CDHFD diet (included in **Figure 4** and **Extended Data 20+21**
and **Rebuttal Figure 27, 28, 32**).

We have functionally strengthened data shown by Dudek et al. that TNF - as a marker of the
inflammatory environment - is crucial for sensitizing the hepatic microenvironment to CD8 T-
cell -mediated cell death by performing anti-TNF with/without PD-1-targeted immunotherapy.
This has allowed the interpretation and has been experimentally demonstrated that only an
inflammatory environment combined with the presence of CD8 T-cells drives increased
hepatocarcinogenesis upon PD-1-targeted immunotherapy (included in **Figure 4, Extended**
**Data 20+21** and **Rebuttal Figure 27, 28, 32**).

Furthermore, to shed new light on potential compensatory immunological mechanisms of
CD4+PD-1+ T-cells in the context of PD-1-targeted immunotherapy, we have performed CD4
depletion with/without PD-1-targeted immunotherapy (included in **Extended Data 22 and 23**
and **Rebuttal Figure 33, 34**). Notably, these experiments indicate that in contrast to CD8+ T-
cells CD4+ T-cells do not play a major effector role in comparison to CD8+ T-cells in anti-PD1
related liver cancer formation in the context of NASH and anti-PD1 treatment (included in
**Figure 32n**).

Rebuttal Figure 36

(a) and (b) multiplex ELISA concentrations of hepatic inflammation-associated cytokines and (c) chemokines of 12 months ND, CD-HFD, CD-HFD-fed mice + 8 weeks treatment of α -PD-1 (ND n = 10 mice; CD-HFD n = 14 mice; CD-HFD + α -PD-1 n = 13 mice).

Rebuttal Figure 37

(a) Histological staining of hepatic tissue by H&E, Sirius Red, PD-1 and CD8 of 12 months ND, CD-HFD or CD-HFD + 8 weeks treatment of α -PD-1 (H&E: ND n= 24 mice; CD-HFD n= 40 mice; CD-HFD + α -PD-1 n= 36 mice; Sirius Red: ND n= 19 mice; CD-HFD n= 31 mice; CD-HFD + α -PD-1 n= 27 mice; PD-1: ND n= 5 mice; CD-HFD n= 5 mice; CD-HFD + α -PD-1 n= 7 mice). Arrowheads indicate PD-1⁺ cells. Scale bar: 50 μ m. (i) ALT and (j) AST levels of 12 months ND, CD-HFD or CD-HFD + 8 weeks treatment of α -PD-1 (ALT: ND n= 22 mice; CD-HFD n= 42 mice; CD-HFD + α -PD-1 n= 30 mice). (k) NAS evaluation by H&E of 12 months ND, CD-HFD or CD-HFD + 8 weeks treatment of α -PD-1 (ND n= 24 mice; CD-HFD n= 40 mice; CD-HFD + α -PD-1 n= 36 mice). (l) Quantification of PD-1 staining of hepatic tissue by immunohistochemistry of 12 months ND, CD-HFD or CD-HFD + 8 weeks treatment of α -PD-1 (ND n= 5 mice; CD-HFD n= 5 mice; CD-HFD + α -CD8 n= 7 mice). (m) Macroscopy of liver of 12 months CD-HFD or CD-HFD + 8 weeks treatment of α -PD-1. Arrowheads indicate tumor/lesions. Scale bar: 10 mm. (n) Fibrosis evaluation of Sirius Red staining of 12 months ND, CD-HFD or CD-HFD + 8 weeks treatment of α -PD-1 (ND n= 19 mice; CD-HFD n= 53 mice; CD-HFD + α -PD-1 n= 33 mice). (o) Quantification of tumor/lesion size and (p) tumor load of 12 months CD-HFD or CD-HFD + 8 weeks treatment of α -PD-1 (tumor/lesion size, tumor load: CD-HFD n= 19 mice; CD-HFD + α -PD-1 n= 29 mice). (q) Quantification of tumor incidence of 12 months CD-HFD or CD-HFD + 8 weeks treatment of α -CD8, co-depletion of α -CD8/NK1, or α -PD-1 (tumor incidence: CD-HFD n= 32 tumors/lesions in 87 mice; CD-HFD + α -CD8 n= 2 tumors/lesions in 31 mice; CD-HFD + α -CD8/NK1.1 n= n= 0 tumors/lesions in 6 mice; CD-HFD + α -PD-1 n= 33 tumors/lesions in 44 mice).

Research for a Life without Cancer

6. Crucially, and related to my previous point, the authors also did not perform CD8 depletion
in the context of anti-PD-1 treatment to show that CD8 cells are indeed the cells that are
responsible for increased carcinogenesis upon anti-PD-1 therapy.

We thank Referee #2 for this important comment and fully agree that anti-PD-1 treatment in
the context of CD8 depletion is crucial for data interpretation and we included this experiment
in a revised manuscript (included in **Figure 4, Extended Data 20 and 21** and **Rebuttal Figure**
**27, 28, 32**).

The combined anti-CD8/anti-PD-1 treatment has allowed an understanding on a functional
level, that indeed increased the hepatic abundance of CD8+PD-1+ T-cells upon PD-1-targeted
immunotherapy is crucial for driving hepato-carcinogenesis. Notably, this treatment reduced
NAS, liver damage and some cytokines (e.g. MCP-1, CCL2, CCL3, MIP-3a) that affect the
pathway of CD8+ T-cell activation by the liver environment (e.g. IL33, IL21).

7. At times, the authors are (highly) selective in the data they choose to discuss and interpret.
As an example, regarding Figure 1i, the authors describe the CD8+ T-cells in CD-HFD mice to
demonstrate profiles of cytotoxicity and effector function because of increased expression of
GzmK/M and Pdcd1. However, in the same plot shows that these cells have reduced
expression of GzmA/B, Klrg1, Il2ra, TNF and Il2; all markers of effector/cytotoxicity. How do
the authors harmonize these observations?

We thank Referee #2 for asking this important question. As Referee #2 highlighted in the
example of **Figure 1**, we think it is of vital importance to display the observed profile of CD8 T-
cells on a broad scale. We believe that this particular character of T cells – that initially appears
to be exhausted (e.g. TOX expression) is actually hyperactivated with a particular pattern of
expression.

Thus, the single-cell technology allows dissecting the expression profile of CD-HFD-fed CD8+
T-cells into a combination of cytotoxicity/exhaustion expression, indicative of a unconventional
activation/effector. To not lose single-cell resolution and how the data translates into proteins,
we have corroborated these data by mass-spectrometry. These data corroborated the scRNA-
data of **Figure 1** with enrichment for effector function (e.g. T-cell activation, T-cell
differentiation, and NK mediated cytotoxicity) in CD-HFD-fed CD8+PD-1+ T-cells (included in
**Extended Data 5** and **Rebuttal Figure 38**). Thus, we decided to display a wide variety of
markers of effector function/cytotoxicity allowing the reader a more sophisticated view into the

phenotype. Moreover, we have compared this pattern with human NASH and indeed could
find that patients with NASH do resemble a similar pattern.

To test this unconventional activation/exhaustion phenotype on a functional level, we
performed all the treatments described in **Figures 2-4** in the absence or in the presence of
anti-PD1-related immunotherapy (anti-CD8, anti-CD8/anti-NK1.1, anti-CD8/anti-PD1, anti-
PD1, anti-PDL1, anti-TNF, anti-TNF/anti-PD1, and as control experiment anti-CD4 and anti-
CD4/anti-PD1), as well as the corroboration with the human data.

For example, an increased anti-inflammatory role by IL-10 expressing CD8+ T-cells upon PD1-
targeted immunotherapy could not be corroborated (included in **Extended Data 19** and
**Rebuttal Figure 39**) (Breuer et al., 2020). Of note, in this publication diet-based NAFLD
induction was achieved by feeding either WD or CD-HFD for 8-10 weeks. This is in strong
contrast to our experimental regime of applying diet for 3, 6, or 12 months as we show, that
the preclinical model presents different stages of NASH pathology severity including
hepatocarcinogenesis (included in **Figure 1** and **Rebuttal Figure 25**).

Furthermore, we would like to draw attention to the improved cross-referencing to the co-
submitted manuscript Dudek et al., which confirmed a CD8 profile of effector
function/exhaustion/cytotoxicity on a functional level (e.g. TNF sensitizing, high Granzyme
expression, TCR-independent mediated cell death). Moreover, we tried to improve the
discussion on recent literature on the role of CD8 T-cells in metabolic diseases.

Rebuttal Figure 38

(a) Selected average marker expression in T-cell subsets of CD8⁺ and (b) CD4⁺ sorted TCRβ⁺
by scRNA-seq of 12 months ND or CD-HFD-fed mice (n= 3 mice/group). (c) Selected marker
expression in hepatic CD8⁺ T-cells by scRNA-seq comparing CD8⁺ with CD8⁺PD-1⁺ T-cells
of 12 months CD-HFD + IgG or CD-HFD-fed mice + 8 weeks treatment of α-PD-1 (n= 3
mice/group). (d) Selected marker expression in hepatic CD4⁺ T-cells by scRNA-seq comparing
CD4⁺ with CD4⁺PD-1⁺ T-cells of 12 months CD-HFD + IgG or CD-HFD-fed mice + 8 weeks
treatment of α-PD-1 fed mice (n= 3 mice/group). (e) Selected marker expression in hepatic
CD8⁺PD-1⁺ T-cells by mass- spectrometry of 12 months ND or CD-HFD-fed mice (ND n= 4
mice, CD-HFD n= 6 mice). (f) Selected marker expression in hepatic CD8⁺PD-1⁺ T-cells
sorted TCRβ⁺ cells by mass- spectrometry of 12 months CD-HFD or CD-HFD-fed + 8 weeks
treatment of α-PD-1 fed mice (n= 6 mice/group). Candidates developing steady in-/decrease
from ND to CD-HFD to CD-HFD-fed mice + 8 weeks treatment of α-PD-1 are indicated in red.
(n= 6 mice/group).

Rebuttal Figure 39

(a) Polarization by flowcytometry of hepatic CD8+PD-1+ T-cells of 12 months ND, CD-HFD or
CD-HFD-fed mice + 8 weeks treatment of α -PD-1 (ND n= 12 mice; CD-HFD n= 7 mice; CD-
HFD + α -PD-1 n= 6 mice).

8. Regarding Figure 1e, the authors state that CD-HFD contain a significantly altered immune
composition that mainly affects the CD8+ T-cell compartment. However, this finding was not
significant ($p=0.09$ for CD8+PD-1+ T-cells and ns for CD8+ T-cells). In this plot, the authors
do show significant differences in frequency of CD4+ T-cells ($p<0.01$), classical monocytes
($p<0.01$) and MDMs Ly6CHigh ($p=0.01$). Why are these cell types not regarded as interesting?
Are these cells responsible for the authors' proposed phenotype? In line 259 the authors state
that there are only minor differences in the CD4 compartment, yet when looking at the data
(Figure S9h and Figure S9f) the difference in the CD4 subset of CD62L-CD44+CD69+ upon
anti-PD-1 blockade is as strong as, if not stronger than, in the same subset of CD8 T-cells,
which the authors do deem interesting.

We thank Referee #2 pointing out these details in our analysis. We agree with Referee #2, that
immunological subsets represented in our data set are well described in the literature (e.g.
reduction of CD4+ T-cells (Ma et al., 2016) and changes in the myeloid compartment, including
classical monocytes and MDMs Ly6CHigh (Malehmir et al., 2019; Nakagawa et al., 2014),
therefore the respective citations are included in our introduction and discussion.

We added new data and have re-analyzed the data displayed in **Figure 1e** according to
Referee's #4 comments also by highlighting NKT cells. These results, in CD8+PD1+ ($p= 0.03$),
significantly changed. Other changed cellular subsets after 12 months of CD-HFD feeding are
CD4+ T-cells ($p= 0.04$), classical monocytes ($p< 0.01$), KC ($p= 0.01$), MDMs ($p=0.02$), MDMs
Ly6C+ ($p< 0.01$). We agree with Referee #2, that CD4 T-cells and their expression of PD-1
might play a crucial role in shaping the liver micro-environment and in the observed phenotype

and thus included analysis of CD4 T-cells to the majority of our experiments (e.g. **Extended**
**Data 3** and **Rebuttal Figure 40**).

However, the magnitude of effects observed in CD4+ T-cells is minor when compared to CD8+
T-cells (e.g. **Extended Data 11** mean (CD8+CD62L-CD44+CD69+) ~12% (%of CD45+) vs
mean (CD4+CD62L-CD44+CD69+) ~4% (%of CD45+) upon PD-1 targeted immunotherapy).
Data obtained from CD4 depletion with/without PD1-targeted immunotherapy indicate, that the
increased hepatocarcinogenesis in the context of immunotherapy is independent of hepatic
abundance of CD4+ T-cells in the preclinical NASH model (included in **Figure 4**, **Extended**
**Data 22 and 23** and **Rebuttal Figure 32n, 33, 34**).

However, CD4+ T-cells might have a diverse set of effector functions (e.g. interpreting tumor
incidence in anti-CD8/anti-PD1 treated animals: although CD4 cells show trends for
decreasing, CD4 are relatively increased in the absence of CD8+ T-cells but immunotherapy,
thus CD4+ T-cells might be responsible for baseline tumor incidence in the context of
immunotherapy (included in **Extended Data 22 and 23** and **Rebuttal Figure 33, 34**); or CD4
might have a tumor controlling role, as there are the trends of increased tumor incidence upon
anti-CD4/anti-PD1 co-treatment (tumor incidence (anti-PD-1 mono-treatment)= 75% vs tumor
incidence (anti-CD4/anti-PD1 co-treatment)= 88%) (included in **Figure 4** and **Rebuttal Figure**
**32n**)).

Of note, CD4+ T-cells might also significantly changed in the human situation, and have also
analyzed human CD4+ cells a by scRNA-Seq (included in **Extended Data 25c** and **Rebuttal**
**Figure 41a**). In addition, we have performed RNA velocity analyses of the scRNA Seq data of
mouse and human CD4 T cells. In mouse, no significant velocity flow was detected in 12
1545 months CD-HFD-fed mice, indicating, that CD4 cells are not transcriptionally activated and
1546 driven by NASH-conditions or PD-1-targeted immunotherapy in NASH. However, we want to
1547 point out, that in the mouse NASH model CD8 T-cells increase statistically significant, and thus
CD4 are relatively fewer cells compared to CD8. Therefore, the velocity analysis of mouse
CD4 T-cells need to be taken with caution, because we included 300-500 cells only per
described subset. As a consequence, we included the negative CD4 T-cell data not in the
manuscript but in the Rebuttal letter as **Rebuttal Figure 42**. Velocity analyses on human CD4
lead to comparable problems like seen in mouse. As a consequence, we included the negative
CD4 T-cell data not in the manuscript but in the Rebuttal letter as **Rebuttal Figure 42**.

Like previously mentioned in point 3 raised by Referee #2 concerning the myeloid cells, our
presented data argue, that myeloid cells are recruited to the liver, extend and fine-tune liver
inflammation. While we see MDMs Ly6C+ cells increased comparing 12 months ND vs CD-
HFD-fed mice, our functional treatments (anti-PD-1, anti-CD8/anti-PD-1, anti-TNF, anti-

TNF/anti-PD-1, anti-CD4 and anti-CD4/anti-PD-1) did not result in significant changes in
CD11b+Ly6C+ cells, indicating a rather minor role in comparison to the changes we observed
in the CD8 compartment (included in **Extended Data 4, 21, 23 and 24** and **Rebuttal Figure**
**23, 24, 28, 34**).

Furthermore, we discuss the myeloid changes and potential role of CD4+ T-cells in greater
detail in the main text.

Finally, we performed an anti-CD4 antibody treatment with or without the combination of anti-
PD1-related immunotherapy. Anti-CD4 antibody treatment successfully depleted or strongly
reduced intrahepatic CD4+ T cells in NASH. However, depletion of CD T cells did not reduce
liver cancer incidence – which is in contrast to CD8+ T cell depletion. Rather, in contrast, CD4
T cell depletion showed a trend in increase of tumor incidence – in line with published data by
1569 Ma et al., 2016 (Nature).

Research for a Life without Cancer

Rebuttal Figure 40

(a) Analysis of 5000 randomly chosen CD45+ cells by flow cytometry to define distinct marker
 expression of 12 months ND or CD-HFD-fed mice (ND n= 4 mice; CD-HFD n= 8 mice). (b)
 Average marker expression of defined CD45+ subsets of 5000 randomly chosen CD45+ cells
 by flow cytometry of 12 months ND or CD-HFD-fed mice (ND n= 4 mice; CD-HFD n= 8 mice).
 (c) Quantification of hepatic CD8+ cells and PD-1+ expressing cells by immunohistochemistry
 of 12 months ND, CD-HFD or WD-HTF-fed mice (PD-1: n= 5 mice/group; CD8: ND n= 6 mice;
 CD-HFD n= 6 mice; WD-HTF n= 5 mice). (d) Immunofluorescence staining of single channel-
 staining PD-1, CD8 and CD4 (ocher) of 12 months ND or CD-HFD-fed mice (n= 3 mice/group).
 Arrowheads indicate CD8+ (red), PD-1+ (green) or CD4+ (ocher) cells. Scale bar: 100 μ m. (e)
 H&E, CD8 and PD-1 staining, evaluation by NAS and quantification of CD8+ cells and PD-1+
 expressing cells by immunohistochemistry of 32-weeks old hURI-tetOFF^{hep} and non-

transgenic litter control mice (n=6 mice/group). Arrowheads indicate specific staining positive
cells. Scale bar: 100 μ m. (f) Quantification of abundance, (g) PD-1 expression and flow
cytometry plots of hepatic CD8⁺ T-cells by flow cytometry of 6 or 12 months ND or CD-HFD-
fed mice (abundance of CD8: 6 months: ND n= 17 mice; CD-HFD n= 10 mice; WD-HTF n= 7
mice; 12 months: ND n= 11 mice; CD-HFD n= 6 mice; WD-HTF n= 5 mice; PD-1 expression
in CD8⁺ T-cells: 6 months: ND n= 15 mice; CD-HFD n= 14 mice; WD-HTF n= 7 mice; 12
1590 months: ND n= 10 mice; CD-HFD n= 6 mice; WD-HTF n= 5 mice). (h) Quantification of
1591 abundance, (i) PD-1 expression and flow cytometry plots of hepatic CD4⁺ T-cells by flow
cytometry of 6 or 12 months ND or CD-HFD fed mice (abundance of CD4: 6 months: ND n=
17 mice; CD-HFD n= 10 mice; WD-HTF n= 7 mice; 12 months: ND n= 11 mice; CD-HFD n= 6
mice; WD-HTF n= 5 mice; PD-1 expression in CD4⁺ T-cells: 6 months: ND n= 15 mice; CD-
HFD n= 14 mice; WD-HTF n= 7 mice; 12 months: ND n= 10 mice; CD-HFD n= 6 mice; WD-
HTF n= 5 mice). (j) Hepatic abundance of TCR $\gamma\delta$ T-cells of 6 or 12 months ND or CD-HFD fed
mice (6 months ND n= 8 mice; CD-HFD n= 6 mice; 12 months ND n= 8 mice; CD-HFD n= 6
mice).

**Rebuttal Figure 41**

(a) Flow cytometry plot of FMO control, (b) quantification of patient-liver-derived PD-1+CD8+
T-cells, and (c) quantification of CD4, CD8, $\gamma\delta$, NK and NKT cells healthy or NAFLD/NASH
patients (Supplementary Table 1: healthy n= 8 patients; NAFLD/NASH n= 16 patients). (d)
Analysis of randomly chosen CD45+ cells and (e) average marker expression of defined
CD45+ subsets by flow cytometry derived from hepatic biopsies of control and NAFLD/NASH
patients to define distinct marker expression (Supplementary Table 2: control n= 6 patients;
NAFLD/NASH n= 11 patients). (f) Definition of cellular subsets, (g) relative quantification of
defined cellular subsets of randomly chosen CD45+ cells, (h) polarization of CD8+ T-cells and
(i) quantification of CD4+CD27+, or $\gamma\delta$ TCR+Eomes+ T-cells by flow cytometry derived from
hepatic biopsies of healthy and NAFLD/NASH patients (Supplementary Table 2: control n= 6
patients; NAFLD/NASH n= 11 patients).

**Rebuttal Figure 42**

(a) RNA Velocity analyses of scRNA-seq data showing expression, and (b) velocity of patient-
liver-derived CD4+ T-cells of control, or NAFLD/NASH patients in comparison to mouse-liver-
derived CD4+ T-cells (patients: NAFLD/NASH n= 3 patients; mouse: n= 3 mice/group).
(c) Correlation of expression along the latent-time of selected genes along the latent-time
(mouse: n= 3 mice/group).

9. Along these lines, in line 387 the authors state that consistent with previous results, effects
on the CD4+PD-1+ T-cell compartment remained minor, yet the differences observed for

matching analyses (i.e. S17a vs S17g, S17b vs S17f, S17i vs S17j) of CD4 and CD8
populations show similar, if not stronger, effects for the CD4 T-cell population. Why are these
differences disregarded by the authors?

We believe that the comment of Referee #2 is important and we are in line that the context of
highlighting potential CD4-mediated effects in the context of PD-1-targeted therapy had to be
investigated in detail (e.g. in **Extended data 5, 18** and **Rebuttal Figure 43**) In line with the
comment of Referee#2, we set out to investigate the character and function of CD4+ T-cells
by scRNA-seq analyses in human and mouse NASH livers, but like raised in point 8 of Referee
#2 strongly suggest to take the velocity analysis of mouse CD4 T-cells with caution, because
we included 300-500 cells only per described subset. Thus, we included these analyses in only
in the **Rebuttal Figure 42**. Moreover, our experiments using an anti-CD4 depleting antibody
alone or in the context of anti-PD1-related immunotherapy indicate a minor role of the CD4
compartment in our model as well (included in **Extended Data 22, 23** and **Rebuttal Figure**
**33, 34**).

As mentioned in point 8 raised by Referee #2, we agree with Referee #2, that similar
phenotypes can be observed when comparing effects in CD4+ and CD8+ T-cell subsets upon
PD-1 targeting immunotherapy. We do not disregard the changes in the CD4 compartment but
would like to draw attention to the magnitude of changes in the setting of chronic hepatic
inflammation – and the functional experiments with anti-CD8, anti-CD8/anti-PD-1, anti-CD4,
and anti-CD4/anti-PD1 antibodies.

We have also discussed the relevant literature as well as our data on CD4+ T cells in the
discussion in detail. We, in addition, believe that the CD4+ T-cell depletion experiments
with/without PD-1 targeted immunotherapy in mice have enabled us to strengthen our
hypothesis on a more functional level: CD4 depletion alone or in the context of anti-PD1-related
immunotherapy in NASH-induced HCC failed to revert/prevent liver cancer formation. In
contrast, anti-CD8 depleting antibody treatment alone reverted/prevented liver cancer
formation.

The role of CD4+ T-cells in the context of immunotherapy remains to be defined in more detail,
as CD4-depletion did not lead to a reversal of the pro-tumorigenic effects of anti-PD1 therapy
in the context of NASH induced HCC. However, CD4+ T-cells might exert a
protective/controlling role in the context of PD1-targeted immunotherapy and presence of
CD8+ T-cells, as combinatorial treatment of anti-CD4 depletion and PD1-targeted
immunotherapy led to an increase of tumor incidence compared to anti-PD1 treatment alone
(included in **Figure 4, Extended Data 22 and 23** and **Rebuttal Figure 32-34**).

**Rebuttal Figure 43**

(a) Marker expression of CD4+ and CD8+ sorted TCR β + cells defining T-cell subsets by single
cell RNA-sequencing of 12 months ND or CD-HFD-fed mice (n= 3 mice/group). (b) Relative
frequency of CD4+ and CD8+ sorted TCR β + cells by single cell RNA-sequencing of 12 months
ND or CD-HFD fed mice (n= 3 mice/group). (c) Selected marker expression in CD4+ T-cells
sorted TCR β + cells by single cell RNA-sequencing of 12 months ND or CD-HFD fed mice (n=
3 mice/group). (d) Selected average marker expression in T-cell subsets of CD4+ and CD8+
sorted TCR β + by scRNA-seq of 12 months ND or CD-HFD-fed mice (n= 3 mice/group).(e)
Differential gene expression of CD4+PD-1+ vs CD4+ T-cells and (f) selected average marker
expression in CD4+ and CD8+ T-cell subsets of by scRNA-seq of control and NAFLD/NASH
patients (control n= 4 patients; NAFLD/NASH n= 7 patients).

10. Similarly, in Figure 5a, the authors claim that a CD8+PD-1+ T-cell population arises upon
NASH. However, there is a, perhaps even stronger, depletion of an Eomes+ gamma-delta T-
cell subset. Additionally, a very strong induction of a CD4+CD27+ population is observed in
NASH samples. Why are these not discussed? Can these populations also be identified in the
authors' murine models? Do these contribute to the authors' described phenotype? The
authors should deplete CD4 T-cells and gamma-delta T-cells in their murine models, as these
cell types may, at the very least, contribute to what occurs in patients.

We thank Referee #2 for raising this important concern. Indeed, we have so far not discussed
the loss of gamma-delta T-cell subsets or a potential increase of CD4+ T-cells and included
this now thoroughly in the revised version of the manuscript (included in **Extended Data 3, 21,**
**23, 25 and 26** and **Rebuttal Figure 28a, 34a, 41, 44**). In line with the comments of Referee#2,
we have now described and discussed these populations in detail, by scRNA-seq and
multicolor flow cytometry in mouse and three distinct human cohorts recruited from 3 different
centers across Europe.

As mentioned in points 8 and 9 raised by Referee #2, we have depleted CD4 T-cells
with/without PD-1 targeted immunotherapy. Of note, CD27 could not be detected in our
scRNA-seq data set obtained from the preclinical mouse model as significantly changed. In
human bulk RNA-seq CD27 expression increased, but CD4 expression decreases with the
severity of pathology. CD27+CD4+ T cells did not reach statistical significance in our cohorts
by flow cytometry (included in **Extended Data 25** and **Rebuttal Figure 41**). Of note, in our
second cohort, CD4+ T-cells are significantly enriched in NAFLD/NASH patients by flow
cytometry, however as this cohort was analyzed retrospectively, we could not analyze CD27
expression (included in **Extended Data 25**). Furthermore, the abundance of CD4+CD27+ cells
was not increased in our human scRNA cohorts (included in **Extended Data 27** and **Rebuttal**
**Figure 44**).

As mentioned in point 8 we have performed a velocity analyses of the scRNA Seq data of
mouse CD4 T cells (see Rebuttal letter below). In mouse, no significant velocity flow was
detected in 12 months CD-HFD-fed mice, indicating, that CD4 cells are not transcriptionally
activated and driven by NASH-conditions or PD-1-targeted immunotherapy in NASH.
However, we again want to point out, that the velocity analysis of mouse CD4 T-cells need to
be taken with caution because we included 300-500 cells only per described subset. As a
consequence, we included the negative CD4 T-cell data not in the manuscript but in the
Rebuttal letter. Velocity analyses on human CD4 lead to comparable problems as seen in
mouse. As a consequence, we included the negative CD4 T-cell data not in the manuscript but
in the Rebuttal letter as **Rebuttal Figure 42**.

We agree that $\gamma\delta$ T-cells might be involved in underlying processes of NASH or NASH to HCC
transition – also in the context of PD1-related immunotherapy. In humans, our data is not
conclusive in all experiments, e.g. our data indicate for $\gamma\delta$ T-cells, if we compare: bulk RNA-
seq indicates a reduced expression in severe NASH pathology of EOMES, TRDC, and TRGC1
(included in **Extended Data 28** and **Rebuttal Figure 41, 44, 45**), however, both flow cytometry
cohorts and the scRNA-seq cohort indicate no change of either $\gamma\delta^+$ T-cells or $\gamma\delta^+$ Eomes+ T-
cells comparing control vs NAFLD/NASH patients (included in **Extended Data 25, 27** and
**Rebuttal Figure 41, 44**).

Corroborating the human flow cytometry data in our mouse model upon NASH establishment,
we detected no difference in hepatic abundance of $\gamma\delta$ -T-cells between chow- or CD-HFD-fed
control mice. Furthermore, data presented in **Figures 1 and 4** and **Extended Data 3** argues
against the major contribution of gamma delta T-cells in the mouse model of NASH. Here, we
did not observe significant differences in the “other leukocytes” subset. In the revised
manuscript, we analyzed $\gamma\delta$ -T-cells separately to strengthen the point, that these cells are not
significantly changed upon diet feeding (included in **Extended Data 3, 20-23** and **Rebuttal**
**Figure 28, 34, 44a**).

Rebuttal Figure 44

(a) Hepatic abundance of TCR $\gamma\delta$ T-cells of 6 or 12 months ND or CD-HFD fed mice (6 months
 ND n = 8 mice; CD-HFD n = 6 mice; 12 months ND n = 8 mice; CD-HFD n = 6 mice).

(b) NAS and BMI of patients used for scRNA-seq analyses of patient-liver-derived T-cells of
 control and NAFLD/NASH patients (control n = 4 patients; NAFLD/NASH n = 7 patients). (c)
 UMAP representation, marker expression, (d) relative quantification and (e), (f), (g) polarization
 of defined T-cell subsets of defined T-cell subsets of patient-liver-derived T-cells by scRNA-
 seq of control and NAFLD/NASH patients (control n = 4 patients; NAFLD/NASH n = 7 patients).

**Rebuttal Figure 45**

(a) RNA-sequencing data comparing NASH with varying fibrosis (F0 – F4 according to Brunt
classification) normalized to NAFLD from a total of n= 206 NAFLD/NASH patients corrected
for batch, gender and center

11. The patient data is not convincing, but also does not match their murine models. In Figure
5a, the authors show that CD8+GzmB+ cells are specifically lost in NASH samples which
seems to counteract the claim made by the authors that inflammatory CD8 T-cells cause liver
inflammation and associated carcinogenesis. The authors similarly show in S19a that IFN γ ,
Ccl3 and PD-L1 are in fact reduced in advanced NASH samples; does the loss of these
inflammatory genes not counteract the claims made in Figure 3g, S4d, S10, S11 and S13a?

We thank Referee #2 for raising this important point and agree, that GzmB+CD8+ population
is decreased as well as GzmB expression in bulk RAN-seq (included in **Extended Data 28**
and **Rebuttal Figure 45**), other populations, on the other hand, are increased. GzmB is a
strong indication for inflammatory CD8+ T-cells. We would like to draw attention to the
improved cross-referencing to the co-submitted manuscript Dudek et al., in which Gzmb along
with other cytotoxic effector molecules (e.g. TNF) are key mediators of a hepatic inflammatory
environment, but not the executing molecules driving hepatocarcinogenesis. However, we
agree with Referee #2, that the data presented in **Figure 5** has limitations due to the small
sample size, although we could reproduce the cellular abundance between healthy vs

NAFLD/NASH patients in a second cohort from a second center (included in **Figure 5** and
**Extended Data 25** and **Rebuttal Figure 41, 46**).

We agree with Referee #2, that certain inflammatory genes (e.g. *Ifny*, *Ccl3*, *Cd274*) show
decreased expression along with NASH progression, however, how this translates into local
hepatic proteins-expression remains elusive (e.g. for human gene expression vs
immunohistochemical staining of *Pdcd1* in NASH F1-3 (included in **Figure 6** and **Rebuttal**
**Figure 47**); or F0-F4 for CD4, or CD274 (included in **Extended Data 28** and **Rebuttal Figure**
**47**)). As an example, human PD-L1 increases with NASH severity on IHC, which is
corroborated by the preclinical model (included in **Extended Data 3, 20, 22** and **Rebuttal**
**Figure 27, 33, 48**).

To shed more light on the phenomena, we focused on our human scRNA-seq on the analyses
of CD8+ T-cells (included in **Figure 5**, **Extended Data 27** and **Rebuttal Figure 43f, 44, 46**)
and correlated these cells to the CD8+ T-cells analyzed from our preclinical model (included
in **Figure 5** and **Rebuttal Figure 46f**). These data match each other very well, strengthening
in our opinion hypotheses and conclusions drawn from the preclinical NASH-model. Therefore,
we do not think the results of the bulk RNA-seq counteracts the claims of previous figures from
the mouse model but allows an in-depth understanding of underlying inflammation in different
NASH stages (e.g. Referee #1: decrease activity of NASH with disease progression to HCC).

Rebuttal Figure 46

(a) Flow cytometry plots, quantification of patient-liver-derived PD-1+CD8+ T-cells, and (b)
 correlation of PD-1+CD8+ T-cells with BMI, NAS and ALT of healthy or NAFLD/NASH patients
 (Supplementary Table 1: healthy n= 8 patients; NAFLD/NASH n= 16 patients). (c) UMAP
 representation of randomly chosen CD45+ cells and (b) flow cytometry plots and quantification
 of CD8+PD-1+CD103+ derived from hepatic biopsies of control, or NAFLD/NASH patients

(Supplementary Table 2: control n= 6 patients; NAFLD/NASH n= 11 patients) Populations:
 CD8+ (violet), CD8+PD-1+CD103+ (red). (e) UMAP representation of CD3+ cells and (f)
 analyses of differential gene expression by scRNA-seq of control, or NAFLD/NASH patients
 (control n= 4 patients; NAFLD/NASH n= 7 patients). (f) Correlation of significant differentially
 expressed genes in liver-derived CD8+PD-1+ compared to CD8+PD-1- T-cells subsets of 12
 1787 months CD-HFD fed mice and NAFLD/NASH patients (mouse: n= 3 mice; human: n= 3
 patients). (g) Velocity analyses of scRNA-seq data showing (h) expression, transcriptional
 activity, (i) gene expression and (j) correlation of expression along the latent-time of selected
 genes along the latent-time of patient-liver-derived CD8+ T-cells of control, or NAFLD/NASH
 patients in comparison to mouse-liver-derived CD8+ T-cells (patients: NAFLD/NASH n= 3
 patients; mouse: n= 3 mice/group).

**Rebuttal Figure 47**

(a) Immunohistochemical staining and (b) quantification of hepatic PD-1, CD8 and CD4
 expressing cells of NAFLD and NASH patients in Supplementary Table 3 with varying stages
 of fibrosis (NAFLD n= 9 patients; NASH F1/0 n= 7 patients; NASH F2 n= 12 patients; NASH
 F3 n= 21 patients; NASH F4 n= 16 patients; CD4: NAFL n= 6 patients; NASH F1/0 n= 4
 patients; NASH F2 n= 8 patients; NASH F3 n= 17 patients; NASH F4 n= 9 patients). (c)

Correlation analysis of PD-1 against fibrosis scoring according to Brunt by
 immunohistochemical staining by RNA-sequencing (NAFLD/NASH n= 65 patients). A total of
 1656 patients were included in all three randomized trials, and 985 patients received a
 checkpoint inhibitor (Supplementary Table 7). (d) Immunohistochemical staining of PD-L1 in
 patient-derived liver samples. Scale bar: 50 μ m.

Rebuttal Figure 48

(a) Quantification of hepatic PD-L1+ expression by RNA in situ hybridization of 6- or 12-months
 ND or CD-HFD-fed mice (6 months: ND n= 13 mice; CD-HFD n= 11 mice; 12 months: ND n=
 7 mice; CD-HFD n= 7 mice). Scale bar: 100 μ m. (b) Quantification of hepatic PD-L1+
 expression by immunohistochemistry of 12 months ND or CD-HFD fed mice (6 months: ND n=
 4 mice; CD-HFD n= 8 mice). Scale bar: 100 μ m.

12. Lastly, the majority of patient data are not significant and show weak effect sizes; is it fair
 to draw strong conclusions on the basis of these data as the authors do?

We agree with Referee #2 and thus recruited additional patients to increase the number of
 patients in our initial clinical cohort from 65 to 130 HCC patients under anti-PD(L)1-targeted
 immunotherapy and validated our results in a second cohort of 118 HCC-patients under PD-
 1-targeted immunotherapy (included in **Figure 6** and **Rebuttal Figure 49**).

We agree with Referee #2, that the presented retrospective PD(L)1 targeted immunotherapy
 treated NAFLD/NASH-associated HCC cohort - although unique for Europe and treatment not
 officially licensed and thus reimbursement - is still small, although we would like to point out,
 that prominent trends or effects can be seen in small retrospective cohorts as well. Thus, our
 analyses of BCLC-C NAFLD/NASH-HCC vs other-etiological-HCC patients indicated, that
 NAFLD/NASH-HCC have significantly reduced overall survival compared to other-etiological-
 HCC in this small retrospective cohort. Of note, multivariate analyses identified NAFLD/NASH
 as an independent factor for treatment response and thus identifying NAFLD/NASH as a
 negative predictor for HCC immunotherapy (included in **Supplementary Table 8** and **Rebuttal**
 **Figure 49**).

We corroborated our hypothesis of non-viral (NASH-related) HCC being less responsive to
immunotherapy by a meta-analysis including 1656 patients of the three most important clinical
trials, identifying immunotherapy vs control for viral HCC as favorable treatment (HR(viral)=
0.64), in contrast, non-viral-HCC showed less benefit (HR(non-viral)= 0.92) for immunotherapy
(included in **Figure 6, Extended Data 30-32, Supplementary Table 9** and **Rebuttal Figure**
**50, 51**)).

Based on these data we want to point out that it is - as indicated by Referee#2 - of the highest
importance to us to specifically define/tone down appropriately the message of our manuscript:
Our manuscript does not indicate that immunotherapy is not beneficial for HCC patients at all.
Our manuscript rather demonstrates that HCC patients with viral etiologies do respond well
and achieve survival benefits - however, that patients with non-viral etiologies (e.g. NASH) do
not achieve a significant outcome benefit.

We thus propose to stratify HCC patients who are very likely to profit from immunotherapy and
strengthen the argumentation to use immunotherapy in specific cohorts of HCC patients. We
agree with Referee#1 that this information needs to be articulated in the paper appropriately
not to deliver wrong messages but to be very specific.

We truly believe that these are important clinical data, also providing the basis to test our
hypotheses in prospective studies on non-significantly beneficial effects in terms of OS for
immunotherapy in HCC patients with non-viral and NAFLD/NASH etiology, in particular.

Moreover, we toned down the conclusions of our retrospective cohort in the manuscript and
would like to point out, that bigger cohorts and prospective clinical trials are of utmost
importance for the scientific community.

Rebuttal Figure 49

(a) Nonalcoholic fatty liver disease (NAFLD) is associated with a worse outcome in patients
 with hepatocellular carcinoma (HCC) treated with PD-(L)1-targeted immunotherapy. A total of
 130 patients with advanced HCC received PD-(L)1-targeted immunotherapy (Supplementary
 Table 8). Kaplan-Meier curve display overall survival of patients with NAFLD vs. those with
 any other etiology; all 130 patients were included in these survival analyses (NAFLD n=13, any
 other etiology n=117). (b) Validation cohort of patients with HCC treated with PD-(L)1-targeted
 immunotherapy. A total of 1180 patients with advanced HCC received PD-(L)1-targeted
 immunotherapy (Supplementary Table 10). Kaplan-Meier curve display overall survival of
 patients with NAFLD vs. those with any other etiology; all 118 patients were included in these

(n=345). Hazard ratios for each trial are represented by squares, the size of the square
represents the weight of the trial in the meta-analysis. The horizontal line crossing the square
represents the 95% confidence interval (CI). The diamonds represent the estimated overall
effect based on the meta-analysis random effect of all trials.

A total of 1243 patients were included in two first-line trials comparing PD-1 or PD-L1 targeted
immunotherapy to sorafenib. 707 patients received an immune checkpoint inhibitor (either PD-
1 or anti-PD-1). (c) HCV and HBV were pooled into a separate category, termed “viral”, and a
subsequent meta-analysis comparing viral (n=754) and non-viral (n=489), mostly NASH and
alcohol intake, was performed. A subgroup analysis studying the specific effects of non-viral
etiologies (n=489) on the magnitude of effect of immunotherapy are presented, when
compared to (d) HBV (n=473) or (e) HCV (n=281). Hazard ratios for each trial are represented
by squares, the size of the square represents the weight of the trial in the meta-analysis. The
horizontal line crossing the square represents the 95% confidence interval (CI). The diamonds
represent the estimated overall effect based on the meta-analysis random effect of all trials.

Minor points:

- Figure 1j lacks a color scale bar and proper description. How does one interpret the difference
between ND and CD-HFD in this plot?

We thank Referee #2 for highlighting the lack of a color bar in this panel, we have added a
color scale bar with a proper description. Figure 1j displays the median expression of selected
genes in the different T-cell populations observed in our scRNA-seq data set (included in
**Figure 1, Extended Data 5 and Rebuttal Figure 43**) and serves as a supplement to the 2-
dimensional tSNE plot. In this panel, we do not compare ND to CD-HFD rather simply allow
the readers to view the gene signatures characterizing the different populations. A comparison
of ND and CD-HFD is visualized using volcano plots in Figure 1. As this heatmap is rather a
technical information, but does not condense scientific explanation in great detail, we decided
to move this heatmap to **Extended Data 5**.

- Where is the ND + PD-1^{-/-} in Figure 3b? Do these mice also get accelerated carcinogenesis?

We thank Referee #2 for highlighting this inconsistency. In line with the point raised by
Referee#2 we have improved this in a revised manuscript including PD-1^{-/-} mice on ND.
Literature does not report accelerated hepatocarcinogenesis
(<http://www.informatics.jax.org/allele/allgenoviews/MGI:4397682>) and we did not observe any
hepatocarcinogenesis in PD1^{-/-} under ND.

- There is no color scale bar in Figure 3e.

We thank Referee #2 for highlighting this inconsistency and improved our manuscript by
adding a scale bar.

- In Figure 5k, shouldn't progression-free survival and time to progression plots yield the exact
same data, but inversed? Why don't these curves match?

We thank Referee #2 for this question. TTP and PFS are different endpoints. TTP is defined
as the time from the date of treatment initiation until the date of first radiological tumor
progression. PFS is a composite endpoint. It is defined as the time from the date of treatment
initiation until radiological progression OR death, whatever comes first (Llovet et al., 2008). We
decided to leave out the non-significant data of TTP and PFS in our manuscript. Moreover,
upon recruiting the validation cohort of 118 HCC-patients under immunotherapy we decided
to not show TTP and PFS, but instead the multivariate analysis (included in **Supplemental**
**Table 9** and **Rebuttal Figure 49**).

- In Figure S1i, what is the parent population?

We thank Referee #2 for highlighting this inconsistency and improved our manuscript by
adding the description of the parent population. In the case of **Extended Data 1** the parental
populations are CD8+ (left) and respective CD4 or CD8 (right) T-cells.

- In Figure S4a, how does one distinguish ND from CD-HFD mice? The y-axis lacks a label.

We thank Referee #2 for highlighting this inconsistency and improved our manuscript by
adding the description of the y-axis.

- Figure 5c is plotted in a confusing manner (as the z-score scale is red independent of whether
it goes up or down), but it seems that the TNF signaling gene sets are actually decreasing in
expression.

We thank Referee #2 for highlighting this inconsistency. We decided after integration of the
new data, to leave that graph out as it communicates similar information already included in
**Extended Data 28**. Of note, if we change the labeling of z-score (similar to **Extended Data**
**28**), it clarifies, that TNF is indeed an increased pathway (similar to **Extended Data 28**).

- Why do the PD-1^{-/-} mice still express PD-1 (Fig. S12e)?

We thank Referee #2 for highlighting this inconsistency and improved our manuscript by re-
analyzing our flow cytometry data set (as gates have been set too loose – leading to a subset
of around 1% PD1 expressing CD4⁺ and CD8⁺ T cells). Analyses revealed that PD1^{-/-} ND-fed
mice have no intrinsic higher immune cell abundance, or activation and hepatocarcinogenesis
compared to ND-fed wt control mice at 6 months under diet (included in **Figure 3** and
**Extended Data 14** and **Rebuttal Figure 52**). Moreover, as indicated no PD1-expression can
be observed.

Rebuttal Figure 52

(a) Histological staining of hepatic tissue by H&E and CD8 of 6 months ND, CD-HFD or PD-1^{-/-} CD-HFD fed mice (H&E: ND n= 8 mice; PD-1^{-/-} ND n= 5 mice; CD-HFD n= 9 mice; PD-1^{-/-} CD-HFD n= 13 mice; CD8: ND n= 4 mice; CD-HFD n= 5 mice; PD-1^{-/-} CD-HFD n= 7 mice).

Arrowheads indicate CD8+ cells. Scale bar: 50 μ m. (b) Cytokine expression of hepatic CD8+
 T-cells of 6 months ND, PD-1-/- ND, CD-HFD or PD-1-/- CD-HFD fed mice (ND n= 4 mice; PD-
 1-/- ND n= 5 mice; CD-HFD n= 5 mice; PD-1-/- CD-HFD n= 6 mice). (c) Tumor/lesion incidence
 of 6 months CD-HFD or PD-1-/- CD-HFD fed mice (tumor incidence: CD-HFD n= 6
 tumors/lesions in 63 mice; PD-1-/- CD-HFD n= 6 tumors/lesions in 13 mice). (d) Body weight
 of 6 months ND, PD-1-/- ND, CD-HFD or PD-1-/- CD-HFD fed mice (ND n= 5 mice; PD-1-/-
 ND n= 3 mice; CD-HFD n= 5 mice; PD-1-/- CD-HFD n= 10 mice). (e) ALT levels of ND, PD-1-/
 -/- ND, CD-HFD or PD-1-/- CD-HFD (ND n= 9 mice; PD-1-/- ND n= 5 mice; CD-HFD n= 9 mice;
 PD-1-/- CD-HFD n= 10 mice). (c) NAS evaluation by H&E of ND, PD-1-/- ND, CD-HFD or PD-
 1-/- CD-HFD fed mice (ND n= 8 mice; PD-1-/- ND n= 5 mice; CD-HFD n= 9 mice; PD-1-/- CD-
 HFD n= 13 mice). (f) CD8 staining of hepatic tissue by immunohistochemistry of 6 months ND,
 PD-1-/- ND, CD-HFD or PD-1-/- CD-HFD fed mice (ND n= 4 mice; PD-1-/- ND n= 5 mice; CD-
 HFD n= 5 mice; PD-1-/- CD-HFD n= 7 mice). (g) – (j) Characterization of hepatic T-cells by
 flow cytometry of 6 months ND, PD-1-/- ND, CD-HFD or PD-1-/- CD-HFD fed mice (ND n= 4
 mice; PD-1-/- ND n= 5 mice; CD-HFD n= 5 mice; PD-1-/- CD-HFD n= 6 mice). (k) Relative
 quantification of hepatic leukocytes of 6 months CD-HFD or PD-1-/- CD-HFD fed mice (ND n= 4
 mice; PD-1-/- ND n= 5 mice; CD-HFD n= 5 mice; PD-1-/- CD-HFD n= 6 mice). (l) Histological
 staining of hepatic tissue by H&E of CD-HFD or PD-1-/- CD-HFD fed mice (ND n= 8 mice; CD-
 HFD n= 9 mice; PD-1-/- CD-HFD n= 13 mice). Dotted line indicates tumor/lesion border. Scale
 1993 bar: 100 μ m.

- In Figure S13k, the authors should present cleaved Caspase 3 and cleaved Caspase 8 if they
 want to conclude something about T-cell death, as total, uncleaved levels of these proteins do
 not indicate cell death.

We thank Referee #2 for highlighting this point. We have accordingly removed these plots and
 demonstrate cleaved caspase 3 by immunohistochemistry, which has the advantage that we
 not only see the Cleaved Caspase 3 directly but also which cells are undergoing apoptosis.
 These data are now included in **Extended Data 16** and **Rebuttal Figure 53**.

**Rebuttal Figure 53**

(a) Histological staining of hepatic tumor tissue by Collagen IV, cleaved Caspase 3, CD8, Ki-
67 of 12 months CD-HFD or CD-HFD-fed mice + 8 weeks treatment of α -PD-1 (Collagen IV,
cleaved Caspase 3: CD-HFD n= 13 mice; CD-HFD + α -PD-1 n= 14 mice; CD8, Ki-67: CD-HFD
n= 5 mice; CD-HFD + α -PD-1 n= 7 mice). Arrowheads indicate positive cells. Dotted line
indicates tumor/lesion rim. Tumor area is indicated by T. Scale bar: 100 μ m. (b) Scoring of
expression by immunohistochemistry staining of intra- and peri-tumoral hepatic tissue of 12
2011 months CD-HFD or CD-HFD-fed mice + 8 weeks treatment of α -PD-1 (CD-HFD n= 13 mice;
CD-HFD + α -PD-1 n= 14 mice). Crossed out boxes indicate not sufficient tissue for analysis.

- In Figure S16f, the FACS plot does not match the quantification on the left.

We thank Referee #2 for bringing this up and apologize for this inconsistency. We would like
to draw the attention, that in the flow cytometry plot the data is displayed as “%of CD8”, in
contrast in the box plot the data is displayed as “%of CD45” to give the reader a more
quantitative analysis.

- Regarding Figure S17b, the authors claim an increase in calcium levels in line 383 of their
manuscript, but this difference is not significant.

We agree with Referee #2. Thus, we have performed additional experiments – supporting our
initial finding that upon PD1-targeted immunotherapy calcium levels were increased on CD8+
but not CD4+ T-cells. This inconsistency was improved our manuscript accordingly.

- In Figure S18b, how does one interpret the difference between healthy, borderline NASH or
NASH patients? There is no explanation of the color scale bar. Also, what are “randomly
chosen CD45+ cells” as mentioned in the corresponding Figure Legend?

We thank Referee #2 for highlighting this inconsistency and improved our manuscript
accordingly by describing differences between patients and highlighting our analysis pipeline
for flow cytometric data according to (Brummelman et al., 2019). Moreover, we have added 2
more cohorts in the main Figure (**Figure 5**) and Extended Data and pooled borderline NASH
and NASH patient into one group of NAFLD/NASH patients after consultation with our
pathologists, who indicated that the difference between borderline NASH and NASH can be
regional – and thus is always is regarded as NASH (**Extended Figure 25** and **Rebuttal Figure**
**41, 44, 46**).

- Figure S19b is not legible.

We thank Referee #2 for this comment. In line, we have now changed the graph size and font
size.

- In lines 237-246 the authors describe that NK1.1-based depletion of immune populations did
not result in changed liver pathology, body weight, fibrosis ALT, hepatic cytokines and hepatic
chemokines. However, the animals who underwent this depletion also completely lacked liver
cancer development. How does this happen if the authors did not detect any changes? The
authors should perform NK1.1 depletion by itself to see if NK1.1+ cells, potentially depending
on CD8 cells, are in fact responsible for the authors' phenotype.

We thank Referee #2 for highlighting this unprecise description of our data and improved our
manuscript by highlighting differences between CD8 depletion and CD8/NK1.1 co-depletion in
greater detail.

We included additional GSEA analysis of RNA-seq data, which display changes in CD8/NK1.1
co-depleted in comparison to CD8 single depleted animals (CD8-single depleted animals
showed enrichment for "cholesterol homeostasis" (included in **Extended Data 9** and **Rebuttal
Figure 54**). Furthermore, we would like to draw attention to a previous study (Wolf et al., 2014),
in which NKT-cells were responsible for metabolic changes and CD8 T-cells driving hepatic
damage. We think, that the lack of liver cancer incidence is a result of CD8 depletion and a
reduction of a pro-tumorigenic environment - e.g. including pro-tumorigenic TNF signaling,
which is similarly enriched (TNF signaling via NFkB) in CD-HFD-fed control animals (NES(CD8
depletion vs control)= -1.6718) and NES(CD8/NK1.1 co-depletion vs control)= -1.6538)
(**Extended Data 8 and 9** and **Rebuttal Figure 31**,). These data were also corroborated by
the analyses of the ICF signature which is strongly abrogated upon CD8 T cells depletion.

Thus, we dissected the role of NK1.1 cells in greater detail by including the GSEA analysis of
RNA-seq data comparing CD8-depleted and CD8/NK1.1 co-depleted animals. Furthermore,
we improved cross-referencing to the co-submitted study Dudek et al. to highlight, that CD8 T-
cells are driving hepatocarcinogenesis.

In line, together with Dudek et al. we generated new data using mouse strains with impaired
NKT cells - namely $J\alpha 18^{-/-}$ and $CD1d^{-/-}$ - under NASH-inducing diet. Both genetic knockout
mouse models develop NASH (including systemic obesity, fibrosis, ALT) and NASH-induced
hepatocarcinogenesis similar to WT control animals at 12-months diet-feeding. These data
argue against an essential role of NKT-cells to drive hepatocarcinogenesis at this time-point.

a

**Rebuttal Figure 54**

(a) Gene set enrichment analysis of RNA sequencing data of hepatic tissue comparing or CD-
 HFD + 8 weeks treatment of α -CD8 fed mice with CD-HFD + co-depletion of α -CD8/NK1.1 of
 12 months ND, CD-HFD, CD-HFD + 8 weeks treatment of α -CD8 fed or CD-HFD + co-
 depletion of α -CD8/NK1.1 (n= 5 mice/group).

- Sentence 289-292 is unclear.

We thank Referee #2 for highlighting the imprecise description and have now improved this in
 the main text of the revised manuscript. The sentence now reads as follows: "Next, we
 investigated the mechanisms underlying the increased occurrence of liver cancer
 incidence/liver tumor formation associated with anti-PD-1 treatment in the context of NASH."

- When discussing GSEA, the authors frequently use the wording 'reduced enrichment (e.g.
 line 241)' when talking about enrichment in the opposite phenotype. This is incorrect, as the
 absolute amount of enrichment is often similar just, as mentioned, in the opposite direction.

We thank Referee #2 for highlighting this imprecise description. We altered this in the revised
 manuscript. The changes read now as follows e.g.: "Gene set enrichment analysis (GSEA) of
 RNA sequencing data from whole liver tissue of CD8⁺ depleted mice revealed enrichment for
 DNA repair, oxidative phosphorylation, complement, and TNF signaling compared to CD-HFD-
 fed control)".

Referee #3 (Remarks to the Author):

This full article manuscript is novel, and the experimentation to support the conclusions is
exhaustive and solid for the most part. In essence, the findings indicate that, in NASH livers,
there is an accumulation/expansion of a pathogenic CD8 T-cell population that expresses PD-
1 and exacerbates NASH pathology and fosters hepatocellular carcinogenesis and
progression. The inflammatory and tissue-damaging functions of this pathogenic CD8 T-cells
are repressed by PD-1 blockade that is common clinical practice for second-line treatment of
advanced HCC and is under clinical trials for earlier stages of the disease. In fact, PD-L1
blockade plus anti-VEGF will soon become the standard of treatment for advanced HCC in
first line. According to the findings in this paper upon PD-1 blockade, authors document an
exacerbation of carcinogenesis and liver damage that questions the indication of PD-1
blockade in NASH-associated liver cancer. A balanced presentation of preclinical and
supportive clinical results in patient specimens very much enhances the significance of this
study.

We thank Referee #3 for the positive feedback and the statement that our study is “novel, and
the experimentation to support the conclusions is exhaustive and solid for the most part” . We
would like to address his/her concerns in the following section point-by-point by presenting
new experimental data sets experiments, rephrasing, and re-analysis of the underlying data-
sets.

Questions and comments:

1. TNF seems to be an actionable therapeutic target for the observed harmful effects of this
CD8 T-cell population. It would be interesting to know if TNF could be blocked preserving anti-
cancer immunity (especially under checkpoint inhibition therapy) but preventing tissue damage
and carcinogenesis promotion.

We thank Referee #3 for raising this important concern and thus have performed anti-TNF
with/without anti-PD-1-related immunotherapy in the context of NASH/HCC. Anti-TNF
treatment alone - without PD1-targeted immunotherapy - leads to liver cancer formation
comparable to control-treated CD-HFD-fed mice.

However, anti-TNF treatment in the context of PD1-targeted immunotherapy leads to a
significant reduction of tumor incidence (tumor incidence(anti-PD-1)= 75% vs tumor
incidence(anti-TNF/anti-PD-1)= 25%, p= 0.0024), liver damage (ALT(anti-PD-1)= 381.6 U/L vs

ALT(anti-TNF/anti-PD-1)= 250 U/L, $p= 0.0072$) and NAFLD-activity score (NAS(anti-PD-1)=
5.875 vs NAS (anti-TNF/anti-PD-1)= 3.1, $p= <0.0001$), when compared to anti-PD1 treated
CD-HFD-fed mice alone. This indicates that TNF exerts key functions of the observed adverse
effects of PD1-targeted immunotherapy, namely contributing to increased
hepatocarcinogenesis (included in **Figure 4, Extended Data 20 and 21** and **Rebuttal Figure**
**55-57**).

Moreover, the combination of anti-PD1 therapy with CD8-T cell depleting antibodies fully
eliminated the adverse, NAS increasing and pro-carcinogenic effects of CD8+ T-cells. These
data emphasize that CD8+ T-cells are a major cell population mediating increased
hepatocarcinogenesis through a TNF-dependent mechanism upon PD1-targeted
immunotherapy (included in **Figure 4, Extended Data 20 and 21** and **Rebuttal Figure 55-57**).
On one hand, the mechanisms could be executed by CD8 T-cell derived TNF itself or by
mechanisms that depend on TNF-signaling on other cells (e.g. myeloid cells). For example,
we see a drastic reduction of myeloid attracting chemokines (MCP-1, CCL3, CCL4, MIP-2) but
also cytokines of liver inflammation (e.g. IL-17A, IL-10, IL-13, IL-33), all cytokines/molecules
which might fuel liver inflammation and thus hepatocarcinogenesis in PD-1-targeted
immunotherapy in NASH mice.

Importantly, comparing mouse-human of CD8+ T-cells isolated from liver tissue of NASH mice
or patients through classical flow cytometry, CYTOF, and on scRNA-seq level we identified
similar populations and transcriptional activation of CD8+ PD1+ in a total of three independent
center patient cohorts (included in **Figure 5, Extended Data 25-27** and **Rebuttal Figure 58-**
**61**). These data indicate that results obtained and hypotheses built from the preclinical NASH
model are relevant for human disease and are in line with published results, where TNF
blockade uncouples mediated toxicity in dual CTLA-4 and PD-1 immunotherapy (Perez-Ruiz
et al., 2019).

Rebuttal Figure 55

(a) ScRNA-seq analysis of hepatic $TCR\beta^+$ cells of 12 months CD-HFD + IgG or CD-HFD-fed mice + 8 weeks treatment by α-PD-1 or α-CD8 antibodies (n= 3 mice/group). (b) Selected marker expression in hepatic CD8⁺ T-cells by scRNA-seq comparing CD8⁺ with CD8⁺PD-1⁺ T-cells of 12 months CD-HFD + IgG or CD-HFD-fed mice + 8 weeks treatment by α-PD-1 antibodies (n= 3 mice/group). (c) Average UMI comparison of hepatic CD8⁺PD-1⁺ T-cells of 12 months CD-HFD + IgG or CD-HFD-fed mice + 8 weeks treatment by α-PD-1 antibodies (n=

3 mice/group). (d) RNA velocity analyses of scRNA-seq data showing expression and (e)
 correlation of expression along the latent-time of selected genes along the latent-time (n= 3
 mice/group). Root cells: yellow cells indicate root cells, blue cells indicate cells farthest away
 from root by RNA velocity. End points: yellow cells indicate end point cells, blue cells indicate
 cells farthest away from defined end point cells by RNA velocity. Latent time: pseudo-time by
 RNA velocity, dark color indicate start of RNA velocity, yellow color indicate end point of latent
 time. RNA velocity flow: Blue cluster defined as start point, orange cluster as intermediate,
 green cluster as end point. Arrows indicate trajectory of cells. (f) PCA plot of hepatic CD8+ or
 CD8+PD-1+ T-cells sorted TCRβ+ cells by mass spectrometry of 12 months ND, CD-HFD or
 CD-HFD-fed mice + 8 weeks treatment by α-PD-1 antibodies (CD8+: ND n= 6 mice, CD-HFD
 + IgG n= 5 mice; CD-HFD + α-PD-1 n= 6 mice; CD8+PD-1+: ND n= 4 mice, CD-HFD + IgG n=
 6 mice; CD-HFD + α-PD-1 n= 6 mice). (g) UMAP representation showing the FlowSOM-guided
 clustering, heatmap showing the median marker expression, and (h) quantification of hepatic
 CD8+ T-cells of 12 months ND, CD-HFD + IgG or CD-HFD-fed mice + 8 weeks treatment by
 α-PD-1 antibodies (ND n= 4 mice; CD-HFD + IgG n= 8 mice; CD-HFD + α-PD-1 n= 6 mice). (i)
 Quantification of CellCNN analyzed flow cytometry data of hepatic CD8+ T-cells of 12 months
 CD-HFD + IgG or CD-HFD-fed mice + 8 weeks treatment by α-PD-1 antibodies (CD-HFD +
 IgG n= 6 mice; CD-HFD + α-PD-1 n= 4 mice). (j) UMAP representation showing the FlowSOM-
 guided clustering, the expression intensity of the indicated marker and heatmap showing the
 median marker expression of flow cytometry data of hepatic CD8+PD-1+ T-cells of 12 months
 ND, CD-HFD or CD-HFD-fed mice + 8 weeks treatment by α-PD-1 antibodies (ND n= 6 mice;
 CD-HFD n= 5 mice; CD-HFD + α-PD-1 n= 6 mice). (k) ALT and (l) NAS evaluation of 12 months
 ND, CD-HFD, CD-HFD-fed mice + 8 weeks treatment by α-PD-1, α-PD-1/α-CD8, α-TNF, or α-
 PD-1/α-TNF antibodies (ND n= 30 mice; CD-HFD n= 47 mice; CD-HFD + α-PD-1 n= 35 mice;
 CD-HFD + α-PD-1/α-CD8 n= 9 mice; CD-HFD + α-TNF n= 10 mice; CD-HFD + α-PD-1/α-TNF
 n= 11 mice). (m) Quantification of hepatic CD8+PD-1+CXCR6+ T-cells ND, CD-HFD, CD-
 HFD-fed mice + 8 weeks treatment by α-PD-1, α-PD-1/α-CD8, α-TNF, α-PD-1/α-TNF, α-CD4,
 or α-PD-1/α-CD4 antibodies (ND n= 30 mice; CD-HFD n= 47 mice; CD-HFD + α-PD-1 n= 35
 mice; CD-HFD + α-PD-1/α-CD8 n= 9 mice; CD-HFD + α-TNF n= 10 mice; CD-HFD + α-PD-
 1/α-TNF n= 11 mice); CD-HFD + α-CD4 n= 8 mice; CD-HFD + α-PD-1/α-CD4 n= 8 mice). (n)
 Quantification of tumor incidence of 12 months CD-HFD or CD-HFD-fed mice + 8 weeks
 treatment by α-CD8, α-CD8/NK1.1, α-PD-1, α-PD-1/α-CD8, α-TNF, α-PD-1/α-TNF, α-CD4, or
 α-PD-1/α-CD4 antibodies (tumor incidence: CD-HFD n= 32 tumors/lesions in 87 mice; CD-
 HFD + α-CD8 n= 2 tumors/lesions in 31 mice; CD-HFD + α-CD8/NK1.1 n= 0 tumors/lesions in
 6 mice; CD-HFD + α-PD-1 n= 33 tumors/lesions in 44 mice; CD-HFD + α-PD-1/α-CD8 n= 2
 tumors/lesions in 9 mice; CD-HFD + α-TNF n= 3 tumors/lesions in 10 mice; CD-HFD + α-PD-
 1/α-TNF n= 3 tumors/lesions in 11 mice); CD-HFD + α-CD4 n= 3 tumors/lesions in 9 mice; CD-
 HFD + α-PD-1/α-CD4 n= 8 tumors/lesions in 9 mice).

Research for a Life without Cancer

Rebuttal Figure 56

(a) Body weight, AST, and histological evaluation by (b) Sirius red, CD4, CD8, PD-1, PD-L1,
F4/80, MHC-II and (c) staining of ND, CD-HFD, or CD-HFD-fed mice + 8 weeks treatment by
α -PD-1, α -PD-1/ α -CD8, α -TNF, α -PD-1/ α -TNF antibodies (body weight: ND n= 16 mice; CD-
HFD n= 29 mice; CD-HFD + α -PD-1 n= 23 mice; CD-HFD + α -PD-1/ α -CD8 n= 9 mice; CD-
HFD + α -TNF n= 10 mice; CD-HFD + α -PD-1/ α -TNF n= 11 mice; AST: body weight: ND n= 30
mice; CD-HFD n= 40 mice; CD-HFD + α -PD-1 n= 30 mice; CD-HFD + α -PD-1/ α -CD8 n= 9
mice; CD-HFD + α -TNF n= 10 mice; CD-HFD + α -PD-1/ α -TNF n= 11 mice; Sirius red: ND n=
11 mice; CD-HFD n= 12 mice; CD-HFD + α -PD-1 n= 12 mice; CD-HFD + α -PD-1/ α -CD8 n= 9
mice; CD-HFD + α -TNF n= 10 mice; CD-HFD + α -PD-1/ α -TNF n= 11 mice; CD4: ND n= 10
mice; CD-HFD n= 11 mice; CD-HFD + α -PD-1 n= 14 mice; CD-HFD + α -PD-1/ α -CD8 n= 9
mice; CD-HFD + α -TNF n= 10 mice; CD-HFD + α -PD-1/ α -TNF n= 11 mice; CD8: ND n= 10
mice; CD-HFD n= 12 mice; CD-HFD + α -PD-1 n= 14 mice; CD-HFD + α -PD-1 n= 14 mice; CD-
HFD + α -PD-1/ α -CD8 n= 9 mice; CD-HFD + α -TNF n= 10 mice; CD-HFD + α -PD-1/ α -TNF n=
11 mice; PD-1: ND n= 12 mice; CD-HFD n= 12 mice; CD-HFD + α -PD-1 n= 14 mice; CD-HFD
+ α -PD-1/ α -CD8 n= 8 mice; CD-HFD + α -TNF n= 10 mice; CD-HFD + α -PD-1/ α -TNF n= 10
mice; PD-L1: ND n= 10 mice; CD-HFD n= 11 mice; CD-HFD + α -PD-1 n= 14 mice; CD-HFD +
α -PD-1/ α -CD8 n= 9 mice; CD-HFD + α -TNF n= 10 mice; CD-HFD + α -PD-1/ α -TNF n= 11 mice;
F4/80: ND n= 11 mice; CD-HFD n= 12 mice; CD-HFD + α -PD-1 n= 14 mice; CD-HFD + α -PD-
1 n= 14 mice; CD-HFD + α -PD-1/ α -CD8 n= 9 mice; CD-HFD + α -TNF n= 10 mice; CD-HFD +
α -PD-1/ α -TNF n= 11 mice; MHC-II: ND n= 11 mice; CD-HFD n= 13 mice; CD-HFD + α -PD-1
n= 14 mice; CD-HFD + α -PD-1 n= 14 mice; CD-HFD + α -PD-1/ α -CD8 n= 9 mice; CD-HFD +
α -TNF n= 10 mice; CD-HFD + α -PD-1/ α -TNF n= 11 mice).

Rebuttal Figure 57

(a) Quantification of hepatic immune cell composition and (b) CD8+PD-1+TNF+ T-cells by flow cytometry of 12 months ND, CD-HFD, or CD-HFD-fed mice + 8 weeks treatment by α-PD-1,

Research for a Life without Cancer

α -PD-1/ α -CD8, α -TNF, α -PD-1/ α -TNF antibodies (Hepatic immune cell composition: ND n= 8
mice; CD-HFD n= 5 mice; CD-HFD + α -PD-1 n= 4 mice; CD-HFD + α -PD-1/ α -CD8 n= 9 mice;
CD-HFD + α -TNF n= 10 mice; CD-HFD + α -PD-1/ α -TNF n= 11 mice; CD8+PD-1+TNF+: ND
n= 8 mice; CD-HFD n= 5 mice; CD-HFD + α -PD-1 n= 3 mice; CD-HFD + α -PD-1/ α -CD8 n= 9
mice; CD-HFD + α -TNF n= 10 mice; CD-HFD + α -PD-1/ α -TNF n= 11 mice). (c) and (d)
multiplex ELISA of hepatic inflammation associated cytokines and (e) chemokines of 12
2243 months ND, CD-HFD or CD-HFD-fed mice + 8 weeks treatment by α -PD-1, α -PD-1/ α -CD8, α -
2244 TNF, α -PD-1/ α -TNF antibodies (ND n= 10 mice; CD-HFD n= 14 mice; CD-HFD + α -PD-1 n=
13 mice; CD-HFD + α -PD-1/ α -CD8 n= 9 mice; CD-HFD + α -TNF n= 10 mice; CD-HFD + α -PD-
1/ α -TNF n= 11 mice).

Research for a Life without Cancer

CD8+PD-1+CD103+ derived from hepatic biopsies of control, or NAFLD/NASH patients
(Supplementary Table 2: control n= 6 patients; NAFLD/NASH n= 11 patients) Populations:
CD8+ (violet), CD8+PD-1+CD103+ (red). (e) UMAP representation of CD3+ cells and analyses
of differential gene expression by scRNA-seq of control, or NAFLD/NASH patients (control n=
4 patients; NAFLD/NASH n= 7 patients). (f) Correlation of significant differentially expressed
genes in liver-derived CD8+PD-1+ compared to CD8+PD-1- T-cells subsets of 12 months CD-
HFD-fed mice and NAFLD/NASH patients (mouse: n= 3 mice; human: n= 3 patients). (g)
Velocity analyses of scRNA-seq data showing (h) expression, transcriptional activity, (i) gene
expression and (j) correlation of expression along the latent-time of selected genes along the
latent-time of patient-liver-derived CD8+ T-cells of control, or NAFLD/NASH patients in
comparison to mouse-liver-derived CD8+ T-cells (patients: NAFLD/NASH n= 3 patients;
mouse: n= 3 mice/group). Root cells: yellow cells indicate root cells, blue cells indicate cells
farthest away from the root by RNA velocity. End points: yellow cells indicate end point cells,
blue cells indicate cells farthest away from defined end point cells by RNA velocity. Latent time:
pseudo-time by RNA velocity, dark color indicate start of RNA velocity, yellow color indicate
end point of latent time. RNA velocity flow: Blue cluster defined as start point, orange cluster
as intermediate, green cluster as end point. Arrows indicate the trajectory of cells.

**Rebuttal Figure 59**

(a) Flow cytometry plot of FMO control, (b) quantification of patient-liver-derived PD-1+CD8+
 T-cells, and (c) quantification of CD4, CD8, $\gamma\delta$, NK and NKT cells healthy or NAFLD/NASH
 patients (Supplementary Table 1: healthy n= 8 patients; NAFLD/NASH n= 16 patients). (d)
 Analysis of randomly chosen CD45+ cells and (e) average marker expression of defined
 CD45+ subsets by flow cytometry derived from hepatic biopsies of control and NAFLD/NASH
 patients to define distinct marker expression (Supplementary Table 2: control n= 6 patients;
 NAFLD/NASH n= 11 patients). (f) Definition of cellular subsets, (g) relative quantification of
 defined cellular subsets of randomly chosen CD45+ cells, (h) polarization of CD8+ T-cells and
 (i) quantification of CD4+CD27+, or $\gamma\delta$ TCR+Eomes+, T-cells by flow cytometry derived from
 hepatic biopsies of healthy and NAFLD/NASH patients (Supplementary Table 2: control n= 6
 patients; NAFLD/NASH n= 11 patients).

 **Rebuttal Figure 60**

(a) tSNE representation, (b) marker expression, (c) average marker expression of defined T-
 cell subsets of patient-liver-derived T-cells analyzed by CyTOF of control and NAFLD/NASH
 patients (control n= 11 patients pooled in 3 analyses; NAFLD/NASH n= 16 patients pooled in
 5 analyses). (d) Composition, (e) HSNE representation of defined T-cell subsets and (f)
 quantification of CD8+CD103+PD-1+ cells of of patient-liver-derived T-cells analyzed by
 CyTOF of control and NAFLD/NASH patients (control n= 11 patients pooled in 3 analyses;
 NAFLD/NASH n= 16 patients pooled in 5 analyses).

**Rebuttal Figure 61**

(a) NAS and BMI of patients used for scRNA-seq analyses of patient-liver-derived T-cells of
 control and NAFLD/NASH patients (control n= 4 patients; NAFLD/NASH n= 7 patients). (b)
 UMAP representation, marker expression, (c) relative quantification and (d), (e), (f) polarization
 of defined T-cell subsets of defined T-cell subsets of patient-liver-derived T-cells by scRNA-
 seq of control and NAFLD/NASH patients (control n= 4 patients; NAFLD/NASH n= 7 patients).
 (g) Differential gene expression of CD4+PD-1+ vs CD4+ T-cells and (h) selected average
 marker expression in CD4+ and CD8+ T-cell subsets of by scRNA-seq of control and
 NAFL/NA2SH patients (control n= 4 patients; NAFLD/NASH n= 7 patients).

2. Would PD-L1 blockade enhance liver cancer and tissue damage as well? Which cells are
 expressing PD-L1 in the system? This becomes important given the recent approval of
 atezolizumab + bevacizumab.

We agree with Referee #3 for raising the point that dissection of anti-PD-L1-targeted
 immunotherapy is of major concern, especially in the light of the recent results of the
 IMBrave150 study. Data we have received from RNA in situ hybridization and
 immunohistochemistry indicate that PD-L1 is expressed with increased level over time – with
 progression of NASH disease (in mice and men). In summary, PDL1 staining in the preclinical
 model is mainly associated with inflammatory cells, positive cells can be observed in the
 sinusoidal space as well (included in **Extended Data 3, 20, 22** and **Rebuttal Figure 56, 62-
 64**). In humans, PDL1 positivity was observed in aggregates of inflammatory cells in the
 parenchyma and the portal tract area. Focally, positivity was also seen in sinusoidal lining cells
 (included in **Extended Data 28** and **Rebuttal Figure 62**).

The cells expressing PD-L1 in NASH-affected mice are mainly lymphocytes but also some
 parenchymal cells (see **Extended Data 3+7, 20+22** and **Rebuttal Figure 63**).

In line with the comment of Referee #3, we have also performed anti-PD-L1 targeted
 immunotherapy in mice with and without established liver cancer (included in **Extended Data
 7** and **Rebuttal Figure 63**). Results from these experiments indicate that similar to anti-PD1 -
 anti-PDL1-treatment does not induce an anti-cancer effect for NASH-induced HCC but induces
 - similar to anti-PD1 treatment - a pro-inflammatory and pro-carcinogenic effect (e.g. increased
 NAS, strong trend in increased hepatic CD8 abundance by IHC (p= 0.0546), cytokines like IL-
 21 and CCL3) (included **Extended Data 7+13** and **Rebuttal Figure 63, 65**). These data
 indicate, that in the preclinical NASH model both PD1 or PDL1-targeted immunotherapy
 induces adverse effects. This is corroborated by our increased, retrospective cohort HCC-
 patients of different etiologies under PD(L)1-targeted immunotherapy, in which multivariate
 analysis results in NAFLD/NASH being an independent negative factor for overall survival and

validated these results in a second cohort of 118 HCC-patients (included in **Figure 6** and
 **Rebuttal Figure 66**). Furthermore, we corroborated our hypothesis of non-viral (NASH-
 related) HCC being less responsive to immunotherapy by a meta-analysis including 1656
 patients of the three most important clinical trials, identifying immunotherapy vs control for viral
 HCC as favorable treatment (HR(viral)= 0.64), in contrast, non-viral-HCC showed less benefit
 (HR(non-viral)= 0.92) for immunotherapy (included in **Figure 6, Extended Data 30-32,**
 **Supplementary Table 9 and Rebuttal Figure 67, 68**)).

**Rebuttal Figure 62**

(a) Immunohistochemical staining and (b) quantification of hepatic PD-1, CD8 and CD4
 expressing cells of NAFLD and NASH patients in Supplementary Table 3 with varying stages
 of fibrosis (NAFLD n= 9 patients; NASH F1/0 n= 7 patients; NASH F2 n= 12 patients; NASH
 F3 n= 21 patients; NASH F4 n= 16 patients; CD4: NAFL n= 6 patients; NASH F1/0 n= 4
 patients; NASH F2 n= 8 patients; NASH F3 n= 17 patients; NASH F4 n= 9 patients).

Correlation analysis of PD-1 against fibrosis scoring according to Brunt by
immunohistochemical staining by RNA-sequencing (NAFLD/NASH n= 65 patients). A total of
1656 patients were included in all three randomized trials, and 985 patients received a
checkpoint inhibitor (Supplementary Table 7). (d) Immunohistochemical staining of PD-L1 in
patient-derived liver samples. Scale bar: 50 μ m.

Rebuttal Figure 63

(a) Quantification of hepatic PD-L1+ expression by RNA in situ hybridization of 6- or 12-months ND or CD-HFD-fed mice (6 months: ND n= 13 mice; CD-HFD n= 11 mice; 12 months: ND n=

Research for a Life without Cancer

7 mice; CD-HFD n= 7 mice). Scale bar: 100 μ m. (b) Quantification of hepatic PD-L1+
expression by immunohistochemistry of 12 months ND or CD-HFD fed mice (6 months: ND n=
4 mice; CD-HFD n= 8 mice). Scale bar: 100 μ m. (c) MRI pictures of liver of mice after 10
2362 months CD-HFD and 7 weeks later after assignment to CD-HFD or CD-HFD-fed mice + 7
2363 weeks treatment of α -PD-L1 (CD-HFD n= 6 mice; CD-HFD + α -PD-L1 n= 8 mice). Lines
indicate tumor nodule. Scale bar: 10 mm. (d) Macroscopy of liver of 12 months ND, CD-HFD
or CD-HFD-fed mice + 8 weeks treatment of α -PD-L1. Arrowheads indicate tumor/lesions.
Scale bar: 10 mm. (e) Body weight, ALT levels of 12 months ND, CD-HFD or CD-HFD-fed
mice + 8 weeks treatment of α -PD-L1 (Body weight, ALT, : ND n= 8 mice; CD-HFD n= 6 mice;
CD-HFD + α -PD-L1 n= 6 mice) (f) and (g) NAS evaluation by H&E, Fibrosis evaluation of Sirius
Red staining, quantification of CD8, PD-1 and PD-L1 staining of hepatic tissue by
immunohistochemistry of 12 months ND, CD-HFD or CD-HFD-fed mice + 8 weeks treatment
of α -PD-L1 (NAS: ND n= 7 mice; CD-HFD n= 6 mice; CD-HFD + α -PD-L1 n= 6 mice; Sirius
Red: ND n= 7 mice; CD-HFD n= 5 mice; CD-HFD + α -PD-L1 n= 6 mice ; CD8, : ND n= 5 mice;
CD-HFD n= 5 mice; CD-HFD + α -PD-L1 n= 5 mice; PD-1, PD-L1: ND n= 5 mice; CD-HFD n=
5 mice; CD-HFD + α -PD-L1 n= 6 mice). Scale bar: 100 μ m. (h) Tumor/Lesion incidence in CD-
HFD or CD-HFD-fed mice + 8 weeks treatment of α -PD-L1 (CD-HFD n= 19 tumors/lesions in
25 mice; CD-HFD + α -PD-L1 n= 7 tumors/lesions in 8 mice)

Rebuttal Figure 64

(a) Body weight, ALT, AST, NAS, and histological evaluation by (b) Sirius Red, CD4, CD8, PD-1, PD-L1, F4/80, MHC-II and (c) staining of ND, CD-HFD, or CD-HFD-fed mice + 8 weeks treatment by α-PD-1, α-CD4, α-PD-1/α-CD4 antibodies (body weight: ND n= 16 mice; CD-HFD n= 29 mice; CD-HFD + α-PD-1 n= 23 mice; CD-HFD + α-CD4 n= 9 mice; CD-HFD + α-PD-1/α-CD4 n= 9 mice; ALT ND n= 30 mice; CD-HFD n= 47 mice; CD-HFD + α-PD-1 n= 35 mice; CD-HFD + α-CD4 n= 9 mice; CD-HFD + α-PD-1/α-CD4 n= 9 mice; AST: ND n= 30 mice; CD-HFD n= 40 mice; CD-HFD + α-PD-1 n= 30 mice; CD-HFD + α-CD4 n= 9 mice; CD-HFD + α-PD-1/α-CD4 n= 9 mice; NAS: ND n= 31 mice; CD-HFD n= 46 mice; CD-HFD + α-PD-1 n= 40 mice; CD-HFD + α-CD4 n= 8 mice; CD-HFD + α-PD-1/α-CD4 n= 8 mice; Sirius red: ND n= 11 mice; CD-HFD n= 12 mice; CD-HFD + α-PD-1 n= 12 mice; CD-HFD + α-CD4 n= 9 mice; CD-HFD + α-PD-1/α-CD4 n= 9 mice; CD4: ND n= 10 mice; CD-HFD n= 11 mice; CD-HFD + α-PD-

2390 1 n= 14 mice; CD-HFD + α -CD4 n= 10 mice; CD-HFD + α -PD-1/ α -CD4 n= 11 mice; CD8: ND
n= 10 mice; CD-HFD n= 12 mice; CD-HFD + α -PD-1 n= 14 mice; CD-HFD + α -CD4 n= 9 mice;
CD-HFD + α -PD-1/ α -CD4 n= 9 mice; PD-1: ND n= 13 mice; CD-HFD n= 12 mice; CD-HFD +
α -PD-1 n= 14 mice; CD-HFD + α -CD4 n= 9 mice; CD-HFD + α -PD-1/ α -CD4 n= 9 mice; PD-L1:
ND n= 12 mice; CD-HFD n= 12 mice; CD-HFD + α -PD-1 n= 14 mice; CD-HFD + α -CD4 n= 9
mice; CD-HFD + α -PD-1/ α -CD4 n= 9 mice; F4/80: ND n= 11 mice; CD-HFD n= 13 mice; CD-
HFD + α -PD-1 n= 14 mice; CD-HFD + α -CD4 n= 8 mice; CD-HFD + α -PD-1/ α -CD4 n= 9 mice;
MHC-II: ND n= 11 mice; CD-HFD n= 13 mice; CD-HFD + α -PD-1 n= 14 mice; CD-HFD + α -
PD-1 n= 14 mice; CD-HFD + α -CD4 n= 9 mice; CD-HFD + α -PD-1/ α -CD4 n= 9 mice). Scale
2399 bar: 100 μ m.

**Rebuttal Figure 65**
(a) and (b) multiplex ELISA concentrations of hepatic inflammation-associated cytokines and
(c) chemokines of 12 months ND, CD-HFD, CD-HFD-fed mice + 8 weeks treatment of α -PD-1
or CD-HFD (ND n= 10 mice; CD-HFD n= 14 mice; CD-HFD + α -PD-1 n= 13 mice).

Rebuttal Figure 66

(a) Nonalcoholic fatty liver disease (NAFLD) is associated with a worse outcome in patients with hepatocellular carcinoma (HCC) treated with PD-(L)1-targeted immunotherapy. A total of 130 patients with advanced HCC received PD-(L)1-targeted immunotherapy (Supplementary Table 8). Kaplan-Meier curve display overall survival of patients with NAFLD vs. those with any other etiology; all 130 patients were included in these survival analyses (NAFLD n=13, any other etiology n=117). (b) Validation cohort of patients with HCC treated with PD-(L)1-targeted immunotherapy. A total of 1180 patients with advanced HCC received PD-(L)1-targeted immunotherapy (Supplementary Table 10). Kaplan-Meier curve display overall survival of patients with NAFLD vs. those with any other etiology; all 118 patients were included in these survival analyses (NAFLD n=11, any other etiology n=107). (c) Multivariate analysis of prognostic factors in HCC patients treated with anti-PD-(L)1-based immunotherapy

represents the 95% confidence interval (CI). The diamonds represent the estimated overall
effect based on the meta-analysis random effect of all trials.

A total of 1243 patients were included in two first-line trials comparing PD-1 or PD-L1 targeted
immunotherapy to sorafenib. 707 patients received an immune checkpoint inhibitor (either PD-
1 or anti-PD-1). (c) HCV and HBV were pooled into a separate category, termed “viral”, and a
subsequent meta-analysis comparing viral (n=754) and non-viral (n=489), mostly NASH and
alcohol intake, was performed. A subgroup analysis studying the specific effects of non-viral
etiologies (n=489) on the magnitude of effect of immunotherapy are presented, when
compared to (d) HBV (n=473) or (e) HCV (n=281). Hazard ratios for each trial are represented
by squares, the size of the square represents the weight of the trial in the meta-analysis. The
horizontal line crossing the square represents the 95% confidence interval (CI). The diamonds
represent the estimated overall effect based on the meta-analysis random effect of all trials.

3. Results on NASH in human samples are compelling and supportive of the relevance of the
findings. It would be interesting to know in such livers which cells express PD-L1.

We thank Referee #3 for highlighting this important aspect of our data – and have consequently
performed PD-L1 expression analyses by immunohistochemistry in human specimens
described in the previous point raised by Referee #3. Although analysis by bulk RNA-seq of
liver tissue indicates a decrease of PDL1/CD274 expression with the severity of NASH
pathology, immunohistochemistry indicates an increase of PDL1 positivity with the severity of
NASH pathology. PDL1 positivity was observed in aggregates of inflammatory cells in the
parenchyma and the portal tract area. Focally, positivity was also seen in sinusoidal lining cells
(included in **Extended Data 28** and **Rebuttal Figure 62d**).

4. What do you think is the fibrogenic factor/s promoted by pathogenic CD8 cells? Any
candidates from the extensive transcriptomic analyses?

We thank Referee #3 for pointing out, that the fibrogenic factor is of major concern to prevent
HCC in subgroups of NASH patients. Our transcriptomic data-set has so far not pointed
towards specific fibrogenic factors, indicating that the chronic inflammatory environment
correlating with pathogenic CD8 cells drives fibrosis in our mice. To strengthen this hypothesis
AI-based analyses of a broad range of parameters of our 12 months CDHFD-fed mice
revealed, that Sirius red staining correlates negatively within CD8 depleted animals, indicating
that CD8-associated inflammation or CD8-dependent mechanisms might be functionally linked
with fibrosis (included in **Figure 1**, **Extended Data 4 and 24** and **Rebuttal Figure 69, 70**).
Moreover, in 12 months CDHFD-fed mice fibrosis correlated positively with CD8 T-cells
abundance, CD8+PD-1+ (%CD8), pDC+MHCII+ polarization, and hepatic TNF concentration.
Therefore, we cannot point out one specific factor driving fibrosis on pathogenic CD8 cells.

Research for a Life without Cancer

a

b

**Rebuttal Figure 69**

(a) UMAP representation of 63 parameters (serology, flow cytometry, histology) indicating
 NASH pathology severity measured of 12 months ND or CD-HFD fed mice (ND n= 22 mice;
 CD-HFD n= 31 mice). (b) Data gathered from hepatic tissue analyses was binary correlated
 with each other of 6- or 12-months ND or CD-HFD fed mice (ND n= 47 mice; CD-HFD n= 72
 mice).

Research for a Life without Cancer

**Rebuttal Figure 70**

(a) UMAP representation of 63 parameters (serology, flow cytometry, histology) and (b)
selected display of analyzed parameters indicating NASH pathology severity measured of 12
2491 months ND or CD-HFD fed mice (ND n= 22 mice; CD-HFD n= 31 mice; CD-HFD + α -PD-1 n=
41 mice; CD-HFD + α -PD-L1 n= 6 mice; CD-HFD + α -CD8 n= 24 mice; CD-HFD + α -
CD8/NK1.1 n= 6 mice; CD-HFD + α -PD-1/ α -CD8 n= 9 mice; CD-HFD + α -TNF n= 10 mice;
CD-HFD + α -PD-1/ α -TNF n= 11 mice; CD-HFD + α -CD4 n= 9 mice; CD-HFD + α -PD-1/ α -CD4
n= 9 mice). (c) Data gathered from hepatic tissue analyses was binary correlated with each
other of 6- or 12-months ND, CD-HFD or CD-HFD + 8 weeks treatment of α -CD8, α -CD8/ α -
NK1.1; α -PD-1, α -PD-1/ α -CD8, α -TNF, α -PD-1/ α -TNF, α -CD4, or α -PD-1/ α -CD4 fed mice (ND
n= 47 mice; CD-HFD n= 72 mice; CD-HFD + α -PD-1 n= 41 mice; CD-HFD + α -PD-L1 n= 6
mice; CD-HFD + α -CD8 n= 29 mice; CD-HFD + α -CD8/NK1.1 n= 6 mice; CD-HFD + α -PD-1/ α -
CD8 n= 9 mice; CD-HFD + α -TNF n= 10 mice; CD-HFD + α -PD-1/ α -TNF n= 11 mice; CD-HFD
+ α -CD4 n= 9 mice; CD-HFD + α -PD-1/ α -CD4 n= 9 mice).

5. Are Kupffer cells involved in the CD8-dependent pathogenesis mechanisms?

We thank Referee #3 for asking the important question about Kupffer cells (KC). A study
(Malehmir et al., 2019) reports, that KCs have a crucial role in the pathogenesis of NASH, but
activation of monocytes and myeloid-derived macrophages correlates with disease
progression. Data presented in **Extended Data 8 and 11** cannot exclude KC-dependent
mechanisms, however, they seem to have a minor role, especially concerning the co-submitted
manuscript Dudek et al. in which CD8+ cells drive pathogenesis in KC-independent ways.

We have further performed analyses on how KC correlate with varying degrees of inflammation
induced by our antibody treatments (anti-CD8, anti-CD8/anti-NK1.1, anti-CD8/anti-PD1, anti-
PD1, anti-PDL1, anti-TNF, anti-TNF/anti-PD1, and as control experiment anti-CD4 and anti-
CD4/anti-PD1) by our AI-based analysis approach (included in **Figure 1, Extended Data 4,**
**20-24** and **Rebuttal Figure 56, 57, 64, 69, 70**). Under baseline conditions (12 months CD-
HFD-fed animals receiving no treatments) KC abundance does not correlate with any
serological or histological marker, but KC activation (measured by MHCII+ polarization)
correlates strongly with tumor size and IL-21 (included in **Extended Data 4** and **Rebuttal**
**Figure 69**). However, when applying treatments (e.g. PD-1-targeted immunotherapy) KC
correlates with treatments as well as activation of hepatic KC (measured by MHCII+) correlate
positively with CD8+PD-1+ (%CD8), Sirius Red staining, tumor incidence, tumor number,
tumor size, and IL-21 (included in **Extended Figure 24** and **Rebuttal Figure 70**).

In summary, we believe in line with our own study (Malehmir et al., 2019) and recent literature
(Remmerie et al., 2020) that Kupffer cells are an important cell type on whose basis not
inflammatory pathologies are initiated and maintained, but also in end-stage disease fresh
KC/KC-like cells (attracted by cytokines e.g. MCP-1, CCL3, MIP-2 (included in **Extended 2,**

**13, 21 and 23 and Rebuttal Figure 57, 65, 71, 72)** activation might be detrimental as indicated
 by our correlation analysis. – laying the ground for adaptive immune cell reactions.

**Rebuttal Figure 71**

(a) Quantification of hepatic immune cell composition and (b) CD8+PD-1+TNF+ T-cells by flow
 cytometry of 12 months ND, CD-HFD, or CD-HFD-fed mice + 8 weeks treatment by α -PD-1,
 α -CD4, α -PD-1/ α -CD4 antibodies (Hepatic immune cell composition: ND n= 8 mice; CD-HFD
 n= 5 mice; CD-HFD + α -PD-1 n= 4 mice; CD-HFD + α -CD4 n= 8 mice; CD-HFD + α -PD-1/ α -
 CD4 n= 8 mice; CD8+PD-1+TNF+: ND n= 8 mice; CD-HFD n= 5 mice; CD-HFD + α -PD-1 n=
 3 mice; CD-HFD + α -CD4 n= 8 mice; CD-HFD + α -PD-1/ α -CD4 n= 8 mice). (c) and (d) multiplex
 ELISA of hepatic inflammation associated cytokines and (e) chemokines of 12 months ND,
 CD-HFD or CD-HFD-fed mice + 8 weeks treatment by α -PD-1, α -CD4, α -PD-1/ α -CD4
 antibodies (ND n= 10 mice; CD-HFD n= 14 mice; CD-HFD + α -PD-1 n= 13 mice; CD-HFD +
 α -CD4 n= 9 mice; CD-HFD + α -PD-1/ α -CD4 n= 9 mice).

 **Rebuttal Figure 72**

(a) and (b) multiplex ELISA of hepatic inflammation associated cytokines and (c) chemokines
 of 12 months ND or CD-HFD-fed mice (ND n= 10 mice; CD-HFD n= 14 mice). All data are
 shown as mean \pm SEM. All data were analyzed by two-tailed Student t test.

6. Obesity and response to PD-1 associations have been reported (PMID: 30420753 and
PMID: 30813970). According to these studies, obesity relates to T-cell dysfunction that PD-1
blockade derepresses and results in better responsiveness. The models of NASH should suffer
overweight as well as perhaps the patients in the reported series. This point should be
addressed if possible and at least discussed. Authors may gain insight with their comparisons
of the models with and without choline in the diet. As a potential consequence, would it be the
case that in HCC patients, obese patients respond worse to treatment contrary to other
indications? Of clinical note, advanced HCC patients frequently experience cachexia but
perhaps less frequently so those with presumed or documented NASH etiology.

We thank Referee #3 for highlighting these important studies of checkpoint inhibition in the
frame of obese cancer patients. (Wang et al., 2018) shows - similar to our study - convincingly
that increased PD-1 expression is a hallmark of diet-induced obesity, thus we cite the study in
our introduction and improved cross-referencing in our discussion. Potential differences in the
outcome of PD-1-targeted immunotherapy might be a consequence of the use of obesity-but,
not NASH-inducing high-fat diet, which we show is crucial to induce hallmarks of NASH by
comparing HFD with CD-HFD in **Extended Data 1**. Moreover, we would like to draw attention
to the different cancer entities, which potentially affect immunotherapy-responsiveness. Wang
et al. use subcutaneous tumor models of lung carcinoma (3LL) and melanoma (B16-F0), but
not spontaneous developed liver cancer in a chronic inflammatory metabolically challenged
hepatic microenvironment. Notably, obese animals have bigger tumor-volumes and anti-PD-1
reactive animals do not control tumor-volume to a smaller absolute tumor-volume compared
to non-obese controls (Figures 2 and 4 in (Wang et al., 2018)).

The second study of (Cortellini et al., 2019) corroborates the preclinical data of (Wang et al.,
2018) nicely in lung-, renal-carcinoma, or melanoma patients, but not liver cancer. No grading
of obese patients was performed (e.g. we report in Supplementary Table 1: healthy/control
liver, NAFLD/NASH), which we show in **Figure 5** is crucial for hepatic CD8 and PD-1
abundance. Supporting our manuscript, (Cortellini et al., 2019) report significantly more
likelihood of obese patients experiencing immune-related-Adverse-Effects (irAEs) “compared
to non-overweight patients (55.6% vs. 25.2%, $p < 0.0001$)”. Unfortunately, no subgroup
analyses about differences of hepatic irAEs between obese/non-obese patients are shown.

We included the study of (Cortellini et al., 2019) in our introduction and discussion.

Our NAFLD/NASH cohort without immunotherapy treatment indicate a correlation of BMI with
CD8+PD-1+ T-cells (included in **Figure 5** and **Rebuttal Figure 58**). In our conducted meta-
analysis, no BMIs were reported, thus statements about treatment response remain

hypothetical. Furthermore, our retrospective HCC-patient cohort under PD(L)1 immunotherapy
was too small for subgroup analysis, however, there was no significant difference in BMI
between NAFLD/NASH-HCC and other etiologies-HCC patients, indicative of obesity (included
in **Supplementary Table 7**).

7. The retrospective series of patients with advanced HCC treated cannot be considered
conclusive at this point and only hypothesis-generating. The wording there needs to be
carefully down-toned.

We agree with Referee #3, that the presented retrospective PD-(L)1 targeted immunotherapy
treated NAFLD/NASH-associated HCC cohort – although unique for Europe and treatment not
officially licensed and thus reimbursement - is still small, although we would like to point out,
that prominent trends or effects can be seen in small retrospective cohorts as well.

Thus, our analyses of BCLC-C NAFLD/NASH-HCC vs other-etiologies-HCC patients
indicated, that NAFLD/NASH-HCC has significantly reduced overall survival compared to
other-etiologies-HCC in this small retrospective cohort, which we validated in a second cohort
of 118 HCC patients under immunotherapy (included in **Figure 6** and **Rebuttal Figure 66**). Of
note, multivariate analyses identified NAFLD/NASH as an independent factor for treatment
response and thus identifying NAFLD/NASH as a negative predictor for HCC immunotherapy
(included in **Supplementary Table 9** and **Rebuttal Figure 66**).

We corroborated our hypothesis of non-viral (NASH-related) HCC being less responsive to
immunotherapy by a meta-analysis including 1656 patients of the three most important clinical
trials (IMbrave 150; Checkmate 459; Keynote 240), identifying immunotherapy vs control for
viral HCC as favorable treatment ($HR(viral) = 0.64$), in contrast, non-viral-HCC showed less
benefit ($HR(non-viral) = 0.92$) for immunotherapy (included in **Figure 6**, **Extended Data 30-32**,
**Supplementary Table 7** and **Rebuttal Figure 67-68**)).

Based on these data we want to point out that it is - as indicated by Referee#3 - of the highest
importance to us to specifically define/tone down appropriately the message of our manuscript:
Our manuscript does not indicate that immunotherapy is not beneficial for HCC patients at all.
Our manuscript rather demonstrates that HCC patients with viral etiologies do respond well
and achieve survival benefits - however, that patients with non-viral etiologies (e.g. NASH) do
not achieve a significant outcome benefit.

We thus propose to stratify HCC patients who are very likely to profit from immunotherapy and
strengthen the argumentation to use immunotherapy in specific cohorts of HCC patients. We

agree with Referee#1 that this information needs to be articulated in the paper appropriately
not to deliver wrong messages but to be very specific.

We truly believe that these are important clinical data, also providing the basis to test our
hypotheses in prospective studies on non-significantly beneficial effects in terms of OS for
immunotherapy in HCC patients with non-viral and NAFLD/NASH etiology, in particular.

Moreover, we toned down the conclusions of our retrospective cohort in the manuscript and
would like to point out, that bigger cohorts and prospective clinical trials are of utmost
importance for the scientific community.

8. An important message of this paper is that progression following PD-(L)1 treatment in NASH
patients could be the development of a second primary malignancy rather than from the same
one. Can this point be addressed in the models? Is multifocal cancer more common in those
cases? The more CD8 pathogenic T-cells in the infiltrate, the more multifocal the tumors?

We thank Referee #3 for asking this important question. In our opinion dissection of
primary/second primary malignancy is overstepping the limitation of the preclinical model,
indicated by the variability of immunohistochemical staining and by the similarity of genomic
aberrations (included in **Extended Data 16** and **Rebuttal Figure 73**).

We further have performed correlation analyses (e.g. CD8, PD-1, PD-L1, NAS, fibrosis, liver
damage, tumor size, and tumor load) to allow readers a more detailed description of the
presented data (included in **Figure 1**, **Extended Data 4+24** and **Rebuttal Figure 69, 70**).

a

b

**Rebuttal Figure 73**

(a) Quantification of genomic aberrations (b) by array comparative genomic hybridization
(aCGH) of tumor tissues of mice after 12 months on CD-HFD (n= 9) or 12 months on CD-HFD-
fed mice + 8 weeks treatment with α -PD-1 (n= 12).

9. The companion back to back paper shows more data on the physiology of the pathogenic
CD8 T-cells that I would otherwise ask to this article. Therefore, proper cross-reference of
those findings is needed at least in discussion.

We thank Referee #3 for highlighting the importance of the co-submitted paper Dudek et al.
and therefore, we improved cross-referencing in the discussion.

**Referee #4 (Remarks to the Author):**

This is an interesting and quite original study of the role of immunity in promoting liver cancer.
There are data from the mouse models presented which show that CD8+ T-cells can contribute
to the pathology of NASH and the risk of cancers. The implication is that checkpoint blockade
which can accentuate the function of CD8 populations can worsen disease. There are also
some human data which are fairly consistent with this idea. It is perhaps not surprising that
checkpoint inhibition might worsen an inflammatory condition, although inducing a cancer risk
is very interesting.

Overall the authors do a very good job in describing the cellular responses and the impact of
depletion/blockade. There seemed to be a bit of a gap around defining the mechanisms in
terms of how the CD8+ T-cell population induced cancer. Also it was somewhat unclear what
the specificity of these T-cells was and what was triggering their initial responsiveness in
NASH. So although a strong case is made for the pro-tumor role the actual pathways to
disease were less concrete.

We thank Referee #4 for appreciating our study's originality in shedding new light on the role
of immunity promoting liver cancer, with fairly consistent human data correlating with the
findings in the preclinical model.

We thank Referee #4 for pointing out the limitations of our study which has helped us to
increase the quality of our manuscript and address the respective points. We would like to
address the concerns of Referee #4 in the following section point-by-point by newly performed
experiments (addressing all questions raised in full), re-phrasing, re-analysis of the underlying
data-sets and would like to draw attention to the improved cross-referencing to the co-
submitted manuscript Dudek et al., which dissect the molecular and cellular mechanism of
CD8+ T-cell dependent pathogenesis in NASH.

Figure 1: There do not appear to be any iNKT-cells in the UMAP or tise plots – these are
discussed latter in the text. That seems a little surprising as they are quite dominant in the
mouse liver and have a clear transcriptional profile. Could the authors clarify where these cells
lie. It would be also useful to know whether other unconventional cell subsets including GD T-
cells and MAIT-cells are incorporated in this, although they are likely much rarer. The latter
may be relevant even if rare as they have been linked to liver fibrosis. The same questions
would also apply to the scRNAseq of the human samples.

We thank Referee #4 for raising this important point. We have now dissected mouse NK1.1+
cells in the revised version of our manuscript into NK1.1+TCRb+ as NKT and NK1.1+TCRb-
as NK cells (included in **Figure 1** and **Rebuttal Figure 74**). Similarly, we highlighted NKT-
cells, MAITs, and $\gamma\delta$ T-cells in our patient-derived hepatic lymphocytes analysis by flow
cytometry, newly performed scRNA-seq, and CYTOF analysis (included in **Figure 5**,
**Extended Data 25-27** and **Rebuttal Figure 74**).

We agree with Referee #4, that MAITs might be important and thus included quantification of
MAITs in our newly performed scRNA-seq and CYTOF analyses of patient-derived hepatic
lymphocytes. In these analyses, no change of relative abundance of MAITs was observed
when comparing control vs. NAFLD/NASH patients. Moreover, we would like to draw attention
to the co-submitted manuscript Dudek et al., which analyzed - together with us - CD-HFD-fed
Ja18^{-/-} and CD1d^{-/-} mice. The latter did not display significant changes in pathology compared
to CD-HFD-fed control mice at time points of established NASH.

We agree with Referee #4, that $\gamma\delta$ T-cells may be important, however in our mouse model
upon NASH establishment, we detected no difference in hepatic abundance of $\gamma\delta$ T-cells
between chow or CD-HFD-fed control mice (included in **Extended Data 3**). Furthermore, data
presented in **Figures 1 and 4** and **Extended Data 3** argue against a major direct contribution
of $\gamma\delta$ T-cells in the preclinical model at time points of 6 or 12 months of diet-feeding.

We agree that $\gamma\delta$ T-cells might be important in the pathogenesis of NASH and NASH to HCC
transition, however, e.g. rather in collaboration with CD8⁺ T cells, also in the context of PD1-
related immunotherapy.

In humans, our data is not conclusive in all experiments, e.g. our data indicate for $\gamma\delta$ T-cells, if
we compare: bulk RNA-seq indicates a reduced expression in severe NASH pathology of
EOMES, TRDC, and TRGC1 (included in **Extended Data 28** and **Rebuttal Figure 75, 76, 77**),
however, both flow cytometry cohorts and the scRNA-seq cohort indicate no change of either
$\gamma\delta$ ⁺ T-cells or $\gamma\delta$ ⁺ Eomes⁺ T-cells comparing control vs NAFLD/NASH patients (included in
**Extended Data 25, 27** and **Rebuttal Figure 75, 76**).

Corroborating the human flow cytometry data in our mouse model upon NASH establishment,
we detected no difference in hepatic abundance of $\gamma\delta$ T-cells between chow- or CD-HFD-fed
control mice. Furthermore, data presented in **Figures 1** and **Extended Data 3** argues against
the major contribution of $\gamma\delta$ T-cells in the mouse model of NASH. Here, we did not observe
significant differences in the “other leukocytes” subset. In the revised manuscript, we analyzed
$\gamma\delta$ -T-cells separately to strengthen the point, that these cells are not significantly changed upon
diet feeding (included in **Extended Data 3, 20-23** and **Rebuttal Figure 76a, 78, 79**).

Rebuttal Figure 74

(a) UMAP representation of 5000 randomly chosen CD45⁺ cells and quantification of hepatic immune cell composition by flow cytometry of 12 months ND or CD-HFD-fed mice (ND n= 4

mice; CD-HFD n= 8 mice). (b) UMAP representation of randomly chosen CD45⁺ cells and

(b) flow cytometry plots and quantification of CD8⁺PD-1⁺CD103⁺ derived from hepatic

biopsies of control, or NAFLD/NASH patients (Supplementary Table 2: control n= 6 patients;

NAFLD/NASH n= 11 patients). (c) UMAP representation of CD3⁺ cells by scRNA-seq of

control, or NAFLD/NASH patients (control n= 4 patients; NAFLD/NASH n= 7 patients). (d)
HSNE representation of defined T-cell subsets of patient-liver-derived T-cells analyzed by
CyTOF of control and NAFLD/NASH patients (control n= 11 patients pooled in 3 analyses;
NAFLD/NASH n= 16 patients pooled in 5 analyses).

**2736 Rebuttal Figure 75**

(a) Flow cytometry plot of FMO control, (b) quantification of patient-liver-derived PD-1+CD8+
T-cells, and (c) quantification of CD4, CD8, $\gamma\delta$, NK and NKT cells healthy or NAFLD/NASH
patients (Supplementary Table 1: healthy n= 8 patients; NAFLD/NASH n= 16 patients). (d)
Analysis of randomly chosen CD45+ cells and (e) average marker expression of defined
CD45+ subsets by flow cytometry derived from hepatic biopsies of control and NAFLD/NASH
patients to define distinct marker expression (Supplementary Table 2: control n= 6 patients;
NAFLD/NASH n= 11 patients). (f) Definition of cellular subsets, (g) relative quantification of
defined cellular subsets of randomly chosen CD45+ cells, (h) polarization of CD8+ T-cells and
(i) quantification of CD4+CD27+, or $\gamma\delta$ TCR+Eomes+, T-cells by flow cytometry derived from
hepatic biopsies of healthy and NAFLD/NASH patients (Supplementary Table 2: control n= 6
patients; NAFLD/NASH n= 11 patients).

Rebuttal Figure 76

(a) Hepatic abundance of TCR $\gamma\delta$ T-cells of 6 or 12 months ND or CD-HFD fed mice (6 months ND n = 8 mice; CD-HFD n = 6 mice; 12 months ND n = 8 mice; CD-HFD n = 6 mice).
 (b) NAS and BMI of patients used for scRNA-seq analyses of patient-liver-derived T-cells of control and NAFLD/NASH patients (control n = 4 patients; NAFLD/NASH n = 7 patients). (c) UMAP representation, marker expression, (d) relative quantification and (e), (f), (g) polarization of defined T-cell subsets of defined T-cell subsets of patient-liver-derived T-cells by scRNA-seq of control and NAFLD/NASH patients (control n = 4 patients; NAFLD/NASH n = 7 patients).

Rebuttal Figure 77

(a) RNA-sequencing data comparing NASH with varying fibrosis (F0 – F4 according to Brunt classification) normalized to NAFLD from a total of n= 206 NAFLD/NASH patients corrected for batch, gender and center

**Rebuttal Figure 78**

(a) Quantification of hepatic immune cell composition and (b) CD8+PD-1+TNF+ T-cells by flow
 cytometry of 12 months ND, CD-HFD, or CD-HFD-fed mice + 8 weeks treatment by α -PD-1,
 α -PD-1/ α -CD8, α -TNF, α -PD-1/ α -TNF antibodies (Hepatic immune cell composition: ND n= 8

mice; CD-HFD n= 5 mice; CD-HFD + α -PD-1 n= 4 mice; CD-HFD + α -PD-1/ α -CD8 n= 9 mice;
CD-HFD + α -TNF n= 10 mice; CD-HFD + α -PD-1/ α -TNF n= 11 mice; CD8+PD-1+TNF+: ND
n= 8 mice; CD-HFD n= 5 mice; CD-HFD + α -PD-1 n= 3 mice; CD-HFD + α -PD-1/ α -CD8 n= 9
mice; CD-HFD + α -TNF n= 10 mice; CD-HFD + α -PD-1/ α -TNF n= 11 mice). (c) and (d)
multiplex ELISA of hepatic inflammation associated cytokines and (e) chemokines of 12
2771 months ND, CD-HFD or CD-HFD-fed mice + 8 weeks treatment by α -PD-1, α -PD-1/ α -CD8, α -
2772 TNF, α -PD-1/ α -TNF antibodies (ND n= 10 mice; CD-HFD n= 14 mice; CD-HFD + α -PD-1 n=
13 mice; CD-HFD + α -PD-1/ α -CD8 n= 9 mice; CD-HFD + α -TNF n= 10 mice; CD-HFD + α -PD-
1/ α -TNF n= 11 mice).

Rebuttal Figure 79

(a) Quantification of hepatic immune cell composition and (b) CD8+PD-1+TNF+ T-cells by flow cytometry of 12 months ND, CD-HFD, or CD-HFD-fed mice + 8 weeks treatment by α-PD-1, α-CD4, α-PD-1/α-CD4 antibodies (Hepatic immune cell composition: ND n= 8 mice; CD-HFD n= 5 mice; CD-HFD + α-PD-1 n= 4 mice; CD-HFD + α-CD4 n= 8 mice; CD-HFD + α-PD-1/α-CD4 n= 8 mice; CD8+PD-1+TNF+: ND n= 8 mice; CD-HFD n= 5 mice; CD-HFD + α-PD-1 n=

3 mice; CD-HFD + α -CD4 n= 8 mice; CD-HFD + α -PD-1/ α -CD4 n= 8 mice). (c) and (d) multiplex
ELISA of hepatic inflammation associated cytokines and (e) chemokines of 12 months ND,
CD-HFD or CD-HFD-fed mice + 8 weeks treatment by α -PD-1, α -CD4, α -PD-1/ α -CD4
antibodies (ND n= 10 mice; CD-HFD n= 14 mice; CD-HFD + α -PD-1 n= 13 mice; CD-HFD +
α -CD4 n= 9 mice; CD-HFD + α -PD-1/ α -CD4 n= 9 mice).

Figure 1e: What are the p values on the right referencing? The difference in the PD1+
population does not appear to be significant. How valid is the PD1+ subset as a subcluster and
also what are the critical significant differences apart from elevated PD1 expression – some
justification for this early on would be helpful. Often PD1 expression is more of a gradient (even
within PD1+ cells) so a binary distinction needs a bit more justification. Does this group of cells
have distinct TCRs from the non-PD1 (or lower PD1) subset or are they the same population
with distinct expression? Some data on this would address the question about specificity –
although this would be better addressed by defining actual TCR-specific (or independent)
functionality.

We thank Referee #4 for raising important points about **Figure 1**. We have now improved our
manuscript by clarifying, that the p-values on the right-side reference to abundance in CD-
HFD-fed mice compared to chow-fed control mice.

We agree with Referee 4, that the CD8+PD-1+ subpopulation was (initially) not significantly
changed ($p= 0.09$). Upon adding novel data, and re-analysis according to the comment of
Referee #4, by highlighting NKT cells, CD8+PD1+ ($p= 0.03$) are significantly changed.
Furthermore, by using AI-based analysis of various parameters displaying our used CD-HFD-
fed cohorts as a total, we observed that pathology severity correlated with the hepatic
abundance of CD8+ T-cells and PD1 polarization of these cells (included in **Figure 1 and 4**,
**Extended Data 4 and 24** and **Rebuttal Figure 80-83**). These analyses indicate, that besides
changes e.g. in myeloid subsets, CD8+PD1+ cells are a key subset in NASH-diseased mice
as well as in human patients (see also **Figure 5** and **Rebuttal Figure 84**). To underline the
importance of a CD8+PD-1+ subset -expressing effector/exhaustion markers correlating with
disease progression- we have connected the data of **Figure 1** more closely to single-cell RNA-
seq data presented in **Figure 1** (e.g. the unique transcriptional activity in NASH-derived CD8+
T-cells (included in **Figure 1** and **Rebuttal Figure 80**) and improved cross-referencing to the
data co-submitted manuscript Dudek et al. in the discussion.

Furthermore, we have included in the revised manuscript, that we did not observe for CD8+ T-
cells a sufficient/non-binary gradient of PD-1 expression, allowing dissection into PD-
1^{negative} /PD-1 $^{\text{intermediate}}$ /PD-1 $^{\text{high}}$ subsets upon 12 months CD-HFD-feeding, (included in

**Extended Data 3**). Moreover, we functionally show that CD8+ T-cell are indeed the drivers of
anti-PD1-related therapy induced liver cancer.

We thank Referee #4 for pointing out the question about TCR dependency and thus would like
to draw the attention to the co-submitted manuscript Dudek et al., which describes TCR-
independent mechanisms on a cellular and molecular level driving CD8+ T cell-mediated
hepatocyte cell death. NASH-diet feeding experiments using mice with impaired TCR-
dependent effector function have been performed in collaboration with Dudek et al.

12-months CD-HFD-fed perforin^{-/-} mice developed NASH (including systemic obesity, fibrosis,
ALT) and NASH-induced hepatocarcinogenesis similar to WT control animals. We have now
addressed the question on TCR-specificity by improved cross-referencing to the co-submitted
manuscript Dudek et al.. In fact, it turns out that the effect of CD8+ T-cells is TCR-effector
function independent.

Furthermore, we have performed combination therapy of 1) anti-TNF with/without PD-1
targeted immunotherapy; 2) anti-CD4 with/without PD-1 targeted immunotherapy; 3) anti-CD8
with PD-1 targeted immunotherapy and 4) PD-L1 targeted immunotherapy, to strengthen
hypotheses about TCR-independent mechanisms (included in **Figure 4, Extended Data 20-**
**23 and Rebuttal Figure 78, 79, 81, 83, 85, 86**).

Rebuttal Figure 80

(a) Histological staining of hepatic tissue by H&E of 3, 6 or 12 months ND, CD-HFD or WD-HTF-fed mice (H&E: 3 months: ND n = 5 mice; CD-HFD n = 5 mice; WD-HTF n = 3 mice; 6 months: ND n = 16 mice; CD-HFD n = 8 mice; WD-HTF n = 8 mice; 12 months: ND n = 9 mice; CD-HFD n = 12 mice; WD-HTF n = 6 mice). Scale bar: 100 μ m. (b) Body weight of 3, 6 or 12 months ND, CD-HFD or WD-HTF-fed mice (3 months: ND n = 8 mice; CD-HFD n = 8 mice; WD-

Research for a Life without Cancer

HTF n= 3 mice; 6 months: ND n= 14 mice; CD-HFD n= 8 mice; WD-HTF n= 8 mice; 12 months:
ND n= 8 mice; CD-HFD n= 8 mice; WD-HTF n= 6 mice). (c) ALT levels of 3, 6 or 12 months
ND, CD-HFD or WD-HTF-fed mice (3 months: ND n= 15 mice; CD-HFD n= 46 mice; WD-HTF
n= 23 mice; 6 months: ND n= 46 mice; CD-HFD n= 59 mice; WD-HTF n= 21 mice; 12 months:
ND n= 25 mice; CD-HFD n= 69 mice; WD-HTF n= 5 mice). (d) NAS evaluation by of 3, 6 or 12
2849 months ND, CD-HFD or WD-HTF-fed mice (3 months: ND n= 5 mice; CD-HFD n= 5 mice; WD-
2850 HTF n= 3 mice; 6 months: ND n= 16 mice; CD-HFD n= 8 mice; WD-HTF n= 8 mice; 12 months:
ND n= 9 mice; CD-HFD n= 12 mice; WD-HTF n= 6 mice). (e) UMAP representation showing
the FlowSOM-guided clustering of randomly chosen CD45+ cells and quantification of hepatic
immune cell composition by flow cytometry of 12 months ND or CD-HFD-fed mice (ND n= 4
mice; CD-HFD n= 8 mice). (f) CD8 and PD-1 staining of hepatic tissue by
immunohistochemistry of 12 months ND, CD-HFD or WD-HTF-fed mice (PD-1: n= 5
mice/group; CD8: ND n= 6 mice; CD-HFD n= 6 mice; WD-HTF n= 5 mice). Scale bar: 100 μ m.
(g) Immunofluorescence staining of PD-1, CD8 and CD4 of 12 months ND or CD-HFD-fed
mice (n= 3 mice/group). Arrowheads indicate CD8+ (red), PD-1+ (green) or CD4+ (ocher) cells.
Scale bar: 100 μ m. (h) UMAP representation of 63 parameters (serology, flow cytometry,
histology) indicating NASH pathology severity measured of 12 months ND or CD-HFD-fed mice
(ND n= 22 mice; CD-HFD n= 31 mice). (i) tSNE representation of TCR β + cells and analyses
of (j) differential gene expression, (k) RNA velocity indicating transcriptional activity, gene
expression and the trajectory of CD8+ cells by scRNA-seq of 12 months ND or CD-HFD-fed
mice (n= 3 mice/group) 53. Root cells: yellow cells indicate root cells, blue cells indicate cells
farthest away from root by RNA velocity. End points: yellow cells indicate end point cells, blue
cells indicate cells farthest away from defined end point cells by RNA velocity. Latent time:
pseudo-time by RNA velocity, dark color indicate start of velocity, yellow color indicate end
point of latent time. RNA velocity flow: Blue cluster defined as start point, orange cluster as
intermediate, green cluster as end point. Arrows indicate the trajectory of cells.

Rebuttal Figure 81

(a) ScRNA- seq analysis of hepatic TCRβ+ cells of 12 months CD-HFD + IgG or CD-HFD-fed mice + 8 weeks treatment by α-PD-1 or α-CD8 antibodies (n= 3 mice/group).

(b) Selected marker expression in hepatic CD8+ T-cells by scRNA-seq comparing CD8+ with CD8+PD-1+ T-cells of 12 months CD-HFD + IgG or CD-HFD-fed mice + 8 weeks treatment by α-PD-1 antibodies (n= 3 mice/group).

(c) Average UMI comparison of hepatic CD8+PD-1+ T-cells of 12 months CD-HFD + IgG or CD-HFD-fed mice + 8 weeks treatment by α-PD-1 antibodies (n=

3 mice/group). (d) RNA velocity analyses of scRNA-seq data showing expression and (e)
 correlation of expression along the latent-time of selected genes along the latent-time (n= 3
 mice/group). Root cells: yellow cells indicate root cells, blue cells indicate cells farthest away
 from root by RNA velocity. End points: yellow cells indicate end point cells, blue cells indicate
 cells farthest away from defined end point cells by RNA velocity. Latent time: pseudo-time by
 RNA velocity, dark color indicate start of RNA velocity, yellow color indicate end point of latent
 time. RNA velocity flow: Blue cluster defined as start point, orange cluster as intermediate,
 green cluster as end point. Arrows indicate trajectory of cells. (f) PCA plot of hepatic CD8+ or
 CD8+PD-1+ T-cells sorted TCR β + cells by mass spectrometry of 12 months ND, CD-HFD or
 CD-HFD-fed mice + 8 weeks treatment by α -PD-1 antibodies (CD8+: ND n= 6 mice, CD-HFD
 + IgG n= 5 mice; CD-HFD + α -PD-1 n= 6 mice; CD8+PD-1+: ND n= 4 mice, CD-HFD + IgG n=
 6 mice; CD-HFD + α -PD-1 n= 6 mice). (g) UMAP representation showing the FlowSOM-guided
 clustering, heatmap showing the median marker expression, and (h) quantification of hepatic
 CD8+ T-cells of 12 months ND, CD-HFD + IgG or CD-HFD-fed mice + 8 weeks treatment by
 α -PD-1 antibodies (ND n= 4 mice; CD-HFD + IgG n= 8 mice; CD-HFD + α -PD-1 n= 6 mice). (i)
 Quantification of CellCNN analyzed flow cytometry data of hepatic CD8+ T-cells of 12 months
 CD-HFD + IgG or CD-HFD-fed mice + 8 weeks treatment by α -PD-1 antibodies (CD-HFD +
 IgG n= 6 mice; CD-HFD + α -PD-1 n= 4 mice). (j) UMAP representation showing the FlowSOM-
 guided clustering, the expression intensity of the indicated marker and heatmap showing the
 median marker expression of flow cytometry data of hepatic CD8+PD-1+ T-cells of 12 months
 ND, CD-HFD or CD-HFD-fed mice + 8 weeks treatment by α -PD-1 antibodies (ND n= 6 mice;
 CD-HFD n= 5 mice; CD-HFD + α -PD-1 n= 6 mice). (k) ALT and (l) NAS evaluation of 12 months
 ND, CD-HFD, CD-HFD-fed mice + 8 weeks treatment by α -PD-1, α -PD-1/ α -CD8, α -TNF, or α -
 PD-1/ α -TNF antibodies (ND n= 30 mice; CD-HFD n= 47 mice; CD-HFD + α -PD-1 n= 35 mice;
 CD-HFD + α -PD-1/ α -CD8 n= 9 mice; CD-HFD + α -TNF n= 10 mice; CD-HFD + α -PD-1/ α -TNF
 n= 11 mice). (m) Quantification of hepatic CD8+PD-1+CXCR6+ T-cells ND, CD-HFD, CD-
 HFD-fed mice + 8 weeks treatment by α -PD-1, α -PD-1/ α -CD8, α -TNF, α -PD-1/ α -TNF, α -CD4,
 or α -PD-1/ α -CD4 antibodies (ND n= 30 mice; CD-HFD n= 47 mice; CD-HFD + α -PD-1 n= 35
 mice; CD-HFD + α -PD-1/ α -CD8 n= 9 mice; CD-HFD + α -TNF n= 10 mice; CD-HFD + α -PD-
 1/ α -TNF n= 11 mice); CD-HFD + α -CD4 n= 8 mice; CD-HFD + α -PD-1/ α -CD4 n= 8 mice). (n)
 Quantification of tumor incidence of 12 months CD-HFD or CD-HFD-fed mice + 8 weeks
 treatment by α -CD8, α -CD8/NK1.1, α -PD-1, α -PD-1/ α -CD8, α -TNF, α -PD-1/ α -TNF, α -CD4, or
 α -PD-1/ α -CD4 antibodies (tumor incidence: CD-HFD n= 32 tumors/lesions in 87 mice; CD-
 HFD + α -CD8 n= 2 tumors/lesions in 31 mice; CD-HFD + α -CD8/NK1.1 n= 0 tumors/lesions in
 6 mice; CD-HFD + α -PD-1 n= 33 tumors/lesions in 44 mice; CD-HFD + α -PD-1/ α -CD8 n= 2
 tumors/lesions in 9 mice; CD-HFD + α -TNF n= 3 tumors/lesions in 10 mice; CD-HFD + α -PD-
 1/ α -TNF n= 3 tumors/lesions in 11 mice); CD-HFD + α -CD4 n= 3 tumors/lesions in 9 mice; CD-
 HFD + α -PD-1/ α -CD4 n= 8 tumors/lesions in 9 mice).

Research for a Life without Cancer

a

b

Rebuttal Figure 82

(a) UMAP representation of 63 parameters (serology, flow cytometry, histology) indicating NASH pathology severity measured of 12 months ND or CD-HFD fed mice (ND n= 22 mice; CD-HFD n= 31 mice). (b) Data gathered from hepatic tissue analyses was binary correlated with each other of 6- or 12-months ND or CD-HFD fed mice (ND n= 47 mice; CD-HFD n= 72 mice).

**Rebuttal Figure 83**

(a) UMAP representation of 63 parameters (serology, flow cytometry, histology) and (b)
 selected display of analyzed parameters indicating NASH pathology severity measured of 12
 2928 months ND or CD-HFD-fed mice (ND n= 22 mice; CD-HFD n= 31 mice; CD-HFD + α -PD-1 n=
 41 mice; CD-HFD + α -PD-L1 n= 6 mice; CD-HFD + α -CD8 n= 24 mice; CD-HFD + α -
 CD8/NK1.1 n= 6 mice; CD-HFD + α -PD-1/ α -CD8 n= 9 mice; CD-HFD + α -TNF n= 10 mice;
 CD-HFD + α -PD-1/ α -TNF n= 11 mice; CD-HFD + α -CD4 n= 9 mice; CD-HFD + α -PD-1/ α -CD4
 n= 9 mice). (c) Data gathered from hepatic tissue analyses was binary correlated with each
 other of 6- or 12-months ND, CD-HFD or CD-HFD-fed mice + 8 weeks treatment of α -CD8, α -
 CD8/ α -NK1.1; α -PD-1, α -PD-1/ α -CD8, α -TNF, α -PD-1/ α -TNF, α -CD4, or α -PD-1/ α -CD4 (ND
 n= 47 mice; CD-HFD n= 72 mice; CD-HFD + α -PD-1 n= 41 mice; CD-HFD + α -PD-L1 n= 6
 mice; CD-HFD + α -CD8 n= 29 mice; CD-HFD + α -CD8/NK1.1 n= 6 mice; CD-HFD + α -PD-1/ α -
 CD8 n= 9 mice; CD-HFD + α -TNF n= 10 mice; CD-HFD + α -PD-1/ α -TNF n= 11 mice; CD-HFD
 + α -CD4 n= 9 mice; CD-HFD + α -PD-1/ α -CD4 n= 9 mice).

Rebuttal Figure 84

(a) Flow cytometry plots, quantification of patient-liver-derived PD-1+CD8+ T-cells, and (b) correlation of PD-1+CD8+ T-cells with BMI, NAS and ALT of healthy or NAFLD/NASH patients

Research for a Life without Cancer

(Supplementary Table 1: healthy n= 8 patients; NAFLD/NASH n= 16 patients). Fluorescence-
minus-one (FMO) defined in Extended Data 25. (c) UMAP representation showing the
FlowSOM-guided clustering of CD45+ cells and (d) flow cytometry plots and quantification of
CD8+PD-1+CD103+ derived from hepatic biopsies of control, or NAFLD/NASH patients
(Supplementary Table 2: control n= 6 patients; NAFLD/NASH n= 11 patients) Populations:
CD8+ (violet), CD8+PD-1+CD103+ (red). (e) UMAP representation of CD3+ cells and analyses
of differential gene expression by scRNA-seq of control, or NAFLD/NASH patients (control n=
4 patients; NAFLD/NASH n= 7 patients). (f) Correlation of significant differentially expressed
genes in liver-derived CD8+PD-1+ compared to CD8+PD-1- T-cells subsets of 12 months CD-
HFD-fed mice and NAFLD/NASH patients (mouse: n= 3 mice; human: n= 3 patients). (g)
Velocity analyses of scRNA-seq data showing (h) expression, transcriptional activity, (i) gene
expression and (j) correlation of expression along the latent-time of selected genes along the
latent-time of patient-liver-derived CD8+ T-cells of control, or NAFLD/NASH patients in
comparison to mouse-liver-derived CD8+ T-cells (patients: NAFLD/NASH n= 3 patients;
mouse: n= 3 mice/group). Root cells: yellow cells indicate root cells, blue cells indicate cells
farthest away from the root by RNA velocity. End points: yellow cells indicate end point cells,
blue cells indicate cells farthest away from defined end point cells by RNA velocity. Latent time:
pseudo-time by RNA velocity, dark color indicate start of RNA velocity, yellow color indicate
end point of latent time. RNA velocity flow: Blue cluster defined as start point, orange cluster
as intermediate, green cluster as end point. Arrows indicate the trajectory of cells.

**2965 Rebuttal Figure 85**

(a) Body weight, AST, and histological evaluation by (b) Sirius red, CD4, CD8, PD-1, PD-L1,
F4/80, MHC-II and (c) staining of ND, CD-HFD, or CD-HFD-fed mice + 8 weeks treatment by
α -PD-1, α -PD-1/ α -CD8, α -TNF, α -PD-1/ α -TNF antibodies (body weight: ND n= 16 mice; CD-
HFD n= 29 mice; CD-HFD + α -PD-1 n= 23 mice; CD-HFD + α -PD-1/ α -CD8 n= 9 mice; CD-
HFD + α -TNF n= 10 mice; CD-HFD + α -PD-1/ α -TNF n= 11 mice; AST: body weight: ND n= 30
mice; CD-HFD n= 40 mice; CD-HFD + α -PD-1 n= 30 mice; CD-HFD + α -PD-1/ α -CD8 n= 9
mice; CD-HFD + α -TNF n= 10 mice; CD-HFD + α -PD-1/ α -TNF n= 11 mice; Sirius red: ND n=
11 mice; CD-HFD n= 12 mice; CD-HFD + α -PD-1 n= 12 mice; CD-HFD + α -PD-1/ α -CD8 n= 9
mice; CD-HFD + α -TNF n= 10 mice; CD-HFD + α -PD-1/ α -TNF n= 11 mice; CD4: ND n= 10
mice; CD-HFD n= 11 mice; CD-HFD + α -PD-1 n= 14 mice; CD-HFD + α -PD-1/ α -CD8 n= 9
mice; CD-HFD + α -TNF n= 10 mice; CD-HFD + α -PD-1/ α -TNF n= 11 mice; CD8: ND n= 10
mice; CD-HFD n= 12 mice; CD-HFD + α -PD-1 n= 14 mice; CD-HFD + α -PD-1 n= 14 mice; CD-
HFD + α -PD-1/ α -CD8 n= 9 mice; CD-HFD + α -TNF n= 10 mice; CD-HFD + α -PD-1/ α -TNF n=
11 mice; PD-1: ND n= 12 mice; CD-HFD n= 12 mice; CD-HFD + α -PD-1 n= 14 mice; CD-HFD
+ α -PD-1/ α -CD8 n= 8 mice; CD-HFD + α -TNF n= 10 mice; CD-HFD + α -PD-1/ α -TNF n= 10
mice; PD-L1: ND n= 10 mice; CD-HFD n= 11 mice; CD-HFD + α -PD-1 n= 14 mice; CD-HFD +
α -PD-1/ α -CD8 n= 9 mice; CD-HFD + α -TNF n= 10 mice; CD-HFD + α -PD-1/ α -TNF n= 11 mice;
F4/80: ND n= 11 mice; CD-HFD n= 12 mice; CD-HFD + α -PD-1 n= 14 mice; CD-HFD + α -PD-
1 n= 14 mice; CD-HFD + α -PD-1/ α -CD8 n= 9 mice; CD-HFD + α -TNF n= 10 mice; CD-HFD +
α -PD-1/ α -TNF n= 11 mice; MHC-II: ND n= 11 mice; CD-HFD n= 13 mice; CD-HFD + α -PD-1
n= 14 mice; CD-HFD + α -PD-1 n= 14 mice; CD-HFD + α -PD-1/ α -CD8 n= 9 mice; CD-HFD +
α -TNF n= 10 mice; CD-HFD + α -PD-1/ α -TNF n= 11 mice). Scale bar: 100 μ m.

**Rebuttal Figure 86**
 (a) Body weight, ALT, AST, NAS, and histological evaluation by (b) Sirius Red, CD4, CD8, PD-
 1, PD-L1, F4/80, MHC-II and (c) staining of ND, CD-HFD, or CD-HFD-fed mice + 8 weeks
 treatment by α-PD-1, α-CD4, α-PD-1/α-CD4 antibodies (body weight: ND n= 16 mice; CD-HFD
 n= 29 mice; CD-HFD + α-PD-1 n= 23 mice; CD-HFD + α-CD4 n= 9 mice; CD-HFD + α-PD-1/α-
 CD4 n= 9 mice; ALT ND n= 30 mice; CD-HFD n= 47 mice; CD-HFD + α-PD-1 n= 35 mice; CD-
 HFD + α-CD4 n= 9 mice; CD-HFD + α-PD-1/α-CD4 n= 9 mice; AST: ND n= 30 mice; CD-HFD
 n= 40 mice; CD-HFD + α-PD-1 n= 30 mice; CD-HFD + α-CD4 n= 9 mice; CD-HFD + α-PD-1/α-
 CD4 n= 9 mice; NAS: ND n= 31 mice; CD-HFD n= 46 mice; CD-HFD + α-PD-1 n= 40 mice;
 CD-HFD + α-CD4 n= 8 mice; CD-HFD + α-PD-1/α-CD4 n= 8 mice; Sirius red: ND n= 11 mice;
 CD-HFD n= 12 mice; CD-HFD + α-PD-1 n= 12 mice; CD-HFD + α-CD4 n= 9 mice; CD-HFD +
 α-PD-1/α-CD4 n= 9 mice; CD4: ND n= 10 mice; CD-HFD n= 11 mice; CD-HFD + α-PD-1 n=
 14 mice; CD-HFD + α-CD4 n= 10 mice; CD-HFD + α-PD-1/α-CD4 n= 11 mice; CD8: ND n= 10

mice; CD-HFD n= 12 mice; CD-HFD + α -PD-1 n= 14 mice; CD-HFD + α -CD4 n= 9 mice; CD-
 HFD + α -PD-1/ α -CD4 n= 9 mice; PD-1: ND n= 13 mice; CD-HFD n= 12 mice; CD-HFD + α -
 PD-1 n= 14 mice; CD-HFD + α -CD4 n= 9 mice; CD-HFD + α -PD-1/ α -CD4 n= 9 mice; PD-L1:
 ND n= 12 mice; CD-HFD n= 12 mice; CD-HFD + α -PD-1 n= 14 mice; CD-HFD + α -CD4 n= 9
 mice; CD-HFD + α -PD-1/ α -CD4 n= 9 mice; F4/80: ND n= 11 mice; CD-HFD n= 13 mice; CD-
 HFD + α -PD-1 n= 14 mice; CD-HFD + α -CD4 n= 8 mice; CD-HFD + α -PD-1/ α -CD4 n= 9 mice;
 MHC-II: ND n= 11 mice; CD-HFD n= 13 mice; CD-HFD + α -PD-1 n= 14 mice; CD-HFD + α -
 PD-1 n= 14 mice; CD-HFD + α -CD4 n= 9 mice; CD-HFD + α -PD-1/ α -CD4 n= 9 mice). Scale
 3012 bar: 100 μ m.

Figure 1f: The stains are both single stains. It should be possible to show a double staining
 CD8+PD1+ population and enumerate them as this seems like the critical part of the study.

We thank Referee #4 for pointing that out. We performed an additional double staining
 corroborating our flow cytometry data in **Figure 1**. In line, we have now included histological
 double staining in a revised manuscript (included in **Figure 1, Extended Data 3, 12,** and
 **Rebuttal Figure 87**). These data indicated that PD1+ expression is indeed associated with
 CD8+ staining.

**Rebuttal Figure 87**

(a) Immunofluorescence staining of PD-1, CD8 and CD4 of 12 months ND or CD-HFD-fed
mice (n= 3 mice/group). Arrowheads indicate CD8+ (red), PD-1+ (green) or CD4+ (ocher) cells.
Scale bar: 100 μ m. (b) Immunofluorescence staining of single channel-staining PD-1, CD8 and
CD4 (ocher) of 12 months ND or CD-HFD-fed mice (n= 3 mice/group). Arrowheads indicate
CD8+ (red), PD-1+ (green) or CD4+ (ocher) cells. Scale bar: 100 μ m. (c) Immunofluorescence
microscopy of 12 months ND, CD-HFD or CD-HFD-fed mice + 8 weeks treatment of α -PD-1
fed mice fed mice (n= 3 mice/group). Scale bar: 100 μ m.

Figure 1j: One of the most upregulated genes in the PD1+ subset is IL-10. Do the authors have
any data on whether this is secreted by this subset. Although the subset is labelled as “PD1+”
it is not the top upregulated gene here (as above). A side-by-side broader functional study
would add a bit of resolution here and if they do secrete IL-10 this may impact on the overall
interpretation. The interpretations about function are all via the screening approaches so some
further specific back up by FACS/ELISA would be helpful in confirming functionality, especially
in the context of an “exhausted” phenotype – this would clarify the statement on line 199 about
“potential effector function”. Such an experiment would also be valuable in the anti-PD1 treated
mice in later parts of the manuscript.

We fully agree and thank Referee #4 for raising this important point of IL-10 expression, which
was also raised in a recent study (Breuer et al., 2020).

We analyzed IL-10+ CD8+PD-1+ T-cells in our revised manuscript (included in **Extended Data**
**19** and **Rebuttal Figure 88a**).

However, we did not see any changes in IL10+ CD8+PD1+ in comparison to CDHFD-fed and
control mice. Moreover, IL10 levels measured by ELISA did neither drop upon CD8-depletion
(included in **Extended Data 10** and **Rebuttal Figure 88b**) nor increase significantly upon anti-
PD1 treatment (included in **Extended Data 13** and **Rebuttal Figure 88c**). Thus, an increased
anti-inflammatory role by IL-10 expressing CD8+ T-cells upon PD1-targeted immunotherapy
could not be corroborated (included in **Extended Data 19** and **Rebuttal Figure 88a**) (Breuer
et al., 2020). Of note, in this publication diet-based NAFLD induction was achieved by feeding
either WD or CD-HFD for 8-10 weeks. This is in strong contrast to our experimental regime of
applying diet for 3, 6, or 12 months as we show, that the preclinical model presents different
stages of NASH pathology severity including hepatocarcinogenesis (data presented in **Figure**
**1** and **Rebuttal Figure 80**). Thus, in our opinion, CD8+PD1+ cells are the main effector
population driving liver inflammation and liver cancer – most likely independent of IL10 being
one of the most upregulated genes in this subset.

In line with our mouse data scRNA-seq of CD8+PD1+ cells derived from control vs
NAFLD/NASH patients did not reveal increased IL10 expression. Besides in bulk RNA-seq of

human liver tissue, we observed a variable expression pattern depending on NASH pathology
 severity (included in **Figure 5, Extended Data 28** and **Rebuttal Figure , 77**).

Rebuttal Figure 88

Polarization by flowcytometry of hepatic CD8+PD-1+ T-cells of 12 months ND, CD-HFD or CD-
 HFD-fed mice + 8 weeks treatment of α -PD-1 (ND n= 12 mice; CD-HFD n= 7 mice; CD-HFD
 + α -PD-1 n= 6 mice). (b) Multiplex ELISA concentrations of hepatic inflammation-associated
 cytokines of 12 months ND, CD-HFD or CD-HFD-fed mice + 8 weeks treatment of α -CD8 or
 CD-HFD-fed mice + co-depletion of α -CD8/NK1.1 (ND n= 10 mice; CD-HFD n= 14 mice; CD-
 HFD + 8 weeks treatment of α -CD8 n= 5 mice; CD-HFD + co-depletion of α -CD8/NK1.1 n= 5
 mice). (c) Multiplex ELISA concentrations of hepatic inflammation-associated cytokines of 12
 3073 months ND, CD-HFD, CD-HFD-fed mice + 8 weeks treatment of α -PD-1 (ND n= 10 mice; CD-
 3074 HFD n= 14 mice; CD-HFD + α -PD-1 n= 13 mice).

Rebuttal Figure 89

(a) Selected average marker expression in T-cell subsets of CD8⁺ and (b) CD4⁺ sorted TCRβ⁺ by scRNA-seq of 12 months ND or CD-HFD-fed mice (n= 3 mice/group). (c) Selected marker expression in hepatic CD8⁺ T-cells by scRNA-seq comparing CD8⁺ with CD8⁺PD-1⁺ T-cells of 12 months CD-HFD + IgG or CD-HFD-fed mice + 8 weeks treatment of α-PD-1 fed mice (n= 3 mice/group). (d) Selected marker expression in hepatic CD4⁺ T-cells by scRNA-seq comparing CD4⁺ with CD4⁺PD-1⁺ T-cells of 12 months CD-HFD + IgG or CD-HFD-fed mice + 8 weeks treatment of α-PD-1 (n= 3 mice/group). (e) Selected marker expression in hepatic CD8⁺PD-1⁺ T-cells by mass- spectrometry of 12 months ND or CD-HFD-fed mice (ND n= 4 mice, CD-HFD n= 6 mice). (f) Selected marker expression in hepatic CD8⁺PD-1⁺ T-cells sorted TCRβ⁺ cells by mass- spectrometry of 12 months CD-HFD or CD-HFD-fed mice + 8 weeks treatment of α-PD-1 (n= 6 mice/group). Candidates developing steady in-/decrease from ND to CD-HFD to CD-HFD-fed mice + 8 weeks treatment of α-PD-1 are indicated in red. (n= 6 mice/group).

Figure 2: It was not that clear why depleting CD8s had no impact on ALT, suggesting they are
not playing a role *in vivo*, while blocking PD1 had some impact (AST is not shown for the anti-
CD8 treatment).

We thank Referee #4 for highlighting that CD8+ T cell depletion in the context of NASH-HCC
transition had no or only minor impact on ALT reduction, an effect that has also come to our
attention and has puzzled us.

On the other hand, we would like to note that in the context of anti-PD1-related immunotherapy
triggered liver damage CD8+ T cell depletion did lead to a significant reduction in liver damage
and NAFLD activity score. Thus, we believe that the anti-PD1 therapy-related damage in NASH
and NASH to HCC transition is mainly triggered by CD8+ T cells. In contrast, in the context of
NASH development without anti-PD1 antibody treatment, other cells than CD8+ T-cell also
contribute to liver damage – and that progressive NASH is characterized by multi-faceted,
collateral damage through myeloid cells, adaptive cells, and cell death.

We think that CD8+ T-cells have an important *in vivo* role driving NASH to HCC transition, as
we strongly decreased or eliminated HCC by CD8+ T-cell depletion (both in NASH or NASH
with anti-PD1 treatment). In line, the co-submitted manuscript by Dudek et al., described
hepatocyte death by a CD8-dependent mechanism.

Notably, ALT can be elevated as a result of the chronic metabolic environment and/or as a
result of the still ongoing hepatic inflammation independent of CD8+ or NK1.1+ cells (included
in **Extended Data 9** and **Rebuttal Figure 90**).

Further, it can be that actually at late time points of co-existence of tumors and NASH – the
collateral damage might be mainly triggered by non-CD8+ T-cells. We have confirmed the
efficient depletion of the CD8 T-cells in our models, excluding that this might be a reason.

AST levels are included in our AI-based analysis (included in **Figure 1 and 4**, **Extended Data**
**4 and 24** and **Rebuttal Figure 80-83**), indicating no change upon CD8 depletion as well.

Rebuttal Figure 90

(a) ALT levels of 12 months ND, CD-HFD, CD-HFD + 8 weeks treatment of α -CD8 or CD-HFD
+ 8 weeks co-depletion of α -CD8/NK1.1 (ALT: ND n= 22 mice; CD-HFD n= 42 mice; CD-HFD
+ α -CD8 n= 31 mice; CD-HFD + α -CD8/NK1.1 n= 6).

Line 202 – lack of impact of anti-PD1. Is there a control for this experiment? The implication is
that this lack of impact is etiology-specific but it may also be that the intervention does not work
well in other HCC models.

We thank Referee #4 for highlighting the etiology-dependent potential outcome of PD-1-
targeted immunotherapy against HCC. We agree with Referee #4, that there might be
bivalence in other HCC models and, more importantly, only a subset of HCC patient react to
PD-1 targeted immunotherapy (El-Khoueiry et al., 2017; Hage et al., 2019). Thus, we have
also performed anti-PD-L1 targeted immunotherapy in CDHFD-fed mice with and without
established liver cancer (included in **Extended Data 7** and **Rebuttal Figure 91**).

The data of our study indicate that similar to anti-PD1 - anti-PDL1-treatment does not induce
an anti-liver cancer effect for NASH-induced HCC but rather induces similar to anti-PD1
treatment a pro-inflammatory and pro-carcinogenic effect. These data further suggest that in
the preclinical NASH models used, both PD1- or PDL1-targeted immunotherapy induces
adverse effects. This is corroborated by our increased, retrospective cohort HCC-patients of
different etiologies under PD(L)1-targeted immunotherapy, in which multivariate analysis
results in NAFLD/NASH being an independent negative factor for overall survival (included in
**Figure 6** and **Rebuttal Figure 92**). Furthermore, we corroborated our hypothesis of non-viral
(NASH-related) HCC being less responsive to immunotherapy by a meta-analysis including
1656 patients of the three most important clinical trials, identifying immunotherapy vs control
for viral HCC as favorable treatment (HR(viral)= 0.64), in contrast, non-viral-HCC showed less
benefit (HR(non-viral)= 0.92) for immunotherapy (included in **Figure 6**, **Extended Data 30-32**,
**Supplementary Table 9** and **Rebuttal Figure 93, 94**)).

Rebuttal Figure 91

(a) MRI pictures of liver of mice after 13 months CD-HFD followed by 7 weeks treatment to
 CD-HFD or CD-HFD-fed mice + 7 weeks by α -PD-L1 antibodies (CD-HFD n= 6 mice; CD-HFD
 + α -PD-L1 n= 8 mice). Lines indicate tumor nodule. Scale bar: 10 mm. (b) Macroscopy of liver
 of ND, CD-HFD or CD-HFD-fed mice + 8 weeks treatment by α -PD-L1 antibodies. Arrowheads
 indicate tumor/lesions. Scale bar: 10 mm. (c) Body weight, ALT levels ND, CD-HFD or CD-
 HFD-fed mice + 8 weeks treatment by α -PD-L1 antibodies (Body weight, ALT, : ND n= 8 mice;
 CD-HFD n= 6 mice; CD-HFD + α -PD-L1 n= 6 mice) (d) and (e) NAS evaluation by H&E, fibrosis
 quantification (Sirius Red), quantification of CD8, PD-1 and PD-L1 staining of hepatic tissue
 by immunohistochemistry of 12 months ND, CD-HFD or CD-HFD-fed mice + 8 weeks
 treatment by α -PD-L1 antibodies (NAS: ND n= 7 mice; CD-HFD n= 6 mice; CD-HFD + α -PD-
 L1 n= 6 mice; Sirius Red: ND n= 7 mice; CD-HFD n= 5 mice; CD-HFD + α -PD-L1 n= 6 mice ;
 CD8, : ND n= 5 mice; CD-HFD n= 5 mice; CD-HFD + α -PD-L1 n= 5 mice; PD-1, PD-L1: ND

n= 5 mice; CD-HFD n= 5 mice; CD-HFD + α -PD-L1 n= 6 mice). Scale bar: 100 μ m. (f)
 Tumor/Lesion incidence in CD-HFD or CD-HFD-fed mice + 8 weeks treatment by α -PD-L1
 antibodies (CD-HFD n= 19 tumors/lesions in 25 mice; CD-HFD + α -PD-L1 n= 7 tumors/lesions
 in 8 mice). Arrowheads indicate specific staining positive cells.

C

		Overall survival		
		HR	95% CI	p-value (Cox regression)
Etiology	Other etiologies	1		0.017
	NAFLD	2.6	1.2-5.6	
Performance Status	0	1		0.049
	≥ 1	1.7	1.0-2.8	
Macrovascular invasion	Absent	1		0.016
	Present	1.8	1.1-3.0	
Extrahepatic metastases	Absent	1		0.121
	Present	0.7	0.4-1.1	
Alpha-fetoprotein (ng/ml)	≤ 200	1		0.019
	> 200	1.8	1.1-2.9	
Child-Pugh class	A	1		0.075
	B	1.6	1.0-2.6	

 **Rebuttal Figure 92**

(a) Nonalcoholic fatty liver disease (NAFLD) is associated with a worse outcome in patients
 with hepatocellular carcinoma (HCC) treated with PD-(L)1-targeted immunotherapy. A total of
 130 patients with advanced HCC received PD-(L)1-targeted immunotherapy (Supplementary
 Table 8). Kaplan-Meier curve display overall survival of patients with NAFLD vs. those with
 any other etiology; all 130 patients were included in these survival analyses (NAFLD n=13, any
 other etiology n=117). (b) Validation cohort of patients with HCC treated with PD-(L)1-targeted
 immunotherapy. A total of 1180 patients with advanced HCC received PD-(L)1-targeted

Rebuttal Figure 94

A total of 1656 patients were included in all three randomized trials, and 985 patients received
a checkpoint inhibitor. Subgroup analysis was performed to study the specific effects of
immunotherapy comparing non-viral etiologies (n=737) with (a) HBV (n=574) or (b) HCV
(n=345). Hazard ratios for each trial are represented by squares, the size of the square
represents the weight of the trial in the meta-analysis. The horizontal line crossing the square
represents the 95% confidence interval (CI). The diamonds represent the estimated overall
effect based on the meta-analysis random effect of all trials.

A total of 1243 patients were included in two first-line trials comparing PD-1 or PD-L1 targeted
immunotherapy to sorafenib. 707 patients received an immune checkpoint inhibitor (either PD-
1 or anti-PD-1). (c) HCV and HBV were pooled into a separate category, termed “viral”, and a
subsequent meta-analysis comparing viral (n=754) and non-viral (n=489), mostly NASH and
alcohol intake, was performed. A subgroup analysis studying the specific effects of non-viral
etiologies (n=489) on the magnitude of effect of immunotherapy are presented, when
compared to (d) HBV (n=473) or (e) HCV (n=281). Hazard ratios for each trial are represented
by squares, the size of the square represents the weight of the trial in the meta-analysis. The
horizontal line crossing the square represents the 95% confidence interval (CI). The diamonds
represent the estimated overall effect based on the meta-analysis random effect of all trials.

Figure 5b and the text are presented in a slightly confusing way. It would be easier to
understand the disease associations of %CD8 (of CD3), and % PD1+ (or MFI) of CD3+CD8+
first. The association of CD103 with tissue residency in the liver is not as good as other tissues,
so a broader look at the CD8+PD1+ population by flow would be better as well as some caution
in interpretation.

We agree with this comment and thank Referee #4 for highlighting this problem. Inline, we
have now improved our manuscript as suggested by Referee#4 (included in **Extended Data**
**25 and 27** and **Rebuttal Figure 75, 76**). Moreover, we corroborated the association of NASH
patients and CD103 in a second patient cohort using CYTOF (included in **Figure 5** and
**Rebuttal Figure 95**).

Rebuttal Figure 95

(a) tSNE representation, (b) marker expression, (c) average marker expression of defined T-cell subsets of patient-liver-derived T-cells analyzed by CyTOF of control and NAFLD/NASH patients (control n= 11 patients pooled in 3 analyses; NAFLD/NASH n= 16 patients pooled in 5 analyses). (d) Composition, (e) HSNE representation of defined T-cell subsets and (f) quantification of CD8+CD103+PD-1+ cells of of patient-liver-derived T-cells analyzed by CyTOF of control and NAFLD/NASH patients (control n= 11 patients pooled in 3 analyses; NAFLD/NASH n= 16 patients pooled in 5 analyses).

Figure 5e could include some study of CD4s as well for reference. That subset has been linked to NASH pathogenesis as well. As above, it should be possible to perform some dual CD8 and PD1 staining to map the subset of interest.

We thank Referee #4 for highlighting this point, that CD4 T-cells and their expression of PD-1 might play a crucial role in the observed phenotype and thus included an in detail analysis of CD4 T-cells to the majority of our experiments (e.g. **Extended Data 3** and **Rebuttal Figure 96**). However, in the preclinical model the magnitude of effects observed in CD4+ T-cells is minor when compared to CD8+ T-cells (e.g. **Extended Data 11** mean (CD8+CD62L-CD44+CD69+) ~12% (%of CD45+) vs mean(CD4+CD62L-CD44+CD69+) ~4% (%of CD45+) upon PD-1 targeted immunotherapy).

Data obtained from CD4 depletion with/without PD1-targeted immunotherapy indicate, that the
increased hepatocarcinogenesis in the context of anti-PD1 related immunotherapy is
independent of hepatic abundance of CD4+ T-cells in the preclinical NASH/HCC model
(included in **Figure 4, Extended Data 22 and 23** and **Rebuttal Figure 79, 81, 86**). However,
CD4+ T-cells might have a diverse set of effector functions (e.g. interpreting tumor incidence
in anti-CD8/anti-PD1 treated animals: in the absence of CD8+ T-cells but immunotherapy, thus
CD4+ T-cells might be responsible for baseline tumor incidence; or the trends of increased
tumor incidence upon anti-CD4/anti-PD1 co-treatment in **Figure 4** and **Rebuttal Figure 81n**).
To allow a wider interpretation of a potential effect of CD4+ T-cells in our preclinical model, we
integrated and correlated the variety and potential changes upon 12 months of diet-feeding in
the AI-based analyses correlating disease parameters with cellular abundance and
polarization (included in **Figure 1, Extended Data 4 and 24** and **Rebuttal Figure 82, 83**).
These data further strengthens that CD4+ T-cells play a minor role, as we see no significant
correlation of CD4-depleted animals with histological, or serological markers.

Of note, CD4+ T-cells are also significantly changed in the human situation by classical flow
cytometry, but in the light of the results obtained in the preclinical model, we decided to not
investigate this result extensively (included in **Extended Data 27** and **Rebuttal Figure 75**). Of
note, CD4+ T-cells are also significantly changed in the human situation and have also
analyzed human CD4+ cells a by scRNASeq (included in **Extended Data 26** and **Rebuttal**
**Figure 75, 76, 89, 97**). In addition, we have performed a velocity analyses of the scRNA Seq
data of mouse and human CD4 T cells (see Rebuttal letter below). In mouse, no significant
velocity flow was detected in 12 months CD-HFD-fed mice, indicating, that CD4 cells are not
transcriptionally activated and driven by NASH-conditions or PD-1-targeted immunotherapy in
NASH. However, we want to point out, that in the mouse NASH model CD8 T-cells increase
statistically significant and thus CD4 are relatively fewer cells compared to CD8. Therefore,
the velocity analysis of mouse CD4 T-cells need to be taken with caution, because we included
300-500 cells only per described subset. As consequence, we included the negative CD4 T-
cell data not in the manuscript but in the Rebuttal letter. Velocity analyses on human CD4 lead
to comparable problems like seen in mouse. As a consequence, we included the negative CD4
T-cell data not in the manuscript but in the Rebuttal letter as **Rebuttal Figure 97**.

However, we discuss the potential role of CD4+ T-cells in greater detail in the main text.

Rebuttal Figure 96

(a) Analysis of 5000 randomly chosen CD45⁺ cells by flow cytometry to define distinct marker expression of 12 months ND or CD-HFD-fed mice (ND n = 4 mice; CD-HFD n = 8 mice). (b) Average marker expression of defined CD45⁺ subsets of 5000 randomly chosen CD45⁺ cells by flow cytometry of 12 months ND or CD-HFD-fed mice (ND n = 4 mice; CD-HFD n = 8 mice). (c) Quantification of hepatic CD8⁺ cells and PD-1⁺ expressing cells by immunohistochemistry of 12 months ND, CD-HFD or WD-HTF-fed mice (PD-1: n = 5 mice/group; CD8: ND n = 6 mice; CD-HFD n = 6 mice; WD-HTF n = 5 mice). (d) Immunofluorescence staining of single channel-staining PD-1, CD8 and CD4 (ocher) of 12 months ND or CD-HFD-fed mice (n = 3 mice/group). Arrowheads indicate CD8⁺ (red), PD-1⁺ (green) or CD4⁺ (ocher) cells. Scale bar: 100 μ m. (e) H&E, CD8 and PD-1 staining, evaluation by NAS and quantification of CD8⁺ cells and PD-1⁺ expressing cells by immunohistochemistry of 32-weeks old hURI-tetOFF^{hsp} and non-transgenic litter control mice (n=6 mice/group). Arrowheads indicate specific staining positive

cells. Scale bar: 100 μ m. (f) Quantification of abundance, (g) PD-1 expression and flow
 cytometry plots of hepatic CD8⁺ T-cells by flow cytometry of 6 or 12 months ND or CD-HFD-
 fed mice (abundance of CD8: 6 months: ND n= 17 mice; CD-HFD n= 10 mice; WD-HTF n= 7
 mice; 12 months: ND n= 11 mice; CD-HFD n= 6 mice; WD-HTF n= 5 mice; PD-1 expression
 in CD8⁺ T-cells: 6 months: ND n= 15 mice; CD-HFD n= 14 mice; WD-HTF n= 7 mice; 12
 3297 months: ND n= 10 mice; CD-HFD n= 6 mice; WD-HTF n= 5 mice). (h) Quantification of
 3298 abundance, (i) PD-1 expression and flow cytometry plots of hepatic CD4⁺ T-cells by flow
 cytometry of 6 or 12 months ND or CD-HFD-fed mice (abundance of CD4: 6 months: ND n=
 17 mice; CD-HFD n= 10 mice; WD-HTF n= 7 mice; 12 months: ND n= 11 mice; CD-HFD n= 6
 mice; WD-HTF n= 5 mice; PD-1 expression in CD4⁺ T-cells: 6 months: ND n= 15 mice; CD-
 HFD n= 14 mice; WD-HTF n= 7 mice; 12 months: ND n= 10 mice; CD-HFD n= 6 mice; WD-
 HTF n= 5 mice).

**Rebuttal Figure 97**

(a) RNA velocity analyses of scRNA-seq data showing expression, and (b) velocity of patient-
 liver-derived CD4⁺ T-cells of control, or NAFLD/NASH patients in comparison to mouse-liver-
 derived CD4⁺ T-cells (patients: NAFLD/NASH n= 3 patients; mouse: n= 3 mice/group).
 (c) Correlation of expression along the latent-time of selected genes along the latent-time
 (mouse: n= 3 mice/group).

Figure 5f is not really that convincing of a relationship with TNF – the r-squared value would
be better to illustrate and would be very low. If the authors think TNF secretion is critical it
would be possible to explore this further in the mouse model.

We thank Referee #4 for highlighting this point. Although TNF is correlated significantly with
PD1 abundance, the correlation is weak as indicated by the r-value and therefore moved the
data to the Extended Data. Moreover, we fully agree with this Referee that further experiments
were needed to underline the role of TNF in NASH/HCC transition in the context of anti-PD1
related immunotherapy.

Thus, we have performed an anti-TNF treatment with or without PD-1- targeted immunotherapy
in the context of NASH/HCC. Anti-TNF treatment without PD1-targeted immunotherapy led to
liver cancer formation comparable to control-treated CD-HFD-fed mice. However, anti-TNF
treatment in the context of PD1-targeted immunotherapy leads to a significant reduction of
tumor incidence compared to anti-PD1 treated CD-HFD-fed mice, indicating that TNF exerts
key functions of the observed adverse effects triggered by PD1-targeted immunotherapy,
namely the increased NAS, liver damage, and hepatocarcinogenesis (included in **Figure 4,**
**Extended Data 20 and 21** and **Rebuttal Figure 78, 81, 85**).

Moreover, the combination of anti-PD1 therapy with anti-CD8 – also ablating the adverse and
pro-carcinogenic effects of CD8+ T-cells emphasize that CD8+ T-cells are a major cell
population mediating increased hepatocarcinogenesis in a TNF-dependent mechanism upon
PD1-targeted immunotherapy (included in **Figure 4, Extended Data 20 and 21** and **Rebuttal**
**Figure 78, 81, 85**).

Importantly, by comparing classical flow cytometry, CYTOF, and on scRNA-seq level of
mouse-human of CD8+ T-cells isolated from liver tissue of NASH mice or patients, we identified
similar populations and transcriptional activation of CD8+ PD1+ in a total of three independent
center patient cohorts (included in **Figure 5, Extended Data 25 and 27** and **Rebuttal Figure**
**75, 76, 84**). These data indicate that results obtained and hypotheses built from the preclinical
NASH model and are in line with published results, where TNF blockade uncouples mediated
toxicity in dual CTLA-4 and PD-1 immunotherapy (Perez-Ruiz et al., 2019).

For Figure 5G some disease controls would be valuable.

We thank Referee #4 for his/her comment for pointing out the lack of appropriate control groups
(e.g. NASH-HCC vs different etiology-induced HCC under Sorafenib/different multi-kinase
inhibitors as a second/third-line therapy). Although of extreme interest for public health and

public knowledge, we described this important issue in our discussion and to the best of our
knowledge there are no NASH-HCC treated cohorts available (apart from, possibly, inside of
the big pharma-industry), which would allow an adequate control arm. Thus, we evaluated
potential disease controls in the manuscript by performing a meta-analysis including 1656
patients of the three major clinical trials (Imbrave 150; Checkmate 459; Keynote 240). Here
we could identify immunotherapy vs control for viral HCC as favorable treatment ($HR(viral)=$
0.64), in contrast non-viral-HCC showed less benefit ($HR(non-viral)= 0.92$) for immunotherapy
(included in **Figure 6, Extended Data 30-32, Supplementary Table 9** and **Rebuttal Figure**
**93, 94**)).

Furthermore, we toned down the conclusions of our retrospective cohort in the manuscript and
would like to point out, that bigger cohorts and prospective clinical trials are of utmost
importance for the scientific community.

Line 493+: This sentence is perhaps overstating the data, which were not significant in all those
parameters. It is likely quite hard to make the firmest comparisons, especially in such a
retrospective analysis, where the heterogeneous group of patients with eg viral aetiologies will
be on effective therapies - the actual aetiologies were not obvious in the supplementary data.
This interpretation could be a bit more cautious throughout (eg. it is in the abstract).

We would like to thank Referee #4 for the important comment and agree. Thus, we toned down
the wording and interpretation of our data. As described previously, we recruited additional
patients to increase the number of patients in our initial clinical cohort from 65 to 130 HCC
patients under anti-PD(L)1-targeted immunotherapy, which we validated in a second cohort
(included in **Figure 6** and **Rebuttal Figure 92**).

We agree with Referee #4, that the presented retrospective PD-(L)1 targeted immunotherapy
treated NAFLD/NASH-associated HCC cohort - although unique for Europe and treatment not
officially licensed and thus reimbursement - is still small, although we would like to point out,
that prominent trends or effects can be seen in small retrospective cohorts as well. Thus, our
analyses of BCLC-C NAFLD/NASH-HCC vs other-etiological-HCC patients indicated, that
NAFLD/NASH-HCC has significantly reduced overall survival compared to other-etiological-
HCC in this small retrospective cohort. Of note, multivariate analyses identified NAFLD/NASH
as an independent factor for treatment response and thus identifying NAFLD/NASH as a
negative predictor for HCC immunotherapy (included in **Supplementary Table 9** and **Rebuttal**
**Figure 92**).

Like previously mentioned, we corroborated our hypothesis of non-viral (NASH-related) HCC
being less responsive to immunotherapy by a meta-analysis including 1656 patients of the
three most important clinical trials (IMbrave 150; Checkmate 459; Keynote 240), identifying
immunotherapy vs control for viral HCC as favorable treatment ($HR(viral) = 0.64$), in contrast,
non-viral-HCC showed less benefit ($HR(non-viral) = 0.92$) for immunotherapy (included in
**Figure 6, Extended Data 30-32, Supplementary Table 7 and Rebuttal Figure 93, 94**)).
Thus, we toned down the conclusions of our retrospective cohort in the manuscript and again
would like to point out, that bigger cohorts and prospective clinical trials are of utmost
importance for the scientific community.

**References**

- 1. Agdashian, D., ElGindi, M., Xie, C., Sandhu, M., Pratt, D., Kleiner, D.E., Figg, W.D.,
 Rytlewski, J.A., Sanders, C., Yusko, E.C., et al. (2019). The effect of anti-CTLA4
 treatment on peripheral and intra-tumoral T cells in patients with hepatocellular
 carcinoma. *Cancer Immunol. Immunother.* 68, 599–608.
- 2. Boege, Y., Malehmir, M., Healy, M.E., Bettermann, K., Lorentzen, A., Vucur, M.,
 Ahuja, A.K., Böhm, F., Mertens, J.C., Shimizu, Y., et al. (2017). A Dual Role of
 Caspase-8 in Triggering and Sensing Proliferation-Associated DNA Damage, a Key
 Determinant of Liver Cancer Development. *Cancer Cell* 32, 342.
- 3. Breuer, D.A., Pacheco, M.C., Washington, M.K., Montgomery, S.A., Hasty, A.H., and
 Kennedy, A.J. (2020). CD8+ T cells regulate liver injury in obesity-related
 nonalcoholic fatty liver disease. *Am. J. Physiol. Liver Physiol.* 318, G211–G224.
- 4. Brummelman, J., Haftmann, C., Núñez, N.G., Alvisi, G., Mazza, E.M.C., Becher, B.,
 and Lugli, E. (2019). Development, application and computational analysis of high-
 dimensional fluorescent antibody panels for single-cell flow cytometry. *Nat. Protoc.*
 14.
- 5. Cortellini, A., Bersanelli, M., Buti, S., Cannita, K., Santini, D., Perrone, F., Giusti, R.,
 Tiseo, M., Michiara, M., Di Marino, P., et al. (2019). A multicenter study of body mass
 index in cancer patients treated with anti-PD-1/PD-L1 immune checkpoint inhibitors:
 when overweight becomes favorable. *J. Immunother. Cancer* 7, 57.
- 6. Cui, P., Li, R., Huang, Z., Wu, Z., Tao, H., Zhang, S., and Hu, Y. (2020). Comparative
 effectiveness of pembrolizumab vs. nivolumab in patients with recurrent or advanced
 NSCLC. *Sci. Rep.* 10, 1–7.
- 7. Duffy, A.G., Ulahannan, S. V, Makorova-rusher, O., Rahma, O., Wedemeyer, H.,
 Pratt, D., Davis, J.L., Hughes, M.S., Heller, T., ElGindi, M., et al. (2016).
 Tremelimumab in Combination with Ablation in Patients with Advanced Hepatocellular
 Carcinoma. *J. Hepatol.* 66, 482–484.
- 8. El-Khoueiry, A.B., Sangro, B., Yau, T., Crocenzi, T.S., Kudo, M., Hsu, C., Kim, T.-
 Y.Y., Choo, S.-P.P., Trojan, J., Welling, T.H., et al. (2017). Nivolumab in patients with
 advanced hepatocellular carcinoma (CheckMate 040): an open-label, non-
 comparative, phase 1/2 dose escalation and expansion trial. *Lancet* 6736, 1–11.
- 9. Finkin, S., Yuan, D., Stein, I., Taniguchi, K., Weber, A., Unger, K., Browning, J.L.,
 Goossens, N., Nakagawa, S., Gunasekaran, G., et al. (2015). Ectopic lymphoid
 structures function as microniches for tumor progenitor cells in hepatocellular
 carcinoma. *Nat. Immunol.* 16, 1235–1244.
- 10. Finn, R., Ryoo, B.-Y., Merle, P., Kudo, M., Bouattour, M., Lim, H.Y., Breder, V.,
 Edeline, J., Chao, Y., Ogasawara, S., et al. (2019). Results of Keynote-240: Phase 3
 Study of Pembrolizumab vs Best Supportive Care for Second-Line Therapy in
 Advanced Hepatocellular Carcinoma. In ASCO Annual Meeting, p.
- 11. Finn, R.S., Qin, S., Ikeda, M., Galle, P.R., Ducreux, M., Kim, T.Y., Kudo, M., Breder,
 3431 V., Merle, P., Kaseb, A.O., et al. (2020). Atezolizumab plus bevacizumab in
 unresectable hepatocellular carcinoma. *N. Engl. J. Med.* 382, 1894–1905.
- 12. Gomes, A.L., Teijeiro, A., Burén, S., Tummala, K.S., Yilmaz, M., Waisman, A.,
 Theurillat, J.P., Perna, C., and Djouder, N. (2016). Metabolic Inflammation-Associated
 IL-17A Causes Non-alcoholic Steatohepatitis and Hepatocellular Carcinoma. *Cancer*
 *Cell* 30, 161–175.
- 13. Hage, C., Hoves, S., Ashoff, M., Schandl, V., Hört, S., Rieder, N., Heichinger, C.,
 Berrera, M., Ries, C.H., Kiessling, F., et al. (2019). Characterizing responsive and
 refractory orthotopic mouse models of hepatocellular carcinoma in cancer
 immunotherapy. *PLoS One* 14, e0219517.
- 14. Kim, C.G., Kim, C., Yoon, S.E., Kim, K.H., Choi, S.J., Kang, B., Kim, H.R., Park, S.-
 H., Shin, E.-C., Kim, Y.-Y., et al. (2020). Hyperprogressive disease during PD-1
 blockade in patients with advanced hepatocellular carcinoma. *J. Hepatol.*
- 15. Kudo, M. (2018). Combination Cancer Immunotherapy in Hepatocellular Carcinoma.
 *Liver Cancer* 7, 20–27.

16. Llovet, J.M., Di Bisceglie, A.M., Bruix, J., Kramer, B.S., Lencioni, R., Zhu, A.X., Sherman, M., Schwartz, M., Lotze, M., Talwalkar, J., et al. (2008). Design and Endpoints of Clinical Trials in Hepatocellular Carcinoma. *JNCI J. Natl. Cancer Inst.* 100, 698–711.
 17. Llovet, J.M., Zucman-Rossi, J., Pikarsky, E., Sangro, B., Schwartz, M., Sherman, M., and Gores, G. (2016). Hepatocellular carcinoma. *Nat. Rev. Dis. Prim.* 2, 16018.
 18. Ma, C., Kesarwala, A.H., Eggert, T., Medina-Echeverz, J., Kleiner, D.E., Jin, P., Stroncek, D.F., Terabe, M., Kapoor, V., ElGindi, M., et al. (2016). NAFLD causes selective CD4⁺ T lymphocyte loss and promotes hepatocarcinogenesis. *Nature* 531, 253–257.
 19. Malehmir, M., Pfister, D., Gallage, S., Szydłowska, M., Inverso, D., Kotsiliti, E., Leone, V., Peiseler, M., Surewaard, B.B.G.J., Rath, D., et al. (2019). Platelet GPIIb α is a mediator and potential interventional target for NASH and subsequent liver cancer. *Nat. Med.* 25, 641–655.
 20. Moser, J.C., Wei, G., Colonna, S. V, Grossmann, K.F., Hyngstrom, J.R., Moser, J.C., Wei, G., Colonna, S. V, Grossmann, K.F., Patel, S., et al. (2020). Comparative-effectiveness of pembrolizumab vs . nivolumab for patients with metastatic melanoma. *Acta Oncol. (Madr).* 59, 434–437.
 21. Nakagawa, H., Umemura, A., Taniguchi, K., Font-Burgada, J., Dhar, D., Ogata, H., Zhong, Z., Valasek, M.A., Seki, E., Hidalgo, J., et al. (2014). ER Stress Cooperates with Hypernutrition to Trigger TNF-Dependent Spontaneous HCC Development. *Cancer Cell* 26, 331–343.
 22. Park, E.J., Lee, J.H., Yu, G., He, G., Ali, S.R., Ryan, G., Holzer, R.G., Österreicher, C.H., Takahashi, H., and Karin, M. (2011). Dietary and genetic obesity promote liver inflammation and tumorigenesis by enhancing IL-6 and TNF expression. *Cell* 140, 197–208.
 23. Perez-Ruiz, E., Minute, L., Otano, I., Alvarez, M., Ochoa, M.C., Belsue, V., de Andrea, C., Rodriguez-Ruiz, M.E., Perez-Gracia, J.L., Marquez-Rodas, I., et al. (2019). Prophylactic TNF blockade uncouples efficacy and toxicity in dual CTLA-4 and PD-1 immunotherapy. *Nature* 569, 428–432.
 24. Pikarsky, E., Porat, R.M., Stein, I., Abramovitch, R., Amit, S., Kasem, S., Gutkovich-Pyest, E., Urieli-Shoval, S., Galun, E., and Ben-Neriah, Y. (2004). NF-kappaB functions as a tumour promoter in inflammation-associated cancer. *Nature* 431, 461–466.
 25. Remmerie, A., Martens, L., Thoné, T., Castoldi, A., Seurinck, R., Pavie, B., Roels, J., Vanneste, B., De Prijck, S., Vanhockerhout, M., et al. (2020). Osteopontin Expression Identifies a Subset of Recruited Macrophages Distinct from Kupffer Cells in the Fatty Liver. *Immunity* 53, 641-657.e14.
 26. Ringelhan, M., Pfister, D., O'Connor, T., Pikarsky, E., and Heikenwalder, M. (2018). The immunology of hepatocellular carcinoma. *Nat. Immunol.* 19.
 27. Scheiner, B., KIRSTEIN, M.M., Hucke, F., Finkelmeier, F., Schulze, K., von Felden, J., Koch, S., Schwabl, P., Hinrichs, J.B., Wanek, F., et al. (2019). Programmed cell death protein-1 (PD-1)-targeted immunotherapy in advanced hepatocellular carcinoma: efficacy and safety data from an international multicentre real-world cohort. *Aliment. Pharmacol. Ther.* 49, 1323–1333.
 28. Tummalala, K.S., Gomes, A.L., Yilmaz, M., Graña, O., Bakiri, L., Ruppen, I., Ximénez-Embún, P., Sheshappanavar, V., Rodríguez-Justo, M., Pisano, D.G., et al. (2014). Inhibition of De Novo NAD⁺ Synthesis by Oncogenic URI Causes Liver Tumorigenesis through DNA Damage. *Cancer Cell* 26, 826–839.
 29. Wang, Z., Aguilar, E.G., Luna, J.I., Dunai, C., Khuat, L.T., Le, C.T., Mirsoian, A., Minnar, C.M., Stoffel, K.M., Sturgill, I.R., et al. (2018). Paradoxical effects of obesity on T cell function during tumor progression and PD-1 checkpoint blockade. *Nat. Med.* 1.
 30. Weiler-Normann, C., and Lohse, A.W. (2016). Nonalcoholic Fatty Liver Disease in Patients with Autoimmune Hepatitis: Further Reason for Teeth GNASHing? *Dig. Dis.*

Research for a Life without Cancer

- Sci. 61, 2462–2464.
31. Wolf, M.J., Adili, A., Piotrowitz, K., Abdullah, Z., Boege, Y., Stemmer, K., Ringelhan,
3503 M., Simonavicius, N., Egger, M., Wohlleber, D., et al. (2014). Metabolic activation of
3504 intrahepatic CD8+ T cells and NKT cells causes nonalcoholic steatohepatitis and liver
cancer via cross-talk with hepatocytes. *Cancer Cell* 26, 549–564.
32. Yau, T., Park, J., Finn, R.S., Cheng, A., Mathurin, P., Edeline, J., Kudo, M., Han, K.,
Harding, J.J., Merle, P., et al. (2019). CheckMate 459 : A Randomized , Multi-Center
Phase 3 Study of Nivolumab vs Sorafenib as First-Line Treatment in Patients With
Advanced Hepatocellular Carcinoma. In ESMO Congress, p.
33. Zen, Y., and Yeh, M.M. (2018). Hepatotoxicity of immune checkpoint inhibitors: a
histology study of seven cases in comparison with autoimmune hepatitis and
idiosyncratic drug-induced liver injury. *Mod. Pathol.* 31, 965–973.
34. Zhu, A.X., Finn, R.S., Edeline, J., Cattan, S., Ogasawara, S., Palmer, D., Verslype,
C., Zagonel, V., Fartoux, L., Vogel, A., et al. (2018). Pembrolizumab in patients with
advanced hepatocellular carcinoma previously treated with sorafenib (KEYNOTE-
224): a non-randomised, open-label phase 2 trial. *Lancet Oncol.* 2018, 1–13.

Author Rebuttals to Initial Comments

SHORTENED AUTHOR REBUTTAL

(please note that the authors have quoted the reviewers in black and responded in blue)

**Referee #1 (Remarks to the Author):**

Using two different mouse models of NASH-induced HCC as well as data from patients with NASH-
associated HCC, the authors suggest the concept that CD8+PD1+ T-cells promote NASH development
and that treatment with checkpoint inhibitors may release the brake in these NASH-promoting cells,
resulting in disease exacerbation and more HCC, which they proposed is confirmed by their findings of
absent response to checkpoint inhibitors Nivolumab and Pembrolizumab in patients with NASH-
associated HCC but not in patients with HCC due to other causes. While the analyses are carefully
performed and raise the question of harmful effects of checkpoints in NASH-associated HCC, both the
mouse and patient studies have major limitations, and it cannot be excluded that this paper sends the
wrong message to the community and will negatively impact the field.

We thank Referee #1 for appreciating that our experiments have been “carefully performed”
experiments as well as for outlining the potential clinical impact of our study on PD-1 targeted
immunotherapy in HCC. Also, we thank Referee #1 for pointing out the current limitations of
the applied mouse models and clinical cohorts of our study, which we have taken utmost
seriously and improved both. Statements on the role of checkpoint inhibitors in non-viral
etiologies in HCC have been tempered, but nonetheless reflect the results of the meta-
analysis, which is aligned with the pre-clinical findings.

**(i)** We have added a third preclinical mouse model of NASH with NASH to HCC transition
(Gomes et al., 2016; Tummala et al., 2014). Analysis of this model corroborated the link
between CD8+PD1+ T-cells and NASH development

**(ii)** We have extended our preclinical experiments with six novel treatment groups and
performed in detail analyses on the mechanism and functional link of liver damage,
inflammation, and responsiveness to anti-PD1-targeted immunotherapy in liver cancer.

**(iii)** We have added human clinical data sets (with 1656 HCC patients on immunotherapy
involving the important clinical trials - IMbrave 150; Checkmate 459; Keynote 240), enlarged
our initial retrospective clinical cohort, and validated results obtained from this cohort in a
second cohort of HCC patients under immunotherapy. Moreover, we corroborated our findings
of CD8+PD1+ increasing by NASH in now in total 3 independent patient cohorts across Europe
by flow cytometry or single-cell RNA-seq. Furthermore, we have performed CYTOF and
scRNA Seq analysis of lymphocytes from livers derived from human NAFLD/NASH and
steatosis and compared these data with our preclinical models - corroborating our data.

In particular, we have now added a meta-analysis including 1656 HCC patients with different
underlying etiologies (viral and non-viral) treated with immunotherapy derived from three large
clinical trials (**Figure 6, Extended Data 30-32, Supplementary Table 7 and Rebuttal Figure**
**1d,e and 2-4**). (Total number of patients in the combined cohort: 1656. One patient in the
CheckMate-459 had unknown etiology, and could therefore not be included in the quantitative
meta-analysis). We conducted this meta-analysis to support the experimental data suggesting

Research for a Life without Cancer

that anti-PD1/anti-PDL1 checkpoint inhibitors would have a distinct effect in non-viral (NASH-
related) HCC as opposed to viral-related HCC (**Figure 6, Extended Data 30-32** and
**Supplementary Table 7** and **Rebuttal Figure 1d,e and 2-4**). Out of eight studies identified in
the search, only three fulfill the pre-established criteria (**Extended data 30** and **Rebuttal**
**Figure 2**), including a total of 1656 HCC patients.

These randomized controlled trials (RCT) included **A**) CheckMate-459 (Yau et al., 2019), a
first-line, randomized, sorafenib-controlled trial testing nivolumab (an anti-PD1 monoclonal
antibody) in monotherapy (n=742), **B**) IMbrave150 (Finn et al., 2020), a first-line, randomized,
sorafenib-controlled trial testing the combination of atezolizumab (an anti-PD-L1 monoclonal
antibody) and bevacizumab (an anti-VEGF-A monoclonal antibody) (n=501), **C**) KEYNOTE-
240 (Finn et al., 2019), a second-line, randomized, placebo-controlled trial testing
pembrolizumab (an anti-PD1 monoclonal antibody) monotherapy. All three trials reported a
subgroup analysis of survival data stratified according to disease etiology: hepatitis B virus
(HBV), hepatitis C virus (HCV), and non-viral, including both NASH and alcohol intake.

**First**, we analyzed whether checkpoint inhibitors were effective in each of three etiologies
(HBV, HCV, and non-viral) and then compared the efficacy by categorizing patients with viral
vs non-viral etiology HCC in all three phase III studies including a total of 1656 patients.
Immunotherapy was superior to the control arm in both HBV (n= 574; p=0.0008) and HCV-
related HCC patients (n= 350; p=0.04), **but not in non-viral** HCCs (n=737; p=0.39). The
magnitude of the benefit with checkpoint treatment according to etiology was significantly
better in viral etiology (pooled HBV and HCV cases) [HR: 0.64; 95%CI 0.48-0.94] than non-
viral etiology [HR: 0.92; 95%CI 0.77-1.11]; p of interaction= 0.03 (**Rebuttal Figure 1e**). Then,
we dissected the specific effect by each viral type in a subgroup analysis. Comparison of
magnitude of effect was significant comparing HBV vs. non- viral etiology (n=1311; p
interaction= 0.03), and there was a non-significant trend for HCV vs. non-viral etiology
(n=1082; p of interaction=0.14) (**Rebuttal Figure 3**).

**Second**, considering that two out of three RCT were conducted in first-line treatment of
advanced HCC with a homogeneous control arm (sorafenib), we conducted a subgroup
analysis specifically with these two studies (n= 1234). This approach allowed us to control for
biases related to the study population and distinct control arms. Immunotherapy was superior
to sorafenib in both HBV (n= 473; p=0.03) and HCV-related HCC patients (n= 281; p=0.03),
but not in non-viral HCC (n=489; p=0.62). (**Rebuttal Figure 4**). The magnitude of the
checkpoint treatment effect vs sorafenib according to etiology showed a non-significant trend
favoring viral etiology (n=754; HR: 0.61 (95%CI 0.40-0.93)] when compared to non-viral
etiology [n=489; HR: 0.94 (95%CI 0.75-1.18)] (p of interaction= 0.08) (**Rebuttal Figure 4a**). As
a result, we have included these data in the resubmitted manuscript (**Figure 6**).

Research for a Life without Cancer

Based on these data we want to point out that it is - as indicated by Referee#1 - of the highest
importance to us to specifically define/tone down appropriately the message of our manuscript:
Our manuscript does not indicate that immunotherapy is not beneficial for HCC patients, rather
demonstrates that HCC patients with viral etiologies do respond well and achieve survival
benefits - however, that patients with non-viral etiologies (e.g. NASH) do not achieve a
significant outcome benefit. We propose to stratify HCC patients who are very likely to profit
from immunotherapy and strengthen the argumentation to use immunotherapy in specific
cohorts of HCC patients. We agree with Referee#1 that this needs to be articulated
appropriately, not to deliver wrong messages but to be very specific.

**Specific points:**

1. The NASH-HCC mouse models represent a major weakness of this paper and may lead to premature
conclusions on the effect of PD-1 therapy in NASH-associated HCC. While the employed mouse models
may be among the best to study various aspects of NASH, several limitations preclude them from
serving as useful preclinical models for HCC:

We thank Referee #1 for appreciating the used NASH-HCC models as “among the best to
study various aspect of NASH”, and we agree in general that studies in preclinical models have
their limitations, especially in the context of chronic inflammation-induced cancer. These
limitations of preclinical models are pronounced if mouse models are not used chronically (e.g.
≥ 1 year). However, we would like to point out that the model(s) used in our paper reflect
sporadic liver cancer development with similar immune cell signature, pathophysiology, and
the heterogeneous genetic landscape found in humans (Ma et al., 2016; Malehmir et al., 2019;
Wolf et al., 2014 - and the data reported in this manuscript). In response to Referee #1, we
have performed synteny analyses comparing HCC nodules from individual mice with human
HCC (**Extended Data 6** and **Rebuttal Figure 5a,b**). These data indicated no significant
changes in genomic aberrations between human HCC and mouse liver tumors.

1a. Many mouse models of cancer are simply not responsive to checkpoint inhibition because of low
mutational load and lacking tumor antigens/neo-antigens. The authors do not provide evidence that the
employed models have a mutational load that is at least as high as in that seen in HCC patients.

We thank and agree with Referee #1 for pointing out the possible unresponsiveness of clinical
models to checkpoint inhibition due to low mutational load. The mutational load HCC of most
conventional preclinical models is indeed very low, or lower compared to human HCC. This is
the case, in particular when taking into account liver cancer models triggered through
transgenesis, e.g. c-myc transgenic mice or preclinical mouse models with hydrodynamic tail
vein injection (HTDVi) of oncogenic drivers and tumor suppressors. In those models, pre-
existing genetic drivers and tumor suppressor deficiencies can be a major drawback
concerning additional mutations and increased mutational load.

Research for a Life without Cancer

In a chronic model of liver inflammation, we could show that mutational load increases over
time - comparing 9, 12, and 15 months (Finkin et al., 2015). Our chronic, spontaneous NASH-
HCC models develop liver cancer in the absence of specific genetic drivers – but rather through
chronic liver damage triggering DNA instability, ER and mitochondrial stress, accumulating
genetic hits over time stochastically triggering liver cancer formation, like has been shown in
human NASH (Boege et al., 2017).

In light of the important question of Referee #1, we have now included a further genetic
screening of 19 mouse HCC nodules in our revised manuscript and compared them to human
HCC nodules and their mutational landscape (**Extended Data 6** and **Rebuttal Figure 5a,b**).
Data from this study confirm that quality, degree of heterogeneity, and load of chromosomal
aberrations (gains and deletions) of the used NASH to HCC mouse model is similar to human
HCC (Wolf et al., 2014 and this manuscript). Furthermore, we would like to point out, that
overall in human HCC so far a responder rate of 17-20% for PD-1-targeted monotherapy was
observed, potentially due to a generally low amount or lack of broad-scale tumor antigens in
HCC (El-Khoueiry et al., 2017; Zhu et al., 2018).

1b. The mouse model - albeit taking over a year - is not comparable to HCC development in patients,
which takes decades and mostly occurs in the setting of advanced fibrosis or cirrhosis (even though a
subset of NASH-associated HCC patients do not have cirrhosis, most of them have advanced fibrosis).
Importantly, in most of these patients, the underlying NASH is much less activate than in earlier disease
stages/burnt out - meaning that the risk of increasing NASH activity and thereby worsening not only
NASH but also increasing NASH-HCC is much lower and possibly not even relevant. The authors'
conclusions would be relevant if one employed checkpoint inhibitors for HCC prevention but are likely
not applicable to patients except for those, in whom HCC develops in the absence of cirrhosis and with
high NAS.

We thank Referee #1 to point out the limitations of preclinical models in comparison to patient-
derived data. We agree that preclinical models do not take decades to develop HCC (averages
mouse life-time ~ 2 years). However, mouse models have helped in the identification of
molecular and cellular mechanisms leading to liver cancer (Ringelhan et al., 2018) - and if used
in a long term fashion - up to 2 years - they do recapitulate in part the chronicity of inflammatory
etiologies driving liver cancer. Moreover, mouse liver cancer occurs in age comparable to the
life-span of patients (we applied 12 - 15 months of NASH-diet feeding months from 2 months
of age onwards), which is comparable with the 4th to 5th life decade in humans regarding the
age of HCC onset/HCC disease (Llovet et al., 2016). We would like to highlight, that preclinical
models implemented in our study develop fibrosis to different degrees (mostly mild peri-cellular
fibrosis to periportal streets and cirrhosis (Malehmir et al., 2019; Wolf et al., 2014)).

Thus, we agree with Referee #1, that the preclinical model might represent a patient subgroup
developing HCC in the background of fibrosis. We agree with Referee#1, that underlying NASH
in HCC patients might be less activated compared to earlier stages and burnt-out.

Research for a Life without Cancer

Of note, clinical state-of-the-art care includes the use of corticosteroids for the treatment of
adverse effects (Weiler-Normann and Lohse, 2016), which can also induce NASH-like
pathologies. Thus, understanding mechanisms of underlying NASH in NASH-HCC in
preclinical models is of vital interest. Furthermore, current studies explore checkpoint inhibitors
for HCC as prevention of recurrence (Kudo, 2018).

We take this point of Referee #1 utmost seriously and devised importance for this critique in
the discussion section. We toned down our interpretations from human cohorts analyzed in a
retrospective design, although we believe the points raised in our manuscript address
important topics like a potential stratification for etiology, the need for biomarkers, and clinical
awareness of potential unfavorable side-effects of checkpoint inhibitor usage (Kim et al., 2020).
In line with the suggestion of Referee #1 to explore the limitations of our mouse models and to
understand the link between liver inflammation and tumor development better, we have re-
analyzed our mouse data sets to dissect potential correlations of fibrosis, tumor size, tumor
nodule number, flow cytometry data of livers, ALT, NAS, CD8, and PD-1 expression using
artificial intelligence, machine learning and neuronal networking (**Figures 1 and Extended**
**Data 4 and 24 and Rebuttal Figure 6 and 7c,d**). Moreover, we have added a third NASH-
HCC mouse model, which corroborates the link between the amount of CD8+, PD1+ T-cells,
and NASH (**Extended Data 3i and Rebuttal Figure 8i**).

Of note, we now underlined that our preclinical NASH models recapitulate in part the alterations
of hepatic immune cells in NASH by performing correlative analyses and machine learning of
liver-derived lymphocytes of NASH patients by CYTOF, classical flow cytometry, and scRNA-
seq (**Figure 5, Extended Data 25-27 and Rebuttal Figure 9-12**). These analyses demonstrate
that the pro-tumorigenic T cell population found in livers of preclinical NASH mouse models
(CD8+PD1+CXCR6+) are also found in / and correlate with NASH in human livers
(CD8+PD1+CD103+).

2. In relation to above-described limitations of the model, the paper does not sufficiently focus on dual
functions of CD8+PD1+ T-cells, promoting NASH but possibly also restricting HCC. These functions are
likely to occur at different stages in patients.

We thank Referee #1 for this important concern. We agree that the effects of CD8+PD1+ cells
are executed at different time points. However, we would like to draw attention to the point that
immunotherapy is considered to boost pre-existing inflammation (determined e.g. by
evaluation of liver infiltration by immune cells using immunohistochemistry or flow cytometry
for CD3, CD8, and PD-L1). Our data rather indicate that this certain population has no impact
in restricting HCC development - in the context of NASH - and even immunotherapy. In fact,
we show that depletion of CD8+ T-cells in NASH prevents NASH to HCC transition.

Thus, CD8+PD1+ T cells drive NASH, which is exacerbated in the context of anti-PD1-related
immunotherapy. We have now pointed this out more clearly, executed novel experiments to

Research for a Life without Cancer

underline this point of early (NASH) and late time points (NASH to HCC transition), analyzed
these cells in the context of human NASH and further discussed this in the discussion section.
To mirror the clinical status of the majority of patients at the time of diagnosis, we performed
PD-1-targeted checkpoint inhibition in mice with pre-existing liver tumors (**Extended Data 6**
**and 7** and **Rebuttal Figure 5 and 13**) and performed now MRI-guided follow up.
Our data clearly show, that anti-PD1 or anti-PDL1-related immunotherapy does not stop or
revert tumor burden but rather supports further tumor abundance. In contrast, when anti-CD8
antibody therapy was applied, it decreased tumor incidence and thus development (**Figure 2,**
**Extended Data 8** and **Rebuttal Figure 14a-g,q** and **15**). Furthermore, we underlined the
importance of hepatic CD8+ T-cells abundance driving NASH-induced hepatocarcinogenesis
by antibody-based treatments in our mouse model (anti-CD8/anti-NK1.1, anti-CD4, anti-TNF;
**Figure 2 and 4, Extended Data 8, 9, 20-23** and **Rebuttal Figure 14, 15, 16k-n, 17-21**), as
well as cross-referencing to the co-submitted manuscript Dudek et al., which describes
molecular mechanisms of CD8+ T-cell-mediated liver damage. Additionally, we dissected
CD8+ T-cell mediated mechanisms driving NASH-induced hepatocarcinogenesis in PD1-
targeted immunotherapy by antibody-based treatments (anti-CD8/anti-PD1, anti-TNF/anti-
PD1, anti-CD4/anti-PD1; **Figure 4, Extended Data 20-23** and **Rebuttal Figure 16k-n, 17-21**).
These data indicated that the abundance of CD8+ T-cells, as well as CD8+ T-cell-derived TNF
plays an important role in boosting liver cancer in the context of NASH/HCC related
immunotherapy. Of note, velocity analyses of scRNA-seq for transcriptional activation, or
proteome analyses of sorted cells could not detect different phenotypes between CD8+PD1+
T-cells derived from mice fed CDHFD with NASH or CDHFD treated with an anti-PD1 related
therapy in the context of HCC development, indicating that the main proportion of CD8+PD1+
T-cells in our preclinical models drive hepatocarcinogenesis and do not restrict HCC (**Figure**
**4, Extended Data 4 and 24** and **Rebuttal Figure 6, 7 and 16**).
Further, our data show that anti-PDL1 therapy lead (**Extended Data 7** and **Rebuttal Figure**
**13**) to the same effects as observed in the anti-PD1 therapy (**Extended Data 6** and **Rebuttal**
**Figure 5**) or in the context of our analyses using PD1 knock-out mice developing NASH/HCC
(**Figure 3, Extended Data 14** and **Rebuttal Figure 22a,b** and **23**).
Data that have not been included in the initial submission of the manuscript indicate that PD-1
targeted immunotherapy-induced hepatic inflammation triggers the enrichment of central
memory-like cells (CD44+CD62L+CD8+) but not T-cells with a naïve character
(CD62L+CD8+) (**Extended Data 6** and **Rebuttal Figure 5n**). This enrichment of memory-like
CD44+CD62L+CD8+ T-cells can be explained by one of two options: these cells might be
expanded and infiltrate the liver upon the anti-PD-1 targeted immunotherapy to either drive
hepatic inflammation or these memory-like T-cells might be indicative of a subset of T-cells
reactive to tumor-associated antigens and thus of CD8+ T-cells of a dual role (**Extended Data**

Research for a Life without Cancer

**6 and Rebuttal Figure 5n**). In respect of the co-submitted manuscript Dudek et al., CD8+ T-
 cells drive liver damage and liver cancer in NASH in an antigen-independent manner. Thus,
 enrichment of memory-like CD44+CD62L+CD8+ T-cells upon PD-1 targeted immunotherapy
 might argue in favor of a dual role of CD8 T-cells. However, tumor size, tumor number per
 liver, and tumor incidence are not affected by increased CD44+CD62L+CD8+ T-cells, arguing
 against a tumor restricting function of CD8 T-cells in this context. We have improved cross-
 referencing of the revised manuscript with the co-submitted manuscript (Dudek et al.).

Data described in this manuscript demonstrate that the NASH-induced microenvironment
 drives hepatic inflammation in a TCR-independent manner and thus rather describes a
 mechanism that activates CD8+T-cells downstream of the TCR through environmental
 signaling (e.g. acetate, IL21 signaling), arguing against a tumor antigen-specific CD8+ T-cells
 mediated HCC restriction in the context of NASH. It is exactly these CD8+ T-cells which –
 altered by the NASH liver microenvironment acquired a pro-tumorigenic phenotype – we can
 detect also by analysis of the ICF signature. The latter is predictive of inflammation triggered
 liver cancer in humans. Notably, CD8 depletion eliminates this signature, strongly underlining
 that CD8 T cells are the main source of driving the pro-tumorigenic environment.

3. The data on the NASH- and NASH-HCC-promoting role of CD8+ T-cells is similar to a previous study
 from the last author (Wolf et al, Cancer Cell). Hence a number of the findings presented in this
 manuscript are incremental with, adding PD1 into this context, with somewhat expected results, as well
 as novel techniques such as scRNA-seq.

We thank Referee #1 for the opinion on the progress we tried to achieve with this manuscript
 as a follow-up study (Wolf et al., 2014). We politely disagree with the statement of Referee #1
 – that indicates “...are incremental with, adding PD1 into this context, with somewhat expected
 results, as well as novel techniques such as scRNA-seq.“, because:

**(i)** Our presented data show for the first time that CD8+PD1+ T-cells and their behavior in
 the context of immunotherapy and metabolic syndrome affect liver cancer in an unexpected
 manner – CD8+PD1+ T cells are pro-tumorigenic in this context – which very likely has clinical
 implications. Identification of increased hepatic abundance of unconventional activated
 resident-like CD8+PD-1+ (e.g. CXCR6+, TOX+, TNF+), but not a change of quality in these
 cells are the hepatocarcinogenesis-driver in the context of NASH is novel – and can be found
 also in the human situation (e.g. two IHC-cohorts across Europe comparing viral vs.
 NAFLD/NASH-HCC, one IHC cohort dissecting the abundance of cells depending on NASH
 pathology severity; also comparing control vs NAFLD/NASH patient samples by scRNA Seq,
 CYTOF and flow cytometry).

**(ii)** Our data expand current knowledge of NASH pathology-associated mechanisms (e.g.
 auto-aggression in a TCR-independent manner with the co-submitted manuscript Dudek et al.,
 corroborating the data in total 3x preclinical models of NASH). Furthermore, we tested this

Research for a Life without Cancer

mechanism hypothesis on a functional level by various antibody-based treatments (PD-L1-
targeted immunotherapy; combination therapy of anti-TNF/anti-PD-1, anti-CD4/anti-PD-1, anti-
CD8/anti-PD1) and now identify that it indeed is TNF and CD8 T cells that promote liver cancer
in the context of PD1-related immunotherapy.

**(iii)** Novel comparison/corroborations and in-depth analysis of T-cell populations in human
and mouse NASH by scRNA, flow cytometry and CYTOF. We did not expect a link between
resident-like CD8+PD1+ cells in the progression of NASH pathology and NASH-induced
hepatocarcinogenesis, as well as the correlation of preclinical model to patient data, identifying
NASH as an etiology of unfavorable predictor of response (e.g. the meta-analysis of 1656
patients corroborates non-viral (NASH-related) HCC compared to viral-HCC as less
responsive to immunotherapy (**Figure 6, Extended Data 30-32 and Rebuttal Figure 1d,e and**
**2-4**), as well as our own small retrospective NASH-HCC vs other-etiological-HCC cohort, which
was validated in a second validation cohort of HCC-patients under immunotherapy (**Figure 6,**
**Supplementary Table 9 and Rebuttal Figure 1f,g**).

4. The human data are based on a very small and poorly analyzed cohort of patients with NASH-
associated HCC (n=10-11). While the underlying question is important, pairing data from this small
cohort with the data from the mouse model with its above-described limitations and confounders may
send a wrong and potentially deleterious message to the community, and much more careful analysis
as well as larger cohorts are needed to put the provided message on a solid scientific foundation: The
authors should analyze outcomes for NASH-HCC patients with or without cirrhosis to account for the
possibility of worsened NASH in patients without cirrhosis (for which the cohort is much too small).

We thank Referee #1 and fully agree, that the presented retrospective
Nivolumab/Pembrolizumab-treated NAFLD/NASH-associated HCC cohort - although unique
for Europe where treatment is not officially licensed - is too small for subgroup analysis for
patients. We have taken this point raised utmost seriously. Thus, we have strengthened our
hypothesis of non-viral (NASH-related) HCC being less responsive to immunotherapy by a
meta-analysis including patients of the three most important clinical trials (1656 patients,
**Figure 6, Extended Data 30-32 and Rebuttal Figure 1d,e and 2-4**).

Moreover, we have increased the number of patients in our initial clinical cohort from 65 to 130
HCC patients under anti-PD(L)1-targeted immunotherapy and validated our results in a second
cohort of 118 HCC patients under PD(L)1-targeted immunotherapy (**Figure 6, Supplementary**
**Table 9 and Rebuttal Figure 1f,g**).

A disadvantage by nature of a retrospective analysis of cohort across multiple centers is, that
clinical material that would have the potential to characterize in patient subgroups (e.g.
worsened NASH) was not sampled. Furthermore, no paired biopsies or other biological
materials (e.g. blood or serum) before/after immunotherapy were taken in this cohorts for HCC
patients, making characterization of treatment response at the single patient resolution and
thus subgroups impossible in this retrospective cohort. Therefore, we decided to investigate

Research for a Life without Cancer

the outcomes for BCLC-C NAFLD/NASH-HCC vs other-etiological-HCC patients with cirrhosis
and observed, that NAFLD/NASH-HCC have significantly reduced overall survival compared
to other-etiological-HCC in this retrospective study. Of note, multivariate analyses identified
NAFLD/NASH as an independent factor for treatment response (**Supplementary Table 9**).
We validated these results in a second independent cohort of 118 under PD1-targeted
immunotherapy based in North America, which included additional n= 11 patients with NASH-
HCC under immunotherapy, corroborating that NASH/NAFLD is a negative predictor to
immunotherapy (main text). We now have toned down the conclusions of our retrospective
cohort in the manuscript and would like to point out, that larger cohorts and prospective clinical
trials are of utmost importance for the scientific community.

311 A. A cohort of n=10-11 NASH-associated HCC patients is unacceptable. Many of the parameters such
as PFS are not significant and it cannot be excluded that inclusion of a larger number of NASH-HCC
patients may change the data significantly.

We agree with Referee #1, however we would like to point out attention, that prominent trends
or effects can also be seen in small retrospective cohorts as well. Although unique for Europe,
where treatment is not officially licensed yet, the complete cohort we have gathered is too small
for subgroup analysis for patients.

We decided to leave out the non-significant data of TTP and PFS in our manuscript. Moreover,
upon recruiting the validation cohort of 118 HCC-patients under immunotherapy we decided
to not show TTP and PFS, but instead the multivariate analysis (**Supplemental Table 9**).
However, we are in line, that an increased patient cohort allows a more sophisticated analysis.
Thus, as mentioned in the previous comment, we increased our patient cohort (from 65 HCC-
patients to 130 HCC-patients) and validated the results in the second cohort of 118 HCC-
patients under PD(L)1-targeted immunotherapy. Furthermore, we would like to highlight the
message from the performed meta-analysis of 1656 patients, also pointing towards identifying
NAFLD/NASH as a negative predictor of immunotherapy response in HCC. Still, the cohorts
are small, and thus, we toned down the conclusions drawn from this retrospective cohort
analyses (added in the main text, **Figure 6**).

B. The authors do not answer the question whether the differences in survival are due to failed
checkpoint therapy or due to other differences between the two cohorts. Most likely, the differences in
survival would persist if the authors removed all responders from the “other etiologies” group. Control
groups that did not receive checkpoint inhibitors are missing to determine if survival is different between
NASH and non-NASH HCC in patients who did not receive checkpoint inhibitors.

We thank Referee #1 for raising this important point of potential differences in survival due to
potential confounders. To address these issues, we have submitted our data to multivariate
analyses, which we included in an updated **Supplementary Table 9**. When we excluded
patients with a complete or partial response from the 112 patients with at least one follow-up

Research for a Life without Cancer

imaging, 86 patients were available for analysis (NAFLD, n=9; other etiologies, n=77). Median
OS was significantly shorter in the NAFLD group (5.4 (95%CI, 1.7-9.1) months vs. 10.3
(95%CI, 8.2-12.4) months; p=0.006), as was median TTP (2.4 (95%CI, 2.1-2.7) months vs. 3.9
(95%CI, 2.5-5.4) months; p=0.008), and median PFS (2.4 (95%CI, 1.9-3.0) months vs. 3.7
(2.3-5.1) months; p=0.035). These data suggest that the improved outcome of non-NAFLD
patients is not only driven by the better response rate observed in these patients. However,
the interpretation of these data due to the size of the underlying cohorts needs to be taken with
caution. Like mentioned before, we have now included a meta-analysis with appropriate
control cohorts, identifying immunotherapy vs control for viral HCC as favorable treatment
(HR(viral)= 0.64), in contrast, non-viral-HCC show less benefit (HR(non-viral)= 0.92). In this
meta-analysis patients with NASH-HCC and Non-NASH HCC who did not receive checkpoint
inhibitors are included as receiving either sorafenib (in RCT of front-line) or placebo (in RCT in
second-line). We thank Referee #1 for pointing out the lack of appropriate control groups (e.g.
NASH-HCC vs. different etiology-induced HCC under Sorafenib/different multi-kinase
inhibitors as a second/third-line therapy). Although of extreme interest for public health and
public knowledge, we described this important issue in our discussion and to the best of our
knowledge there are no NASH-HCC treated cohorts available (apart from, possibly, inside of
the big pharma-industry), which would allow an adequate control arm. Available cohorts (El-
Khoueiry et al., 2017; Finn et al., 2019, 2020) are only differentiating between viral vs. non-
viral etiologies, which combine ASH and NASH-induced HCC.

C. Is there any indication of increase NASH activity in patients receiving Pembro or Nivo?

We thank Referee #1 for this important comment. We have added baseline AST and ALT in
the pre-existing and novel cohorts (**Supplementary Table 8**). Like previously mentioned, the
character of the retrospective studies did not allow to obtain paired biopsies before/after
immunotherapy, and bigger cohorts of prospective clinical trials are needed.

D. There is no proper analysis of confounding factors.

We thank Referee #1 for pointing out this lack of analyses in our initial submission. We have
now performed multivariate analyses, which we included in the main text and in an updated
**Supplementary Tables 8 and 9**. In short: Macrovascular invasion, a negative prognostic
factor in HCC, was less frequent in NAFLD patients (23% vs 49%). NAFLD patients received
immunotherapy more often as first-line therapy (46% vs. 23%), and the proportion of patients
receiving the combination of atezolizumab plus bevacizumab, the only immunotherapy-based
treatment that has succeeded in a phase III trial of advanced-stage HCC so far, was higher in
the NAFLD cohort (23% vs. 5%). Despite these more favorable characteristics, immunotherapy
was less effective in patients with NAFLD, which translated into a worse overall survival (OS)
for the NAFLD cohort: 5.4 (95%CI, 1.8-9.0) months vs. 11.0 (95%CI, 7.5-14.5) months

Research for a Life without Cancer

(p=0.023). Adjusting for other well-known prognostic factors (Child-Pugh class, macrovascular
invasion, extrahepatic metastases, performance status, and alpha-fetoprotein (AFP)), NAFLD
remained independently associated with worse survival (HR 2.6 (95%CI, 1.2-5.6; p=0.017).
These data indicate that PD-1-targeted immunotherapy in HCC patients with concomitant
NASH might lead to unfavorable effects.

E. Another problem is mixing Pembro and Nivo groups. Even though the target is the same, the authors
need to provide subgroup analysis for this and increase the number far beyond what they have to make
any meaningful conclusions in these subgroups.

We thank Referee#1 for this comment. Nivolumab and pembrolizumab are mostly considered
comparable in solid tumors. Performing a subgroup analysis based on Nivolumab and
pembrolizumab is simply not feasible nor realistic in HCC, even more so in NASH-HCC.

We would like to draw attention to other studies performed in solid tumors (NSCLC (Cui et al.,
2020), and Melanoma (Moser et al., 2020)) that show a similar efficacy (although the overall
level of evidence is low): We agree with this point of Referee #1, which we so far have not
been able to make clear. Similar to the previous point (4A.), our retrospective analyses of the
patient cohorts is too small to address these concerns in an in-depth manner.

We agree with Referee #1, that both Nivolumab and Pembrolizumab are targeting the molecule
PD-1, with similar response rates of 17-20% as monotherapy in HCC (El-Khoueiry et al., 2017;
Zhu et al., 2018). The consensus in the literature is to combine both PD-1 targeting antibodies
and pool their results. Moreover, we validated these results in the second cohort of 118 treated
immunotherapy treated HCC-patients, including n= 11 NASH-HCC patients.

F. Characterization of patients is insufficient - how were other liver diseases excluded, including ALD,
which is not trivial, and especially important in such small cohorts?

We thank Referee #1 for raising this important point and would like to draw the attention, that
criteria for the retrospective patient cohort are described elsewhere (Scheiner et al., 2019).

We have especially analyzed the parameters to identify NAFLD/NASH from viral (e.g. patient
history, liver histology, MRI, obesity). It should be indicated that the differences between NASH
and BASH are indeed difficult to account for – less so when differentiating between NASH and
ASH. Furthermore, we toned down our statement regarding the effects of immunotherapy in
our patient cohorts/case reports in the revised manuscript.

5. Do the authors get the same results when blocking CTLA-4 - which was, even though not approved
for HCC - the first approach and published study to show efficacy of checkpoint inhibitors in HCC?

We thank Referee #1 for this important question and would like to draw the attention to a phase
II trial combining TACE with Tremelimumab that did not differentiate between underlying
etiology for the patient outcome or immune population (Agdashian et al., 2019; Duffy et al.,
2016). This phase II trial showed a similar response rate (21-26%) compared to the 17-20%

Research for a Life without Cancer

response rate for PD-1 targeted monotherapy (El-Khoueiry et al., 2017; Zhu et al., 2018).
Clinical consensus for immunotherapy indicates increased hepatotoxicity of CTLA-4-
compared to PD-1-targeting immunotherapy (Zen and Yeh, 2018), arguing in favor of PD-
1/PD-L1-targeting immunotherapies for the future.

Although we observed in human Tregs cells CTLA-4 positivity by scRNA-seq and flow
cytometry, in our manuscript CTLA-4 expression was not identified as significantly different
between treatments as shown by scRNA-seq (**Figure 1**: CTLA-4 expression in CD8+ T-cells
comparing ND vs CD-HFD: FC= 0.1894, p= 0.0642; **Extended Data 5**: CTLA-4 expression in
CD4+ T-cells comparing ND vs CD-HFD: FC= 0.2173, p= 0.1431; **Figure 4 and Extended**
**Data 18**). In our mass spectrometry-based data set, we found no significant change of CTLA-
4 abundance (**Extended Data 5 and 18 and Rebuttal Figure 24e and 25e**), corroborating our
flow cytometry-based analysis, which had also low CTLA-4 expression in mouse or human
(**Extended Data 18 and 25 and Rebuttal Figure 10d,e and 25h**). Thus, we believe that the
application of CTLA-4-targeted immunotherapy is unlikely to cause a positive effect in our
preclinical model. We have discussed the potential use of targeting rather T-cell activation
(anti-CTLA-4) than exhaustion (anti-PD-1 or anti-PD-L1) in combination, or together with a
potential generation of tumor antigens by ablation strategies (e.g. TACE).

Research for a Life without Cancer

Referee #2 (Remarks to the Author):

In their manuscript, Pfister and colleagues aim to show that CD8+PD-1+ T-cells expand during
progressing, diet-induced NAFLD and, upon treatment with anti-PD-1 antibodies, that these cells can
promote carcinogenesis by establishing an inflammatory tumor microenvironment in a diet-induced,
murine model of advanced NAFLD. Additionally, the authors observe a similar, intratumoral
CD8+CD103+PD-1+ T-cell subset in NASH-induced human HCC patients and claim that patients with
NASH-induced HCC respond worse to anti-PD-1 therapy compared to HCC of other origin. While the
seminal observation in this paper is intriguing, namely that anti-PD-1 treatment can exacerbate
tumorigenesis in a murine model of NASH-induced HCC, the authors fail to demonstrate clear causal
relationships between the implicated cell types, liver inflammation and tumor development in the vast
amount of the data they present, which therefore remain largely correlative. I will highlight my major
concerns below.

We thank Referee #2 for the concise and detailed comments and understanding of our aimed
key points to be delivered in the manuscript. Also, we thank Referee #2 for pointing out the
limitations of our study of correlative data interpretation rather than functional dissection. We
appreciate Referee`s #2 opinion, that our human cohort results lead to indications of a worse
response rate of NAFLD/NASH-induced HCC compared to non-NAFLD/NASH-HCC upon PD-
1 targeted immunotherapy. We would like to address the referee`s concerns in the following
section point-by-point:

1. In the reporting summary, the authors state that “Exclusion criteria was pre-established and the CD-
HFD fed mice which did not show the NASH phenotype, high ALT, AST and body weight, were excluded
from the analysis”. I fail to understand why this decision was taken as these mice offer valuable insight
in the author’s proposed mechanism. Do CD-HFD mice without overt signs of NASH have reduced
CD8+PD-1+ T-cells? Do these mice also less frequently grow tumors upon anti-PD-1 blockade? Do the
T-cells in the livers of these mice fail display an enhanced effector phenotype? Aside from the valuable
experimental insights that could be gained from these mice, the decision to exclude these CD-HFD but
non-NASH mice from analysis also invalidates any claim that links a given diet to a given phenotype
since mice that did not fit the authors’ desired phenotype were excluded.

We thank Referee #2 for the above questions. All mice were included in the respective
treatment – as stated in the paper, indicated by the large mouse data sets in **Figure 1-4** in
NAS, ALT, AST, and body weight. Thus, the statement “Exclusion criteria” is inappropriate
and a mistake made on our side and is corrected in an updated Reporting Summary. We fully
agree with Referee #2 that these mice “offer valuable insight in the proposed mechanism” and
this is actually why we have included all of them in our analyses.

To display the experimental range of mice fed 12 months CD-HFD, we have now performed
correlations of a large number of integrated parameters of each mouse (e.g. tumor incidence,
tumor size, tumor nodule number, immune-histochemistry, serology, flow cytometry data;
**Figures 1 and 4, Extended Data 4 and 24 and Rebuttal Figure 6, 7c-e, 16, and 26**): In more

Research for a Life without Cancer

detail, we have re-analyzed our data sets to dissect the potential correlations of CD8+ T-cells,
PD-1+ T-cells, ALT, fibrosis, NAS, tumor incidence, tumor nodule size, and effector phenotype
- by artificial intelligence and machine learning clustering.

We did not analyze the hepatic environment at time points 10, but after 12 months under diet,
after treatment finished, thus a paired analysis of mice with reduced CD8+PD-1+ T-cells and
their reaction to PD-1-targeted immunotherapy is not possible. In 12 months, CD-HFD-fed
mice CD8 (%CD45) and effector CD8 cells (CD8+CD44+CD62L-) correlate positively with
markers of severity of NASH pathology (e.g. ALT, AST, NAS), as well as tumor incidence
(**Extended Data 4** and **Rebuttal Figure 6**). In 12 months CD-HFD-fed mice polarization by
PD-1 of these CD8+ T-cells (CD8+PD-1+(%CD8)) correlate positively with ALT, AST, but not
significantly with NAS or tumor incidence, indicating that the hepatic abundance of CD8+PD-
1+ cells is important for NASH (e.g. CD8+PD-1+ (%CD45) correlates (Spearman correlation
$r= 0.3844$, $p= 0.0058$) with NAS, not reported in the paper).

Correlation data included in **Extended Data 24** and **Rebuttal Figure 7c-e** shows, that PD-1-
targeted immunotherapy correlates positively with markers of severity of NASH pathology (e.g.
ALT, AST, NAS), with tumor incidence and tumor numbers per liver, and hepatic CD8 T-cells
(e.g. by histology and flow cytometry), effector CD8 cells (CD8+CD44+CD62L-), as well as the
polarization of CD8+PD-1+(%CD8). These data indicate similar to the Referee's comment,
that mice with reduced hepatic CD8 T-cells and thus also less effector CD8 cells
(CD8+CD44+CD62L-) develop fewer tumors, and that in our data set reduced numbers of
hepatic CD8+PD1+ T-cells result in lower NAS and lower tumor incidence upon PD-1-targeted
immunotherapy (**Extended Data 24** and **Rebuttal Figure 7c-e**).

We agree with Referee #2, that these data allowed us to gain valuable insights understanding
the phenotype, why some mice develop milder NAFLD/NASH when compared to experimental
controls submitted to similar times of diet feeding, and how this affected PD-1 blockade. We
would like to point out that mice develop NAFLD/NASH at 12 months post-diet start with an
incidence of 100% (please also see **Figures 1** and **Rebuttal Figure 26a-d**).

2. The data presented by the authors fail to demonstrate clear causal relationships. As an example, the
authors note in lines 341-343 that a pro-inflammatory hepatic environment is created by TNF upon anti-
PD-1 treatment, yet fail to show supporting evidence that this indeed drives "necro-inflammation" and
accelerated hepatocarcinogenesis. The authors should neutralize TNF in their in vivo models to
determine whether this molecule is indeed required for their phenotype, i.e., inflammatory
microenvironment, liver damage and increased tumorigenicity.

We thank Referee #2 for this very important point. We agree with the comment of Referee #2
and therefore have performed anti-TNF treatment in NASH mice with/without PD-1 targeted
immunotherapy (**Figure 4**, **Extended Data 20 and 21** and **Rebuttal Figure 16k-n, 18 and 19**).

Research for a Life without Cancer

Of note, data from these experiments demonstrate that TNF, derived from CD8+ T-cells is the
main driver of the pro-tumorigenic effects of T-cells in the context of immunotherapy in NASH
(**Figure 3** and **Rebuttal Figure 22e**).

Furthermore, we would like to highlight, that our manuscript correlates increased hepatic
abundance of CD8+PD-1+ T-cells upon PD-1-targeted immunotherapy as crucial for driving
hepatocarcinogenesis. Besides, we have now performed additional scRNA-seq and velocity
blot analyses from human patients with NAFLD/NASH or steatosis and compared those with
mouse immune cells. These data demonstrate high similarities between CD8+ PD1+ T-cells
derived from human and mouse NASH livers. Moreover, we would like to draw the attention of
this Referee to the improved cross-referencing to the co-submitted manuscript Dudek et al., in
which the authors also show that TNF is one key molecule driving increased CD8-dependent
hepatic pathogenesis.

3. Based on the authors' presented data, this problem can be further expanded. In Figure S9d and S9m,
the authors show an increase in the number of antigen-presenting cells and increased MHC-II
expression. Are these recruited upon liver inflammation? Are they required for liver inflammation?

We thank Referee #2 for raising the point about myeloid cells in the context of chronic
inflammation and would like to interpret the data shown in **Extended Data 11** and **Rebuttal**
**Figure 27** in comparison to **Extended Data 8** and **Rebuttal Figure 15**, which now indicates,
that antigen-presenting cells and increased MHC-II expression are a result of increased liver
inflammation upon PD-1 targeted immunotherapy. We would like to highlight our previous
study (Malehmir et al., 2019), which demonstrated, that myeloid cells are correlated with liver
inflammation and are recruited as a consequence of NASH development. Moreover, we have
shown by depletion of antigen-presenting cells, including Kupffer cells (by chlodronate
encapsulating liposomes) abrogates or prevents NASH development.

To address the point raised by Referee #2 more experimentally, we analyzed our mouse
cohorts in total by AI, which indicates that hepatic MHCII+ cells correlate positively with NASH
pathology (weight, NAS, ALT, AST, cholesterol, fibrosis by Sirius Red staining, hepatic
concentrations of MCP-1, CCL3, MIP-2, and IL-21) and MHCII+ as a marker of myeloid
activation on different subsets correlated predominantly in CD11b+CD11c+ (myeloid dendritic
cells (CD11b+CD11c+) with ALT, GOT, NAS in 12 months CD-HFD-fed mice (**Extended Data**
**4** and **Rebuttal Figure 6**). To dissect the Referees question in our experimental functional
antibody-treatment experiments (**Extended Data 24** and **Rebuttal Figure 7c-e**). MHCII+ cells
correlate positively with CD-HFD and CD-HFD+PD-1-targeted immunotherapy, as well as
NASH pathology (weight, NAS, ALT, AST, cholesterol, fibrosis by Sirius Red staining, hepatic
concentrations of MCP-1, CCL3, CCL4, MIP-2, and IL-21) in 12 months old mice. Moreover,
MHCII+ as a marker of myeloid activation on different subsets correlated for CD11b+MHCII+
and mDC+MHCII+ positive with PD-1-targeted immunotherapy, ALT, AST, NAS CCL4, and

Research for a Life without Cancer

MIP-2. pDC+MHCII+ and KC+MHCII+ cells correlated negatively in CD8-depleted and
CD8+NK1.1 co-depleted animals. The latter myeloid subset correlates positively with fibrosis
and tumor incidence when pooling the data of all treatments.

We would like to highlight our previous study (Malehmir et al., 2019), which showed, that
myeloid cells are correlated with liver inflammation and are recruited as a consequence of
NASH development. However, a genetic study using CCR2^{-/-} mice (impaired myeloid
recruitment upon inflammation) developed NASH and NASH-induced tumors; in contrast,
Rag1^{-/-} mice with functional myeloid but impaired adaptive immune compartments were
protected from NASH and NASH-induced tumors (Wolf et al., 2014). These data argue, that
myeloid cells are recruited to the liver, extend, and fine-tune liver inflammation.

4. In Figure S11 the authors show an increase in many inflammatory mediators upon anti-PD-1 therapy;
which of these are required for the accelerated carcinogenesis? While the authors propose a mechanism
based on liver inflammation leading to increased hepatocarcinogenesis upon anti-PD-1 blockade, they
provide little if any conclusive evidence for this hypothesis.

We thank Referee #2 for asking this important question. We believe that the inflammatory
mediators for increased hepatocarcinogenesis stem from the increase of CD8+ T-cells upon
anti-PD1 immunotherapy. Importantly, by performing depletion experiments of different T-cell
subsets – anti-CD8 or anti-CD4, we can demonstrate that the CD8+ T-cells but not CD4+ T-
cells are needed for driving hepatocarcinogenesis and driving the pro-tumorigenic effect of
anti-PD1-related immunotherapy (**Figure 4, Extended Data 20-23 and Rebuttal Figure 16,**
**18-21**). Of note, PD-1-targeted immunotherapy increases the hepatic abundance of
CD8+PD1+ T-cells in vivo (e.g. **Extended Data 11 and Rebuttal Figure 27d,e**), as well as
increases the number of CD8+PD1+ cells in vitro (**Extended Data 18 and Rebuttal Figure**
**25I**). To understand the nuances of the observed necro-inflammation, anti-PD1-related
immunotherapy, and liver cancer formation, we perform correlations analysis of fibrosis, tumor
nodule number, tumor size, ALT, NAS, CD8, and PD-1 expression by machine learning and
neuronal networking (**Figures 1 and 4, Extended Data 4 and 24 and Rebuttal Figure 6, 7c-**
**e, 16, 26h**).

We have analyzed the inflammatory environment looking into a specific signature (ICF) on the
transcriptional level in NASH mice with and without anti-PD1-related immunotherapy (**Figure**
**3 and Rebuttal Figure 22d**). This transcriptional ICF signature is a predictor of liver cancer
formation triggered through inflammation in humans. It can be stated that the altered
inflammatory signature of NASH livers in the context of anti-PD1-related immunotherapy
overlaps with a signature that from human patients is known to have a bad prognosis and high
correlation with inflammation triggered liver cancer. Importantly, upon CD8+ T cell depletion
the intrahepatic ICF signature is downregulated - demonstrating that CD8+ T cell-derived
inflammatory mediators might be linked with liver cancer formation.

Research for a Life without Cancer

Moreover, to identify factors secreted in relation to CD8+ T-cells in NASH livers (as identified
by their reduction upon anti-CD8 treatment) we have performed *in situ* RNA hybridization
analyses for several cytokines. Further, we have performed flow cytometry and RNA-seq of
hepatic tissues as well as scRNA-seq from human and mouse immune cells. Doing so, we
have identified T-cell derived TNF as a possible, important candidate for increased
hepatocarcinogenesis upon PD1-targeted immunotherapy.

To test this hypothesis on a functional level, we performed an anti-PD1/anti-TNF as well as an
anti-TNF treatment alone. These experiments demonstrate that TNF is a functionally important
cytokine contributing to the anti-PD1 antibody treatment mediated pro-carcinogenic effect.

Besides, we would like to draw attention to the improved cross-referencing to the co-submitted
manuscript Dudek et al., which shows that TNF and IL-15, a target downstream of IL-21 - both
upregulated upon anti-PD-1 therapy - are crucial mediators of CD8-mediated hepatic cell
death. In line, literature highlights the crucial role of TNF for hepatocarcinogenesis (Nakagawa
et al., 2014; Park et al., 2011; Pikarsky et al., 2004) and that anti-TNF treatment uncouples the
toxicity of CTLA-4/PD-1-targeted immunotherapy (Perez-Ruiz et al., 2019).

5. Some of the data the authors present seems internally inconsistent. As an example, the authors
postulate that the pro-inflammatory hepatic environment is responsible for the increase in liver cancer
incidence in anti-PD-1-treated mice, which they underscore by an increase in inflammatory cytokines in
the liver microenvironment (Figure S11). However, they also show that upon CD8 depletion, which
reduces cancer incidence, the inflammatory cytokines do not significantly reduce compared to the CD-
HFD diet mice alone. This implies that the inflammatory microenvironment is not actually responsible
for increased cancer incidence. How do the authors harmonize these findings?

We thank Referee #2 for his comment on the bivalence of cellular and micro-environmental
induced cell death, inflammation, and liver cancer formation. However, we firmly state, that our
data is not internally inconsistent, and have added several experiments that clarify the
mechanisms of action. We state, that anti-PD-1 therapy induces an increased hepatic
inflammatory microenvironment, indicated by a) increased abundance of hepatic immune cells
(mainly CD8+ and CD8+PD-1+ cells) (**Figure 2 and Extended Data 11 and Rebuttal Figure**
**14, 27**); b) by increased inflammation-associated cytokines (e.g. IFN γ , TNF, IL-21, IP10, MCP-
1, CCL3) (**Extended Data 13 and Rebuttal Figure 28**); c) on mRNA expression levels we
actually clearly see the increase in all pathways relevant for inflammation induced liver cancer
- as analyzed by the ICF-signature (**Figure 3 and Rebuttal Figure 22d**). Thus, we think, that
there are 2 components (first cells, like CD8+ T-cells and second, the inflammatory liver
environment) responsible for (increased) liver cancer incidence.

We agree with Referee #2 that initially this appears not logic - but we believe that a liver tissue
homogenate analysis cannot uncover the CD8+-T cell restricted cytokine changes, as other
immune cells will still produce inflammatory immune cells. This is indicated for example in

Research for a Life without Cancer

**Figure 3** and **Rebuttal Figure 22e**, which shows, that upon CD8 depletion TNF+ cells are
significantly reduced by *in situ* hybridization. Again, effects of the CD8 depletion manifests
strongly on mRNA expression level as pathways relevant for inflammation induced liver cancer
are strongly reduced - as analyzed by the ICF-signature (**Figure 3** and **Rebuttal Figure 22d**).
Moreover, as stated by the Referee it appears that anti-CD8 treatment alone did not reduce,
but anti-CD8/anti-PD-1 did reduce several chemokines indicative of a hepatic inflammatory
environment on protein level, that are responsible for myeloid cell attraction like MCP-1, CCL2,
CCL3, MIP-3a, or alarmins like IL-33 when compared to anti-PD1 alone (**Extended Data 10**
and **21** and **Rebuttal Figure 19c-e** and **29**).

Moreover, we want to point out that our data are also confirmed by the co-submitted manuscript
Dudek et al., revealing that the mechanisms of CD8+ T-cell mediated cell death is 1) CD8+ T-
cell dependent, 2) TCR independent, and 3) TNF is a crucial cytokine sensitizing the CD8+ T-
cell to get auto-aggressive and thus starts to mediate cell death.

We demonstrate that TNF is a marker of a pro-inflammatory, pro-carcinogenic hepatic
environment and that it is increased upon PD-1-targeted immunotherapy and remains high in
CD8+ depleted mice (**Extended Data 10** and **Rebuttal Figure 29**). However, CD8 depleted
mice lack tumor development (**Figure 2** and **Rebuttal Figure 14q**). In line with Referee #2 and
the co-submitted manuscript Dudek et al., we think, that the presence of CD8+ T-cells is
essential to drive hepatocarcinogenesis. We thus have performed the above mentioned CD8
depletion combined with PD-1 targeted immunotherapy to underline that CD8+ T-cells are
essential for increased hepatocarcinogenesis upon PD-1-targeted immunotherapy compared
to control mice under CDHFD diet (**Figure 4** and **Extended Data 20+21** and **Rebuttal Figure**
**16, 18** and **19**).

We have functionally strengthened data shown by Dudek et al. that TNF - as a marker of the
inflammatory environment - is crucial for sensitizing the hepatic microenvironment to CD8 T-
cell -mediated cell death by performing anti-TNF with/without PD-1-targeted immunotherapy.
This has allowed the interpretation and has been experimentally demonstrated that only an
inflammatory environment combined with the presence of CD8 T-cells drives increased
hepatocarcinogenesis upon PD-1-targeted immunotherapy (**Figure 4, Extended Data 20+21**
and **Rebuttal Figure 16, 18** and **19**).

Furthermore, to shed new light on potential compensatory immunological mechanisms of
CD4+PD-1+ T-cells in the context of PD-1-targeted immunotherapy, we have performed CD4
depletion with/without PD-1-targeted immunotherapy (**Extended Data 22 and 23** and **Rebuttal**
**Figure 20** and **21**). Notably, these experiments indicate that in contrast to CD8+ T-cells CD4+
T-cells do not play a major effector role in comparison to CD8+ T-cells in anti-PD1 related liver
cancer formation in the context of NASH and anti-PD1 treatment (**Figure 16n**).

Research for a Life without Cancer

6. Crucially, and related to my previous point, the authors also did not perform CD8 depletion
in the context of anti-PD-1 treatment to show that CD8 cells are indeed the cells that are
responsible for increased carcinogenesis upon anti-PD-1 therapy.

We thank Referee #2 for this important comment and fully agree that anti-PD-1 treatment in
the context of CD8 depletion is crucial for data interpretation and we included this experiment
in a revised manuscript (**Figure 4, Extended Data 20 and 21 and Rebuttal Figure 16, 18 and**
**19**). The combined anti-CD8/anti-PD-1 treatment has allowed an understanding on a functional
level, that indeed increased the hepatic abundance of CD8+PD-1+ T-cells upon PD-1-targeted
immunotherapy is crucial for driving hepato-carcinogenesis. Notably, this treatment reduced
NAS, liver damage and some cytokines (e.g. MCP-1, CCL2, CCL3, MIP-3a) that affect the
pathway of CD8+ T-cell activation by the liver environment (e.g. IL33, IL21).

7. At times, the authors are (highly) selective in the data they choose to discuss and interpret. As an
example, regarding Figure 1i, the authors describe the CD8+ T-cells in CD-HFD mice to demonstrate
profiles of cytotoxicity and effector function because of increased expression of GzmK/M and Pdcd1.
However, in the same plot shows that these cells have reduced expression of GzmA/B, Klrg1, Il2ra, TNF
and Il2; all markers of effector/cytotoxicity. How do the authors harmonize these observations?

We thank Referee #2 for asking this important question. As Referee #2 highlighted in the
example of **Figure 1**, we think it is of vital importance to display the observed profile of CD8 T-
cells on a broad scale. We believe that this particular character of T cells – that initially appears
to be exhausted (e.g. TOX expression) is actually hyperactivated with a particular pattern of
expression.

Thus, the single-cell technology allows dissecting the expression profile of CD-HFD-fed CD8+
T-cells into a combination of cytotoxicity/exhaustion expression, indicative of a unconventional
activation/effector. To not lose single-cell resolution and how the data translates into proteins,
we have corroborated these data by mass-spectrometry. These data corroborated the scRNA-
data of **Figure 1** with enrichment for effector function (e.g. T-cell activation, T-cell
differentiation, and NK mediated cytotoxicity) in CD-HFD-fed CD8+PD-1+ T-cells (**Extended**
**Data 5 and Rebuttal Figure 24**). Thus, we decided to display a wide variety of markers of
effector function/cytotoxicity allowing the reader a more sophisticated view into the phenotype.
Moreover, we have compared this pattern with human NASH and indeed could find that
patients with NASH do resemble a similar pattern.

To test this unconventional activation/exhaustion phenotype on a functional level, we
performed all the treatments described in **Figures 2-4** in the absence or in the presence of
anti-PD1-related immunotherapy (anti-CD8, anti-CD8/anti-NK1.1, anti-CD8/anti-PD1, anti-
PD1, anti-PDL1, anti-TNF, anti-TNF/anti-PD1, and as control experiment anti-CD4 and anti-
CD4/anti-PD1), as well as the corroboration with the human data.

Research for a Life without Cancer

For example, an increased anti-inflammatory role by IL-10 expressing CD8+ T-cells upon PD1-
targeted immunotherapy could not be corroborated (**Extended Data 19** and **Rebuttal Figure**
**30k**) (Breuer et al., 2020). Of note, in this publication diet-based NAFLD induction was
achieved by feeding either WD or CD-HFD for 8-10 weeks. This is in strong contrast to our
experimental regime of applying diet for 3, 6, or 12 months as we show, that the preclinical
model presents different stages of NASH pathology severity including hepatocarcinogenesis
(**Figure 1** and **Rebuttal Figure 26a-d**).

Furthermore, we would like to draw attention to the improved cross-referencing to the co-
submitted manuscript Dudek et al., which confirmed a CD8 profile of effector
function/exhaustion/cytotoxicity on a functional level (e.g. TNF sensitizing, high Granzyme
expression, TCR-independent mediated cell death). Moreover, we tried to improve the
discussion on recent literature on the role of CD8 T-cells in metabolic diseases.

8. Regarding Figure 1e, the authors state that CD-HFD contain a significantly altered immune
composition that mainly affects the CD8+ T-cell compartment. However, this finding was not
significant ($p=0.09$ for CD8+PD-1+ T-cells and ns for CD8+ T-cells). In this plot, the authors
do show significant differences in frequency of CD4+ T-cells ($p<0.01$), classical monocytes
($p<0.01$) and MDMs Ly6CHigh ($p=0.01$). Why are these cell types not regarded as interesting?
Are these cells responsible for the authors' proposed phenotype? In line 259 the authors state
that there are only minor differences in the CD4 compartment, yet when looking at the data
(Figure S9h and Figure S9f) the difference in the CD4 subset of CD62L-CD44+CD69+ upon
anti-PD-1 blockade is as strong as, if not stronger than, in the same subset of CD8 T-cells,
which the authors do deem interesting.

We thank Referee #2 pointing out these details in our analysis. We agree with Referee #2, that
immunological subsets represented in our data set are well described in the literature (e.g.
reduction of CD4+ T-cells (Ma et al., 2016) and changes in the myeloid compartment, including
classical monocytes and MDMs Ly6CHigh (Malehmir et al., 2019; Nakagawa et al., 2014),
therefore the respective citations are included in our introduction and discussion.

We added new data and have re-analyzed the data displayed in **Figure 1e** according to
Referee`s #4 comments also by highlighting NKT cells. These results, in CD8+PD1+ ($p= 0.03$),
significantly changed. Other changed cellular subsets after 12 months of CD-HFD feeding are
CD4+ T-cells ($p= 0.04$), classical monocytes ($p< 0.01$), KC ($p= 0.01$), MDMs ($p=0.02$), MDMs
Ly6C+ ($p< 0.01$). We agree with Referee #2, that CD4 T-cells and their expression of PD-1
might play a crucial role in shaping the liver micro-environment and in the observed phenotype
and thus included analysis of CD4 T-cells to the majority of our experiments (e.g. **Extended**
**Data 3** and **Rebuttal Figure 8c-h**).

Research for a Life without Cancer

However, the magnitude of effects observed in CD4+ T-cells is minor when compared to CD8+
T-cells (e.g. **Extended Data 11** mean (CD8+CD62L-CD44+CD69+) ~12% (%of CD45+) vs
mean (CD4+CD62L-CD44+CD69+) ~4% (%of CD45+) upon PD-1 targeted immunotherapy).
Data obtained from CD4 depletion with/without PD1-targeted immunotherapy indicate, that the
increased hepatocarcinogenesis in the context of immunotherapy is independent of hepatic
abundance of CD4+ T-cells in the preclinical NASH model (**Figure 4, Extended Data 22 and**
**23 and Rebuttal Figure 16n, 20 and 21**).

However, CD4+ T-cells might have a diverse set of effector functions (e.g. interpreting tumor
incidence in anti-CD8/anti-PD1 treated animals: although CD4 cells show trends for
decreasing, CD4 are relatively increased in the absence of CD8+ T-cells but immunotherapy,
thus CD4+ T-cells might be responsible for baseline tumor incidence in the context of
immunotherapy (**Extended Data 22 and 23 and Rebuttal Figure 20 and 21**); or CD4 might
have a tumor controlling role, as there are the trends of increased tumor incidence upon anti-
CD4/anti-PD1 co-treatment (tumor incidence (anti-PD-1 mono-treatment)= 75% vs tumor
incidence (anti-CD4/anti-PD1 co-treatment)= 88%) (**Figure 4 and Rebuttal Figure 16n**)).

Of note, CD4+ T-cells might also significantly changed in the human situation, and have also
analyzed human CD4+ cells a by scRNA-Seq (**Extended Data 25c and Rebuttal Figure 10c**).
In addition, we have performed RNA velocity analyses of the scRNA Seq data of mouse and
human CD4 T cells. In mouse, no significant velocity flow was detected in 12 months CD-HFD-
fed mice, indicating, that CD4 cells are not transcriptionally activated and driven by NASH-
conditions or PD-1-targeted immunotherapy in NASH. However, we want to point out, that in
the mouse NASH model CD8 T-cells increase statistically significant, and thus CD4 are
relatively fewer cells compared to CD8. Therefore, the velocity analysis of mouse CD4 T-cells
need to be taken with caution, because we included 300-500 cells only per described subset.
As a consequence, we included the negative CD4 T-cell data not in the manuscript but in the
Rebuttal letter as **Rebuttal Figure 31**. Velocity analyses on human CD4 lead to comparable
problems like seen in mouse. As a consequence, we included the negative CD4 T-cell data
not in the manuscript but in the Rebuttal letter as **Rebuttal Figure 31**.

Like previously mentioned in point 3 raised by Referee #2 concerning the myeloid cells, our
presented data argue, that myeloid cells are recruited to the liver, extend and fine-tune liver
inflammation. While we see MDMs Ly6C+ cells increased comparing 12 months ND vs CD-
HFD-fed mice, our functional treatments (anti-PD-1, anti-CD8/anti-PD-1, anti-TNF, anti-
TNF/anti-PD-1, anti-CD4 and anti-CD4/anti-PD-1) did not result in significant changes in
CD11b+Ly6C+ cells, indicating a rather minor role in comparison to the changes we observed
in the CD8 compartment (**Extended Data 4, 21, 23 and 24 and Rebuttal Figure 6, 7c-e, 19**
**and 21**).

Research for a Life without Cancer

Furthermore, we discuss the myeloid changes and potential role of CD4+ T-cells in greater
detail in the main text.

Finally, we performed an anti-CD4 antibody treatment with or without the combination of anti-
PD1-related immunotherapy. Anti-CD4 antibody treatment successfully depleted or strongly
reduced intrahepatic CD4+ T cells in NASH. However, depletion of CD T cells did not reduce
liver cancer incidence – which is in contrast to CD8+ T cell depletion. Rather, in contrast, CD4
T cell depletion showed a trend in increase of tumor incidence – in line with published data by
(Ma et al., 2016).

9. Along these lines, in line 387 the authors state that consistent with previous results, effects on the
CD4+PD-1+ T-cell compartment remained minor, yet the differences observed for matching analyses
(i.e. S17a vs S17g, S17b vs S17f, S17i vs S17j) of CD4 and CD8 populations show similar, if not
stronger, effects for the CD4 T-cell population. Why are these differences disregarded by the authors?

We believe that the comment of Referee #2 is important and we are in line that the context of
highlighting potential CD4-mediated effects in the context of PD-1-targeted therapy had to be
investigated in detail (e.g. in **Extended data 5, 18** and **Rebuttal Figure 15** and **24**). In line with
the comment of Referee#2, we set out to investigate the character and function of CD4+ T-
cells by scRNA-seq analyses in human and mouse NASH livers, but like raised in point 8 of
Referee #2 strongly suggest to take the velocity analysis of mouse CD4 T-cells with caution,
because we included 300-500 cells only per described subset. Thus, we included these
analyses in only in the **Rebuttal Figure 31**. Moreover, our experiments using an anti-CD4
depleting antibody alone or in the context of anti-PD1-related immunotherapy indicate a minor
role of the CD4 compartment in our model as well (**Extended Data 22, 23** and **Rebuttal Figure**
**20** and **21**).

As mentioned in point 8 raised by Referee #2, we agree with Referee #2, that similar
phenotypes can be observed when comparing effects in CD4+ and CD8+ T-cell subsets upon
PD-1 targeting immunotherapy. We do not disregard the changes in the CD4 compartment but
would like to draw attention to the magnitude of changes in the setting of chronic hepatic
inflammation – and the functional experiments with anti-CD8, anti-CD8/anti-PD-1, anti-CD4,
and anti-CD4/anti-PD1 antibodies.

We have also discussed the relevant literature as well as our data on CD4+ T cells in the
discussion in detail. We, in addition, believe that the CD4+ T-cell depletion experiments
with/without PD-1 targeted immunotherapy in mice have enabled us to strengthen our
hypothesis on a more functional level: CD4 depletion alone or in the context of anti-PD1-related
immunotherapy in NASH-induced HCC failed to revert/prevent liver cancer formation. In
contrast, anti-CD8 depleting antibody treatment alone reverted/prevented liver cancer
formation. The role of CD4+ T-cells in the context of immunotherapy remains to be defined in
more detail, as CD4-depletion did not lead to a reversal of the pro-tumorigenic effects of anti-

Research for a Life without Cancer

PD1 therapy in the context of NASH induced HCC. However, CD4+ T-cells might exert a
protective/controlling role in the context of PD1-targeted immunotherapy and presence of
CD8+ T-cells, as combinatorial treatment of anti-CD4 depletion and PD1-targeted
immunotherapy led to an increase of tumor incidence compared to anti-PD1 treatment alone
**(Figure 4, Extended Data 22 and 23 and Rebuttal Figure 16n, 20 and 21)**.

10. Similarly, in Figure 5a, the authors claim that a CD8+PD-1+ T-cell population arises upon NASH.
However, there is a, perhaps even stronger, depletion of an Eomes+ gamma-delta T-cell subset.
Additionally, a very strong induction of a CD4+CD27+ population is observed in NASH samples. Why
are these not discussed? Can these populations also be identified in the authors' murine models? Do
these contribute to the authors' described phenotype? The authors should deplete CD4 T-cells and
gamma-delta T-cells in their murine models, as these cell types may, at the very least, contribute to
what occurs in patients.

We thank Referee #2 for raising this important concern. Indeed, we have so far not discussed
the loss of gamma-delta T-cell subsets or a potential increase of CD4+ T-cells and included
this now thoroughly in the revised version of the manuscript **(Extended Data 3, 21, 23, 25 and**
**26 and Rebuttal Figure 8, 19, 21 and 10, 11)**. In line with the comments of Referee#2, we
have now described and discussed these populations in detail, by scRNA-seq and multicolor
flow cytometry in mouse and three distinct human cohorts recruited from 3 different centers
across Europe.

As mentioned in points 8 and 9 raised by Referee #2, we have depleted CD4 T-cells
with/without PD-1 targeted immunotherapy. Of note, CD27 could not be detected in our
scRNA-seq data set obtained from the preclinical mouse model as significantly changed. In
human bulk RNA-seq CD27 expression increased, but CD4 expression decreases with the
severity of pathology. CD27+CD4+ T cells did not reach statistical significance in our cohorts
by flow cytometry **(Extended Data 25 and Rebuttal Figure 10)**. Of note, in our second cohort,
CD4+ T-cells are significantly enriched in NAFLD/NASH patients by flow cytometry, however
as this cohort was analyzed retrospectively, we could not analyze CD27 expression **(Extended**
**Data 25)**. Furthermore, the abundance of CD4+CD27+ cells was not increased in our human
scRNA cohorts **(Extended Data 27 and Rebuttal Figure 12)**.

As mentioned in point 8 we have performed a velocity analyses of the scRNA Seq data of
mouse CD4 T cells (see Rebuttal letter below). In mouse, no significant velocity flow was
detected in 12 months CD-HFD-fed mice, indicating, that CD4 cells are not transcriptionally
activated and driven by NASH-conditions or PD-1-targeted immunotherapy in NASH.
However, we again want to point out, that the velocity analysis of mouse CD4 T-cells need to
be taken with caution because we included 300-500 cells only per described subset. As a
consequence, we included the negative CD4 T-cell data not in the manuscript but in the
Rebuttal letter. Velocity analyses on human CD4 lead to comparable problems as seen in

Research for a Life without Cancer

mouse. As a consequence, we included the negative CD4 T-cell data not in the manuscript but
in the Rebuttal letter as **Rebuttal Figure 31**.

We agree that $\gamma\delta$ T-cells might be involved in underlying processes of NASH or NASH to HCC
transition – also in the context of PD1-related immunotherapy. In humans, our data is not
conclusive in all experiments, e.g. our data indicate for $\gamma\delta$ T-cells, if we compare: bulk RNA-
seq indicates a reduced expression in severe NASH pathology of EOMES, TRDC, and TRGC1
(**Extended Data 28** and **Rebuttal Figure 32**), however, both flow cytometry cohorts and the
scRNA-seq cohort indicate no change of either $\gamma\delta^+$ T-cells or $\gamma\delta^+$ Eomes⁺ T-cells comparing
control vs NAFLD/NASH patients (**Extended Data 25, 27** and **Rebuttal Figure 10** and **12**).

Corroborating the human flow cytometry data in our mouse model upon NASH establishment,
we detected no difference in hepatic abundance of $\gamma\delta$ -T-cells between chow- or CD-HFD-fed
control mice. Furthermore, data presented in **Figures 1 and 4** and **Extended Data 3** argues
against the major contribution of gamma delta T-cells in the mouse model of NASH. Here, we
did not observe significant differences in the “other leukocytes” subset. In the revised
manuscript, we analyzed $\gamma\delta$ -T-cells separately to strengthen the point, that these cells are not
significantly changed upon diet feeding (included in **Extended Data 3, 20-23** and **Rebuttal**
**Figure 8j, 18-21**).

11. The patient data is not convincing, but also does not match their murine models. In Figure 5a, the
authors show that CD8+GzmB+ cells are specifically lost in NASH samples which seems to counteract
the claim made by the authors that inflammatory CD8 T-cells cause liver inflammation and associated
carcinogenesis. The authors similarly show in S19a that IFN γ , Ccl3 and PD-L1 are in fact reduced in
advanced NASH samples; does the loss of these inflammatory genes not counteract the claims made
in Figure 3g, S4d, S10, S11 and S13a?

We thank Referee #2 for raising this important point and agree, that GzmB+CD8+ population
is decreased as well as GzmB expression in bulk RAN-seq (**Extended Data 28** and **Rebuttal**
**Figure 32a**), other populations, on the other hand, are increased. GzmB is a strong indication
for inflammatory CD8+ T-cells. We would like to draw attention to the improved cross-
referencing to the co-submitted manuscript Dudek et al., in which Gzmb along with other
cytotoxic effector molecules (e.g. TNF) are key mediators of a hepatic inflammatory
environment, but not the executing molecules driving hepatocarcinogenesis. However, we
agree with Referee #2, that the data presented in **Figure 5** has limitations due to the small
sample size, although we could reproduce the cellular abundance between healthy vs
NAFLD/NASH patients in a second cohort from a second center (**Figure 5** and **Extended Data**
**25** and **Rebuttal Figure 9** and **10**).

We agree with Referee #2, that certain inflammatory genes (e.g. Ifny, Ccl3, Cd274) show
decreased expression along with NASH progression, however, how this translates into local
hepatic proteins-expression remains elusive (e.g. for human gene expression vs

Research for a Life without Cancer

immunohistochemical staining of Pdccl1 in NASH F1-3 (**Figure 6** and **Rebuttal Figure 1a,b**);
or F0-F4 for CD4, or CD274 (**Extended Data 28** and **Rebuttal Figure 32**). As an example,
human PD-L1 increases with NASH severity on IHC, which is corroborated by the preclinical
model (**Extended Data 3, 20, 22** and **Rebuttal Figure 8k,l, 18** and **20**).

To shed more light on the phenomena, we focused on our human scRNA-seq on the analyses
of CD8+ T-cells (**Figure 5, Extended Data 27** and **Rebuttal Figure 9** and **12**) and correlated
these cells to the CD8+ T-cells analyzed from our preclinical model (**Figure 5** and **Rebuttal**
**Figure 9f,j**). These data match each other very well, strengthening in our opinion hypotheses
and conclusions drawn from the preclinical NASH-model. Therefore, we do not think the results
of the bulk RNA-seq counteracts the claims of previous figures from the mouse model but
allows an in-depth understanding of underlying inflammation in different NASH stages (e.g.
Referee #1: decrease activity of NASH with disease progression to HCC).

12. Lastly, the majority of patient data are not significant and show weak effect sizes; is it fair to draw
strong conclusions on the basis of these data as the authors do?

We agree with Referee #2 and thus recruited additional patients to increase the number of
patients in our initial clinical cohort from 65 to 130 HCC patients under anti-PD(L)1-targeted
immunotherapy and validated our results in a second cohort of 118 HCC-patients under PD-
1-targeted immunotherapy (**Figure 6** and **Rebuttal Figure 1f,g**).

We agree with Referee #2, that the presented retrospective PD(L)1 targeted immunotherapy
treated NAFLD/NASH-associated HCC cohort - although unique for Europe and treatment not
officially licensed and thus reimbursement - is still small, although we would like to point out,
that prominent trends or effects can be seen in small retrospective cohorts as well. Thus, our
analyses of BCLC-C NAFLD/NASH-HCC vs. other-etiology-HCC patients indicated, that
NAFLD/NASH-HCC have significantly reduced overall survival compared to other-etiology-
HCC in this small retrospective cohort. Multivariate analyses identified NAFLD/NASH as an
independent factor for treatment response and thus identifying NAFLD/NASH as a negative
predictor for HCC immunotherapy (**Supplementary Table 8**).

We corroborated our hypothesis of non-viral (NASH-related) HCC being less responsive to
immunotherapy by a meta-analysis including 1656 patients of the three most important clinical
trials, identifying immunotherapy vs control for viral HCC as favorable treatment ($HR(viral)=$
0.64), in contrast, non-viral-HCC showed less benefit ($HR(non-viral)= 0.92$) for immunotherapy
(**Figure 6, Extended Data 30-32, Supplementary Table 9** and **Rebuttal Figure 1-4**).

Based on these data we want to point out that it is - as indicated by Referee#2 - of the highest
importance to us to specifically define/tone down appropriately the message of our manuscript:
Our manuscript did not intend to indicate that immunotherapy is not beneficial for HCC patients.
It rather demonstrates that HCC patients with viral etiologies do respond well and achieve

Research for a Life without Cancer

survival benefits - however, that patients with non-viral etiologies (e.g. NASH) do not achieve
a significant outcome benefit.

We thus propose to stratify HCC patients who are very likely to profit from immunotherapy and
strengthen the argumentation to use immunotherapy in specific cohorts of HCC patients. We
agree with Referee#1 that this information needs to be articulated in the paper appropriately
not to deliver wrong messages but to be very specific. We truly believe that these are important
clinical data, also providing the basis to test our hypotheses in prospective studies on non-
significantly beneficial effects in terms of OS for immunotherapy in HCC patients with non-viral
and NAFLD/NASH etiology, in particular. Moreover, we toned down the conclusions of our
retrospective cohort in the manuscript and would like to point out, that bigger cohorts and
prospective clinical trials are of utmost importance for the scientific community.

Minor points:

- Figure 1j lacks a color scale bar and proper description. How does one interpret the difference between
ND and CD-HFD in this plot?

We thank Referee #2 for highlighting the lack of a color bar in this panel, we have added a
color scale bar with a proper description. Figure 1j displays the median expression of selected
genes in the different T-cell populations observed in our scRNA-seq data set (**Figure 1,**
**Extended Data 5** and **Rebuttal Figure 24** and **26**) and serves as a supplement to the 2-
dimensional tSNE plot. In this panel, we do not compare ND to CD-HFD rather simply allow
the readers to view the gene signatures characterizing the different populations. A comparison
of ND and CD-HFD is visualized using volcano plots in Figure 1. As this heatmap is rather a
technical information, but does not condense scientific explanation in great detail, we decided
to move this heatmap to **Extended Data 5**.

- Where is the ND + PD-1^{-/-} in Figure 3b? Do these mice also get accelerated carcinogenesis?

We thank Referee #2 for highlighting this inconsistency. In line with the point raised by
Referee#2 we have improved this in a revised manuscript including PD-1^{-/-} mice on ND.
Literature does not report accelerated hepatocarcinogenesis
(<http://www.informatics.jax.org/allele/allgenoviews/MGI:4397682>) and we did not observe any
hepatocarcinogenesis in PD1^{-/-} under ND.

- There is no color scale bar in Figure 3e.

We thank Referee #2 for highlighting this inconsistency and improved our manuscript by
adding a scale bar.

- In Figure 5k, shouldn't progression-free survival and time to progression plots yield the exact same
data, but inversed? Why don't these curves match?

Research for a Life without Cancer

We thank Referee #2 for this question. TTP and PFS are different endpoints. TTP is defined
as the time from the date of treatment initiation until the date of first radiological tumor
progression. PFS is a composite endpoint. It is defined as the time from the date of treatment
initiation until radiological progression OR death, whatever comes first (Llovet et al., 2008). We
decided to leave out the non-significant data of TTP and PFS in our manuscript. Moreover,
upon recruiting the validation cohort of 118 HCC-patients under immunotherapy we decided
to not show TTP and PFS, but instead the multivariate analysis (**Supplemental Table 9**).

- In Figure S1i, what is the parent population?

We thank Referee #2 for highlighting this inconsistency and improved our manuscript by
adding the description of the parent population. In the case of **Extended Data 1** the parental
populations are CD8+ (left) and respective CD4 or CD8 (right) T-cells.

- In Figure S4a, how does one distinguish ND from CD-HFD mice? The y-axis lacks a label.

We thank Referee #2 for highlighting this inconsistency and improved our manuscript by
adding the description of the y-axis.

- Figure 5c is plotted in a confusing manner (as the z-score scale is red independent of whether it goes
up or down), but it seems that the TNF signaling gene sets are actually decreasing in expression.

We thank Referee #2 for highlighting this inconsistency. We decided after integration of the
new data, to leave that graph out as it communicates similar information already included in
**Extended Data 28**. Of note, if we change the labeling of z-score, it clarifies, that TNF is indeed
an increased pathway (similar to **Extended Data 28**).

- Why do the PD-1^{-/-} mice still express PD-1 (Fig. S12e)?

We thank Referee #2 for highlighting this inconsistency and improved our manuscript by re-
analyzing our flow cytometry data set (as gates have been set too loose – leading to a subset
of around 1% PD1 expressing CD4+ and CD8+ T cells). Analyses revealed that PD1^{-/-} ND-fed
mice have no intrinsic higher immune cell abundance, or activation and hepatocarcinogenesis
compared to ND-fed wt control mice at 6 months under diet (**Figure 3** and **Extended Data 14**
and **Rebuttal Figure 22, 23**). Moreover, as indicated no PD1-expression can be observed.

- In Figure S13k, the authors should present cleaved Caspase 3 and cleaved Caspase 8 if they want to
conclude something about T-cell death, as total, uncleaved levels of these proteins do not indicate cell
death.

We thank Referee #2 for highlighting this point. We have removed these plots and demonstrate
cleaved caspase 3 by immunohistochemistry, which has the advantage that we not only see
the Cleaved Caspase 3 directly but also which cells are undergoing apoptosis. These data are
now included in **Extended Data 16** and **Rebuttal Figure 33**.

Research for a Life without Cancer

- In Figure S16f, the FACS plot does not match the quantification on the left.

We thank Referee #2 for bringing this up and apologize for this inconsistency. We would like
to draw the attention, that in the flow cytometry plot the data is displayed as “%of CD8”, in
contrast in the box plot the data is displayed as “%of CD45” to give the reader a more
quantitative analysis.

- Regarding Figure S17b, the authors claim an increase in calcium levels in line 383 of their manuscript,
but this difference is not significant.

We agree with Referee #2. Thus, we have performed additional experiments – supporting our
initial finding that upon PD1-targeted immunotherapy calcium levels were increased on CD8+
but not CD4+ T-cells. This inconsistency was improved our manuscript accordingly.

- In Figure S18b, how does one interpret the difference between healthy, borderline NASH or NASH
patients? There is no explanation of the color scale bar. Also, what are “randomly chosen CD45+ cells”
as mentioned in the corresponding Figure Legend?

We thank Referee #2 for highlighting this inconsistency and improved our manuscript
accordingly by describing differences between patients and highlighting our analysis pipeline
for flow cytometric data according to (Brummelman et al., 2019). Moreover, we have added 2
more cohorts in the main Figure (**Figure 5**) and Extended Data and pooled borderline NASH
and NASH patient into one group of NAFLD/NASH patients after consultation with our
pathologists, who indicated that the difference between borderline NASH and NASH can be
regional – and thus is always is regarded as NASH (**Extended Figure 25** and **Rebuttal Figure**
**10**).

- Figure S19b is not legible.

We thank Referee #2 for this comment. In line, we have now changed the graph size and font
size.

- In lines 237-246 the authors describe that NK1.1-based depletion of immune populations did not result
in changed liver pathology, body weight, fibrosis ALT, hepatic cytokines and hepatic chemokines.
However, the animals who underwent this depletion also completely lacked liver cancer development.
How does this happen if the authors did not detect any changes? The authors should perform NK1.1
depletion by itself to see if NK1.1+ cells, potentially depending on CD8 cells, are in fact responsible for
the authors' phenotype.

We thank Referee #2 for highlighting this unprecise description of our data. We improved our
manuscript by highlighting differences between CD8 depletion and CD8/NK1.1 co-depletion.
We included additional GSEA analysis of RNA-seq data, which display changes in CD8/NK1.1
co-depleted in comparison to CD8 single depleted animals (CD8-single depleted animals
showed enrichment for “cholesterol homeostasis” (**Extended Data 9** and **Rebuttal Figure 17**)).

Research for a Life without Cancer

Furthermore, we would like to draw attention to a previous study (Wolf et al., 2014), in which
NKT-cells were responsible for metabolic changes and CD8 T-cells driving hepatic damage.
We think, that the lack of liver cancer incidence is a result of CD8 depletion and a reduction of
a pro-tumorigenic environment - e.g. including pro-tumorigenic TNF signaling, which is
similarly enriched (TNF signaling via NFKB) in CD-HFD-fed control animals (NES(CD8
depletion vs control)= -1.6718) and NES(CD8/NK1.1 co-depletion vs control)= -1.6538)
(**Extended Data 8 and 9** and **Rebuttal Figure 15 and 17**). These data were also corroborated
by the analyses of the ICF signature which is strongly abrogated upon CD8 T cells depletion.
Thus, we dissected the role of NK1.1 cells in greater detail by including the GSEA analysis of
RNA-seq data comparing CD8-depleted and CD8/NK1.1 co-depleted animals. Furthermore,
we improved cross-referencing to the co-submitted study Dudek et al. to highlight, that CD8 T-
cells are driving hepatocarcinogenesis. In line, together with Dudek et al. we generated new
data using mouse strains with impaired NKT cells - namely $J\alpha 18^{-/-}$ and $CD1d^{-/-}$ - under NASH-
inducing diet. Both genetic knockout mouse models develop NASH (including systemic
obesity, fibrosis, ALT) and NASH-induced hepatocarcinogenesis similar to WT control animals
at 12-months diet-feeding. These data argue against an essential role of NKT-cells to drive
hepatocarcinogenesis at this time-point.

- Sentence 289-292 is unclear.

We thank Referee #2 for highlighting the imprecise description and have now improved this in
the main text of the revised manuscript. The sentence now reads as follows: "Next, we
investigated the mechanisms underlying the increased occurrence of liver cancer
incidence/liver tumor formation associated with anti-PD-1 treatment in the context of NASH."

- When discussing GSEA, the authors frequently use the wording 'reduced enrichment (e.g. line 241)
when talking about enrichment in the opposite phenotype. This is incorrect, as the absolute amount of
enrichment is often similar just, as mentioned, in the opposite direction.

We thank Referee #2 for highlighting this imprecise description. We altered this in the revised
manuscript. The changes read now as follows e.g.: "Gene set enrichment analysis (GSEA) of
RNA sequencing data from whole liver tissue of CD8⁺ depleted mice revealed enrichment for
DNA repair, oxidative phosphorylation, complement, and TNF signaling compared to CD-HFD-
fed control)".

Research for a Life without Cancer

Referee #3 (Remarks to the Author):

This full article manuscript is novel, and the experimentation to support the conclusions is exhaustive
and solid for the most part. In essence, the findings indicate that, in NASH livers, there is an
accumulation/expansion of a pathogenic CD8 T-cell population that expresses PD-1 and exacerbates
NASH pathology and fosters hepatocellular carcinogenesis and progression. The inflammatory and
tissue-damaging functions of this pathogenic CD8 T-cells are repressed by PD-1 blockade that is
common clinical practice for second-line treatment of advanced HCC and is under clinical trials for earlier
stages of the disease. In fact, PD-L1 blockade plus anti-VEGF will soon become the standard of
treatment for advanced HCC in first line. According to the findings in this paper upon PD-1 blockade,
authors document an exacerbation of carcinogenesis and liver damage that questions the indication of
PD-1 blockade in NASH-associated liver cancer. A balanced presentation of preclinical and supportive
clinical results in patient specimens very much enhances the significance of this study.

We thank Referee #3 for the positive feedback and the statement that our study is “novel, and
the experimentation to support the conclusions is exhaustive and solid for the most part”. We
would like to address his/her concerns in the following section point-by-point by presenting
new experimental data sets experiments, rephrasing, and re-analysis of the underlying data-
sets.

Questions and comments:

1. TNF seems to be an actionable therapeutic target for the observed harmful effects of this CD8 T-cell
population. It would be interesting to know if TNF could be blocked preserving anti-cancer immunity
(especially under checkpoint inhibition therapy) but preventing tissue damage and carcinogenesis
promotion.

We thank Referee #3 for raising this important concern and thus have performed anti-TNF
with/without anti-PD-1-related immunotherapy in the context of NASH/HCC. Anti-TNF
treatment alone - without PD1-targeted immunotherapy - leads to liver cancer formation
comparable to control-treated CD-HFD-fed mice. However, anti-TNF treatment in the context
of PD1-targeted immunotherapy leads to a significant reduction of tumor incidence (tumor
incidence(anti-PD-1)= 75% vs tumor incidence(anti-TNF/anti-PD-1)= 25%, p= 0.0024), liver
damage (ALT(anti-PD-1)= 381.6 U/L vs ALT(anti-TNF/anti-PD-1)= 250 U/L, p= 0.0072) and
NAFLD-activity score (NAS(anti-PD-1)= 5.875 vs NAS (anti-TNF/anti-PD-1)= 3.1, p= <0.0001),
when compared to anti-PD1 treated CD-HFD-fed mice alone. This indicates that TNF exerts
key functions of the observed adverse effects of PD1-targeted immunotherapy, namely
contributing to increased hepatocarcinogenesis (**Figure 4, Extended Data 20 and 21 and**
Rebuttal **Figure 16, 18 and 19**).

Moreover, the combination of anti-PD1 therapy with CD8-T cell depleting antibodies fully
eliminated the adverse, NAS increasing and pro-carcinogenic effects of CD8+ T-cells. These

Research for a Life without Cancer

data emphasize that CD8+ T-cells are a major cell population mediating increased
hepatocarcinogenesis through a TNF-dependent mechanism upon PD1-targeted
immunotherapy (**Figure 4, Extended Data 20 and 21 and Rebuttal Figure 16, 18 and 19**).
On one hand, the mechanisms could be executed by CD8 T-cell derived TNF itself or by
mechanisms that depend on TNF-signaling on other cells (e.g. myeloid cells). For example,
we see a drastic reduction of myeloid attracting chemokines but also cytokines of liver
inflammation (e.g. IL-17A, IL-10), all cytokines/molecules which might fuel liver inflammation
and thus hepatocarcinogenesis in PD-1-targeted immunotherapy in NASH mice.

Importantly, comparing mouse-human of CD8+ T-cells isolated from liver tissue of NASH mice
or patients through classical flow cytometry, CYTOF, and on scRNA-seq level we identified
similar populations and transcriptional activation of CD8+ PD1+ in a total of three independent
center patient cohorts (**Figure 5, Extended Data 25-27 and Rebuttal Figure 9-13**). These
data indicate that results obtained and hypotheses built from the preclinical NASH model are
relevant for human disease and are in line with published results, where TNF blockade
uncouples mediated toxicity in dual CTLA-4 and PD-1 immunotherapy (Perez-Ruiz et al.,
2019).

2. Would PD-L1 blockade enhance liver cancer and tissue damage as well? Which cells are expressing
PD-L1 in the system? This becomes important given the recent approval of atezolizumab +
bevacizumab.

We agree with Referee #3 for raising the point that dissection of anti-PD-L1-targeted
immunotherapy is of major concern, especially in the light of the recent results of the
IMBrave150 study. Data we have received from RNA in situ hybridization and
immunohistochemistry indicate that PD-L1 is expressed with increased level over time – with
progression of NASH disease (in mice and men). In summary, PDL1 staining in the preclinical
model is mainly associated with inflammatory cells, positive cells can be observed in the
sinusoidal space as well (**Extended Data 3, 20, 22 and Rebuttal Figure 8, 18 and 20**). In
humans, PDL1 positivity was observed in aggregates of inflammatory cells in the parenchyma
and the portal tract area. Focally, positivity was also seen in sinusoidal lining cells (**Extended
Data 28 and Rebuttal Figure 32**).

The cells expressing PD-L1 in NASH-affected mice are mainly lymphocytes but also some
parenchymal cells (**Extended Data 3+7, 20+22 and Rebuttal Figure 8, 13, 18 and 20**).

In line with the comment of Referee #3, we have also performed anti-PD-L1 targeted
immunotherapy in mice with and without established liver cancer (**Extended Data 7 and
Rebuttal Figure 13**). Results from these experiments indicate that similar to anti-PD1 - anti-
PDL1-treatment does not induce an anti-cancer effect for NASH-induced HCC but induces -
similar to anti-PD1 treatment - a pro-inflammatory and pro-carcinogenic effect (e.g. increased
NAS, strong trend in increased hepatic CD8 abundance by IHC ($p=0.0546$), cytokines like IL-

Research for a Life without Cancer

21 and CCL3) (**Extended Data 7+13** and **Rebuttal Figure 13** and **28**). These data indicate,
that in the preclinical NASH model both PD1 or PDL1-targeted immunotherapy induces
adverse effects. This is corroborated by our increased, retrospective cohort HCC-patients of
different etiologies under PD(L)1-targeted immunotherapy, in which multivariate analysis
results in NAFLD/NASH being an independent negative factor for overall survival and validated
these results in a second cohort of 118 HCC-patients (**Figure 6** and **Rebuttal Figure 1g,f**).
Furthermore, we corroborated our hypothesis of non-viral (NASH-related) HCC being less
responsive to immunotherapy by a meta-analysis including 1656 patients of the three most
important clinical trials, identifying immunotherapy vs control for viral HCC as favorable
treatment (HR(viral)= 0.64), in contrast, non-viral-HCC showed less benefit (HR(non-viral)=
0.92) for immunotherapy (**Figure 6, Extended Data 30-32, Supplementary Table 9** and
**Rebuttal Figure 1-4**)).

3. Results on NASH in human samples are compelling and supportive of the relevance of the findings.
It would be interesting to know in such livers which cells express PD-L1.

We thank Referee #3 for highlighting this important aspect of our data – and have consequently
performed PD-L1 expression analyses by immunohistochemistry in human specimens
described in the previous point raised by Referee #3. Although analysis by bulk RNA-seq of
liver tissue indicates a decrease of PDL1/CD274 expression with the severity of NASH
pathology, immunohistochemistry indicates an increase of PDL1 positivity with the severity of
NASH pathology. PDL1 positivity was observed in aggregates of inflammatory cells in the
parenchyma and the portal tract area. Focally, positivity was also seen in sinusoidal lining cells
(**Extended Data 28** and **Rebuttal Figure 32**).

4. What do you think is the fibrogenic factor/s promoted by pathogenic CD8 cells? Any candidates from
the extensive transcriptomic analyses?

We thank Referee #3 for pointing out, that the fibrogenic factor is of major concern to prevent
HCC in subgroups of NASH patients. Our transcriptomic data-set has so far not pointed
towards specific fibrogenic factors, indicating that the chronic inflammatory environment
correlating with pathogenic CD8 cells drives fibrosis in our mice. To strengthen this hypothesis
AI-based analyses of a broad range of parameters of our 12 months CDHFD-fed mice
revealed, that Sirius red staining correlates negatively within CD8 depleted animals, indicating
that CD8-associated inflammation or CD8-dependent mechanisms might be functionally linked
with fibrosis (included in **Figure 1, Extended Data 4 and 24** and **Rebuttal Figure 6, 7** and
**26**). Moreover, in 12 months CDHFD-fed mice fibrosis correlated positively with CD8 T-cells
abundance, CD8+PD-1+ (%CD8), pDC+MHCII+ polarization, and hepatic TNF concentration.
Therefore, we cannot point out one specific factor driving fibrosis on pathogenic CD8 cells.

Research for a Life without Cancer

5. Are Kupffer cells involved in the CD8-dependent pathogenesis mechanisms?

We thank Referee #3 for asking the important question about Kupffer cells (KC). A study
(Malehmir et al., 2019) reports, that KCs have a crucial role in the pathogenesis of NASH, but
activation of monocytes and myeloid-derived macrophages correlates with disease
progression. Data presented in **Extended Data 8 and 11** cannot exclude KC-dependent
mechanisms, however, they seem to have a minor role, especially concerning the co-submitted
manuscript Dudek et al. in which CD8+ cells drive pathogenesis in KC-independent ways.

We have further performed analyses on how KC correlate with varying degrees of inflammation
induced by our antibody treatments (anti-CD8, anti-CD8/anti-NK1.1, anti-CD8/anti-PD1, anti-
PD1, anti-PDL1, anti-TNF, anti-TNF/anti-PD1, and as control experiment anti-CD4 and anti-
CD4/anti-PD1) by our AI-based analysis approach (**Figure 1, Extended Data 4, 20-24** and
**Rebuttal Figure 6, 18-21** and **26**). Under baseline conditions (12 months CD-HFD-fed animals
receiving no treatments) KC abundance does not correlate with any serological or histological
marker, but KC activation (measured by MHCII+ polarization) correlates strongly with tumor
size and IL-21 (**Extended Data 4** and **Rebuttal Figure 6**). However, when applying treatments
(e.g. PD-1-targeted immunotherapy) KC correlates with treatments as well as activation of
hepatic KC (measured by MHCII+) correlate positively with CD8+PD-1+ (%CD8), Sirius Red
staining, tumor incidence, tumor number, tumor size, and IL-21 (**Extended Figure 24** and
**Rebuttal Figure 7**).

In summary, we believe in line with our own study (Malehmir et al., 2019) and recent literature
(Remmerie et al., 2020) that Kupffer cells are an important cell type on whose basis not
inflammatory pathologies are initiated and maintained, but also in end-stage disease fresh
KC/KC-like cells (attracted by cytokines e.g. MCP-1, CCL3, MIP-2 (**Extended 2, 13, 21** and
**23** and **Rebuttal Figure 19, 21, 28** and **34**) activation might be detrimental as indicated by our
correlation analysis. – laying the ground for adaptive immune cell reactions.

6. Obesity and response to PD-1 associations have been reported (PMID: 30420753 and PMID:
30813970). According to these studies, obesity relates to T-cell dysfunction that PD-1 blockade
derepresses and results in better responsiveness. The models of NASH should suffer overweight as
well as perhaps the patients in the reported series. This point should be addressed if possible and at
least discussed. Authors may gain insight with their comparisons of the models with and without choline
in the diet. As a potential consequence, would it be the case that in HCC patients, obese patients
respond worse to treatment contrary to other indications? Of clinical note, advanced HCC patients
frequently experience cachexia but perhaps less frequently so those with presumed or documented
NASH etiology.

We thank Referee #3 for highlighting these important studies of checkpoint inhibition in the
frame of obese cancer patients. (Wang et al., 2018) shows - similar to our study - convincingly
that increased PD-1 expression is a hallmark of diet-induced obesity, thus we cite the study in

Research for a Life without Cancer

our introduction and improved cross-referencing in our discussion. Potential differences in the outcome of PD-1-targeted immunotherapy might be a consequence of the use of obesity-but, not NASH-inducing high-fat diet, which we show is crucial to induce hallmarks of NASH by comparing HFD with CD-HFD in **Extended Data 1**. Moreover, we would like to draw attention to the different cancer entities, which potentially affect immunotherapy-responsiveness. Wang et al. use subcutaneous tumor models of lung carcinoma (3LL) and melanoma (B16-F0), but not spontaneous developed liver cancer in a chronic inflammatory metabolically challenged hepatic microenvironment. Notably, obese animals have bigger tumor-volumes and anti-PD-1 reactive animals do not control tumor-volume to a smaller absolute tumor-volume compared to non-obese controls (Figures 2 and 4 in (Wang et al., 2018)).

The second study of (Cortellini et al., 2019) corroborates the preclinical data of (Wang et al., 2018) nicely in lung-, renal-carcinoma, or melanoma patients, but not liver cancer. No grading of obese patients was performed (e.g. we report in Supplementary Table 1: healthy/control liver, NAFLD/NASH), which we show in **Figure 5** is crucial for hepatic CD8 and PD-1 abundance. Supporting our manuscript, (Cortellini et al., 2019) report significantly more likelihood of obese patients experiencing immune-related-Adverse-Effects (irAEs) “compared to non-overweight patients (55.6% vs. 25.2%, $p < 0.0001$)”. Unfortunately, no subgroup analyses about differences of hepatic irAEs between obese/non-obese patients are shown.

We included the study of (Cortellini et al., 2019) in our introduction/ discussion.

Our NAFLD/NASH cohort without immunotherapy treatment indicate a correlation of BMI with CD8+PD-1+ T-cells (**Figure 5** and **Rebuttal Figure 9**). In our conducted meta-analysis, no BMIs were reported, thus statements about treatment response remain hypothetical. Furthermore, our retrospective HCC-patient cohort under PD(L)1 immunotherapy was too small for subgroup analysis, however, there was no significant difference in BMI between NAFLD/NASH-HCC and other etiologies-HCC patients, indicative of obesity (**Supplementary Table 7**).

7. The retrospective series of patients with advanced HCC treated cannot be considered conclusive at this point and only hypothesis-generating. The wording there needs to be carefully down-toned.

We agree with Referee #3, that the presented retrospective PD-(L)1 targeted immunotherapy treated NAFLD/NASH-associated HCC cohort – although unique for Europe and treatment not officially licensed and thus reimbursement - is still small, although we would like to point out, that prominent trends or effects can be seen in small retrospective cohorts as well.

Thus, our analyses of BCLC-C NAFLD/NASH-HCC vs other-etiological-HCC patients indicated, that NAFLD/NASH-HCC has significantly reduced overall survival compared to other-etiological-HCC in this small retrospective cohort, which we validated in a second cohort of 118 HCC patients under immunotherapy (included in **Figure 6** and **Rebuttal Figure 1f,g**). Of note, multivariate analyses identified NAFLD/NASH as an independent factor for treatment

Research for a Life without Cancer

response and thus identifying NAFLD/NASH as a negative predictor for HCC immunotherapy
(included in **Supplementary Table 9**).

We corroborated our hypothesis of non-viral (NASH-related) HCC being less responsive to
immunotherapy by a meta-analysis including 1656 patients of the three most important clinical
trials (IMbrave 150; Checkmate 459; Keynote 240), identifying immunotherapy vs control for
viral HCC as favorable treatment (HR(viral)= 0.64), in contrast, non-viral-HCC showed less
benefit (HR(non-viral)= 0.92) for immunotherapy (included in **Figure 6, Extended Data 30-32,**
**Supplementary Table 7 and Rebuttal Figure 1-4**).

Based on these data we want to point out that it is - as indicated by Referee#3 - of the highest
importance to us to specifically define/tone down appropriately the message of our manuscript:
Our manuscript does not indicate that immunotherapy is not beneficial for HCC patients at all.
Our manuscript rather demonstrates that HCC patients with viral etiologies do respond well
and achieve survival benefits - however, that patients with non-viral etiologies (e.g. NASH) do
not achieve a significant outcome benefit.

We thus propose to stratify HCC patients who are very likely to profit from immunotherapy and
strengthen the argumentation to use immunotherapy in specific cohorts of HCC patients. We
agree with Referee#1 that this information needs to be articulated in the paper appropriately
not to deliver wrong messages but to be very specific. We truly believe that these are important
clinical data, also providing the basis to test our hypotheses in prospective studies on non-
significantly beneficial effects in terms of OS for immunotherapy in HCC patients with non-viral
and NAFLD/NASH etiology, in particular. Moreover, we toned down the conclusions of our
retrospective cohort in the manuscript and would like to point out, that bigger cohorts and
prospective clinical trials are of utmost importance for the scientific community.

8. An important message of this paper is that progression following PD-(L)1 treatment in NASH patients
could be the development of a second primary malignancy rather than from the same one. Can this
point be addressed in the models? Is multifocal cancer more common in those cases? The more CD8
pathogenic T-cells in the infiltrate, the more multifocal the tumors?

We thank Referee #3 for asking this important question. In our opinion dissection of
primary/second primary malignancy is overstepping the limitation of the preclinical model,
indicated by the variability of immunohistochemical staining and by the similarity of genomic
aberrations (**Extended Data 16 and Rebuttal Figure 33**).

We further have performed correlation analyses (e.g. CD8, PD-1, PD-L1, NAS, fibrosis, liver
damage, tumor size, and tumor load) to allow readers a more detailed description of the
presented data (**Figure 1, Extended Data 4+24 and Rebuttal Figure 6, 7c-e and 26**).

Research for a Life without Cancer

9. The companion back to back paper shows more data on the physiology of the pathogenic CD8 T-
cells that I would otherwise ask to this article. Therefore, proper cross-reference of those findings is
needed at least in discussion.

We thank Referee #3 for highlighting the importance of the co-submitted paper Dudek et al.
and therefore, we improved cross-referencing in the discussion.

Research for a Life without Cancer

Referee #4 (Remarks to the Author):

This is an interesting and quite original study of the role of immunity in promoting liver cancer. There are
data from the mouse models presented which show that CD8+ T-cells can contribute to the pathology
of NASH and the risk of cancers. The implication is that checkpoint blockade which can accentuate the
function of CD8 populations can worsen disease. There are also some human data which are fairly
consistent with this idea. It is perhaps not surprising that checkpoint inhibition might worsen an
inflammatory condition, although inducing a cancer risk is very interesting. Overall the authors do a very
good job in describing the cellular responses and the impact of depletion/blockade. There seemed to be
a bit of a gap around defining the mechanisms in terms of how the CD8+ T-cell population induced
cancer. Also it was somewhat unclear what the specificity of these T-cells was and what was triggering
their initial responsiveness in NASH. So although a strong case is made for the pro-tumor role the actual
pathways to disease were less concrete.

We thank Referee #4 for appreciating our study's originality in shedding new light on the role
of immunity promoting liver cancer, with fairly consistent human data correlating with the
findings in the preclinical model. We thank Referee #4 for pointing out the limitations of our
study which has helped us to increase the quality of our manuscript and address the respective
points. We would like to address the concerns of Referee #4 in the following section point-by-
point by newly performed experiments, re-phrasing, re-analysis of the underlying data-sets and
would like to draw attention to the improved cross-referencing to the co-submitted manuscript
Dudek et al., which dissect the molecular and cellular mechanism of CD8+ T-cell dependent
pathogenesis in NASH.

Figure 1: There do not appear to be any iNKT-cells in the UMAP or tise plots – these are discussed
latter in the text. That seems a little surprising as they are quite dominant in the mouse liver and have a
clear transcriptional profile. Could the authors clarify where these cells lie. It would be also useful to
know whether other unconventional cell subsets including GD T-cells and MAIT-cells are incorporated
in this, although they are likely much rarer. The latter may be relevant even if rare as they have been
linked to liver fibrosis. The same questions would also apply to the scRNAseq of the human samples.

We thank Referee #4 for raising this important point. We have now dissected mouse NK1.1+
cells in the revised version of our manuscript into NK1.1+TCRb+ as NKT and NK1.1+TCRb-
as NK cells (**Figure 1** and **Rebuttal Figure 26**). Similarly, we highlighted NKT-cells, MAITs,
and $\gamma\delta$ T-cells in our patient-derived hepatic lymphocytes analysis by flow cytometry, newly
performed scRNA-seq, and CYTOF analysis (**Figure 5**, **Extended Data 25-27** and **Rebuttal**
**Figure 9-12**).

We agree with Referee #4, that MAITs might be important and thus included quantification of
MAITs in our newly performed scRNA-seq and CYTOF analyses of patient-derived hepatic
lymphocytes. In these analyses, no change of relative abundance of MAITs was observed
when comparing control vs. NAFLD/NASH patients. Moreover, we would like to draw attention
to the co-submitted manuscript Dudek et al., which analyzed - together with us - CD-HFD-fed

Research for a Life without Cancer

Ja18^{-/-} and CD1d^{-/-} mice. The latter did not display significant changes in pathology compared
to CD-HFD-fed control mice at time points of established NASH.

We agree with Referee #4, that $\gamma\delta$ T-cells may be important, however in our mouse model
upon NASH establishment, we detected no difference in hepatic abundance of $\gamma\delta$ T-cells
between chow or CD-HFD-fed control mice (**Extended Data 3**). Furthermore, data presented
in **Figures 1 and 4** and **Extended Data 3** argue against a major direct contribution of $\gamma\delta$ T-
cells in the preclinical model at time points of 6 or 12 months of diet-feeding. We agree that $\gamma\delta$
T-cells might be important in the pathogenesis of NASH and NASH to HCC transition, however,
e.g. rather in collaboration with CD8⁺ T cells, also in the context of PD1-related
immunotherapy. In humans, our data is not conclusive in all experiments, e.g. our data indicate
for $\gamma\delta$ T-cells, if we compare: bulk RNA-seq indicates a reduced expression in severe NASH
pathology of EOMES, TRDC, and TRGC1 (**Extended Data 28** and **Rebuttal Figure 32**),
however, both flow cytometry cohorts and the scRNA-seq cohort indicate no change of either
$\gamma\delta^+$ T-cells or $\gamma\delta^+$ Eomes⁺ T-cells comparing control vs NAFLD/NASH patients (**Extended**
**Data 25, 27** and **Rebuttal Figure 10 and 12**).

Corroborating the human flow cytometry data in our mouse model upon NASH establishment,
we detected no difference in hepatic abundance of $\gamma\delta$ T-cells between chow- or CD-HFD-fed
control mice. Furthermore, data presented in **Figures 1** and **Extended Data 3** argues against
the major contribution of $\gamma\delta$ T-cells in the mouse model of NASH. We did not observe significant
differences in the “other leukocytes” subset. In the revised manuscript, we analyzed $\gamma\delta$ -T-cells
separately to strengthen the point, that these cells are not significantly changed (**Extended**
**Data 3, 20-23** and **Rebuttal Figure 8** and **18-21**).

Figure 1e: What are the p values on the right referencing? The difference in the PD1⁺ population does
not appear to be significant. How valid is the PD1⁺ subset as a subcluster and also what are the critical
significant differences apart from elevated PD1 expression – some justification for this early on would
be helpful. Often PD1 expression is more of a gradient (even within PD1⁺ cells) so a binary distinction
needs a bit more justification. Does this group of cells have distinct TCRs from the non-PD1 (or lower
PD1) subset or are they the same population with distinct expression? Some data on this would address
the question about specificity – although this would be better addressed by defining actual TCR-specific
(or independent) functionality.

We thank Referee #4 for raising important points about **Figure 1**. We have now improved our
manuscript by clarifying, that the p-values on the right-side reference to abundance in CD-
HFD-fed mice compared to chow-fed control mice. We agree with Referee 4, that the CD8⁺PD-
1⁺ subpopulation was (initially) not significantly changed (p= 0.09). Upon adding novel data,
and re-analysis according to the comment of Referee #4, by highlighting NKT cells, CD8⁺PD1⁺
(p= 0.03) are significantly changed. Furthermore, by using AI-based analysis of various
parameters displaying our used CD-HFD-fed cohorts as a total, we observed that pathology

Research for a Life without Cancer

severity correlated with the hepatic abundance of CD8+ T-cells and PD1 polarization of these
cells (**Figure 1 and 4, Extended Data 4 and 24 and Rebuttal Figure 6, 7c-e, 16 and 26**).
These analyses indicate, that besides changes e.g. in myeloid subsets, CD8+PD1+ cells are
a key subset in NASH-diseased mice as well as in human patients (**Figure 5 and Rebuttal**
**Figure 9**). To underline the importance of a CD8+PD-1+ subset -expressing
effector/exhaustion markers correlating with disease progression- we have connected the data
of **Figure 1** more closely to single-cell RNA-seq data presented in **Figure 1** (e.g. unique
transcriptional activity in NASH-derived CD8+ T-cells (**Figure 1 and Rebuttal Figure 26**) and
improved cross-referencing to the data co-submitted manuscript Dudek et al. in the discussion.
Furthermore, we have included in the revised manuscript, that we did not observe for CD8+ T-
cells a sufficient/non-binary gradient of PD-1 expression, allowing dissection into PD-
1^{negative} /PD-1 $^{\text{intermediate}}$ /PD-1 $^{\text{high}}$ subsets upon 12 months CD-HFD-feeding, (**Extended Data 3**).
Moreover, we functionally show that CD8+ T-cell are indeed the drivers of anti-PD1-related
therapy induced liver cancer.

We thank Referee #4 for pointing out the question about TCR dependency and thus would like
to draw the attention to the co-submitted manuscript Dudek et al., which describes TCR-
independent mechanisms on a cellular and molecular level driving CD8+ T cell-mediated
hepatocyte cell death. NASH-diet feeding experiments using mice with impaired TCR-
dependent effector function have been performed in collaboration with Dudek et al. 12-months
CD-HFD-fed perforin $^{-/-}$ mice developed NASH (including systemic obesity, fibrosis, ALT) and
NASH-induced hepatocarcinogenesis similar to WT control animals. We have now addressed
the question on TCR-specificity by improved cross-referencing to the co-submitted manuscript
Dudek et al.. In fact, it turns out that the effect of CD8+ T-cells is TCR-effector function
independent. Furthermore, we have performed combination therapy of 1) anti-TNF with/without
PD-1 targeted immunotherapy; 2) anti-CD4 with/without PD-1 targeted immunotherapy; 3) anti-
CD8 with PD-1 targeted immunotherapy and 4) PD-L1 targeted immunotherapy, to strengthen
hypotheses about TCR-independent mechanisms (**Figure 4, Extended Data 20-23 and**
**Rebuttal Figure 16 and 18-21**).

Figure 1f: The stains are both single stains. It should be possible to show a double staining CD8+PD1+
population and enumerate them as this seems like the critical part of the study.

We thank Referee #4 for pointing that out. We performed an additional double staining
corroborating our flow cytometry data in **Figure 1**. In line, we have now included histological
double staining in a revised manuscript (**Figure 1, Extended Data 3, 12, and Rebuttal Figure**
**8, 26 and 35**). These data indicated that PD1+ expression is indeed associated with CD8+
staining.

Research for a Life without Cancer

Figure 1j: One of the most upregulated genes in the PD1+ subset is IL-10. Do the authors have any data
on whether this is secreted by this subset. Although the subset is labelled as “PD1+” it is not the top
upregulated gene here (as above). A side-by-side broader functional study would add a bit of resolution
here and if they do secrete IL-10 this may impact on the overall interpretation. The interpretations about
function are all via the screening approaches so some further specific back up by FACS/ELISA would
be helpful in confirming functionality, especially in the context of an “exhausted” phenotype – this would
clarify the statement on line 199 about “potential effector function”. Such an experiment would also be
valuable in the anti-PD1 treated mice in later parts of the manuscript.

We fully agree and thank Referee #4 for raising this important point of IL-10 expression, which
was also raised in a recent study (Breuer et al., 2020). We analyzed IL-10+ CD8+PD-1+ T-
cells in our revised manuscript (**Extended Data 19** and **Rebuttal Figure 30**).

However, we did not see any changes in IL10+ CD8+PD1+ in comparison to CDHFD-fed and
control mice. Moreover, IL10 levels measured by ELISA did neither drop upon CD8-depletion
(**Extended Data 10** and **Rebuttal Figure 29**) nor increase significantly upon anti-PD1
treatment (**Extended Data 13** and **Rebuttal Figure 28**). Thus, an increased anti-inflammatory
role by IL-10 expressing CD8+ T-cells upon PD1-targeted immunotherapy could not be
corroborated (**Extended Data 19** and **Rebuttal Figure 30k**) (Breuer et al., 2020). Of note, in
this publication diet-based NAFLD induction was achieved by feeding either WD or CD-HFD
for 8-10 weeks. This is in strong contrast to our experimental regime of applying diet for 3, 6,
or 12 months as we show, that the preclinical model presents different stages of NASH
pathology severity including hepatocarcinogenesis (**Figure 1** and **Rebuttal Figure 26**). Thus,
in our opinion, CD8+PD1+ cells are the main effector population driving liver inflammation and
liver cancer – most likely independent of IL10 being one of the most upregulated genes in this
subset. In line with our mouse data scRNA-seq of CD8+PD1+ cells derived from control vs
NAFLD/NASH patients did not reveal increased IL10 expression. Besides in bulk RNA-seq of
human liver tissue, we observed a variable expression pattern depending on NASH pathology
severity (**Figure 5**, **Extended Data 28** and **Rebuttal Figure 9, 28**).

Figure 2: It was not that clear why depleting CD8s had no impact on ALT, suggesting they are not playing
a role in vivo, while blocking PD1 had some impact (AST is not shown for the anti-CD8 treatment).

We thank Referee #4 for highlighting that CD8+-T cell depletion in the context of NASH-HCC
transition had no or only minor impact on ALT reduction, an effect that has also come to our
attention and has puzzled us. On the other hand, we would like to note that in the context of
anti-PD1-related immunotherapy triggered liver damage CD8+ T cell depletion did lead to a
significant reduction in liver damage and NAFLD activity score. Thus, we believe that the anti-
PD1 therapy-related damage in NASH and NASH to HCC transition is mainly triggered by
CD8+ T cells. In contrast, in the context of NASH development without anti-PD1 antibody
treatment, other cells than CD8+ T-cell also contribute to liver damage – and that progressive

Research for a Life without Cancer

NASH is characterized by multi-faceted, collateral damage through myeloid cells, adaptive
cells, and cell death.

We think that CD8+ T-cells have an important *in vivo* role driving NASH to HCC transition, as
we strongly decreased or eliminated HCC by CD8+ T-cell depletion (both in NASH or NASH
with anti-PD1 treatment). In line, the co-submitted manuscript by Dudek et al., described
hepatocyte death by a CD8-dependent mechanism.

Notably, ALT can be elevated as a result of the chronic metabolic environment and/or as a
result of the still ongoing hepatic inflammation independent of CD8+ or NK1.1+ cells
(**Extended Data 9** and **Rebuttal Figure 17**).

Further, it can be that actually at late time points of co-existence of tumors and NASH – the
collateral damage might be mainly triggered by non-CD8+ T-cells. We have confirmed the
efficient depletion of the CD8 T-cells in our models, excluding that this might be a reason. AST
levels are included in our AI-based analysis (**Figure 1 and 4, Extended Data 4 and 24** and
**Rebuttal Figure 6, 7c-e, 16** and **26**), indicating no change upon CD8 depletion as well.

Line 202 – lack of impact of anti-PD1. Is there a control for this experiment? The implication is that this
lack of impact is etiology-specific but it may also be that the intervention does not work well in other
HCC models.

We thank Referee #4 for highlighting the etiology-dependent potential outcome of PD-1-
targeted immunotherapy against HCC. We agree with Referee #4, that there might be
bivalence in other HCC models and, more importantly, only a subset of HCC patient react to
PD-1 targeted immunotherapy (El-Khoueiry et al., 2017; Hage et al., 2019). Thus, we have
also performed anti-PD-L1 targeted immunotherapy in CDHFD-fed mice with and without
established liver cancer (**Extended Data 7** and **Rebuttal Figure 13**).

The data of our study indicate that similar to anti-PD1 - anti-PDL1-treatment does not induce
an anti-liver cancer effect for NASH-induced HCC but rather induces similar to anti-PD1
treatment a pro-inflammatory and pro-carcinogenic effect. These data further suggest that in
the preclinical NASH models used, both PD1- or PDL1-targeted immunotherapy induces
adverse effects. This is corroborated by our increased, retrospective cohort HCC-patients of
different etiologies under PD(L)1-targeted immunotherapy, in which multivariate analysis
results in NAFLD/NASH being an independent negative factor for overall survival.
Furthermore, we corroborated our hypothesis of non-viral (NASH-related) HCC being less
responsive to immunotherapy by a meta-analysis including 1656 patients of the three most
important clinical trials, identifying immunotherapy vs control for viral HCC as favorable
treatment (HR(viral)= 0.64), in contrast, non-viral-HCC showed less benefit (HR(non-viral)=
0.92) for immunotherapy (**Figure 6, Extended Data 30-32, Supplementary Table 9** and
**Rebuttal Figure 1-4**)).

Research for a Life without Cancer

Figure 5b and the text are presented in a slightly confusing way. It would be easier to understand the
disease associations of %CD8 (of CD3), and % PD1+ (or MFI) of CD3+CD8+ first. The association of
CD103 with tissue residency in the liver is not as good as other tissues, so a broader look at the
CD8+PD1+ population by flow would be better as well as some caution in interpretation.

We agree with this comment and thank Referee #4 for highlighting this problem. Inline, we
have now improved our manuscript as suggested by Referee#4 (included in **Extended Data**
**25 and 27** and **Rebuttal Figure 10 and 12**). Moreover, we corroborated the association of
NASH patients and CD103 in a second patient cohort using CYTOF (included in **Figure 5** and
**Rebuttal Figure 9**).

Figure 5e could include some study of CD4s as well for reference. That subset has been linked to NASH
pathogenesis as well. As above, it should be possible to perform some dual CD8 and PD1 staining to
map the subset of interest.

We thank Referee #4 for highlighting this point, that CD4 T-cells and their expression of PD-1
might play a crucial role in the observed phenotype and thus included an in detail analysis of
CD4 T-cells to the majority of our experiments (e.g. **Extended Data 3** and **Rebuttal Figure 8**).
However, in the preclinical model the magnitude of effects observed in CD4+ T-cells is minor
when compared to CD8+ T-cells (e.g. **Extended Data 11** mean (CD8+CD62L-CD44+CD69+) ~12% (%of CD45+) vs mean(CD4+CD62L-CD44+CD69+) ~4% (%of CD45+) upon PD-1
targeted immunotherapy).

Data obtained from CD4 depletion with/without PD1-targeted immunotherapy indicate, that the
increased hepatocarcinogenesis in the context of anti-PD1 related immunotherapy is
independent of hepatic abundance of CD4+ T-cells in the preclinical NASH/HCC model
(included in **Figure 4**, **Extended Data 22 and 23** and **Rebuttal Figure 16, 20 and 21**).
However, CD4+ T-cells might have a diverse set of effector functions (e.g. interpreting tumor
incidence in anti-CD8/anti-PD1 treated animals: in the absence of CD8+ T-cells but
immunotherapy, thus CD4+ T-cells might be responsible for baseline tumor incidence; or the
trends of increased tumor incidence upon anti-CD4/anti-PD1 co-treatment in **Figure 4** and
**Rebuttal Figure 16n**). To allow a wider interpretation of a potential effect of CD4+ T-cells in
our preclinical model, we integrated and correlated the variety and potential changes upon 12
1536 months of diet-feeding in the AI-based analyses correlating disease parameters with cellular
abundance and polarization (**Figure 1**, **Extended Data 4 and 24** and **Rebuttal Figure 6, 7c-
e and 26**). These data further strengthens that CD4+ T-cells play a minor role, as we see no
significant correlation of CD4-depleted animals with histological, or serological markers.

Of note, CD4+ T-cells are also significantly changed in the human situation by classical flow
cytometry, but in the light of the results obtained in the preclinical model, we decided to not
investigate this result extensively (**Extended Data 25** and **Rebuttal Figure 10**). Of note, CD4+
T-cells are also significantly changed in the human situation and have also analyzed human

Research for a Life without Cancer

CD4+ cells a by scRNASeq (**Extended Data 27** and **Rebuttal Figure 12**). In addition, we have
performed a velocity analyses of the scRNA Seq data of mouse and human CD4 T cells (see
**Rebuttal Figure 31**). In mouse, no significant velocity flow was detected in 12 months CD-
HFD-fed mice, indicating, that CD4 cells are not transcriptionally activated and driven by
NASH-conditions or PD-1-targeted immunotherapy in NASH. However, we want to point out,
that in the mouse NASH model CD8 T-cells increase statistically significant and thus CD4 are
relatively fewer cells compared to CD8. Therefore, the velocity analysis of mouse CD4 T-cells
need to be taken with caution, because we included 300-500 cells only per described subset.
As consequence, we included the negative CD4 T-cell data not in the manuscript but in the
Rebuttal letter. Velocity analyses on human CD4 lead to comparable problems like seen in
mouse. As a consequence, we included the negative CD4 T-cell data not in the manuscript but
in the Rebuttal letter as **Rebuttal Figure 31**.

However, we discuss the potential role of CD4+ T-cells in greater detail in the main text.

Figure 5f is not really that convincing of a relationship with TNF – the r-squared value would be better
to illustrate and would be very low. If the authors think TNF secretion is critical it would be possible to
explore this further in the mouse model.

We thank Referee #4 for highlighting this point. Although TNF is correlated significantly with
PD1 abundance, the correlation is weak as indicated by the r-value and therefore moved the
data to the Extended Data. Moreover, we fully agree with this Referee that further experiments
were needed to underline the role of TNF in NASH/HCC transition in the context of anti-PD1
related immunotherapy.

Thus, we have performed an anti-TNF treatment with or without PD-1- targeted
immunotherapy in the context of NASH/HCC. Anti-TNF treatment without PD1-targeted
immunotherapy led to liver cancer formation comparable to control-treated CD-HFD-fed mice.
However, anti-TNF treatment in the context of PD1-targeted immunotherapy leads to a
significant reduction of tumor incidence compared to anti-PD1 treated CD-HFD-fed mice,
indicating that TNF exerts key functions of the observed adverse effects triggered by PD1-
targeted immunotherapy, namely the increased NAS, liver damage, and hepatocarcinogenesis
(**Figure 4, Extended Data 20 and 21** and **Rebuttal Figure 16, 18** and **19**).

Moreover, the combination of anti-PD1 therapy with anti-CD8 – also ablating the adverse and
pro-carcinogenic effects of CD8+ T-cells emphasize that CD8+ T-cells are a major cell
population mediating increased hepatocarcinogenesis in a TNF-dependent mechanism upon
PD1-targeted immunotherapy (included in **Figure 4, Extended Data 20 and 21** and **Rebuttal**
**Figure 16, 18** and **19**).

Importantly, by comparing classical flow cytometry, CYTOF, and on scRNA-seq level of
mouse-human of CD8+ T-cells isolated from liver tissue of NASH mice or patients, we

Research for a Life without Cancer

identified similar populations and transcriptional activation of CD8+ PD1+ in a total of three
independent center patient cohorts (**Figure 5, Extended Data 25 and 27 and Rebuttal Figure**
**9, 10 and 12**). These data indicate that results obtained and hypotheses built from the
preclinical NASH model and are in line with published results, where TNF blockade uncouples
mediated toxicity in dual CTLA-4 and PD-1 immunotherapy (Perez-Ruiz et al., 2019).

For Figure 5G some disease controls would be valuable.

We thank Referee #4 for his/her comment for pointing out the lack of appropriate control groups
(e.g. NASH-HCC vs different etiology-induced HCC under Sorafenib/different multi-kinase
inhibitors as a second/third-line therapy). Although of extreme interest for public health and
public knowledge, we described this important issue in our discussion and to the best of our
knowledge there are no NASH-HCC treated cohorts available (apart from, possibly, inside of
the big pharma-industry), which would allow an adequate control arm. Thus, we evaluated
potential disease controls in the manuscript by performing a meta-analysis including 1656
patients of the three major clinical trials (Imbrave 150; Checkmate 459; Keynote 240). Here
we could identify immunotherapy vs control for viral HCC as favorable treatment ($HR(viral)=$
0.64), in contrast non-viral-HCC showed less benefit ($HR(non-viral)= 0.92$) for immunotherapy
(**Figure 6, Extended Data 30-32, Supplementary Table 9 and Rebuttal Figure 1-4**)).

Furthermore, we toned down the conclusions of our retrospective cohort in the manuscript and
would like to point out, that bigger cohorts and prospective clinical trials are of utmost
importance for the scientific community.

Line 493+: This sentence is perhaps overstating the data, which were not significant in all those
parameters. It is likely quite hard to make the firmest comparisons, especially in such a retrospective
analysis, where the heterogeneous group of patients with eg viral aetiologies will be on effective
therapies - the actual aetiologies were not obvious in the supplementary data. This interpretation could
be a bit more cautious throughout (eg. it is in the abstract).

We would like to thank Referee #4 for the important comment and agree. Thus, we toned down
the wording and interpretation of our data. As described previously, we recruited additional
patients to increase the number of patients in our initial clinical cohort from 65 to 130 HCC
patients under anti-PD(L)1-targeted immunotherapy, which we validated in a second cohort
(**Figure 6 and Rebuttal Figure 1f,g**).

We agree with Referee #4, that the presented retrospective PD-(L)1 targeted immunotherapy
treated NAFLD/NASH-associated HCC cohort - although unique for Europe and treatment not
officially licensed and thus reimbursement - is still small, although we would like to point out,
that prominent trends or effects can be seen in small retrospective cohorts as well. Thus, our
analyses of BCLC-C NAFLD/NASH-HCC vs other-etiological-HCC patients indicated, that
NAFLD/NASH-HCC has significantly reduced overall survival compared to other-etiological-

Research for a Life without Cancer

HCC in this small retrospective cohort. Of note, multivariate analyses identified NAFLD/NASH
as an independent factor for treatment response and thus identifying NAFLD/NASH as a
negative predictor for HCC immunotherapy (**Supplementary Table 9**).

Like previously mentioned, we corroborated our hypothesis of non-viral (NASH-related) HCC
being less responsive to immunotherapy by a meta-analysis including 1656 patients of the
three most important clinical trials (IMbrave 150; Checkmate 459; Keynote 240), identifying
immunotherapy vs control for viral HCC as favorable treatment (HR(viral)= 0.64), in contrast,
non-viral-HCC showed less benefit (HR(non-viral)= 0.92) for immunotherapy (**Figure 6,**
**Extended Data 30-32, Supplementary Table 7 and Rebuttal Figure 1-4**)).

Thus, we toned down the conclusions of our retrospective cohort in the manuscript and again
would like to point out, that bigger cohorts and prospective clinical trials are of utmost
importance for the scientific community.

Research for a Life without Cancer

**References**

- 1. Agdashian, D., ElGindi, M., Xie, C., Sandhu, M., Pratt, D., Kleiner, D.E., Figg, W.D., Rytlewski, J.A.,
Sanders, C., Yusko, E.C., et al. (2019). The effect of anti-CTLA4 treatment on peripheral and intra-tumoral
T cells in patients with hepatocellular carcinoma. *Cancer Immunol. Immunother.* 68, 599–608.
- 2. Boege, Y., Malehmir, M., Healy, M.E., Bettermann, K., Lorentzen, A., Vucur, M., Ahuja, A.K., Böhm, F.,
Mertens, J.C., Shimizu, Y., et al. (2017). A Dual Role of Caspase-8 in Triggering and Sensing Proliferation-
Associated DNA Damage, a Key Determinant of Liver Cancer Development. *Cancer Cell* 32, 342.
- 3. Breuer, D.A., Pacheco, M.C., Washington, M.K., Montgomery, S.A., Hasty, A.H., and Kennedy, A.J.
(2020). CD8+ T cells regulate liver injury in obesity-related nonalcoholic fatty liver disease. *Am. J. Physiol.*
*Liver Physiol.* 318, G211–G224.
- 4. Brummelman, J., Haftmann, C., Núñez, N.G., Alvisi, G., Mazza, E.M.C., Becher, B., and Lugli, E. (2019).
Development, application and computational analysis of high-dimensional fluorescent antibody panels for
single-cell flow cytometry. *Nat. Protoc.* 14.
- 5. Cortellini, A., Bersanelli, M., Buti, S., Cannita, K., Santini, D., Perrone, F., Giusti, R., Tiseo, M., Michiara,
1647 M., Di Marino, P., et al. (2019). A multicenter study of body mass index in cancer patients treated with anti-
1648 PD-1/PD-L1 immune checkpoint inhibitors: when overweight becomes favorable. *J. Immunother. Cancer*
7, 57.
- 6. Cui, P., Li, R., Huang, Z., Wu, Z., Tao, H., Zhang, S., and Hu, Y. (2020). Comparative effectiveness of
pembrolizumab vs. nivolumab in patients with recurrent or advanced NSCLC. *Sci. Rep.* 10, 1–7.
- 7. Duffy, A.G., Ulahannan, S. V, Makorova-rusher, O., Rahma, O., Wedemeyer, H., Pratt, D., Davis, J.L.,
Hughes, M.S., Heller, T., ElGindi, M., et al. (2016). Tremelimumab in Combination with Ablation in Patients
with Advanced Hepatocellular Carcinoma. *J. Hepatol.* 66, 482–484.
- 8. El-Khoueiry, A.B., Sangro, B., Yau, T., Crocenzi, T.S., Kudo, M., Hsu, C., Kim, T.-Y.Y., Choo, S.-P.P.,
Trojan, J., Welling, T.H., et al. (2017). Nivolumab in patients with advanced hepatocellular carcinoma
(CheckMate 040): an open-label, non-comparative, phase 1/2 dose escalation and expansion trial. *Lancet*
6736, 1–11.
- 9. Finkin, S., Yuan, D., Stein, I., Taniguchi, K., Weber, A., Unger, K., Browning, J.L., Goossens, N.,
Nakagawa, S., Gunasekaran, G., et al. (2015). Ectopic lymphoid structures function as microniches for
tumor progenitor cells in hepatocellular carcinoma. *Nat. Immunol.* 16, 1235–1244.
- 10. Finn, R., Ryoo, B.-Y., Merle, P., Kudo, M., Bouattour, M., Lim, H.Y., Breder, V., Edeline, J., Chao, Y.,
Ogasawara, S., et al. (2019). Results of Keynote-240: Phase 3 Study of Pembrolizumab vs Best
Supportive Care for Second-Line Therapy in Advanced Hepatocellular Carcinoma. In ASCO Annual
Meeting, p.
- 11. Finn, R.S., Qin, S., Ikeda, M., Galle, P.R., Ducreux, M., Kim, T.Y., Kudo, M., Breder, V., Merle, P., Kaseb,
1667 A.O., et al. (2020). Atezolizumab plus bevacizumab in unresectable hepatocellular carcinoma. *N. Engl. J.*
*Med.* 382, 1894–1905.
- 12. Gomes, A.L., Teijeiro, A., Burén, S., Tummala, K.S., Yilmaz, M., Waisman, A., Theurillat, J.P., Perna, C.,
and Djouder, N. (2016). Metabolic Inflammation-Associated IL-17A Causes Non-alcoholic Steatohepatitis
and Hepatocellular Carcinoma. *Cancer Cell* 30, 161–175.
- 13. Hage, C., Hoves, S., Ashoff, M., Schandl, V., Hört, S., Rieder, N., Heichinger, C., Berrera, M., Ries, C.H.,
Kiessling, F., et al. (2019). Characterizing responsive and refractory orthotopic mouse models of
hepatocellular carcinoma in cancer immunotherapy. *PLoS One* 14, e0219517.
- 14. Kim, C.G., Kim, C., Yoon, S.E., Kim, K.H., Choi, S.J., Kang, B., Kim, H.R., Park, S.-H., Shin, E.-C., Kim,
Y.-Y., et al. (2020). Hyperprogressive disease during PD-1 blockade in patients with advanced
hepatocellular carcinoma. *J. Hepatol.*
- 15. Kudo, M. (2018). Combination Cancer Immunotherapy in Hepatocellular Carcinoma. *Liver Cancer* 7, 20–
27.
- 16. Llovet, J.M., Di Bisceglie, A.M., Bruix, J., Kramer, B.S., Lencioni, R., Zhu, A.X., Sherman, M., Schwartz,
1681 M., Lotze, M., Talwalkar, J., et al. (2008). Design and Endpoints of Clinical Trials in Hepatocellular
Carcinoma. *JNCI J. Natl. Cancer Inst.* 100, 698–711.
- 17. Llovet, J.M., Zucman-Rossi, J., Pikarsky, E., Sangro, B., Schwartz, M., Sherman, M., and Gores, G.
(2016). Hepatocellular carcinoma. *Nat. Rev. Dis. Prim.* 2, 16018.
- 18. Ma, C., Kesarwala, A.H., Eggert, T., Medina-Echeverez, J., Kleiner, D.E., Jin, P., Stroncek, D.F., Terabe,
1686 M., Kapoor, V., ElGindi, M., et al. (2016). NAFLD causes selective CD4+ T lymphocyte loss and promotes
hepatocarcinogenesis. *Nature* 531, 253–257.
- 19. Malehmir, M., Pfister, D., Gallage, S., Szydlowska, M., Inverso, D., Kotsiliti, E., Leone, V., Peiseler, M.,
Surewaard, B.B.G.J., Rath, D., et al. (2019). Platelet GPIIb is a mediator and potential interventional target
for NASH and subsequent liver cancer. *Nat. Med.* 25, 641–655.

Research for a Life without Cancer

20. Moser, J.C., Wei, G., Colonna, S. V, Grossmann, K.F., Hyngstrom, J.R., Moser, J.C., Wei, G., Colonna, S. V, Grossmann, K.F., Patel, S., et al. (2020). Comparative-effectiveness of pembrolizumab vs . nivolumab for patients with metastatic melanoma. *Acta Oncol. (Madr)*. 59, 434–437.
 21. Nakagawa, H., Umemura, A., Taniguchi, K., Font-Burgada, J., Dhar, D., Ogata, H., Zhong, Z., Valasek, M.A., Seki, E., Hidalgo, J., et al. (2014). ER Stress Cooperates with Hypernutrition to Trigger TNF-Dependent Spontaneous HCC Development. *Cancer Cell* 26, 331–343.
 22. Park, E.J., Lee, J.H., Yu, G., He, G., Ali, S.R., Ryan, G., Holzer, R.G., Österreicher, C.H., Takahashi, H., and Karin, M. (2011). Dietary and genetic obesity promote liver inflammation and tumorigenesis by enhancing IL-6 and TNF expression *Euk. Cell* 140, 197–208.
 23. Perez-Ruiz, E., Minute, L., Otano, I., Alvarez, M., Ochoa, M.C., Belsue, V., de Andrea, C., Rodriguez-Ruiz, M.E., Perez-Gracia, J.L., Marquez-Rodas, I., et al. (2019). Prophylactic TNF blockade uncouples efficacy and toxicity in dual CTLA-4 and PD-1 immunotherapy. *Nature* 569, 428–432.
 24. Pikarsky, E., Porat, R.M., Stein, I., Abramovitch, R., Amit, S., Kasem, S., Gutkovich-Pyest, E., Urieli-Shoval, S., Galun, E., and Ben-Neriah, Y. (2004). NF-kappaB functions as a tumour promoter in inflammation-associated cancer. *Nature* 431, 461–466.
 25. Remmerie, A., Martens, L., Thoné, T., Castoldi, A., Seurinck, R., Pavie, B., Roels, J., Vanneste, B., De Prijck, S., Vanhockerhout, M., et al. (2020). Osteopontin Expression Identifies a Subset of Recruited Macrophages Distinct from Kupffer Cells in the Fatty Liver. *Immunity* 53, 641-657.e14.
 26. Ringelhan, M., Pfister, D., O'Connor, T., Pikarsky, E., and Heikenwalder, M. (2018). The immunology of hepatocellular carcinoma. *Nat. Immunol.* 19.
 27. Scheiner, B., Kirstein, M.M., Hucke, F., Finkelmeier, F., Schulze, K., von Felden, J., Koch, S., Schwabl, P., Hinrichs, J.B., Waneck, F., et al. (2019). Programmed cell death protein-1 (PD-1)-targeted immunotherapy in advanced hepatocellular carcinoma: efficacy and safety data from an international multicentre real-world cohort. *Aliment. Pharmacol. Ther.* 49, 1323–1333.
 28. Tummala, K.S., Gomes, A.L., Yilmaz, M., Graña, O., Bakiri, L., Ruppen, I., Ximénez-Embún, P., Sheshappanavar, V., Rodriguez-Justo, M., Pisano, D.G., et al. (2014). Inhibition of De Novo NAD+ Synthesis by Oncogenic URI Causes Liver Tumorigenesis through DNA Damage. *Cancer Cell* 26, 826–839.
 29. Wang, Z., Aguilar, E.G., Luna, J.I., Dunai, C., Khuat, L.T., Le, C.T., Mirsoian, A., Minnar, C.M., Stoffel, K.M., Sturgill, I.R., et al. (2018). Paradoxical effects of obesity on T cell function during tumor progression and PD-1 checkpoint blockade. *Nat. Med.* 1.
 30. Weiler-Normann, C., and Lohse, A.W. (2016). Nonalcoholic Fatty Liver Disease in Patients with Autoimmune Hepatitis: Further Reason for Teeth GNASHing? *Dig. Dis. Sci.* 61, 2462–2464.
 31. Wolf, M.J., Adili, A., Piotrowitz, K., Abdullah, Z., Boege, Y., Stemmer, K., Ringelhan, M., Simonavicius, N., Egger, M., Wohlleber, D., et al. (2014). Metabolic activation of intrahepatic CD8+ T cells and NKT cells causes nonalcoholic steatohepatitis and liver cancer via cross-talk with hepatocytes. *Cancer Cell* 26, 549–564.
 32. Yau, T., Park, J., Finn, R.S., Cheng, A., Mathurin, P., Edeline, J., Kudo, M., Han, K., Harding, J.J., Merle, P., et al. (2019). CheckMate 459 : A Randomized , Multi-Center Phase 3 Study of Nivolumab vs Sorafenib as First-Line Treatment in Patients With Advanced Hepatocellular Carcinoma. In *ESMO Congress*, p.
 33. Zen, Y., and Yeh, M.M. (2018). Hepatotoxicity of immune checkpoint inhibitors: a histology study of seven cases in comparison with autoimmune hepatitis and idiosyncratic drug-induced liver injury. *Mod. Pathol.* 31, 965–973.
 34. Zhu, A.X., Finn, R.S., Edeline, J., Cattan, S., Ogasawara, S., Palmer, D., Verslype, C., Zagonel, V., Fartoux, L., Vogel, A., et al. (2018). Pembrolizumab in patients with advanced hepatocellular carcinoma previously treated with sorafenib (KEYNOTE-224): a non-randomised, open-label phase 2 trial. *Lancet Oncol.* 2045, 1–13.

1739 **Rebuttal Figures**
Rebuttal Figure 1: PD-1 and PD-L1 targeted immunotherapy in advanced HCC has a distinct effect depending on disease etiology

(a) Immunohistochemical staining and (b) quantification of hepatic PD-1, CD8 and CD4 expressing cells of NAFLD and NASH patients in Supplementary Table 3 with varying stages of fibrosis (NAFLD n= 9 patients; NASH F1/0 n= 7 patients; NASH F2 n= 12 patients; NASH F3 n= 21 patients; NASH F4 n= 16 patients; CD4: NAFL n= 6 patients; NASH F1/0 n= 4 patients; NASH F2 n= 8 patients; NASH F3 n= 17 patients; NASH F4 n= 9 patients). (c) Correlation analysis of PD-1 against fibrosis scoring according to Brunt by immunohistochemical staining by RNA-sequencing (NAFLD/NASH n= 65 patients). (d) A total of 1656 patients were included in all three randomized trials, and 985 patients received a checkpoint inhibitor (Supplementary Table 7). Separate meta-analyses were performed for each of the three etiologies: non-viral (including mostly NASH and alcohol intake), HCV and HBV. (e) HCV and HBV were pooled into a separate category, termed "viral", and a subsequent meta-analysis comparing viral (n=919) and non-viral, including mostly NASH and alcohol intake (n=737) was performed. Hazard ratios for each trial are represented by squares, the size of the square represents the weight of the trial in the meta-analysis.

Research for a Life without Cancer

The horizontal line crossing the square represents the 95% confidence interval (CI). The diamonds represent the
estimated overall effect based on the meta-analysis random effect of all trials. Inverse variance and random effects
methods were used to calculate HRs, 95% CIs, P values, and the test for overall effect; these calculations were
two-sided. The Cochran's Q-test and I^2 were used to calculate heterogeneity. Random = random effects method,
IV = Inverse variance. **(f)** Nonalcoholic fatty liver disease (NAFLD) is associated with a worse outcome in patients
with hepatocellular carcinoma (HCC) treated with PD-(L)1-targeted immunotherapy. A total of 130 patients with
advanced HCC received PD-(L)1-targeted immunotherapy (**Supplementary Table 8**). Kaplan-Meier curve display
overall survival of patients with NAFLD vs. those with any other etiology; all 130 patients were included in these
survival analyses (NAFLD n=13, any other etiology n=117). **(g)** Validation cohort of patients with HCC treated with
PD-(L)1-targeted immunotherapy. A total of 118 patients with advanced HCC received PD-(L)1-targeted
immunotherapy (**Supplementary Table 10**). Kaplan-Meier curve display overall survival of patients with NAFLD vs.
those with any other etiology; all 118 patients were included in these survival analyses (NAFLD n=11, any other
etiology n=107).

PRISMA 2009 Flow Diagram

Rebuttal Figure 2: PRISMA Flow chart of the systematic review of targeted immunotherapy in HCC.

Selection of articles assessing the clinical outcome of immune checkpoint inhibitors in advanced HCC for inclusion in the systematic review and meta-analysis. ICPI: Immune checkpoint inhibitor.

Research for a Life without Cancer

Rebuttal Figure 3: PD-1 and PD-L1 targeted immunotherapy in advanced HCC has a distinct effect depending on disease etiology

A total of 1656 patients were included in all three randomized trials, and 985 patients received a checkpoint inhibitor. Subgroup analysis was performed to study the specific effects of immunotherapy comparing non-viral etiologies (n=737) with (a) HBV (n=574) or (b) HCV (n=345). Hazard ratios for each trial are represented by squares, the size of the square represents the weight of the trial in the meta-analysis. The horizontal line crossing the square represents the 95% confidence interval (CI). The diamonds represent the estimated overall effect based on the meta-analysis random effect of all trials.

Research for a Life without Cancer

Rebuttal Figure 4: Subgroup analysis of PD-1 and PD-L1 targeted immunotherapy in first-line trials of advanced HCC

A total of 1243 patients were included in two first-line trials comparing PD-1 or PD-L1 targeted immunotherapy to sorafenib. 707 patients received an immune checkpoint inhibitor (either PD-1 or anti-PD-1). (a) HCV and HBV were pooled into a separate category, termed “viral”, and a subsequent meta-analysis comparing viral (n=754) and non-viral (n=489), mostly NASH and alcohol intake, was performed. A subgroup analysis studying the specific effects of non-viral etiologies (n=489) on the magnitude of effect of immunotherapy are presented, when compared to (b) HBV (n=473) or (c) HCV (n=281). Hazard ratios for each trial are represented by squares, the size of the square represents the weight of the trial in the meta-analysis. The horizontal line crossing the square represents the 95% confidence interval (CI). The diamonds represent the estimated overall effect based on the meta-analysis random effect of all trials.

Research for a Life without Cancer

Rebuttal Figure 5: α -PD-1 treatment does not achieve anti-tumor effects in NASH-induced tumors

(a) Synteny analysis of mouse-HCC and (b) quantification of genomic aberrations by array-based Comparative Genomic Hybridization (aCGH) after 12 months on CD-HFD (n= 19) and human NALFD/NASH-HCC (n= 78). (c) MRI pictures of liver of mice after 13- months CD-HFD-fed mice followed by 7 weeks treatment of CD-HFD or CD-HFD + 7 weeks by α -PD-1 antibodies (CD-HFD n= 6 mice; CD-HFD + α -PD-1 n= 4 mice). Lines indicate tumor nodule. Scale bar: 10 mm. (d) Histological staining of hepatic tissue by H&E, Sirius Red and CD8 of 15 months ND, CD-HFD or CD-HFD-fed mice + 8 weeks treatment by α -PD-1 antibodies (H&E: ND n= 3 mice; CD-HFD n= 10 mice; CD-HFD + α -PD-1 n= 8 mice; Sirius Red: ND n= 3 mice; CD-HFD n= 5 mice; CD-HFD + α -PD-1 n= 9 mice; CD8: ND n= 5 mice; CD-HFD n= 5 mice; CD-HFD + α -PD-1 n= 3 mice). Scale bar: 50 μ m. Arrowheads indicate CD8⁺ cells. (e) Body weight of 15 months ND, CD-HFD or CD-HFD-fed mice + 8 weeks treatment by α -PD-1

Research for a Life without Cancer

antibodies (ND n= 5 mice; CD-HFD n= 4 mice; CD-HFD + α -PD-1 n= 9 mice). **(f)** NAS evaluation by H&E of 15
1803 months ND, CD-HFD or CD-HFD-fed mice + 8 weeks treatment by α -PD-1 antibodies (ND n= 3 mice; CD-HFD n=
10 mice; CD-HFD + α -PD-1 n= 8 mice). **(g)** Fibrosis quantification (Sirius Red) of 15 months ND, CD-HFD or CD-
HFD-fed mice + 8 weeks treatment by α -PD-1 antibodies (ND n= 3 mice; CD-HFD n= 5 mice; CD-HFD + α -PD-1
n= 9 mice). **(h)** ALT levels of 15 months ND, CD-HFD or CD-HFD-fed mice + 8 weeks treatment by α -PD-1
antibodies (ND n= 3 mice; CD-HFD n= 4 mice; CD-HFD + α -PD-1 n= 8 mice). **(i)** Quantification of CD8⁺ and **(j)** PD-
1⁺ cells in hepatic tissue by immunohistochemistry of 15 months ND, CD-HFD or CD-HFD-fed mice + 8 weeks
treatment by α -PD-1 antibodies (ND n= 3 mice; CD-HFD n= 4 mice; CD-HFD + α -PD-1 n= 8 mice; intra-tumoral
staining: CD-HFD n= 3 mice; CD-HFD + α -PD-1 n= 8 mice). **(k)** Quantification and **(l)** expression of PD-1 of hepatic
CD4⁺ and CD8⁺ T-cells by flow cytometry of 15 months CD-HFD or CD-HFD-fed mice + 8 weeks treatment by α -
PD-1 antibodies (CD-HFD n= 4 mice; CD-HFD + α -PD-1 n= 8 mice). **(m)** Macroscopy of liver of 15 months ND, CD-
HFD or CD-HFD-fed mice + 8 weeks treatment by α -PD-1 antibodies. Arrowheads indicate tumor/lesions. Scale
1814 bar: 10 mm. **(n)** Quantification of CD8⁺ T-cells by flow cytometry of 15 months CD-HFD or CD-HFD-fed mice + 8
1815 weeks treatment by α -PD-1 antibodies (ND n= 3 mice; CD-HFD n= 4 mice; CD-HFD + α -PD-1 n= 8 mice). **(o)**
Quantification of tumor/lesion size, **(p)** tumor load and **(q)** tumor incidence of 15 months CD-HFD or CD-HFD-fed
mice + 8 weeks treatment by α -PD-1 antibodies (tumor/lesion size and tumor load: CD-HFD n= 9 mice; CD-HFD +
α -PD-1 n= 7 mice; tumor incidence: CD-HFD n= 17 tumors/lesions in 22 mice; CD-HFD + α -PD-1 n= 10
tumors/lesions in 10 mice).

Rebuttal Figure 6: Hepatic immune cell environment, including CD8⁺ T-cells abundance and effector phenotype correlate with NASH pathology and liver cancer incidence

(a) Data gathered from hepatic tissue analyses was binary correlated with each other of 6- or 12-months ND or CD-HFD-fed mice (ND n= 47 mice; CD-HFD n= 72 mice). NAS correlated with diet, weight, ALT, GOT, cholesterol, Sirius red, CD8 cells/mm², PD-1 cells/mm², F4/80 cluster/mm², MHCII cluster/mm², PD-L1 (%area), CD8 (%CD45), CD8⁺CD44⁺CD62L⁻ (%CD45), mDC MHC II⁺ (%parent), TNF (pg/ml), IL-1β (pg/ml), IL-10 (pg/ml), IL-13 (pg/ml), IP-10 (pg/ml), MCP-1 (pg/ml), CCL3 (pg/ml), CCL4 (pg/ml), MIP-2 (pg/ml). Tumor incidence correlated with diet, weight, ALT, cholesterol, NAS, Sirius red, CD8 cells/mm², PD-1 cells/mm², F4/80 cluster/mm², MHCII cluster/mm², CD8 (%CD45), CD8⁺CD44⁺CD62L⁻ (%CD45), TNF (pg/ml), IL-1β (pg/ml), IP-10 (pg/ml), MCP-1 (pg/ml), CCL3 (pg/ml), CCL4 (pg/ml), MIP-2 (pg/ml).

Research for a Life without Cancer

Rebuttal Figure 7: PD-1-targeted immunotherapy induces hepatic inflammation, which drives hepatocarcinogenesis in a CD8⁺ T-cell dependent manner
(a) Tumor/lesion load and **(b)** tumor/lesion size of 12-months CD-HFD or CD-HFD-fed mice + 8 weeks treatment by α -PD-1, α -PD-1/ α -CD8, α -TNF, α -PD-1/ α -TNF, α -CD4, or α -PD-1/ α -CD4 antibodies (CD-HFD n= 19 mice; CD-HFD + α -PD-1 n= 29 mice; CD-HFD + α -PD-1/ α -CD8 n= 2 mice; CD-HFD + α -TNF n= 3 mice; CD-HFD + α -PD-1/ α -TNF n= 3 mice; CD-HFD + α -CD4 n= 3 mice; CD-HFD + α -PD-1/ α -CD4 n= 8 mice). **(c)** UMAP representation

Research for a Life without Cancer

of 63 parameters (serology, flow cytometry, histology) and **(d)** selected display of analyzed parameters indicating
NASH pathology severity measured of 12 months ND, CD-HFD or CD-HFD-fed mice+ 8 weeks treatment by α -
CD8, α -CD8/ α -NK1.1; α -PD-1, α -PD-1/ α -CD8, α -TNF, α -PD-1/ α -TNF, α -CD4, or α -PD-1/ α -CD4 antibodies (ND n=
22 mice; CD-HFD n= 31 mice; CD-HFD + α -PD-1 n= 41 mice; CD-HFD + α -PD-L1 n= 6 mice; CD-HFD + α -CD8 n=
24 mice; CD-HFD + α -CD8/NK1.1 n= 6 mice; CD-HFD + α -PD-1/ α -CD8 n= 9 mice; CD-HFD + α -TNF n= 10 mice;
CD-HFD + α -PD-1/ α -TNF n= 11 mice; CD-HFD + α -CD4 n= 9 mice; CD-HFD + α -PD-1/ α -CD4 n= 9 mice). **(e)** Data
gathered from hepatic tissue analyses was binary correlated with each other of 6- or 12-months ND, CD-HFD or
CD-HFD-fed mice + 8 weeks treatment by α -CD8, α -CD8/ α -NK1.1; α -PD-1, α -PD-1/ α -CD8, α -TNF, α -PD-1/ α -TNF,
α -CD4, or α -PD-1/ α -CD4 antibodies (ND n= 47 mice; CD-HFD n= 72 mice; CD-HFD + α -PD-1 n= 41 mice; CD-HFD
+ α -PD-L1 n= 6 mice; CD-HFD + α -CD8 n= 29 mice; CD-HFD + α -CD8/NK1.1 n= 6 mice; CD-HFD + α -PD-1/ α -CD8
n= 9 mice; CD-HFD + α -TNF n= 10 mice; CD-HFD + α -PD-1/ α -TNF n= 11 mice; CD-HFD + α -CD4 n= 9 mice; CD-
HFD + α -PD-1/ α -CD4 n= 9 mice).

Research for a Life without Cancer

Rebuttal Figure 8: T-cell activation and hepatic abundance correlate with NASH pathology

(a) Umap showing the expression intensity of the indicated marker of scholastically selected CD45⁺ cells define distinct marker expression of 12 months ND or CD-HFD-fed mice (ND n = 4 mice; CD-HFD n = 8 mice). (b) Heatmap showing the median marker expression of the defined CD45⁺ subsets displayed in (a) by flow cytometry of 12 months ND or CD-HFD-fed mice (ND n = 4 mice; CD-HFD n = 8 mice). (c) Quantification of hepatic CD8⁺ cells and PD-1⁺ expressing cells by immunohistochemistry of 12 months ND, CD-HFD or WD-HTF-fed mice (PD-1: n = 5

Research for a Life without Cancer

mice/group; CD8: ND n= 6 mice; CD-HFD n= 6 mice; WD-HTF n= 5 mice). **(d)** Immunofluorescence staining of
single channel-staining PD-1, CD8 and CD4 of 12 months ND or CD-HFD-fed mice (n= 3 mice/group). Arrowheads
indicate CD8⁺ (red), PD-1⁺ (green) or CD4⁺ (ocher) cells. Scale bar: 100 μ m. **(e)** Quantification of abundance, **(f)**
PD-1 expression and flow cytometry plots of hepatic CD8⁺ T-cells by flow cytometry of 6 or 12 months ND or CD-
HFD-fed mice (abundance of CD8: 6 months: ND n= 17 mice; CD-HFD n= 10 mice; WD-HTF n= 7 mice; 12 months:
ND n= 11 mice; CD-HFD n= 6 mice; WD-HTF n= 5 mice; PD-1 expression in CD8⁺ T-cells: 6 months: ND n= 15
mice; CD-HFD n= 14 mice; WD-HTF n= 7 mice; 12 months: ND n= 10 mice; CD-HFD n= 6 mice; WD-HTF n= 5
mice). **(g)** Quantification of abundance, **(h)** PD-1 expression and flow cytometry plots of hepatic CD4⁺ T-cells by
flow cytometry of 6 or 12 months ND or CD-HFD-fed mice (abundance of CD4: 6 months: ND n= 17 mice; CD-HFD
n= 10 mice; WD-HTF n= 7 mice; 12 months: ND n= 11 mice; CD-HFD n= 6 mice; WD-HTF n= 5 mice; PD-1
expression in CD4⁺ T-cells: 6 months: ND n= 15 mice; CD-HFD n= 14 mice; WD-HTF n= 7 mice; 12 months: ND
n= 10 mice; CD-HFD n= 6 mice; WD-HTF n= 5 mice). **(i)** H&E, CD8 and PD-1 staining, evaluation by NAS and
quantification of CD8⁺ cells and PD-1⁺ expressing cells by immunohistochemistry of 32-weeks old hURI-tetOFFhep
and non-transgenic litter control mice (n=6 mice/group). Arrowheads indicate specific staining positive cells. Scale
1872 bar: 100 μ m. **(j)** Hepatic abundance of TCR $\gamma\delta$ T-cells of 6 or 12 months ND or CD-HFD-fed mice (6 months ND n=
8 mice; CD-HFD n= 6 mice; 12 months ND n= 8 mice; CD-HFD n= 6 mice). **(k)** Quantification of hepatic PD-L1⁺
expression by mRNA *in situ* hybridization of 6- or 12-months ND or CD-HFD-fed mice (6 months: ND n= 13 mice;
CD-HFD n= 11 mice; 12 months: ND n= 7 mice; CD-HFD n= 7 mice). Scale bar: 100 μ m. **(l)** Quantification of hepatic
PD-L1⁺ expression by immunohistochemistry of 12 months ND or CD-HFD-fed mice (6 months: ND n= 4 mice; CD-
HFD n= 8 mice). Scale bar: 100 μ m.

Research for a Life without Cancer

Rebuttal Figure 9: Hepatic resident-like CD8⁺PD-1⁺ T-cells are increased in livers of non-alcoholic fatty liver disease (NAFLD) patients

(a) Flow cytometry plots, quantification of patient-liver-derived PD-1⁺CD8⁺ T-cells, and (b) correlation of PD-1⁺CD8⁺ T-cells with BMI, NAS and ALT of healthy or NAFLD/NASH patients (**Supplementary Table 1**: healthy n= 8 patients; NAFLD/NASH n= 16 patients). Fluorescence-minus-one (FMO). (c) UMAP representation showing the FlowSOM-guided clustering of CD45⁺ cells and (d) flow cytometry plots and quantification of CD8⁺PD-1⁺CD103⁺ derived from hepatic biopsies of control, or NAFLD/NASH patients (**Supplementary Table 2**: control n= 6 patients; NAFLD/NASH n= 11 patients) Populations: CD8⁺ (violet), CD8⁺PD-1⁺CD103⁺ (red). (e) UMAP representation of CD3⁺ cells and analyses of differential gene expression by scRNA-seq of control, or NAFLD/NASH patients (control

Research for a Life without Cancer

n= 4 patients; NAFLD/NASH n= 7 patients). **(f)** Correlation of significant differentially expressed genes in liver-
derived CD8⁺PD-1⁺ compared to CD8⁺PD-1⁻ T-cells subsets of 12 months CD-HFD-fed mice and NAFLD/NASH
patients (mouse: n= 3 mice; human: n= 3 patients). **(g)** Velocity analyses of scRNA-seq data showing **(h)**
expression, transcriptional activity, **(i)** gene expression and **(j)** correlation of expression along the latent-time of
selected genes along the latent-time of patient-liver-derived CD8⁺ T-cells of control, or NAFLD/NASH patients in
comparison to mouse-liver-derived CD8⁺ T-cells (patients: NAFLD/NASH n= 3 patients; mouse: n= 3 mice/group).
Root cells: yellow cells indicate root cells, blue cells indicate cells farthest away from the root by RNA velocity. End
points: yellow cells indicate end point cells, blue cells indicate cells farthest away from defined end point cells by
RNA velocity. Latent time: pseudo-time by RNA velocity, dark color indicate start of RNA velocity, yellow color
indicate end point of latent time. RNA velocity flow: Blue cluster defined as start point, orange cluster as
intermediate, green cluster as end point. Arrows indicate the trajectory of cells.

Research for a Life without Cancer

Rebuttal Figure 10: An inflammatory cellular polarization of T-cells can be found in liver biopsies of NAFLD/NASH patients
(a) Flow cytometry plot of FMO control, **(b)** quantification of patient-liver-derived PD-1⁺CD8⁺ T-cells, and **(c)** quantification of CD4, CD8, $\gamma\delta$, NK and NKT cells healthy or NAFLD/NASH patients (**Supplementary Table 1**: healthy n= 8 patients; NAFLD/NASH n= 16 patients). **(d)** Umap showing the expression intensity of the indicated marker on scholastically selected CD45⁺ cells and **(e)** Heatmap showing the median marker expression of the defined CD45+ subsets of figure 5c by flow cytometry derived from hepatic biopsies of control and NAFLD/NASH

Research for a Life without Cancer

patients to define distinct marker expression (**Supplementary Table 2**: control n= 6 patients; NAFLD/NASH n= 11
patients). **(f)** Definition of cellular subsets, **(g)** relative quantification of defined cellular subsets of randomly chosen
CD45⁺ cells, **(h)** polarization of CD8⁺ T-cells and **(i)** quantification of CD4⁺CD27⁺, or $\gamma\delta$ TCR⁺Eomes⁺ T-cells by
flow cytometry derived from hepatic biopsies of healthy and NAFLD/NASH patients (**Supplementary Table 2**:
control n= 6 patients; NAFLD/NASH n= 11 patients).

Research for a Life without Cancer

Rebuttal Figure 11: CyTOF analyses of T-cells from liver biopsies of NAFLD/NASH patients reveals co-expression of PD-1 and CD103 in CD8⁺ T-cells

(a) tSNE representation, (b) marker expression, (c) average marker expression of defined T-cell subsets of patient-liver-derived T-cells analyzed by CyTOF of control and NAFLD/NASH patients (control n= 11 patients pooled in 3 analyses; NAFLD/NASH n= 16 patients pooled in 5 analyses). (d) Composition, (e) HSNE representation of defined T-cell subsets and (f) quantification of CD8⁺CD103⁺PD-1⁺ cells of patient-liver-derived T-cells analyzed by CyTOF of control and NAFLD/NASH patients (control n= 11 patients pooled in 3 analyses; NAFLD/NASH n= 16 patients pooled in 5 analyses).

Research for a Life without Cancer

Rebuttal Figure 12: Single cell RNA-sequencing of T-cells found in patient liver biopsies of NAFLD/NASH corroborate mouse gene expression inflammatory

(a) NAS and BMI of patients used for scRNA-seq analyses of patient-liver-derived T-cells of control and NAFLD/NASH patients (control n = 4 patients; NAFLD/NASH n = 7 patients). (b) UMAP representation, marker expression, (c) relative quantification and (d), (e), (f) polarization of defined T-cell subsets of defined T-cell subsets of patient-liver-derived T-cells by scRNA-seq of control and NAFLD/NASH patients (control n = 4 patients;

Research for a Life without Cancer

1927 NAFLD/NASH n= 7 patients). **(g)** Differential gene expression of CD4+PD-1+ vs CD4+ T-cells and **(h)** selected
1928 average marker expression in CD4+ and CD8+ T-cell subsets of by scRNA-seq of control and NAFLD/NASH patients
1929 (control n= 4 patients; NAFLD/NASH n= 7 patients).

Rebuttal Figure 13: α -PD-L1 treatment does not achieve anti-tumor effects in NASH-induced tumors

(a) MRI pictures of liver of mice after 13 months CD-HFD followed by 7 weeks treatment to CD-HFD or CD-HFD-fed mice + 7 weeks by α -PD-L1 antibodies (CD-HFD n= 6 mice; CD-HFD + α -PD-L1 n= 8 mice). Lines indicate tumor nodule. Scale bar: 10 mm. (b) Macroscopy of liver of ND, CD-HFD or CD-HFD-fed mice + 8 weeks treatment by α -PD-L1 antibodies. Arrowheads indicate tumor/lesions. Scale bar: 10 mm. (c) Body weight, ALT levels ND, CD-HFD or CD-HFD-fed mice + 8 weeks treatment by α -PD-L1 antibodies (Body weight, ALT, : ND n= 8 mice; CD-HFD n= 6 mice; CD-HFD + α -PD-L1 n= 6 mice) (d) and (e) NAS evaluation by H&E, fibrosis quantification (Sirius Red), quantification of CD8, PD-1 and PD-L1 staining of hepatic tissue by immunohistochemistry of 12 months ND, CD-HFD or CD-HFD-fed mice + 8 weeks treatment by α -PD-L1 antibodies (NAS: ND n= 7 mice; CD-HFD n= 6 mice; CD-HFD + α -PD-L1 n= 6 mice; Sirius Red: ND n= 7 mice; CD-HFD n= 5 mice; CD-HFD + α -PD-L1 n= 6 mice; CD8, : ND n= 5 mice; CD-HFD n= 5 mice; CD-HFD + α -PD-L1 n= 5 mice; PD-1, PD-L1: ND n= 5 mice; CD-HFD n= 5 mice; CD-HFD + α -PD-L1 n= 6 mice). Scale bar: 100 μ m. (f) Tumor/Lesion incidence in CD-HFD or CD-HFD-fed mice + 8 weeks treatment by α -PD-L1 antibodies (CD-HFD n= 19 tumors/lesions in 25 mice; CD-HFD + α -PD-L1 n= 7 tumors/lesions in 8 mice). Arrowheads indicate specific staining positive cells.

Research for a Life without Cancer

Rebuttal Figure 14: Figure 2: Anti-PD-1 treatment drives hepatocarcinogenesis in a CD8-dependent manner in NASH
(a) Histological staining of hepatic tissue by H&E, Sirius Red, PD-1 and CD8 of 12 months ND, CD-HFD or CD-HFD-fed mice + 8 weeks treatment by α-CD8 antibodies (H&E: ND n= 24 mice; CD-HFD n= 40 mice; CD-HFD + α-CD8 n= 29 mice; Sirius Red: ND n= 19 mice; CD-HFD n= 31 mice; CD-HFD + α-CD8 n= 24 mice; PD-1: n= 5 mice/group; CD8: ND n= 6 mice; CD-HFD n= 6 mice; CD-HFD + α-CD8 n= 5 mice). Arrowheads indicate CD8⁺ or PD-1⁺ cells. Scale bar: 50 μm. (b) ALT levels of 12 months ND, CD-HFD or CD-HFD-fed mice + 8 weeks treatment by α-CD8 antibodies (ND n= 22 mice; CD-HFD n= 42 mice; CD-HFD + α-CD8 n= 31 mice). (c) NAS evaluation by H&E of 12 months ND, CD-HFD or CD-HFD-fed mice + 8 weeks treatment by α-CD8 antibodies (ND n= 24 mice;

Research for a Life without Cancer

CD-HFD n= 40 mice; CD-HFD + α -CD8 n= 29 mice). **(d)** Fibrosis quantification (Sirius Red) of 12 months ND, CD-
 HFD or CD-HFD-fed mice + 8 weeks treatment by α -CD8 antibodies (ND n= 19 mice; CD-HFD n= 53 mice; CD-
 HFD + α -CD8 n= 27 mice) **(e)** Flow cytometry analysis for polarization of hepatic CD8⁺ T-cells of 12 months CD-
 HFD or CD-HFD-fed mice + 8 weeks treatment by α -PD-1 antibodies (CD-HFD n= 12 mice; CD-HFD + α -CD8 n=
 17 mice). **(f)** Flow cytometry plots of 12 months ND, CD-HFD or CD-HFD + 8 weeks treatment by α -CD8 antibodies.
 **(g)** Quantification of PD-1⁺ cells of hepatic tissue by immunohistochemistry of 12 months ND, CD-HFD or CD-HFD
 + 8 weeks treatment by α -CD8 antibodies (n= 5 mice/group). **(h)** Histological staining of hepatic tissue by H&E,
 Sirius Red, PD-1 and CD8 of 12 months ND, CD-HFD or CD-HFD-fed mice + 8 weeks treatment by α -PD-1
 antibodies (H&E: ND n= 24 mice; CD-HFD n= 40 mice; CD-HFD + α -PD-1 n= 36 mice; Sirius Red: ND n= 19 mice;
 CD-HFD n= 31 mice; CD-HFD + α -PD-1 n= 27 mice; PD-1: ND n= 5 mice; CD-HFD n= 5 mice; CD-HFD + α -PD-1
 n= 7 mice). Arrowheads indicate PD-1⁺ cells. Scale bar: 50 μ m. **(i)** ALT and **(j)** AST levels of 12 months ND, CD-
 HFD or CD-HFD-fed mice + 8 weeks treatment by α -PD-1 antibodies (ALT: ND n= 22 mice; CD-HFD n= 42 mice;
 CD-HFD + α -PD-1 n= 30 mice). **(k)** NAS evaluation by H&E of 12 months ND, CD-HFD or CD-HFD + 8 weeks
 treatment by α -PD-1 antibodies (ND n= 24 mice; CD-HFD n= 40 mice; CD-HFD + α -PD-1 n= 36 mice). **(l)**
 Quantification of PD-1⁺ cells of hepatic tissue by immunohistochemistry of 12 months ND, CD-HFD or CD-HFD-fed
 mice + 8 weeks treatment by α -PD-1 antibodies (ND n= 5 mice; CD-HFD n= 5 mice; CD-HFD + α -PD-1 n= 7 mice).
 **(m)** Macroscopy of liver of 12 months CD-HFD or CD-HFD-fed mice + 8 weeks treatment by α -PD-1 antibodies.
 Arrowheads indicate tumor/lesions. Scale bar: 10 mm. **(n)** Fibrosis quantification (Sirius Red) of 12 months ND,
 CD-HFD or CD-HFD-fed mice + 8 weeks treatment by α -PD-1 antibodies (ND n= 19 mice; CD-HFD n= 53 mice;
 CD-HFD + α -PD-1 n= 33 mice). **(o)** Quantification of tumor/lesion size and **(p)** tumor load of 12 months CD-HFD or
 CD-HFD-fed mice + 8 weeks treatment by α -PD-1 antibodies (tumor/lesion size, tumor load: CD-HFD n= 19 mice;
 CD-HFD + α -PD-1 n= 29 mice). **(q)** Quantification of tumor incidence of 12 months CD-HFD or CD-HFD-fed mice
 + 8 weeks treatment by α -CD8, co-depletion of α -CD8/NK1, or α -PD-1 antibodies (tumor incidence: CD-HFD n= 32
 tumors/lesions in 87 mice; CD-HFD + α -CD8 n= 2 tumors/lesions in 31 mice; CD-HFD + α -CD8/NK1.1 n= 0
 tumors/lesions in 6 mice; CD-HFD + α -PD-1 n= 33 tumors/lesions in 44 mice).

Research for a Life without Cancer

Rebuttal Figure 15: CD8 T-cell depletion in NASH does not induce compensatory immunological reactions

(a) Body weight of 12 months ND, CD-HFD or CD-HFD-fed mice + 8 weeks treatment by α -CD8 antibodies (ND n= 15 mice; CD-HFD n= 28 mice; CD-HFD + α -CD8 n= 28 mice). (b) Assessment of metabolic tolerance by intra peritoneal glucose tolerance test of 12 months CD-HFD or CD-HFD-fed mice + 8 weeks treatment by α -CD8 antibodies (CD-HFD n= 8 mice; CD-HFD + α -CD8 n= 10 mice). (c) Quantification of CD8 staining of hepatic tissue by immunohistochemistry of 12 months ND, CD-HFD or CD-HFD-fed mice + 8 weeks treatment by α -CD8 antibodies (ND n= 6 mice; CD-HFD n= 6 mice; CD-HFD + α -CD8 n= 5 mice). (d) Absolute and (e) relative quantification of hepatic leukocytes of 12 months CD-HFD or CD-HFD-fed mice + 8 weeks treatment by α -CD8

Research for a Life without Cancer

antibodies (CD-HFD n= 9 mice; CD-HFD + α -CD8 n= 12 mice). **(f)** Analyses of cytokine expression for polarization
 of hepatic CD8⁺ T-cells of 12 months CD-HFD or CD-HFD-fed mice + 8 weeks treatment by α -CD8 antibodies
 (GzmB, IFN γ , TNF: CD-HFD n= 13 mice; α -CD8 + CD-HFD n= 17 mice; IL-10: CD-HFD n= 7 mice; α -CD8 + CD-
 HFD n= 9 mice). **(g)** Expression of PD-1 of hepatic CD4⁺ and CD8⁺ T-cells by flow cytometry of 12 months CD-HFD
 or CD-HFD-fed mice + 8 weeks treatment by α -CD8 antibodies (CD-HFD n= 11 mice; α -CD8 + CD-HFD n= 17
 mice). **(h)** Flow cytometry analysis for polarization of hepatic myeloid cells of 12 months CD-HFD or CD-HFD-fed
 mice + 8 weeks treatment by α -CD8 antibodies (CD-HFD n= 8 mice; α -CD8 + CD-HFD n= 12 mice). **(i)** Flow
 cytometric analysis for polarization of hepatic CD4⁺ T-cells of 12 months CD-HFD or CD-HFD-fed mice + 8 weeks
 treatment by α -CD8 antibodies (CD-HFD n= 12 mice; α -CD8 + CD-HFD n= 17 mice). **(j)** Cytokine expression of
 hepatic CD4⁺ T-cells of 12 months CD-HFD or CD-HFD-fed mice + 8 weeks treatment by α -CD8 antibodies (GzmB,
 IFN γ , TNF: CD-HFD n= 13 mice; CD-HFD + α -CD8 n= 17 mice; IL-10, Foxp3: CD-HFD n= 7 mice; CD-HFD + α -
 CD8 n= 9 mice). **(k)** Cytokine expression for polarization of hepatic NK and NKT-cells of 12 months CD-HFD or
 CD-HFD-fed mice + 8 weeks treatment by α -CD8 antibodies (CD-HFD n= 4 mice; α -CD8 + CD-HFD n= 5 mice). **(l)**
 Gene set enrichment analysis of RNA sequencing data of hepatic tissue comparing CD-HFD with CD-HFD-fed mice
 + α -CD8 of 12 months ND, CD-HFD or CD-HFD-fed mice + 8 weeks treatment by α -CD8 antibodies (n= 5
 mice/group).

Research for a Life without Cancer

Rebuttal Figure 16: Resident-like CD8⁺PD-1⁺ T-cells drive hepatocarcinogenesis in a TNF-dependent manner upon anti-PD-1 treatment in NASH

(a) ScRNA- seq analysis of hepatic TCRβ⁺ cells of 12 months CD-HFD + IgG or CD-HFD-fed mice + 8 weeks treatment by α-PD-1 or α-CD8 antibodies (n= 3 mice/group). (b) Selected marker expression in hepatic CD8⁺ T-cells by scRNA-seq comparing CD8⁺ with CD8⁺PD-1⁺ T-cells of 12 months CD-HFD + IgG or CD-HFD-fed mice + 8 weeks treatment by α-PD-1 antibodies (n= 3 mice/group). (c) Average UMI comparison of hepatic CD8⁺PD-1⁺ T-cells of 12 months CD-HFD + IgG or CD-HFD-fed mice + 8 weeks treatment by α-PD-1 antibodies (n= 3 mice/group). (d) RNA velocity analyses of scRNA-seq data showing expression and (e) correlation of expression along the latent-time of selected genes along the latent-time (n= 3 mice/group). Root cells: yellow cells indicate root cells, blue cells indicate cells farthest away from root by RNA velocity. End points: yellow cells indicate end point cells, blue cells indicate cells farthest away from defined end point cells by RNA velocity. Latent time: pseudo-time by RNA velocity, dark color indicate start of RNA velocity, yellow color indicate end point of latent time. RNA velocity flow: Blue cluster defined as start point, orange cluster as intermediate, green cluster as end point. Arrows indicate trajectory of cells.

Research for a Life without Cancer

**(f)** PCA plot of hepatic CD8⁺ or CD8⁺PD-1⁺ T-cells sorted TCRβ⁺ cells by mass spectrometry of 12 months ND, CD-
 HFD or CD-HFD-fed mice + 8 weeks treatment by α-PD-1 antibodies (CD8⁺: ND n= 6 mice, CD-HFD + IgG n= 5
 mice; CD-HFD + α-PD-1 n= 6 mice; CD8⁺PD-1⁺: ND n= 4 mice, CD-HFD + IgG n= 6 mice; CD-HFD + α-PD-1 n= 6
 mice). **(g)** UMAP representation showing the FlowSOM-guided clustering, heatmap showing the median marker
 expression, and **(h)** quantification of hepatic CD8⁺ T-cells of 12 months ND, CD-HFD + IgG or CD-HFD-fed mice +
 8 weeks treatment by α-PD-1 antibodies (ND n= 4 mice; CD-HFD + IgG n= 8 mice; CD-HFD + α-PD-1 n= 6 mice).
 **(i)** Quantification of CellCNN analyzed flow cytometry data of hepatic CD8⁺ T-cells of 12 months CD-HFD + IgG or
 CD-HFD-fed mice + 8 weeks treatment by α-PD-1 antibodies (CD-HFD + IgG n= 6 mice; CD-HFD + α-PD-1 n= 4
 mice). **(j)** UMAP representation showing the FlowSOM-guided clustering, the expression intensity of the indicated
 marker and heatmap showing the median marker expression of flow cytometry data of hepatic CD8⁺PD-1⁺ T-cells
 of 12 months ND, CD-HFD or CD-HFD-fed mice + 8 weeks treatment by α-PD-1 antibodies (ND n= 6 mice; CD-
 HFD n= 5 mice; CD-HFD + α-PD-1 n= 6 mice). **(k)** ALT and **(l)** NAS evaluation of 12 months ND, CD-HFD, CD-
 HFD-fed mice + 8 weeks treatment by α-PD-1, α-PD-1/α-CD8, α-TNF, or α-PD-1/α-TNF antibodies (ND n= 30 mice;
 CD-HFD n= 47 mice; CD-HFD + α-PD-1 n= 35 mice; CD-HFD + α-PD-1/α-CD8 n= 9 mice; CD-HFD + α-TNF n= 10
 mice; CD-HFD + α-PD-1/α-TNF n= 11 mice). **(m)** Quantification of hepatic CD8⁺PD-1⁺CXCR6⁺ T-cells ND, CD-
 HFD, CD-HFD-fed mice + 8 weeks treatment by α-PD-1, α-PD-1/α-CD8, α-TNF, α-PD-1/α-TNF, α-CD4, or α-PD-
 1/α-CD4 antibodies (ND n= 30 mice; CD-HFD n= 47 mice; CD-HFD + α-PD-1 n= 35 mice; CD-HFD + α-PD-1/α-
 CD8 n= 9 mice; CD-HFD + α-TNF n= 10 mice; CD-HFD + α-PD-1/α-TNF n= 11 mice); CD-HFD + α-CD4 n= 8 mice;
 CD-HFD + α-PD-1/α-CD4 n= 8 mice). **(n)** Quantification of tumor incidence of 12 months CD-HFD or CD-HFD-fed
 mice + 8 weeks treatment by α-CD8, α-CD8/NK1.1, α-PD-1, α-PD-1/α-CD8, α-TNF, α-PD-1/α-TNF, α-CD4, or α-
 PD-1/α-CD4 antibodies (tumor incidence: CD-HFD n= 32 tumors/lesions in 87 mice; CD-HFD + α-CD8 n= 2
 tumors/lesions in 31 mice; CD-HFD + α-CD8/NK1.1 n= 0 tumors/lesions in 6 mice; CD-HFD + α-PD-1 n= 33
 tumors/lesions in 44 mice; CD-HFD + α-PD-1/α-CD8 n= 2 tumors/lesions in 9 mice; CD-HFD + α-TNF n= 3
 tumors/lesions in 10 mice; CD-HFD + α-PD-1/α-TNF n= 3 tumors/lesions in 11 mice); CD-HFD + α-CD4 n= 3
 tumors/lesions in 9 mice; CD-HFD + α-PD-1/α-CD4 n= 8 tumors/lesions in 9 mice).

Research for a Life without Cancer

Rebuttal Figure 17: α -CD8/NK1.1 co-depletion does not further ameliorate NASH pathology compared to CD8 T-cell depletion alone

(a) H&E and Sirius Red staining, (b) body weight, (c) NASH evaluation by H&E, (d) fibrosis quantification (Sirius Red) and (e) ALT levels of 12 months ND, CD-HFD, CD-HFD-fed mice + 8 weeks treatment by α -CD8 or CD-HFD-fed mice + 8 weeks co-depletion of α -CD8/NK1.1 antibodies (body weight: ND n = 15 mice; CD-HFD n = 28 mice; CD-HFD + α -CD8 n = 28 mice; fibrosis ND n = 19 mice; CD-HFD n = 53 mice; CD-HFD + α -CD8 n = 27 mice; CD-HFD + α -CD8/NK1.1 n = 6 mice; NASH: ND n = 24 mice; CD-HFD n = 40 mice; CD-HFD + α -CD8 n = 29 mice; CD-HFD + α -CD8/NK1.1 n = 6); ALT: ND n = 22 mice; CD-HFD n = 42 mice; CD-HFD + α -CD8 n = 31 mice; CD-HFD + α -CD8/NK1.1 n = 6). Scale bar: 100 μ m. (f) Flow cytometry plots and (g) quantification of hepatic NK1.1 abundance of 12 months ND, CD-HFD, CD-HFD-fed mice + 8 weeks treatment by α -CD8 or CD-HFD-fed mice + 8 weeks co-depletion of α -CD8/NK1.1 antibodies (ND n = 4 mice; CD-HFD n = 8 mice; CD-HFD + α -CD8 n = 7 mice; CD-HFD + α -CD8/NK1.1 n = 6 mice). (h) Gene set enrichment analysis of RNA sequencing data of hepatic tissue comparing CD-HFD with CD-HFD-fed mice + co-depletion of α -CD8/NK1.1 of 12 months ND, CD-HFD or CD-HFD-fed mice + co-depletion of α -CD8/NK1.1 antibodies (n = 5 mice/group). (i) Gene set enrichment analysis of RNA sequencing data of hepatic tissue comparing or CD-HFD-fed mice + 8 weeks treatment by α -CD8 with CD-HFD-fed mice + co-depletion of α -CD8/NK1.1 of 12 months ND, CD-HFD, CD-HFD-fed mice + 8 weeks treatment by α -CD8 or CD-HFD-fed mice + co-depletion of α -CD8/NK1.1 antibodies (n = 5 mice/group).

Research for a Life without Cancer

**Rebuttal Figure 18: CD8⁺ T-cells drive hepatic inflammation and subsequent liver cancer in a TNF-dependent manner upon**
**PD-1-targeted immunotherapy**
**(a)** Body weight, AST, and histological evaluation by **(b)** Sirius red, CD4, CD8, PD-1, PD-L1, F4/80, MHC-II and **(c)**
staining of ND, CD-HFD, or CD-HFD-fed mice + 8 weeks treatment by α -PD-1, α -PD-1/ α -CD8, α -TNF, α -PD-1/ α -
TNF antibodies (body weight: ND n= 16 mice; CD-HFD n= 29 mice; CD-HFD + α -PD-1 n= 23 mice; CD-HFD + α -
PD-1/ α -CD8 n= 9 mice; CD-HFD + α -TNF n= 10 mice; CD-HFD + α -PD-1/ α -TNF n= 11 mice; AST: body weight:
ND n= 30 mice; CD-HFD n= 40 mice; CD-HFD + α -PD-1 n= 30 mice; CD-HFD + α -PD-1/ α -CD8 n= 9 mice; CD-
HFD + α -TNF n= 10 mice; CD-HFD + α -PD-1/ α -TNF n= 11 mice; Sirius red: ND n= 11 mice; CD-HFD n= 12 mice;

Research for a Life without Cancer

CD-HFD + α -PD-1 n= 12 mice; CD-HFD + α -PD-1/ α -CD8 n= 9 mice; CD-HFD + α -TNF n= 10 mice; CD-HFD + α -
 PD-1/ α -TNF n= 11 mice; CD4: ND n= 10 mice; CD-HFD n= 11 mice; CD-HFD + α -PD-1 n= 14 mice; CD-HFD + α -
 PD-1/ α -CD8 n= 9 mice; CD-HFD + α -TNF n= 10 mice; CD-HFD + α -PD-1/ α -TNF n= 11 mice; CD8: ND n= 10 mice;
 CD-HFD n= 12 mice; CD-HFD + α -PD-1 n= 14 mice; CD-HFD + α -PD-1 n= 14 mice; CD-HFD + α -PD-1/ α -CD8 n=
 9 mice; CD-HFD + α -TNF n= 10 mice; CD-HFD + α -PD-1/ α -TNF n= 11 mice; PD-1: ND n= 12 mice; CD-HFD n= 12
 mice; CD-HFD + α -PD-1 n= 14 mice; CD-HFD + α -PD-1/ α -CD8 n= 8 mice; CD-HFD + α -TNF n= 10 mice; CD-HFD
 + α -PD-1/ α -TNF n= 10 mice; PD-L1: ND n= 10 mice; CD-HFD n= 11 mice; CD-HFD + α -PD-1 n= 14 mice; CD-HFD
 + α -PD-1/ α -CD8 n= 9 mice; CD-HFD + α -TNF n= 10 mice; CD-HFD + α -PD-1/ α -TNF n= 11 mice; F4/80: ND n= 11
 mice; CD-HFD n= 12 mice; CD-HFD + α -PD-1 n= 14 mice; CD-HFD + α -PD-1 n= 14 mice; CD-HFD + α -PD-1/ α -
 CD8 n= 9 mice; CD-HFD + α -TNF n= 10 mice; CD-HFD + α -PD-1/ α -TNF n= 11 mice; MHC-II: ND n= 11 mice; CD-
 HFD n= 13 mice; CD-HFD + α -PD-1 n= 14 mice; CD-HFD + α -PD-1 n= 14 mice; CD-HFD + α -PD-1/ α -CD8 n= 9
 mice; CD-HFD + α -TNF n= 10 mice; CD-HFD + α -PD-1/ α -TNF n= 11 mice). Scale bar: 100 μ m.

Research for a Life without Cancer

Rebuttal Figure 19: Inflammation associated hepatic cytokine and chemokine environment in CD8⁺ T-cells driven hepatic inflammation upon PD-1-targeted immunotherapy

(a) Quantification of hepatic immune cell composition and (b) CD8⁺PD-1⁺TNF⁺ T-cells by flow cytometry of 12 months ND, CD-HFD, or CD-HFD-fed mice + 8 weeks treatment by α-PD-1, α-PD-1/α-CD8, α-TNF, α-PD-1/α-TNF antibodies (Hepatic immune cell composition: ND n= 8 mice; CD-HFD n= 5 mice; CD-HFD + α-PD-1 n= 4 mice; CD-HFD + α-PD-1/α-CD8 n= 9 mice; CD-HFD + α-TNF n= 10 mice; CD-HFD + α-PD-1/α-TNF n= 11 mice; CD8⁺PD-1⁺TNF⁺: ND n= 8 mice; CD-HFD n= 5 mice; CD-HFD + α-PD-1 n= 3 mice; CD-HFD + α-PD-1/α-CD8 n= 9 mice; CD-HFD + α-TNF n= 10 mice; CD-HFD + α-PD-1/α-TNF n= 11 mice). (c) and (d) multiplex ELISA of hepatic inflammation associated cytokines and (e) chemokines of 12 months ND, CD-HFD or CD-HFD-fed mice + 8 weeks treatment by α-PD-1, α-PD-1/α-CD8, α-TNF, α-PD-1/α-TNF antibodies (ND n= 10 mice; CD-HFD n= 14 mice; CD-HFD + α-PD-1 n= 13 mice; CD-HFD + α-PD-1/α-CD8 n= 9 mice; CD-HFD + α-TNF n= 10 mice; CD-HFD + α-PD-1/α-TNF n= 11 mice).

Research for a Life without Cancer

Rebuttal Figure 20: Depletion of CD4⁺ T-cells does not impair hepatic inflammation in NASH upon PD-1-targeted immunotherapy

(a) Body weight, ALT, AST, NAS, and histological evaluation by **(b)** Sirius Red, CD4, CD8, PD-1, PD-L1, F4/80, MHC-II and **(c)** staining of ND, CD-HFD, or CD-HFD-fed mice + 8 weeks treatment by α -PD-1, α -CD4, α -PD-1/ α -CD4 antibodies (body weight: ND n= 16 mice; CD-HFD n= 29 mice; CD-HFD + α -PD-1 n= 23 mice; CD-HFD + α -CD4 n= 9 mice; CD-HFD + α -PD-1/ α -CD4 n= 9 mice; ALT ND n= 30 mice; CD-HFD n= 47 mice; CD-HFD + α -PD-1 n= 35 mice; CD-HFD + α -CD4 n= 9 mice; CD-HFD + α -PD-1/ α -CD4 n= 9 mice; AST: ND n= 30 mice; CD-HFD n= 40 mice; CD-HFD + α -PD-1 n= 30 mice; CD-HFD + α -CD4 n= 9 mice; CD-HFD + α -PD-1/ α -CD4 n= 9 mice; NAS: ND n= 31 mice; CD-HFD n= 46 mice; CD-HFD + α -PD-1 n= 40 mice; CD-HFD + α -CD4 n= 8 mice; CD-HFD + α -PD-1/ α -CD4 n= 8 mice; Sirius red: ND n= 11 mice; CD-HFD n= 12 mice; CD-HFD + α -PD-1 n= 12 mice; CD-HFD + α -CD4 n= 9 mice; CD-HFD + α -PD-1/ α -CD4 n= 9 mice; CD4: ND n= 10 mice; CD-HFD n= 11 mice; CD-HFD + α -PD-1 n= 14 mice; CD-HFD + α -CD4 n= 10 mice; CD-HFD + α -PD-1/ α -CD4 n= 11 mice; CD8: ND n= 10 mice; CD-HFD n= 12 mice; CD-HFD + α -PD-1 n= 14 mice; CD-HFD + α -CD4 n= 9 mice; CD-HFD + α -PD-1/ α -CD4 n= 9 mice; PD-1: ND n= 13 mice; CD-HFD n= 12 mice; CD-HFD + α -PD-1 n= 14 mice; CD-HFD + α -CD4 n= 9 mice; CD-HFD + α -PD-1/ α -CD4 n= 9 mice; PD-L1: ND n= 12 mice; CD-HFD n= 12 mice; CD-HFD + α -PD-1 n= 14 mice; CD-HFD + α -CD4 n= 9 mice; CD-HFD + α -PD-1/ α -CD4 n= 9 mice; F4/80: ND n= 11 mice; CD-HFD n= 13 mice; CD-HFD + α -PD-1 n= 14 mice; CD-HFD + α -CD4 n= 8 mice; CD-HFD + α -PD-1/ α -CD4 n= 9 mice; MHC-II: ND n= 11 mice; CD-HFD n= 13 mice; CD-HFD + α -PD-1 n= 14 mice; CD-HFD + α -CD4 n= 9 mice; CD-HFD + α -PD-1/ α -CD4 n= 9 mice). Scale bar: 100 μ m.

Research for a Life without Cancer

Rebuttal Figure 21: Inflammation associated hepatic cytokine and chemokine environment in CD4-depleted animals with or without PD-1-targeted immunotherapy

(a) Quantification of hepatic immune cell composition and (b) CD8⁺PD1⁺TNF⁺ T-cells by flow cytometry of 12 months ND, CD-HFD, or CD-HFD-fed mice + 8 weeks treatment by α-PD-1, α-CD4, α-PD-1/α-CD4 antibodies (Hepatic immune cell composition: ND n= 8 mice; CD-HFD n= 5 mice; CD-HFD + α-PD-1 n= 4 mice; CD-HFD + α-CD4 n= 8 mice; CD-HFD + α-PD-1/α-CD4 n= 8 mice; CD8⁺PD1⁺TNF⁺: ND n= 8 mice; CD-HFD n= 5 mice; CD-HFD + α-PD-1 n= 3 mice; CD-HFD + α-CD4 n= 8 mice; CD-HFD + α-PD-1/α-CD4 n= 8 mice). (c) and (d) multiplex ELISA of hepatic inflammation associated cytokines and (e) chemokines of 12 months ND, CD-HFD or CD-HFD-fed mice + 8 weeks treatment by α-PD-1, α-CD4, α-PD-1/α-CD4 antibodies (ND n= 10 mice; CD-HFD n= 14 mice; CD-HFD + α-PD-1 n= 13 mice; CD-HFD + α-CD4 n= 9 mice; CD-HFD + α-PD-1/α-CD4 n= 9 mice).

Research for a Life without Cancer

Rebuttal Figure 22: Anti-PD-1 treatment drives hepatocarcinogenesis by enhancing an inflammatory and pro-tumorigenic liver microenvironment

(a) Histological staining of hepatic tissue by H&E and CD8 of 6 months ND, CD-HFD or PD-1^{-/-} CD-HFD-fed mice (H&E: ND n= 8 mice; PD-1^{-/-} ND n= 5 mice; CD-HFD n= 9 mice; PD-1^{-/-} CD-HFD n= 13 mice; CD8: ND n= 4 mice; CD-HFD n= 5 mice; PD-1^{-/-} CD-HFD n= 7 mice). Arrowheads indicate CD8⁺ cells. Scale bar: 50 μ m. **(b)** Cytokine expression of hepatic CD8⁺ T-cells of 6 months ND, PD-1^{-/-} ND, CD-HFD or PD-1^{-/-} CD-HFD-fed mice (ND n= 4 mice; PD-1^{-/-} ND n= 5 mice; CD-HFD n= 5 mice; PD-1^{-/-} CD-HFD n= 6 mice). **(c)** Tumor/lesion incidence of 6 months CD-HFD or PD-1^{-/-} CD-HFD-fed mice (tumor incidence: CD-HFD n= 6 tumors/lesions in 63 mice; PD-1^{-/-} CD-HFD n= 6 tumors/lesions in 13 mice). **(d)** Immune cancer field and ICF³⁸- patterns of RNA sequencing data of hepatic tissue of 12 months ND, CD-HFD or CD-HFD-fed mice + 8 weeks treatment by α -PD-1 or α -CD8 antibodies (ND, CD-HFD + α -PD-1, CD-HFD + α -CD8 n= 5 mice/group; CD-HFD n= 4 mice) through single-sample Gene Set Enrichment Analysis (ssGSEA). **(e)** Quantification of mRNA *in situ* hybridization for hepatic TNF⁺ cells of 12 months ND, CD-HFD or CD-HFD-fed mice + 8 weeks treatment by α -CD8 or α -PD-1 antibodies (ND n= 25 fields of view (FOV) in 3 mice; CD-HFD n= 27 FOV in 3 mice; CD-HFD + α -PD-1 n= 40 FOV in 3 mice; CD-HFD + α -CD8 n= 55 FOV in 3 mice). Arrowheads indicate TNF⁺ cells. Scale bar: 20 μ m. **(f)** Histological staining of liver tumor tissue by p62 of 12 months ND, CD-HFD or CD-HFD + 8 weeks treatment by α -PD-1 antibodies or CD-HFD-fed mice + 8 weeks treatment by α -CD8 antibodies (n= 5 mice/group). **(g)** Immunofluorescence staining for Collagen IV, CD8 and Cleaved Caspase 3 of liver tissue of 12 months ND, CD-HFD + IgG or CD-HFD-fed mice + 8 weeks treatment by α -PD-1 antibodies (n= 27 FOV in 3 mice/group). Scale bar: 30 μ m.

Research for a Life without Cancer

Rebuttal Figure 23: PD-1^{-/-} mice fed NASH-inducing diet have an increased inflammatory liver environment

(a) Body weight of 6 months ND, PD-1^{-/-} ND, CD-HFD or PD-1^{-/-} CD-HFD-fed mice (ND n= 5 mice; PD-1^{-/-} ND n= 3 mice; CD-HFD n= 5 mice; PD-1^{-/-} CD-HFD n= 10 mice). (b) ALT levels of ND, PD-1^{-/-} ND, CD-HFD or PD-1^{-/-} CD-HFD-fed mice (ND n= 9 mice; PD-1^{-/-} ND n= 5 mice; CD-HFD n= 9 mice; PD-1^{-/-} CD-HFD n= 10 mice). (c) NASH evaluation by H&E of ND, PD-1^{-/-} ND, CD-HFD or PD-1^{-/-} CD-HFD-fed mice (ND n= 8 mice; PD-1^{-/-} ND n= 5 mice; CD-HFD n= 9 mice; PD-1^{-/-} CD-HFD n= 13 mice). (d) Quantification of CD8⁺ cells in hepatic tissue by immunohistochemistry of 6 months ND, PD-1^{-/-} ND, CD-HFD or PD-1^{-/-} CD-HFD-fed mice (ND n= 4 mice; PD-1^{-/-} ND n= 5 mice; CD-HFD n= 5 mice; PD-1^{-/-} CD-HFD n= 7 mice). (e) – (g) Characterization of hepatic T-cells by flow cytometry of 6 months ND, PD-1^{-/-} ND, CD-HFD or PD-1^{-/-} CD-HFD-fed mice (ND n= 4 mice; PD-1^{-/-} ND n= 5 mice; CD-HFD n= 5 mice; PD-1^{-/-} CD-HFD n= 6 mice). (h) Relative quantification of hepatic leukocytes of 6 months CD-HFD or PD-1^{-/-} CD-HFD-fed mice (ND n= 4 mice; PD-1^{-/-} ND n= 5 mice; CD-HFD n= 5 mice; PD-1^{-/-} CD-HFD n= 6 mice). (i) Histological staining of hepatic tissue by H&E of CD-HFD or PD-1^{-/-} CD-HFD-fed mice (ND n= 8 mice; CD-HFD n= 9 mice; PD-1^{-/-} CD-HFD n= 13 mice). Dotted line indicates tumor/lesion border. Scale bar: 100 μm.

Research for a Life without Cancer

Rebuttal Figure 24: In depth characterization of hepatic immune cell compartment focusing on T-cells

(a) Marker expression of CD4⁺ and CD8⁺ sorted TCRβ⁺ cells defining T-cell subsets by single cell RNA-sequencing of 12 months ND or CD-HFD-fed mice (n = 3 mice/group). (b) Relative frequency of CD4⁺ and CD8⁺ sorted TCRβ⁺ cells by single cell RNA-sequencing of 12 months ND or CD-HFD-fed mice (n = 3 mice/group). (c) Selected marker expression in CD4⁺ T-cells sorted TCRβ⁺ cells by single cell RNA-sequencing of 12 months ND or CD-HFD-fed mice (n = 3 mice/group). (d) Selected average marker expression in T-cell subsets of CD4⁺ and CD8⁺ sorted TCRβ⁺ by scRNA-seq of 12 months ND or CD-HFD-fed mice (n = 3 mice/group). (e) Selected marker expression in hepatic CD8⁺PD-1⁺ T-cells by mass spectrometry of 12 months ND or CD-HFD-fed mice (ND n = 4 mice, CD-HFD n = 6 mice). (f) Gene set enrichment analysis of hepatic CD8⁺PD-1⁺ T-cells sorted TCRβ⁺ cells by mass spectrometry of 12 months ND or CD-HFD-fed mice (ND n = 4 mice, CD-HFD n = 6 mice).

Research for a Life without Cancer

Rebuttal Figure 25: Cellular drivers of hepatic necroinflammation- and increased hepatocarcinogenesis upon α -PD-1 treatment in NASH

(a) Analysis of hepatic TCR β^+ cells by single cell RNA-sequencing of 12 months CD-HFD-fed mice + 8 weeks treatment by α -CD8 antibodies (n= 3 mice). (b) Velocity analyses on scRNA-seq data CD8 $^+$ cells of 12 months ND or CD- HFD-fed mice + 8 weeks treatment by α -PD-1 antibodies (n= 3 mice). (c) Velocity analyses of scRNA-seq data showing correlation of expression along the latent-time of selected genes along the latent-time of ND-fed mice (n= 3 mice). (d) RNA velocity analyses indicating transcriptional activity and gene expression of CD8 $^+$ cells by

Research for a Life without Cancer

scRNA-seq of 12 months ND, CD-HFD or CD- HFD-fed mice + 8 weeks treatment by α -PD-1 antibodies (n= 3
 mice/group). **(e)** Gene set enrichment analysis of mass spectrometry data comparing hepatic CD8⁺PD-1⁺ T-cells
 sorted TCR β ⁺ cells from CD-HFD with CD-HFD-fed mice + α -PD-1. Selected marker expression in hepatic CD8⁺PD-
 1⁺ T-cells sorted TCR β ⁺ cells by mass spectrometry of 12 months CD-HFD or CD-HFD-fed mice + 8 weeks
 treatment by α -PD-1 antibodies (n= 6 mice/group). Candidates developing steady in-/decrease from ND to CD-HFD
 to CD-HFD + 8 weeks treatment by α -PD-1 are indicated in red. (n= 6 mice/group). **(f)** Selected marker expression
 in hepatic CD4⁺ T-cells sorted TCR β ⁺ cells by single cell RNA-sequencing comparing CD4⁺ with CD4⁺PD-1⁺ T-
 cells of 12 months CD-HFD + IgG or CD-HFD-fed mice + 8 weeks treatment by α -PD-1 or α -CD8 antibodies (n= 3
 mice/group). **(g)** Average UMI comparison of hepatic CD4⁺ T-cells of 12 months CD-HFD + IgG or CD-HFD-fed
 mice + 8 weeks treatment by α -PD-1 antibodies (n= 3 mice/group). **(h)** Umap showing the expression intensity of
 the indicated marker on scholastically selected TCR β ⁺ CD8⁺ cells of flow cytometry data to define distinct marker
 expression of 12 months ND, CD-HFD + IgG, CD-HFD-fed mice + 8 weeks treatment by α -PD-1 antibodies (ND n=
 4 mice; CD-HFD n= 8 mice; CD-HFD + α -PD-1 n= 6 mice). **(i)** Quantification of manual gating and flow cytometry
 plots for hepatic CD8⁺PD-1⁺ TNF⁺ abundance of 12 months CD-HFD + IgG, CD-HFD-fed mice + 8 weeks treatment
 by α -PD-1 antibodies (CD-HFD n= 8 mice; CD-HFD + α -PD-1 n= 6 mice). **(j)** CellCNN analyzed flow cytometry data
 of hepatic CD8⁺ T-cells of 12 months CD-HFD + IgG or CD-HFD-fed mice + 8 weeks treatment by α -PD-1 antibodies
 (CD-HFD + IgG n= 6 mice; CD-HFD + α -PD-1 n= 4 mice). **(k)** Immunofluorescence staining for PD-1, CD8 and Ki-
 67 of liver tissue of 12 months ND, CD-HFD + IgG or CD-HFD-fed mice + 8 weeks treatment by α -PD-1 antibodies
 (n= 2 mice/group). Scale bar: 100 μ m. **(l)** In vitro stimulated splenic CD8 T cells from C57Bl/6 mice were treated
 with α -PD-1 antibody for 72 hours (cell count: n= 5 experiments/group; Ki-67: n= 4 experiments/group).

Research for a Life without Cancer

Rebuttal Figure 26: Progression of NASH pathology is associated with increased, and transcriptionally activated hepatic CD8+PD-1+ T-cells

(a) Histological staining of hepatic tissue by H&E of 3, 6 or 12 months ND, CD-HFD or WD-HTF-fed mice (H&E: 3 months: ND n= 5 mice; CD-HFD n= 5 mice; WD-HTF n= 3 mice; 6 months: ND n= 16 mice; CD-HFD n= 8 mice; WD-HTF n= 8 mice; 12 months: ND n= 9 mice; CD-HFD n= 12 mice; WD-HTF n= 6 mice). Scale bar: 100 μ m. (b) Body weight of 3, 6 or 12 months ND, CD-HFD or WD-HTF-fed mice (3 months: ND n= 8 mice; CD-HFD n= 8 mice; WD-HTF n= 3 mice; 6 months: ND n= 14 mice; CD-HFD n= 8 mice; WD-HTF n= 8 mice; 12 months: ND n= 8 mice; CD-HFD n= 8 mice; WD-HTF n= 6 mice). (c) ALT levels of 3, 6 or 12 months ND, CD-HFD or WD-HTF-fed mice (3 months: ND n= 15 mice; CD-HFD n= 46 mice; WD-HTF n= 23 mice; 6 months: ND n= 46 mice; CD-HFD n= 59 mice; WD-HTF n= 21 mice; 12 months: ND n= 25 mice; CD-HFD n= 69 mice; WD-HTF n= 5 mice). (d) NASH evaluation by of 3, 6 or 12 months ND, CD-HFD or WD-HTF-fed mice (3 months: ND n= 5 mice; CD-HFD n= 5 mice; WD-HTF n= 3 mice; 6 months: ND n= 16 mice; CD-HFD n= 8 mice; WD-HTF n= 8 mice; 12 months: ND n=

Research for a Life without Cancer

9 mice; CD-HFD n= 12 mice; WD-HTF n= 6 mice). **(e)** UMAP representation showing the FlowSOM-guided
 clustering of randomly chosen CD45⁺ cells and quantification of hepatic immune cell composition by flow cytometry
 of 12 months ND or CD-HFD-fed mice (ND n= 4 mice; CD-HFD n= 8 mice). **(f)** CD8 and PD-1 staining of hepatic
 tissue by immunohistochemistry of 12 months ND, CD-HFD or WD-HTF-fed mice (PD-1: n= 5 mice/group; CD8:
 ND n= 6 mice; CD-HFD n= 6 mice; WD-HTF n= 5 mice). Scale bar: 100 μ m. **(g)** Immunofluorescence staining of
 PD-1, CD8 and CD4 of 12 months ND or CD-HFD-fed mice (n= 3 mice/group). Arrowheads indicate CD8⁺ (red),
 PD-1⁺ (green) or CD4⁺ (ocher) cells. Scale bar: 100 μ m. **(h)** UMAP representation of 63 parameters (serology, flow
 cytometry, histology) indicating NASH pathology severity measured of 12 months ND or CD-HFD-fed mice (ND n=
 22 mice; CD-HFD n= 31 mice). **(i)** tSNE representation of TCR β ⁺ cells and analyses of **(j)** differential gene
 expression, **(k)** RNA velocity indicating transcriptional activity, gene expression and the trajectory of CD8⁺ cells by
 scRNA-seq of 12 months ND or CD-HFD-fed mice (n= 3 mice/group)⁵³. Root cells: yellow cells indicate root cells,
 blue cells indicate cells farthest away from root by RNA velocity. End points: yellow cells indicate end point cells,
 blue cells indicate cells farthest away from defined end point cells by RNA velocity. Latent time: pseudo-time by
 RNA velocity, dark color indicate start of velocity, yellow color indicate end point of latent time. RNA velocity flow:
 Blue cluster defined as start point, orange cluster as intermediate, green cluster as end point. Arrows indicate the
 trajectory of cells.

Research for a Life without Cancer

Rebuttal Figure 27: α-PD-1 treatment in NASH does increase intrahepatic CD8 T-cells and PD-1 expression, and only leads to minor changes in other T-cell subsets

(a) Body weight of 12 months ND, CD-HFD or CD-HFD-fed mice + 8 weeks treatment by α-PD-1 antibodies (ND n = 15 mice; CD-HFD n = 28 mice; CD-HFD + α-PD-1 n = 26 mice). (b) Assessment of metabolic tolerance by intra peritoneal glucose tolerance test of 12 months CD-HFD or CD-HFD-fed mice + 8 weeks treatment by α-PD-1 antibodies (n = 9 mice/group). (c) Expression of PD-1 of hepatic CD4⁺ and PD-1⁺ T-cells by flow cytometry of 12 months CD-HFD or CD-HFD-fed mice + 8 weeks treatment by α-PD-1 antibodies (CD-HFD n = 10 mice; α-PD-1 + CD-HFD n = 13 mice). (d) Absolute and (e) relative quantification of hepatic leukocytes of 12 months CD-HFD or CD-HFD-fed mice + 8 weeks treatment by α-PD-1 antibodies (CD3: CD-HFD n = 6 mice; CD-HFD + α-PD-1 n = 10 mice; CD4, CD8, CD19, NK, NKT, CD11b⁺, mDC, pDC: CD-HFD n = 10 mice; CD-HFD + α-PD-1 n = 12 mice, KC: CD-HFD n = 6 mice; CD-HFD + α-PD-1 n = 4 mice). (f) Flow cytometric analysis for polarization of hepatic CD8⁺ T-cells of 12 months CD-HFD or CD-HFD-fed mice + 8 weeks treatment by α-PD-1 antibodies (CD-HFD n = 10 mice; α-PD-1 + CD-HFD n = 14 mice). (g) Cytokine expression of hepatic CD4⁺ T-cells of 12 months CD-HFD or CD-HFD-fed mice + 8 weeks treatment by α-PD-1 antibodies (CD-HFD n = 13 mice; CD-HFD + α-PD-1 n = 14 mice). (h) Flow cytometry analysis for polarization of hepatic CD4⁺ T-cells of 12 months CD-HFD or CD-HFD-fed mice + 8 weeks treatment by α-PD-1 antibodies (CD-HFD n = 12 mice; α-PD-1 + CD-HFD n = 17 mice). (i) Cytokine expression of

Research for a Life without Cancer

hepatic CD4⁺ T-cells of 12 months CD-HFD or CD-HFD-fed mice + 8 weeks treatment by α -PD-1 antibodies (GzmB,
IFN γ , TNF: CD-HFD n= 13 mice; CD-HFD + α -PD-1 n= 14 mice; IL-10, Foxp3: CD-HFD n= 7 mice; CD-HFD + α -
PD-1 n= 9 mice). **(j)** Expression of Tim-3 of hepatic CD4⁺ and CD8⁺ T-cells by flow cytometry of 12 months CD-
HFD or CD-HFD-fed mice + 8 weeks treatment by α -PD-1 antibodies (CD-HFD n= 4 mice; α -PD-1 + CD-HFD n= 9
mice). **(k)** Cytokine expression for polarization of hepatic NK and **(l)** NKT-cells of 12 months CD-HFD or CD-HFD-
fed mice + 8 weeks treatment by α -PD-1 antibodies (n= 5 mice/group). **(m)** Flow cytometric analysis for polarization
of hepatic myeloid cells of 12 months CD-HFD or CD-HFD-fed mice + 8 weeks treatment by α -PD-1 antibodies
(CD-HFD n= 8 mice; α -PD-1 + CD-HFD n= 12 mice).

Rebuttal Figure 28: α -PD-1 treatment causes increased inflammation-associated hepatic cytokine and chemokine environment in NASH

(a) and (b) multiplex ELISA concentrations of inflammation-associated hepatic cytokines and **(c)** chemokines of mice submitted to 12 months of ND, CD-HFD, CD-HFD-fed mice + 8 weeks treatment by α -PD-1 antibodies (ND n= 10 mice; CD-HFD n= 14 mice; CD-HFD + α -PD-1 n= 13 mice).

Rebuttal Figure 29: Minor changes in inflammation-associated hepatic cytokine and chemokine environment in NASH under CD8 T-cell depletion or CD8/NK1.1 co-depletion treatment
(a) and (b) multiplex ELISA concentrations of hepatic inflammation-associated cytokines and **(c)** chemokines of 12 months ND, CD-HFD or CD-HFD-fed mice + 8 weeks treatment by α -CD8 or CD-HFD-fed mice + co-depletion of α -CD8/NK1.1 antibodies (ND n= 10 mice; CD-HFD n= 14 mice; CD-HFD + 8 weeks treatment by α -CD8 n= 5 mice; CD-HFD + co-depletion of α -CD8/NK1.1 n= 5 mice).

Research for a Life without Cancer

Rebuttal Figure 30: CD8⁺PD-1⁺ are TOX^{high}, have a resident-like character and are enriched upon α-PD-1 treatment in NASH
(a) Quantification of intracellular Foxo1 and **(b)** calcium levels in CD8⁺ T-cells by flow cytometry of 12 months ND, CD-HFD or CD-HFD-fed mice + 8 weeks treatment by α-PD-1 antibodies (Foxo1: ND n= 6 mice; CD-HFD n= 5 mice; CD-HFD + α-PD-1 n= 7 mice; calcium: ND n= 13 mice; CD-HFD n= 10 mice; CD-HFD + α-PD-1 n= 10 mice). Polarization of CD8⁺PD-1⁺ T-cells **(c)**, as well as Umap showing FlowSOM-guided clustering **(d)** and the expression intensity of the indicated marker **(e)** on stochastically selected hepatic CD8⁺ T-cells of 12 months ND, CD-HFD or CD-HFD-fed mice + 8 weeks treatment by α-PD-1 antibodies (ND n= 6 mice; CD-HFD n= 5 mice; CD-HFD + α-PD-1 n= 6 mice). **(f)** Quantification of intracellular Calcium and **(g)** Foxo1 levels in CD4⁺ T-cells by flow cytometry of 12 months ND, CD-HFD or CD-HFD-fed mice + 8 weeks treatment by α-PD-1 antibodies ((Foxo1: ND n= 6 mice; CD-HFD n= 5 mice; CD-HFD + α-PD-1 n= 7 mice; calcium: ND n= 13 mice; CD-HFD n= 10 mice; CD-HFD + α-PD-1 n= 10 mice). **(h)** Polarization analysis by flow cytometry of hepatic CD4⁺PD-1⁺ T-cells of 12 months ND, CD-HFD or CD-HFD-fed mice + 8 weeks treatment by α-PD-1 antibodies (ND n= 6 mice; CD-HFD n= 5 mice; CD-HFD + α-PD-1 n= 6 mice). **(i)** Relative quantification of hepatic CD8⁺PD-1⁺ and **(j)** CD4⁺PD-1⁺ T-cells by flow cytometry of 12 months ND, CD-HFD or CD-HFD-fed mice + 8 weeks treatment by α-PD-1 antibodies (ND n= 6 mice; CD-HFD n= 5 mice; CD-HFD + α-PD-1 n= 6 mice). **(k)** Polarization by flowcytometry of hepatic CD8⁺PD-1⁺ T-cells of 12 months ND, CD-HFD or CD-HFD-fed mice + 8 weeks treatment by α-PD-1 antibodies (ND n= 12 mice; CD-HFD n= 7 mice; CD-HFD + α-PD-1 n= 6 mice).

Rebuttal Figure 31: RNA velocity analyses on CD4 T-cells in NASH

(a) RNA Velocity analyses of scRNA-seq data showing expression, and (b) velocity of patient-liver-derived CD4+
 T-cells of control, or NAFLD/NASH patients in comparison to mouse-liver-derived CD4+ T-cells (patients:
 NAFLD/NASH n= 3 patients; mouse: n= 3 mice/group). (c) Correlation of expression along the latent-time of
 selected genes along the latent-time (mouse: n= 3 mice/group).

Rebuttal Figure 32: Severity of NAS is associated with CD8 and PD-1 expression
(a) RNA-sequencing data comparing NASH with varying fibrosis (F0 – F4 according to Brunt classification) normalized to NAFLD from a total of n= 206 NAFLD/NASH patients corrected for batch, gender and center. **(b)** Single gene PD-1 correlation analysis of RNA-sequencing data from a total of n= 206 NAFLD/NASH patients corrected for batch, gender and center. **(c)** Quantification of hepatic parenchymal PD-1, parenchymal CD8, parenchymal CD4 parenchymal and **(d)** portal tract TNF expressing cells of NAFL and NASH with varying fibrosis patients (NAFL n= 9 patients; NASH F1/0 n= 7 patients; NASH F2 n= 12 patients; NASH F3 n= 21 patients; NASH F4 n= 16 patients; CD4: NAFL n= 6 patients; NASH F1/0 n= 4 patients; NASH F2 n= 8 patients; NASH F3 n= 17

Research for a Life without Cancer

patients; NASH F4 n= 9 patients). **(e)** (c) Correlation analysis of PD-1 against TNF by RNA-sequencing or NAS by
immunohistochemical staining (NAFLD/NASH n= 65 patients). **(f)** Immunofluorescence staining of PD-1 and CD8
of NAFL and NASH with varying fibrosis patients. Arrowheads indicate CD8⁺PD-1⁺ cells. Scale bar: 50 μm. (g)
Immunohistochemical staining of PD-L1 in patient-derived liver samples. Scale bar: 50 μm.

Research for a Life without Cancer

Rebuttal Figure 33: CD8⁺PD-1⁺ T-cells drive necro-inflammation induced hepatocarcinogenesis in NASH

(a) Quantification of mRNA *in situ* hybridization for hepatic TNF⁺ cells of 12 months CD-HFD or CD-HFD-fed mice + 8 weeks treatment by α-PD-1 antibodies with or without tumor (without tumor: CD-HFD n = 30 field of view (FOV) in 3 mice; CD-HFD + α-PD-1 n = 40 FOV in 3 mice; peri-tumoral: CD-HFD n = 20 FOV in 3 mice; CD-HFD + α-PD-1 n = 21 FOV in 3 mice; intra-tumoral: CD-HFD n = 19 FOV in 3 mice; CD-HFD + α-PD-1 n = 22 FOV in 3 mice). Arrowheads indicate TNF⁺ cells. Scale bar: 20 μm. (b) Quantification of CD8 staining of peri- and intra-tumoral hepatic tissue by immunohistochemistry of 12 months CD-HFD or CD-HFD-fed mice + 8 weeks treatment by α-PD-1 antibodies (peri-tumoral: CD-HFD n = 11 mice; CD-HFD + α-PD-1 n = 10 mice; intra-tumoral: CD-HFD n = 5 mice;

Research for a Life without Cancer

CD-HFD + α -PD-1 n= 7 mice). **(c)** Histological staining of hepatic tumor tissue by Collagen IV, cleaved Caspase 3,
CD8, Ki-67 of 12 months CD-HFD or CD-HFD-fed mice + 8 weeks treatment by α -PD-1 antibodies (Collagen IV,
cleaved Caspase 3: CD-HFD n= 13 mice; CD-HFD + α -PD-1 n= 14 mice; CD8, Ki-67: CD-HFD n= 5 mice; CD-HFD
+ α -PD-1 n= 7 mice). Arrowheads indicate positive cells. Dotted line indicates tumor/lesion rim. Tumor area is
indicated by T. Scale bar: 100 μ m. **(d)** Quantification of Ki-67 staining of peri- and intra-tumoral hepatic tissue by
immunohistochemistry of 12 months CD-HFD or CD-HFD-fed mice + 8 weeks treatment by α -PD-1 antibodies (CD-
HFD n= 11 mice; CD-HFD + α -PD-1 n= 10 mice). **(e)** Scoring of expression by immunohistochemistry staining of
intra- and **(d)** peri-tumoral hepatic tissue of 12 months CD-HFD or CD-HFD-fed mice + 8 weeks treatment by α -PD-
1 antibodies (CD-HFD n= 13 mice; CD-HFD + α -PD-1 n= 14 mice). Crossed out boxes indicate not sufficient tissue
for analysis. **(f)** Histological staining of intra-tumoral hepatic tissue by pSTAT1, or pSTAT3 of 12 months CD-HFD
or CD-HFD-fed mice + 8 weeks treatment by α -PD-1 antibodies (CD-HFD n= 13 mice; CD-HFD + α -PD-1 n= 14
mice). Arrowheads indicate staining positive cells. Scale bar: 50 μ m **(g)** Quantification of **(h)** genomic aberrations
by array comparative genomic hybridization (aCGH) of tumor tissues of mice after 12 months on CD-HFD-fed mice
(n= 9) or 12 months on CD-HFD-fed mice + 8 weeks treatment with α -PD-1 antibodies (n= 12).

Research for a Life without Cancer

Rebuttal Figure 34: Inflammation associated hepatic cytokine and chemokine environment in NASH
(a) and (b) multiplex ELISA of hepatic inflammation associated cytokines and (c) chemokines of 12 months ND or CD-HFD-fed mice (ND n= 10 mice; CD-HFD n= 14 mice).

Rebuttal Figure 35: α -PD-1 treatment causes enrichment of inflammation- and apoptosis-associated pathways in NASH
(a) Immunofluorescence microscopy of 12 months ND, CD-HFD or CD-HFD-fed mice + 8 weeks treatment by α -PD-1 antibodies (n= 3 mice/group). Scale bar: 100 μ m. **(b)** Gene set enrichment analysis of RNA sequencing data of hepatic tissue comparing CD-HFD with CD-HFD-fed mice + α -PD-1 of 12 months ND, CD-HFD or CD-HFD-fed mice + 8 weeks treatment by α -PD-1 antibodies (n= 5 mice/group).

Reviewer Reports on the First Revision:

Referee #1 (Remarks to the Author):

The authors have added an incredible amount of additional data from mouse models and clinical studies, and further strengthened their underlying hypotheses. The underlying message is important to the basic and clinical HCC communities. However, the concerns about manuscript with two underlying hypotheses, that are not sufficiently linked, remain.

I. The mouse studies in NASH-HCC models showing a tumor-promoting role of CD8+ PD-1+ T cells by increasing NASH, which is particularly strong when PD1-blockade is done early. This is corroborated by correlative human scRNA-seq, FACS and IHC studies showing immune alterations associated with NASH in patients confirming above data. These studies are well performed; however not strongly linked to the title of paper or the second part; as this is not relevant to immunotherapy which is given to patient with advanced unresectable HCC, and not to NASH patients or for HCC prevention. The concept of immunosurveillance would be an opportunity to link the two parts, but is not sufficiently investigated.

II. The mouse studies show a lack of response to checkpoint inhibition with a trend towards increased HCC as well as increased fibrosis when given as therapeutic treatment and increase HCC and NASH when given as prevention treatment. This is paralleled by human data showing lack of response to checkpoint inhibitors in patients with non-viral HCC. There are concerns about mouse studies as therapeutic checkpoint inhibitor approaches are performed when there is still underlying NASH activity; and there are limitations in the human section as all analyzed focus on non-viral HCC.

Main criticism:

1. The authors mingle above two concepts in the paper that are not necessarily linked and do not sufficiently separate these ideas: 1. The idea of immune-mediated NASH promotion. 2. The idea of failing restoration of anti-tumor immunity in NASH-HCC. The dual role of the immune systems in this long disease process is highlighted in the paper, but the fact that these are likely stage-specific functions, mentioned in the previous review, is not addressed. The authors failed to separate these in the mouse models, where they study the effect of checkpoint inhibitors in settings where NASH is still maintained via continued CD-HFD, both for prevention and therapeutic treatments. Not only does this not reflect the situation in most patients, who have advanced HCC and no to little ongoing NASH, but is also makes it difficult to interpret the data, i.e. is a potential beneficial of checkpoint inhibitors on tumors covered up by the increased NASH activity under checkpoint inhibition; or is it simply not present; or is there even the opposite?

2. The studies suggesting failure of checkpoint inhibition in NASH-HCC in mice and patients each have limitations. The provided information are incredibly important and potentially impactful but for this reason studies and data need to be very carefully performed, analyzed and interpreted to guide the field into the right direction.

2a. It is doubtful whether the employed mouse HCC studies are helpful to guide the field towards new concepts in NASH-HCC therapy - as the models first of all seem to rely on continued NASH with ongoing CD-HFD when therapies are given, making the factors driving HCC progression different from patients; it cannot be excluded that the HCC growth in these models still requires ongoing NASH; and second, these are not sufficiently established and characterized preclinical models, making it unclear whether the failure is due to model or species specific, such as lacking similarities like sufficiently high mutational load and other characteristic that dictate response to immunotherapy in HCC patients. It also needs to be emphasized that all patients receiving checkpoint inhibitors have advanced HCC, usually outside of Milan criteria, which again may not be

achieved in mouse models. The authors provided additional data on their model, but mutational load is not specifically addressed. A key factor would be to have a model in which tumors continue to grow in the absence of NASH diet, within an environment that is similar to human HCC (cirrhotic). As this may be impossible and as mice could still respond differently, the strong focus on mouse models is not ideal to address the essence of this question, and more emphasis should have been on human trials. The main point of mouse models would be to address underlying mechanisms after having established key data in patients - the reverse approach is inherently less convincing and more difficult, requiring nearly perfect models.

2b. With the provided concept, suggesting that CD8+PD1+ cells promote NASH, and thereby driving HCC, but do not provide tumor-suppressing immune surveillance, the paper does not reflect that NASH carries a significantly lower risk for HCC development than other chronic liver diseases. The proposed concept of a failing immunosurveillance by CD8+PD1+ cells, based on mouse models, appears to not fully reflect clinical reality.

2b. The analysis of human checkpoint inhibitor trials is greatly expanded, but it is almost a paper within a paper that could be published on its own. Rather than trying to address some of the inherent human study weaknesses through analysis in mouse models, the authors should have attempted a deeper analysis of these data. One of the key points are whether the non-viral HCC is indeed mostly NASH-HCC; whether NASH-HCC has a different prognosis due to the underlying metabolic complications affecting other organ systems; moreover other differences such as age (a key factor in HCC outcomes) and gender are not consistently analyzed in Tables S9 and S10.

2c. Related to 2b, the authors did not ask and answer the question whether the divergent data on checkpoint inhibition in different patient groups are because of (A) presence of viral antigens that enhance anti-tumor immunity, especially in HBV-associated HCC; or whether (B) the lack of response in non-viral HCC is indeed related to specific NASH-HCC and its liver and immune environment as suggested by the entire paper.

2d. Subanalysis of cirrhotic NASH-HCC (lacking significant NASH activity) and non-cirrhotic NASH-HCC (where some patients may still have some NASH-activity) is not addressed, and admittedly extremely difficult (but the combination with a mouse model with NASH activity does trigger this question).

In summary, the data in the manuscript are relevant, highlighting the dual role of the immune system. However, mouse models are not sufficiently strong to answer questions on CD8+PD1+ cells and anti-tumor immunity independent of the associated NASH-driving functions; patient data lack in-depth analysis, lacking explanation of the data and addressing the question whether viral antigens may explain many of the observed differences rather than specifics of NASH-HCC and its environment; and the concepts of immune-mediated NASH promotion vs the failure of checkpoint inhibition in advanced NASH-HCC tumors lack a sufficient distinction due to the lacking analysis of stage-specific functions of these two underlying immune-mediated tumor-promoting and restricting mechanisms throughout the paper, due to lacking focus on this question as well as lacking mouse models that could satisfactorily answer this question. Hence, major overhaul and rewrite of the paper is needed, highlighting its strengths, and addressing above points so that the field benefits. One point that could link the two hypotheses would be failing immunosurveillance in the precancerous NASH liver. The paper will raise the important question whether NASH-HCC benefits from immunotherapy, but it will not be able to fully answer it (which is ok - it can be the basis for new prospective studies). However, the authors should make this message clearer to the audience by separating the two topics, i.e. (A) the role of the immune system in driving NASH and (B) the potential failure of checkpoint inhibitors in advanced NASH-HCC. Without this, the authors will not sufficiently reach and impact the basic or clinical research community.

Referee #3 (Remarks to the Author):

I do congratulate authors for an amazing effort to satisfy the many questions and concerns asked and posed by the reviewers. The revised version is much improved and in my view advances our knowledge on this matter a very considerable extent. The interconnection with the back-to-back paper is excellent. I am glad that my suggestion of TNF blockade leads to experimental postulation of a clinically actionable target. Moreover, in the companion manuscript this has been mechanistically explored as well.

Referee #4 (Remarks to the Author):

The rebuttal is very thorough. All of the technical points seem to have been addressed thoroughly. However, I'm left a bit concerned about the actual conclusions of the paper.

On the one hand there is a very thorough mechanistic study in mice which reveals a clear in vivo phenotype in vivo that CPI therapy promotes carcinogenesis. On the other there is clinical data - which is much improved - that different aetiologies of HCC respond differently to CPI. So it seems on the face of it that HCCs in NASH are harder to treat with the current monotherapy - that is a fine conclusion from the data and should certainly help design new stratified trials as suggested (and maybe would be the starting point for the paper). But almost all of the rest of the paper is about pathways to cancer development - there did not seem to be a strong signal from the human trial data or from the mouse in vivo experiment that CPI therapy of an established tumour made things worse.

It seemed to me currently the two bits are not that well connected - really addressing different issues - and the only experiment which clearly linked them is a negative result tucked away in extended figure 6/7 and is not really explored any further. For that it would be really helpful to show that this lack of response was linked to the CD8 phenotype seen. So a parallel experiment with a distinct non-NASH model of HCC which did show an impact of CPI would seem important. In other models published, even if responsive to dual checkpoint inhibition, there is limited response to PD1/PDL1 alone. So making the link between the underlying pathogenesis/CD8 phenotype and the responsiveness to CPI still needs to be clarified.

Author Rebuttals to First Revision

(please note that the authors have quoted the reviewers in black and responded in blue)

Referee #1 (Remarks to the Author):

The authors have added an incredible amount of additional data from mouse models and clinical studies, and further strengthened their underlying hypotheses. The underlying message is important to the basic and clinical HCC communities. However, the concerns about manuscript with two underlying hypotheses, that are not sufficiently linked, remain.

I. The mouse studies in NASH-HCC models showing a tumor-promoting role of CD8+ PD-1+ T cells by increasing NASH, which is particularly strong when PD1-blockade is done early. This is corroborated by correlative human scRNA-seq, FACS and IHC studies showing immune alterations associated with NASH in patients confirming above data. These studies are well performed; however not strongly linked to the title of paper or the second part; as this is not relevant to immunotherapy which is given to patient with advanced unresectable HCC, and not to NASH patients or for HCC prevention. The concept of immunosurveillance would be an opportunity to link the two parts, but is not sufficiently investigated.

We thank Referee #1 for appreciating our well performed studies and we agree with this Referee that the concept of immunosurveillance is appealing and could underline the connection between our mouse and the human data in a new light. To accommodate the message in this fresh light, we have changed the title of our manuscript into **"Immune-checkpoint blockade stalls immunosurveillance and anti-tumor effects in NASH-HCC"**.

We will further explain this concept in the text.

Of note, in the light of the IMbrave150 study PD-(L)1, targeted immunotherapy will most likely become the new standard of care first line therapy not only for advanced HCC, but also in a preventive fashion upon surgical intervention to avoid liver cancer recurrence – thus we believe that also the aspect of chemoprevention is an important issue.

II. The mouse studies show a lack of response to checkpoint inhibition with a trend towards increased HCC as well as increased fibrosis when given as therapeutic treatment and increase HCC and NASH when given as prevention treatment. This is paralleled by human data showing lack of response to checkpoint inhibitors in patients with non-viral HCC. There are concerns about mouse studies as therapeutic checkpoint inhibitor approaches are performed when there is still underlying NASH activity; and there are limitations in the human section as all analyzed focus on non-viral HCC.

We thank Referee #1 for highlighting ongoing feeding of NASH-inducing diet in our therapeutic PD-1-targeted immunotherapy scheme. One interesting concept for immunotherapy treatment for the future might be *"to first treat the metabolic disorder and normalize the hepatic and systemic dyslipidemia, then tackle HCC/tumor surveillance"*, which we will now address our revised manuscript.

We are in line with Referee#1, that the human non-viral HCC data leaves questions unanswered. Given that we did not have access to the raw data of the 3 phase III trials (led by

companies) included in the meta-analysis, we cannot provide information about the distribution of alcohol-related vs. NASH-related HCC within the non-viral HCC group.

Notably, we have added two 2 retrospective cohorts of HCC patients treated with immunotherapy, in which we addressed this issue by comparing NASH-related HCC vs. non-NASH etiologies. Here, we were able to demonstrate that NASH-related HCC was associated with a worse outcome in 2 independent cohorts. We agree that these retrospective cohorts still have their limitations, but will highlight that they could pave the way for prospective studies.

Main criticism:

1. The authors mingle above two concepts in the paper that are not necessarily linked and do not sufficiently separate these ideas:

1. The idea of immune-mediated NASH promotion. 2. The idea of failing restoration of anti-tumor immunity in NASH-HCC.

The dual role of the immune systems in this long disease process is highlighted in the paper, but the fact that these are likely stage-specific functions, mentioned in the previous review, is not addressed. The authors failed to separate these in the mouse models, where they study the effect of checkpoint inhibitors in settings where NASH is still maintained via continued CD-HFD, both for prevention and therapeutic treatments. Not only does this not reflect the situation in most patients, who have advanced HCC and no to little ongoing NASH, but is also makes it difficult to interpret the data, i.e. is a potential beneficial of checkpoint inhibitors on tumors covered up by the increased NASH activity under checkpoint inhibition; or is it simply not present; or is there even the opposite?

We thank Referee #1 for raising this point. The idea of immune-mediated NASH promotion can be indeed demonstrated by our data. To address the second point of failing restoration of anti-tumor immunity in NASH-HCC - we have analyzed liver tissue from established NASH-HCC that were treated with immune-check point blockade by flow cytometry markers of exhaustion, cytotoxicity, or re-activation. In none of the analyzed markers, we observed any differences between CD-HFD and CD-HFD + anti-PD-1 treatment, arguing against a tumor restricting role of CD8+ T-cells with ongoing CD-HFD feeding in the therapeutic setting. Like written in our previous response we saw a small but significant increase of CD44^{CD62L}CD8 T cells (infiltrating memory cells), however we do not observe an anti-tumor effect.

This corroborates our immunohistochemistry data, in which we did not observe any difference of PD-1+ or CD8+ T-cells in peri- or intra-tumoral liver tissue. These data are now included in a revised manuscript (**Supplementary Information**).

We have initially shown that over time CD8 T cells fuel steatosis and NASH development. Over time generation of hepatic, resident CXCR6+ PD1+CD8+ T cells that are dysfunctional for effective tumor-surveillance can be detected. However, these CD8 T

cells are hyperactivated (as analyzed by single cell RNA Seq, flow cytometry) and auto-aggressive as shown by Dudek et al.. Thus, PD1CD8+ T cells are generated progressively filling up the pool of hepatic CD8+ T cells over time in the context of NASH. Experiments of CD8+ T cell depletion in established precancerous NASH show that the CD8+ T cell population contributes to liver cancer development – as CD8+ T cell depletion reduced HCC incidence.

Immune-checkpoint blockade either given at established NASH or at NASH with HCC exacerbates this phenotype by increasing the number of these hyper-activated PD1CD8+ T cells. At early stages of NASH/HCC immune check point blockade increases HCC incidence - at late stage HCC immune check point blockade cannot function anymore as the pre-existing T cells **cannot execute efficiently tumor surveillance**. Thus, a potential beneficial role of checkpoint inhibitors on tumors is overruled mainly by auto-aggressive T-cells. We cannot exclude on single cell level that there are single tumor specific cells present but if so they not only kill tumor cells and thus will induce also compensatory proliferation by randomly killing other cells (as shown by Dudek et al.).

Our single cell RNA seq. data as well as our flow cytometry analyses of PD1+ CD8+ T cells in the context of NASH as well as in the context of NASH/HCC clearly show that hepatic PD1CD8+ T cells do not change in quality but rather in number over time. Our velocity analysis indicates one final, endpoint CD8 +T cell population that increases over time progressively with NASH development which is a PD1+CD8+CXCR6+ T cell population - reminiscent of the auto-aggressive T cell population by Dudek et al. Moreover, in this co-submitted manuscript this stable character of auto-aggressive T cells over time is described, corroborating our findings. We will further discuss the concept in the revised manuscript that potential treatment of the underlying chronic inflammation (e.g. reverting NASH) may precede immunotherapy.

2. The studies suggesting failure of checkpoint inhibition in NASH-HCC in mice and patients each have limitations. The provided information are incredibly important and potentially impactful but for this reason studies and data need to be very carefully performed, analyzed and interpreted to guide the field into the right direction.

We thank Referee #1 for acknowledging the information of our data as “incredibly important and potentially impactful” and agree, that a careful handling needs to be done – this is what we have intended in the revised version of our manuscript.

2a. It is doubtful whether the employed mouse HCC studies are helpful to guide the field towards new concepts in NASH-HCC therapy - as the models first of all seem to rely on continued NASH with ongoing CD-HFD when therapies are given, making the factors driving HCC progression different from patients; it cannot be excluded that the HCC growth in these models still requires ongoing NASH; and second,

these are not sufficiently established and characterized preclinical models, making it unclear whether the failure is due to model or species specific, such as lacking similarities like sufficiently high mutational load and other characteristic that dictate response to immunotherapy in HCC patients. It also needs to be emphasized that all patients receiving checkpoint inhibitors have advanced HCC, usually outside of Milan criteria, which again may not be achieved in mouse models. The authors provided additional data on their model, but mutational load is not specifically addressed. A key factor would be to have a model in which tumors continue to grow in the absence of NASH diet, within an environment that is similar to human HCC (cirrhotic). As this may be impossible and as mice could still respond differently, the strong focus on mouse models is not ideal to address the essence of this question, and more emphasis should have been on human trials. The main point of mouse models would be to address underlying mechanisms after having established key data in patients - the reverse approach is inherently less convincing and more difficult, requiring nearly perfect models.

We thank Referee #1 for his/her comment and would like to state, that although CD8-depleted animals (former Figure 2, now Extended Data 7) have elevated NAS, and ALT and are metabolically impaired with all features of NAFLD/NAS, they lack liver cancer in the preventive treatment regimen. We now included distinct non-NASH liver cancer models into the manuscript to demonstrate that in the absence of NASH - immunotherapy does prolong animal survival and reduces liver cancer development.

Moreover, we cannot rule out entirely that species specific contribute to lack of response in the mouse models. However, our clinical data are in line with those obtained from our animal studies, as patients with NASH-related HCC treated with immunotherapy also had a poorer outcome.

Nevertheless, we acknowledge the limitations of such retrospective analyses and will clearly state in the discussion section that prospective validation of these findings is warranted. Our data could pave the way for the design of such prospective protocols. Additionally, we want to emphasize that currently there is no established and validated biomarker that predicts response to immunotherapy in HCC (as reviewed in Pinter M et al. JAMA Oncol 2020). Tumor mutational load is generally low in HCC and its use as a predictive biomarker in HCC is not supported by available clinical data (Ang C et al. Oncotarget 2019;10:4018. Harding JJ et al. Clin Cancer Res 2019; 25:2116. Wong CN et al. Liver Int 2020;Epub ahead of print). We agree with the reviewer that most patients with NASH-related HCC suffer from concomitant cirrhosis – but they also in general suffer from systemic dyslipidemia and obesity.

Thus, we do not believe that continuation of NASH-inducing diet in mice is in conflict with the clinical scenario in humans. On the contrary, patients with NASH usually remain exposed to metabolic risk factors (e.g., overweight, unhealthy diet, hypertension, hyperlipidemia, lack of exercise) even after they developed HCC. Only a minority of patients is able to dramatically

change their lifestyle. Thus, we agree that the mouse models are not perfect but they are still representative for our conducted clinical observation in large parts. We discuss this issue and potential complications in our discussion.

2b. With the provided concept, suggesting that CD8+PD1+ cells promote NASH, and thereby driving HCC, but do not provide tumor-suppressing immune surveillance, the paper does not reflect that NASH carries a significantly lower risk for HCC development than other chronic liver diseases. The proposed concept of a failing immunosurveillance by CD8+PD1+ cells, based on mouse models, appears to not fully reflect clinical reality.

We agree with Referee #1 that the used mouse model might only represent subsets of patients, as some metabolic-impaired patients with liver features reminiscent of NAFLD/NASH pathology react in a long-lasting manner to immunotherapy. **We rather even state that NASH does not allow CD8+ T cells to exert their function of immune-surveillance and this is exacerbated by immune check point blockade (see our point auto-aggression in human and mouse NASH).** So far, no clinical consensus could be achieved for biomarkers to discriminate those patients, who might benefit from immunotherapy. Moreover, we would like to draw attention to the suboptimal setup of the retrospective clinical design and the need for prospective validation of the proposed concept. We tone down our interpretation and discuss that the need for stratification of patients, who might benefit from immunotherapy, cannot only rely on the underlying inflammatory etiology.

2b. The analysis of human checkpoint inhibitor trials is greatly expanded, but it is almost a paper within a paper that could be published on its own. Rather than trying to address some of the inherent human study weaknesses through analysis in mouse models, the authors should have attempted a deeper analysis of these data. One of the key points are whether the non-viral HCC is indeed mostly NASH-HCC; whether NASH-HCC has a different prognosis due to the underlying metabolic complications affecting other organ systems; moreover other differences such as age (a key factor in HCC outcomes) and gender are not consistently analyzed in Tables S9 and S10.

We agree with the comment of Referee #1 and would like to raise awareness, that the clinical data were obtained retrospectively and outside of a prospective clinical trial protocol, biopsies before initiation of immunotherapy were not mandatory and thus are not available in the majority of patients. Therefore, we were not able to obtain data regarding the immune microenvironment with respect to different underlying etiologies. We now analyzed age and gender in our multivariate analysis (**Supplementary Table 9**) and these two factors were not a confounder of HCC treatment outcome.

Research for a Life without Cancer

2c. Related to 2b, the authors did not ask and answer the question whether the divergent data on checkpoint inhibition in different patients groups are because of (A) presence of viral antigens that enhance anti-tumor immunity, especially in HBV-associated HCC; or whether (B) the lack of response in non-viral HCC is indeed related to specific NASH-HCC and its liver and immune environment as suggested by the entire paper.

We thank Referee #1 for this important point and would like to refer to our answer previously given, that the human non-viral HCC data leaves unanswered questions behind. Given that we did not have access to the raw data of the 3 phase III trials included in the meta-analysis, we cannot provide information about the distribution of alcohol-related vs. NASH-related HCC within the non-viral HCC group, or level of viral antigens, that potentially enhance anti-tumor immunity. However, in our 2 retrospective cohorts of HCC patients treated with immunotherapy, we addressed this issue by comparing NASH-related HCC vs. non-NASH etiologies. We were able to demonstrate that NASH-related HCC was associated with a worse outcome in 2 independent cohorts. We agree that these retrospective cohorts still have their limitations, but they could pave the way for prospective studies.

2d. Subanalysis of cirrhotic NASH-HCC (lacking significant NASH activity) and non-cirrhotic NASH-HCC (where some patients may still have some NASH-activity) is not addressed, and admittedly extremely difficult (but the combination with a mouse model with NASH activity does trigger this question).

We agree with Referee #1, that analyses of non-cirrhotic NASH-HCCs might give valuable insights but is extremely difficult. Thus we were not able to perform further meaningful sub-analyses.

In summary, the data in the manuscript are relevant, highlighting the dual role of the immune system. However, mouse models are not sufficiently strong to answer questions on CD8+PD1+ cells and anti-tumor immunity independent of the associated NASH-driving functions; patient data lack in-depth analysis, lacking explanation of the data and addressing the question whether viral antigens may explain many of the observed differences rather than specifics of NASH-HCC and its environment; and the concepts of immune-mediated NASH promotion vs the failure of checkpoint inhibition in advanced NASH-HCC tumors lack a sufficient distinction due to the lacking analysis of stage-specific functions of these two underlying immune-mediated tumor-promoting and restricting mechanisms throughout the paper, due to lacking focus on this question as well as lacking mouse models that could satisfactorily answer this question. Hence, major overhaul and rewrite of the paper is needed, highlighting its strengths, and addressing above points so that the field benefits. One point that could link the two hypotheses would be failing immunosurveillance in the precancerous NASH liver. The paper will raise the important question whether NASH-HCC benefits from immunotherapy, but it will not be able to fully answer it (which is ok -

it can be the basis for new prospective studies). However, the authors should make this message clearer to the audience by separating the two topics, i.e. (A) the role of the immune system in driving NASH and (B) the potential failure of checkpoint inhibitors in advanced NASH-HCC. Without this, the authors will not sufficiently reach and impact the basic or clinical research community.

We thank Referee #1 for the constructive points and thus re-organized our manuscript to underline/highlighting its strengths while critically raising the point of prospective validation. **We will link the 2 hypotheses indicated above through failing immuosurveillance in the precancerous NASH liver in the revised version of our manuscript.**

Referee #2 (Remarks to the Author):

In their (too) lengthy rebuttal, Pfister et al., provide two key additions to their original manuscript. First, they increase the size of their original patient cohort and add a validation cohort and secondly, they perform in vivo depletion experiments to determine the involvement of several cell types and inflammatory mediators in establishing anti-PD-1-accelerated hepatocarcinogenesis. While the number of NAFLD patients included in this manuscript is still low (n=13 and n=11 respectively), it is encouraging to see that the data obtained was similar between the two datasets. I agree with the authors that this is an interesting observation.

We thank Referee #2 for acknowledging our work and for the notion that we have provided key additions to our original manuscript.

However, despite the overabundance of data, the mechanism by which NASH (and/or NAFLD) predisposes to anti-PD-1-accelerated hepatocarcinogenesis remains largely unclear. As I said in my original rebuttal, the data presented by the authors fail to demonstrate clear causal relationships. As an example, the authors present cytokine measurements after several antibody-based interventions in Extended Data 21, but fail to determine which of these are important. They state that liver inflammation is reduced upon CD8 depletion (which is a solid and interesting result), yet, for example, IFN γ , IL-21 and IL-31 remain unchanged. What do the authors base their statements on?

We acknowledge the comment of Referee #2 but we would like to add that not all inflammatory mediators are reduced upon CD8 depletion - as indicted by our ICF signature analysis. We base our statement on immunohistochemistry describing reduced T-cell infiltration into the liver. We have now specified this further in the revised manuscript.

Are these mediators not inflammatory? Which mediators instead would indicate an inflammatory environment?

These are all inflammatory mediators, but many of them do not correlate with disease. We have now specified this further in our revised manuscript. IL31 for example does not correlate with NAS, ALT,

sirius red or tumor incidence. In contrast, our correlative analysis as well as our convolutional neural network analysis identified levels of TNF to correlate with liver cancer incidence.

Also, why can significant amounts of TNF still be found in conditions of TNF depletion? Similarly, why is TNF not significantly down in CD-HFD + anti-PD-1/anti-TNF when compared to CD-HFD + anti-PD-1? Does this not indicate that the authors' intervention did not work? And if that is the case, why is there a significant effect of anti-PD-1 + anti-TNF treatment on tumor lesions relative to anti-PD-1 only?

We politely disagree with Referee # 2. The TNF inhibition has worked as we have observed significant decrease of CD8+ T cells, PD1+ T cells, MHCII+ and F480+cells.

Along the same lines, I had asked in my original review about the involvement of CD4 T cells. The authors have now performed CD4 depletion experiments, and they state that this 'did not decrease liver pathology or liver inflammation (lines 471-472)', yet they show that TNF, the molecule they say is responsible for causing liver inflammation, is in fact significantly less abundant (Extended Data 23c, CD-HFD + anti-PD-1 vs CD-HFD + anti-PD-1/anti-CD4), as is IL-21, IL-33, IL-1B and IL-13 (amongst others). Are these mediators not inflammatory? Which mediators then indicate an inflammatory environment?

We thank Referee #2 for this notion and we are happy that this Referee has acknowledged our data on the role of CD4 T cells. Again as stated above we will be more precise when stating the term inflammatory environment and will write this more specifically in the main text and discussion.

To compound these issues, I believe the authors have actually stumbled upon an interesting finding regarding these CD4 cells, which they seem to have overlooked. While tumor incidence is similar upon depletion of CD4 cells in the context of CD-HFD + anti-PD-1 (Fig. 4n), the number of tumors per liver and the individual lesion size is actually reduced (Extended Data 24a, b). This would imply that the CD4 cells actually do play a role in the authors' proposed mechanism. It is unclear to me why the authors would not follow up on this important aspect of their mechanism, especially since they put a lot of emphasis on the CD4 cells when discussing their patient data.

We thank Referee# 2 for this statement and agree that this is an interesting finding and might be in line with CTLA4 mediated T-reg depletion (a particular subset of T cells) - and indeed we had already noted and discussed this in the revised paper that CD4 T cells might contribute to tumor progression – but not initiation – as tumor incidence was unchanged in a CD4 T cell depletion setting. Thus, we believe that in the mouse and the human setting CD4 T cells might play a role in tumor progression and we have now re-enforced this statement in the discussion section.

Lastly, the authors remain highly selective in the data they choose to discuss and how they interpret

it (also see my points before). To illustrate more examples:
line 247 and Figure 1e, more populations are affected than just the CD8 compartment;
line 250 Extended Data 3g, CD4 cells actually increase significantly;
line 264-266 and Figures 1i-j and Extended Data 5a-d, CD8 cells both lose and gain cytotoxic function by RNA seq (see Gzma/Gzmb for example) and the effect size of CD4 cells seems the same if not larger than the CD8 T cells;
line 273-276 and Extended Data 5e, not everything was validated by mass spec including importantly the finding of enhanced Tox expression;
line 342-343 and Extended Data 11f, CD4 cells do also change significantly,
line 343-344 and Extended Data 11k-l, While NKT and NK cells do not increase effector cytokine production neither do CD8 T cells;
line 431-433 and Extended Data 18e, the authors actually present significant data;
line 434-436 and Extended Data 18f, the authors actually present significant data;
line 456-457 and Extended Data 19f-h, the authors present significant data for CD4 cells and largely in the same order of magnitude as their findings for CD8 cells

We thank Referee# 2 for the statements. As the functional data show a very strong effect with CD8 T cells we focus on this cell population in this manuscript – still we do cite and discuss the other populations as well. We thank Referee# 2 for these points and will adjust those accordingly.

Referee #3 (Remarks to the Author):

I do congratulate authors for an amazing effort to satisfy the many questions and concerns asked and posed by the reviewers. The revised version is much improved and in my view advances our knowledge on this matter a very considerable extent. The interconnection with the back-to-back paper is excellent. I am glad that my suggestion of TNF blockade leads to experimental postulation of a clinically actionable target. Moreover, in the companion manuscript this has been mechanistically explored as well.

We thank Referee #3 for his insights and constructive comments throughout the review process.

Referee #4 (Remarks to the Author):

The rebuttal is very thorough. All of the technical points seem to have been addressed thoroughly. However, I'm left a bit concerned about the actual conclusions of the paper. On the one hand there is a very thorough mechanistic study in mice which reveals a clear in vivo phenotype in vivo that CPI therapy promotes carcinogenesis. On the other there is clinical data - which is much improved - that different aetiologies of HCC respond differently to CPI. So it seems on the face of it that HCCs in NASH are harder to treat with the current monotherapy - that is a fine conclusion from the data and should certainly help design new stratified trials as suggested (and maybe would be the starting point for the paper). But almost

all of the rest of the paper is about pathways to cancer development - there did not seem to be a strong signal from the human trial data or from the mouse in vivo experiment that CPI therapy of an established tumour made things worse.

We agree that our clinical data clearly indicate that immunotherapy seems to be less effective in patients with NASH-related HCC. We acknowledge that these retrospective analyses still have their limitations and need prospective validation. Our data could pave the way for the design of such prospective protocols.

It seemed to me currently the two bits are not that well connected - really addressing different issues - and the only experiment which clearly linked them is a negative result tucked away in extended figure 6/7 and is not really explored any further. For that it would be really helpful to show that this lack of response was linked to the CD8 phenotype seen. So a parallel experiment with a distinct non-NASH model of HCC which did show an impact of CPI would seem important. In other models published, even if responsive to dual checkpoint inhibition, there is limited response to PD1/PDL1 alone. So making the link between the underlying pathogenesis/CD8 phenotype and the responsiveness to CPI still needs to be clarified.

We thank Referee 4 for this important comment. We have now added distinct non-NASH liver cancer models that do respond to CPI – as a positive control and have included this into the Supplemental material.

Reviewer Reports on the Second Revision:

Referee #1 (Remarks to the Author):

The authors have addressed my concerns through rewriting of the manuscript, which has become more coherent, with mouse and human parts now fitting much better together. I only have two remaining comments/suggestions, which can be mostly addressed through editorial changes, but which are important for the overall interpretation of the manuscript:

1. The paper is not sufficiently clear on whether anti-PD1 affects injury (which promotes HCC) and thereby might overshadow its anti-tumor effects when given as “therapeutic” approach in NASH-HCC mice at later stages (13 months). The authors state author state that there is more pronounced liver damage - but show unaltered NAS score and decreased serum ALT in the anti-PD1-treated group in Extended Data 2F, i.e. no increase in damage. This data is in fact supporting a failure of checkpoint inhibitors in a therapeutic setting, and not just an increased injury (which is likely not occurring in most patients receiving checkpoint inhibitor therapy) overshadowing anti-tumor immunity in this mouse model. This would in the end be more similar to what is seen in patients and should be clearly pointed out in the results and discussion (and the statement on damage should be corrected). As there is more fibrosis, which is often the result of more injury but could also be the result of more inflammation in response to checkpoint inhibitors, the authors might want to additional check injury by TUNEL staining to be 100% sure.

1b. The authors have shown several additional mouse models in Extended Data 3, in which which anti-PD1 was effective, i.e. different from the NASH model - but the comparison is not fair as none of these mouse models had a chronic injury component. Fortunately, a recent paper addressed this point (Chung et al, PMID: 32839204), showing decreased HCC in a DEN+CCl4 HCC model when anti-PD1 was given in a preventative manner from week 10-20. This paper should be cited and discussed. The discussion of the Chung et al paper in the context of PD1 inhibitor-mediated NASH injury promotion and lack of therapeutic effects is important and will further improve the paper, providing more evidence that these are NASH-specific phenomena.

Referee #2 (Remarks to the Author):

The authors have improved the paper significantly, with better explanations, fewer overstatements and more focus.

A few final remarks:

-as far as I could see, the %age of TNF producing cells was not quantified, only images were shown (Figure 1n,o; Extended Data 2i-n).

-same for TNF expression (Extended Data 5a-g).

Referee #4 (Remarks to the Author):

The comments have all been addressed. In the title could the authors maybe use impacts on or something similar rather than precludes as that sounds a bit absolute.

Author Rebuttals to Second Revision

(please note that the authors have quoted the reviewers in black and responded in blue)

Referees' comments:

Referee #1 (Remarks to the Author):

The authors have addressed my concerns through rewriting of the manuscript, which has become more coherent, with mouse and human parts now fitting much better together. I only have two remaining comments/suggestions, which can be mostly addressed through editorial changes, but which are important for the overall interpretation of the manuscript:

1. The paper is not sufficiently clear on whether anti-PD1 affects injury (which promotes HCC) and thereby might overshadow its anti-tumor effects when given as “therapeutic” approach in NASH-HCC mice at later stages (13 months). The authors state that there is more pronounced liver damage - but show unaltered NAS score and decreased serum ALT in the anti-PD1-treated group in Extended Data 2F, i.e. no increase in damage. This data is in fact supporting a failure of checkpoint inhibitors in a therapeutic setting, and not just an increased injury (which is likely not occurring in most patients receiving checkpoint inhibitor therapy) overshadowing anti-tumor immunity in this mouse model. This would in the end be more similar to what is seen in patients and should be clearly pointed out in the results and discussion (and the statement on damage should be corrected). As there is more fibrosis, which is often the result of more injury but could also be the result of more inflammation in response to checkpoint inhibitors, the authors might want to additional check injury by TUNEL staining to be 100% sure.

We thank Reviewer #1 for his positive and constructive input and his help to make this paper more conclusive and focused. We agree with the point of Referee#1 that a failure of checkpoint inhibitors is strengthened by the reduced liver damage. Indeed, we did not observe an increase in liver damage (by Cl. Casp 3 staining) upon anti-PD1 treatment - corroborating the argument of Referee#1. We have now underlined this in the text.

1b. The authors have shown several additional mouse models in Extended Data 3, in which which anti-PD1 was effective, i.e. different from the NASH model - but the comparison is not fair as none of these mouse models had a chronic injury component. Fortunately, a recent paper addressed this point (Chung et al, PMID: 32839204), showing decreased HCC in a DEN+CCI4 HCC model when anti-PD1 was given in a preventative manner from week 10-20. This paper should be cited and discussed. The discussion of the Chung et al paper in the context of PD1 inhibitor-mediated NASH injury promotion and lack of therapeutic effects is important and will further improve the paper, providing more evidence that these are NASH-specific phenomena.

We thank Reviewer #1 for this comment. We have now cited the manuscript by Chung et al., and have discussed it in the main text.

Referee #2

The authors have improved the paper significantly, with better explanations, fewer overstatements, and more focus.

We thank Reviewer #2 for commenting that our manuscript has improved significantly.

A few final remarks:

-as far as I could see, the %age of TNF producing cells was not quantified, only images were shown (Figure 1n,o; Extended Data 2i-n).

We thank Referee #2 for the comment. We have highlighted the quantification in Figure 1n and added a quantification for former Figure 1o (which was now moved into the Extended data 2) - which is now also included in Extended data 2.

-same for TNF expression (Extended Data 5a-g).

TNF expression was quantified and is indicated in Extended data 5.